# Finding All $\epsilon$-Good Arms in Stochastic Bandits

**Blake Mason**
University of Wisconsin
Madison, WI 53706
bmason3@wisc.edu

**Lalit Jain**
University of Washington
Seattle, WA 98115
lalitj@uw.edu

**Ardhendu Tripathy**
University of Wisconsin
Madison, WI 53706
astripathy@wisc.edu

**Robert Nowak**
University of Wisconsin
Madison, WI 53706
rdnowak@wisc.edu

## Abstract

The pure-exploration problem in stochastic multi-armed bandits aims to find one or more arms with the largest (or near largest) means. Examples include finding an $\epsilon$-good arm, best-arm identification, top-$k$ arm identification, and finding all arms with means above a specified threshold. However, the problem of finding *all* $\epsilon$-good arms has been overlooked in past work, although arguably this may be the most natural objective in many applications. For example, a virologist may conduct preliminary laboratory experiments on a large candidate set of treatments and move all $\epsilon$-good treatments into more expensive clinical trials. Since the ultimate clinical efficacy is uncertain, it is important to identify all $\epsilon$-good candidates. Mathematically, the all-$\epsilon$-good arm identification problem presents significant new challenges and surprises that do not arise in the pure-exploration objectives studied in the past. We introduce two algorithms to overcome these and demonstrate their great empirical performance on a large-scale crowd-sourced dataset of 2.2M ratings collected by the New Yorker Caption Contest as well as a dataset testing hundreds of possible cancer drugs.

## 1 Introduction

We propose a new multi-armed bandit problem where the objective is to return *all* arms that are $\epsilon$-good relative to the best-arm. Concretely, if the arms have means $\mu_1, \cdots, \mu_n$, with $\mu_1 = \max_{1 \le i \le n} \mu_i$, then the goal is to return the set $\{i : \mu_i \ge \mu_1 - \epsilon\}$ in the additive case, and $\{i : \mu_i \ge (1-\epsilon)\mu_1\}$ in the multiplicative case. The ALL-$\epsilon$ problem is a novel setting in the bandits literature, adjacent to two other methods for finding many good arms: TOP-$k$ where the goal is to return the arms with the $k$ highest means, and threshold bandits where the goal is to identify all arms above a fixed threshold. Building on a metaphor given by [1], if TOP-$k$ is a "contest" and thresholding bandits is an "exam", ALL-$\epsilon$ organically decides which arms are "above the bar" relative to the highest score. We argue that the ALL-$\epsilon$ problem formulation is more appropriate in many applications, and we show that it presents some unique challenges that make its solution distinct from TOP-$k$ and threshold bandits.

**A Natural and Robust Objective.** A motivating example is drug discovery, where pharmacologists want to identify a set of highly-potent drug candidates from potentially millions of compounds using various *in vitro* and *in silico* assays, and only the selected undergo more expansive testing [2]. Since performing the assays can be costly, one would like to use an adaptive, sequential experiment design that requires fewer experiments than a fixed experiment design. In sequential experiment design, it is important to fix the objective at the beginning as that choice affects the experimentation process. Both the objectives of finding the top-$k$ performing drugs, or all drugs above a threshold can result

in failure. In TOP-$k$, choosing $k$ too small may miss potent compounds, and choosing $k$ too large may yield many ineffective compounds and require an excessively large number of experiments. Setting a threshold suffers from the same issues - with the additional concern that if it is set too high, potentially no drug discoveries are made. In contrast, the ALL-$\epsilon$ objective of finding all arms whose potency is within 20% of the best avoids these concerns by giving a robust and natural guarantee: *no* significantly suboptimal arms will be returned and but *every* near-optimal arm will be discovered.

We emphasize that unlike TOP-$k$ or thresholding which require some prior knowledge about the distribution of arms to guarantee a good set of returned arms, choosing the arms relative to the best is a natural, distribution-free metric for finding good arms. As an example, we consider the New Yorker Cartoon Caption Contest (NYCCC). Each week, contestants submit thousands of supposedly funny captions for a cartoon (see Appendix A), which are rated from 1 (unfunny) to 3 (funny) through a crowdsourcing process. The New Yorker editors select final winners from a set with the highest average crowd-ratings (typically over 1 million ratings per contest).

The number of truly funny captions varies from week to week, and this makes setting a choice of $k$ or fixed threshold difficult. In Figure 1, we plot the distribution of ratings from 3 different contests. Horizontal lines depict a reasonable threshold of $0.8\mu_1$ in each and vertical lines show the number of arms that exceed this threshold. Both of these quantities vary over weeks and these differences can be stark. In contest 627, only $k = 27$ arms are within 20% of $\mu_1$, but $k = 748$ are in contest 651. Additionally, a fixed threshold of $\tau = 1.5$, admits captions within 30% of the best in contest 627, but only those within 15% of the best in contest 651. These examples show that it would be imprudent, and indeed, incorrect to choose a value

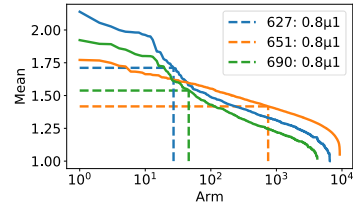

Figure 1: Mean ratings from contests 627, 651, 690

of $k$ or a threshold based on past contests– the far more principled decision is to optimize for the objective of finding the captions that are within a percentage of the best every week.

Though the ALL-$\epsilon$ objective is natural and easy to state, it has not been studied in the literature. As we will show, admitting arms relative to the best makes the ALL-$\epsilon$ problem inherently more challenging than either TOP-$k$ or thresholding. In particular, it is not easily possible to adapt TOP-$k$ or thresholding algorithms to achieve the instance dependent lower bound for ALL-$\epsilon$. In this work, we provide a careful investigation of the ALL-$\epsilon$ problem including theoretical and empirical guarantees.

## 1.1 Problem Statement and Notation

Fix $\epsilon > 0$ and a failure probability $\delta > 0$. Let $\nu := \{\rho_1, \cdots, \rho_n\}$ be an instance of $n$ distributions (or arms) with 1-sub-Gaussian distributions having *unknown* means $\mu_1 \geq \cdots \geq \mu_n$. We now formally define our notions of additive and multiplicative $\epsilon$-good arms.

**Definition 1** (additive $\epsilon$-good). *For a given $\epsilon > 0$, arm $i$ is additive $\epsilon$-good if $\mu_i \geq \mu_1 - \epsilon$.*

**Definition 2** (multiplicative $\epsilon$-good). *For a given $\epsilon > 0$, arm $i$ is multiplicative $\epsilon$-good if $\mu_i \geq (1 - \epsilon)\mu_1$.*

Additionally, we define the sets

$$G_\epsilon(\nu) := \{i : \mu_i \geq \mu_1 - \epsilon\} \text{ and } M_\epsilon(\nu) := \{i : \mu_i \geq (1 - \epsilon)\mu_1\} \tag{1}$$

to be the sets of additive and multiplicative $\epsilon$-good arms respectively. Where clear, we take $G_\epsilon = G_\epsilon(\nu)$ and $M_\epsilon = M_\epsilon(\nu)$. Consider an algorithm that at each time $s$ selects an arm $I_s \in [n]$ based on the history $\mathcal{F}_{s-1} = \sigma(I_1, X_1, \cdots, I_{s-1}, X_{s-1})$, and observes a reward $X_s \overset{\text{iid}}{\sim} \rho_{I_s}$. The objective of the algorithm is to return $G_\epsilon$ or $M_\epsilon$ using as few total samples as possible.

**Definition 3.** (ALL-$\epsilon$ problem). *An algorithm for the ALL-$\epsilon$ problem is $\delta$-PAC if (a) the algorithm has a finite stopping time $\tau$ with respect to $\mathcal{F}_t$, (b) at time $\tau$ it recommends a set $\widehat{G}$ such that with probability at least $1 - \delta$, $\widehat{G} = G_\epsilon$ in the additive case, or $\widehat{G} = M_\epsilon$ in the multiplicative case.*

**Notation:** For any arm $i \in [n]$, let $\widehat{\mu}_i(t)$ denote the empirical mean after $t$ pulls. For all $i \in [n]$, define the suboptimality gap $\Delta_i := \mu_1 - \mu_i$. Without loss of generality, we denote $k = |G_\epsilon|$ (resp. $k = |M_\epsilon|$). Throughout, we will keep track of the quantity $\alpha_\epsilon := \min_{i \in G_\epsilon} \mu_i - (\mu_1 - \epsilon)$ which is the distance from the smallest additive $\epsilon$-good arm, denoted $\mu_k$, to the threshold $\mu_1 - \epsilon$. Additionally, if

$G_\epsilon^c$ is non-empty, we consider $\beta_\epsilon = \min_{i \in G_\epsilon^c}(\mu_1 - \epsilon) - \mu_i$, the distance of the largest arm that is not additive $\epsilon$-good, denoted $\mu_{k+1}$, to the threshold. Equivalently, in the case of returning multiplicative $\epsilon$ arms, we define $\tilde{\alpha}_\epsilon := \min_{i \in M_\epsilon} \mu_i - (1 - \epsilon)\mu_1$, $\tilde{\beta}_\epsilon := \min_{i \in M_\epsilon^c}(1 - \epsilon)\mu_1 - \mu_i$, $\mu_k$, and $\mu_{k+1}$ to be the smallest differences of arms in $M_\epsilon$ and $M_\epsilon^c$ to $(1 - \epsilon)\mu_1$ respectively. For our sample complexity results, we also consider a relaxed version of the ALL-$\epsilon$ problem, where for a user-given slack $\gamma \geq 0$, we allow our algorithm to return $\widehat{G}$ that satisfies $G_\epsilon \subset \widehat{G} \subset G_{\epsilon+\gamma}$ in the <span style="color:red">additive</span> case, or $M_\epsilon \subset \widehat{G} \subset M_{\epsilon+\gamma}$ in the <span style="color:blue">multiplicative</span> case. As we will see, this prevents large or potentially unbounded sample complexities when arms' means are very close to or equal $\mu_1 - \epsilon$.

## 1.2 Contributions and Summary of Main Results

In this paper we propose the new problem of finding *all* $\epsilon$-good arms and give a precise characterization of its complexity. Our contribution is threefold:

- Information-theoretic lower bounds for the ALL-$\epsilon$ problem.

- A novel algorithm, $(\mathtt{ST})^2$, that is nearly optimal, is easy to implement, and has excellent empirical performance on real-world data.

- An instance optimal algorithm, FAREAST.

We now summarize our results in the <span style="color:red">additive</span> setting (the <span style="color:blue">multiplicative</span> setting is analogous).

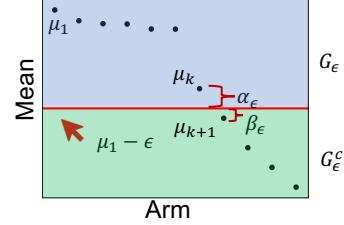

Figure 2: An example instance

**Lower Bound and Algorithms.** As a preview of our results, we highlight the impact of three key quantities that affect the sample complexity: the user provided $\epsilon$ and the instance dependent quantities $\alpha_\epsilon$ and $\beta_\epsilon$, (see Figure 2). In this case, Theorem 2.1 implies that any $\delta$-PAC algorithm requires an expected number of samples exceeding

$$\sum_{i=1}^{n} \max\left\{ \frac{1}{(\mu_1 - \epsilon - \mu_i)^2}, \frac{1}{(\mu_1 + \alpha_\epsilon - \mu_i)^2} \right\} \log\left(\frac{1}{\delta}\right). \tag{2}$$

We provide two algorithms, $(\mathtt{ST})^2$ and FAREAST for the ALL-$\epsilon$ problem. Our starting point, $(\mathtt{ST})^2$ is a novel combination of UCB [3] and LUCB [4] and is easier to implement and has good empirical performance. $(\mathtt{ST})^2$ is nearly optimal, however in some instances does not achieve the lower bound. To overcome this gap, we provide an instance optimal algorithm FAREAST which achieves the lower bound, however suffers from larger constants and is not always better in practical applications.

To highlight the difficulty of developing optimal algorithms for the ALL-$\epsilon$ problem, we quickly discuss a naive elimination approach that uniformly samples all arms and eliminates arms once they are known to be above or below $\mu_1 - \epsilon$ and not the best arm. Intuitively, such an algorithm would keep pulling arms until $\mu_1 - \epsilon$ is estimated to an accuracy of $O(\min(\alpha_\epsilon, \beta_\epsilon))$ to resolve the arms around the threshold (see Figure 2). An elimination algorithm pays a high cost of exploration - potentially over pulling arms close to $\mu_1$ compared to the lower bound until a time when $\mu_1 - \epsilon$ is estimated sufficiently well. Our algorithm FAREAST provides a novel approach to overcome the issues with this approach. However, as we will show in Section 4, in certain instances a dependence on $\sum_{i=1}^{n}(\mu_1 + \beta_\epsilon - \mu_i)^{-2}$ is present in *moderate confidence*, i.e., it is not multiplied by $\log(1/\delta)$, unlike the lower bound in equation (2) and becomes negligible compared to other terms as $\delta \to 0$.

**Empirical results.** We demonstrate the empirical success of $(\mathtt{ST})^2$ on a real world dataset of 9250 captions from the NYCCC. In Fig. 4a, we compare $(\mathtt{ST})^2$ to other methods that have been used to run this contest. We show that $(\mathtt{ST})^2$ is better able to detect which arms have means within $10\%$ of the best. The plot demonstrates the sub-optimality of using existing sampling schemes such as UCB or LUCB with an incorrect $k$ for the ALL-$\epsilon$ problem, providing an additional empirical validation for the study of this paper.

## 1.3 Connections to prior Bandit art

Our problem is related to several prior pure-exploration settings in the multi-armed bandit literature, including TOP-$k$ bandits, and threshold bandits.

**TOP-$k$.** In the TOP-$k$ problem, the goal is to identify the set $\{\mu_1, \cdots, \mu_k\}$ with probability greater than $1 - \delta$ [4–9]. The ALL-$\epsilon$ problem reduces to the setting of the TOP-$k$ problem with $k = |G_\epsilon|$ when $|G_\epsilon|$ is known. In particular, lower bounds for the TOP-$k$ problem apply to our setting. A lower bound (with precise logarithmic factors) given in [9] is $\sum_{i=1}^{k} (\mu_i - \mu_{k+1})^{-2} \log((n - k)/\delta) + \sum_{i=k+1}^{n} (\mu_i - \mu_k)^{-2} \log(k/\delta)$. In general, this is smaller than our lower bound in Theorem 2.1 since $\mu_k \geq \mu_1 - \epsilon \geq \mu_{k+1}$. A particular case of this problem is best-arm identification when $k = 1$.

Approximate versions of the TOP-$k$ problem have also been considered where the goal is to return a set of arms $\mathcal{S}$ with $|\mathcal{S}| = k$ and such that with probability greater than $1 - \delta$, each $i \in \mathcal{S}$ satisfies $\mu_i \geq \mu_k - \epsilon$ [4, 10]. In the case where $k = 1$, this is also known as the problem of identifying an (single) $\epsilon$-good arm [4, 7, 9–17] which has received a large amount of interest. If $|G_\epsilon| = k$, [6], demonstrate a lower bound of $O((k\epsilon^{-2} + \sum_{i=k+1}^{n} (\mu_1 - \mu_i)^{-2}) \log(1/\delta))$ samples in expectation to find such an arm and [10] provide an algorithm that matches this to doubly logarithmic factors, though methods such as [4, 9, 18, 19] achieve better empirical performance. A particular instance of interest is when it is known that one arm is at mean $\epsilon$, and the rest are at mean zero. In this setting, [11] show a lower bound on the sample complexity of $O(n/\epsilon^2 + 1/\epsilon^2 \log(1/\delta))$ highlighting that the dependence on $n$ only occurs in *moderate confidence*, i.e., for a fixed value of $\delta$. They also provide a matching upper bound that motivates our procedure in FAREAST. Finally [15] considers the *unverifiable* regime where there are potentially many $\epsilon$-good arms. In such cases, sample-efficient algorithms exist that return an $\epsilon$-good arm with high probability, but *verifying* it is $\epsilon$-good requires far more samples. Extending these ideas to the setting of ALL-$\epsilon$ is a goal of future work.

**Threshold Bandits.** In the threshold bandit problem, we are given a threshold $\tau$ and the goal is to identify the set of arms whose means are greater than the threshold [1, 20]. If the value of $\mu_1$ were known, then ALL-$\epsilon$ problem would reduce to a threshold bandit with $\tau = \mu_1 - \epsilon$. A naive sequential sampling scheme that stops sampling an arm when its upper or lower confidence bound clears the threshold has sample complexity $O(\sum_{i=1}^{n} (\mu_i - \tau)^{-2} \log(n/\delta))$. Up to factors of $\log(n)$, this can be shown to be a lower bound for threshold bandits as well, and as a result is bounded above by the result Theorem 2.1. Hence, ALL-$\epsilon$ is intrinsically more difficult than threshold bandits. A naive approach to the ALL-$\epsilon$ problem is to first identify the index and mean of the best arm using a best-arm identification algorithm and then utilize it to build an estimate of the threshold $\mu_1 - \epsilon$. In general, this two-step procedure is sub-optimal if there are many arms close to the best-arm in which case identifying the best-arm is both unnecessary and expends unnecessary samples. In the fixed confidence setting, threshold bandits is closely related to that of multiple hypothesis testing, and recent work [21] achieves tight upper and lower bounds for this problem including tighter logarithmic factors similar to those for TOP-$k$. If $\mu_1$ is known, then the additive ALL-$\epsilon$ problem reduces to the FWER (family-wise error rate) and FWPD (family-wise probability of detection) setting in [21]. Finally, in the fixed budget setting, [1] proposes an optimal anytime method APT whose sampling strategy we use as a comparison in Section 5.

## 2  Lower Bound

**Theorem 2.1.** *(additive and multiplicative lower bounds) Fix $\delta, \epsilon > 0$. Consider $n$ arms, such that the $i^{th}$ is distributed according to $\mathcal{N}(\mu_i, 1)$. Any $\delta$-PAC algorithm for the additive setting satisfies*

$$\mathbb{E}[\tau] \geq 2 \sum_{i=1}^{n} \max \left\{ \frac{1}{(\mu_1 - \epsilon - \mu_i)^2}, \frac{1}{(\mu_1 + \alpha_\epsilon - \mu_i)^2} \right\} \log \left( \frac{1}{2.4\delta} \right)$$

*and if $\mu_1 > 0$, any $\delta$-PAC algorithm for the multiplicative algorithm satisfies,*

$$\mathbb{E}[\tau] \geq 2 \sum_{i=1}^{n} \max \left\{ \frac{1}{((1-\epsilon)\mu_1 - \mu_i)^2}, \frac{1}{(\mu_1 + \frac{\tilde{\alpha}_\epsilon}{1-\epsilon} - \mu_i)^2} \right\} \log \left( \frac{1}{2.4\delta} \right).$$

The bounds are different but share a common interpretation. Consider the additive case. First, every arm must be sampled inversely proportional to its squared distance to $\mu_1 - \epsilon$. In a manner similar to thresholding [1], even if $\mu_1 - \epsilon$ was known, these number of samples are necessary to decide if an arm's mean is above or below that quantity. This leads to the first term in the $\max\{\cdot, \cdot\}$. The second term in the $\max\{\cdot, \cdot\}$ states that every arm must be sampled inversely proportional to its squared distance to $\mu_1 + \alpha_\epsilon$. Recall that $\alpha_\epsilon = \mu_k - (\mu_1 - \epsilon)$ is the margin by which arm $k$ is good.

Hence, to verify that $k \in G_\epsilon$, it is also necessary to confirm that all means are below $\mu_1 + \alpha_\epsilon$, as $\mu_1 + \alpha_\epsilon - \epsilon \geq \mu_k$ which would imply that $k$ is bad. This represents the necessity of estimating the threshold, and leads to the second term. For arms in $G_\epsilon^c$, comparing against $\mu_1 - \epsilon$ is always more difficult, but for arms in $G_\epsilon$, either constraint may be more challenging to ensure. We state the bound for Gaussian distributions, but the same technique can be used to prove equivalent results for other distributions. Lastly, we note that it is possible to prove bounds with tighter logarithmic terms. For an instance where $O(n^\phi)$ arms have mean $2\epsilon$ for $\phi \in (0, 1)$, and the remaining have mean 0, Theorem 1 of [22] suggests that $\Omega(n/\epsilon^2 \log(n/\delta))$ samples are necessary, exceeding the above bounds by a factor of $\log(n)$.

# 3 An Optimism Algorithm for ALL-$\epsilon$

We propose algorithm 1 called $(\text{ST})^2$, (**S**ample the **T**hreshold, **S**plit the **T**hreshold) to return a set containing all $\epsilon$-good arms and none worse than $(\epsilon + \gamma)$-good with probability $1 - \delta$. Intuitively, $(\text{ST})^2$ runs UCB and LUCB1 in parallel. At all times, $(\text{ST})^2$ pulls three arms. We pull the arm with the highest upper confidence bound, similarly to the UCB algorithm, [3], to refine an estimate of the threshold using the highest empirical mean (**S**ample the **T**hreshold). Using the empirical estimate of the threshold, we pull an arm above it and an arm below it whose confidence bounds cross it, similar to LUCB1, [4] (**S**plit the **T**hreshold). Using these bounds, $(\text{ST})^2$ forms upper and lower bounds on the true threshold, i.e. $\mu_1 - \epsilon$ (resp. $(1 - \epsilon)\mu_1$) and terminates when it can declare that all arms are either in $G_{\epsilon+\gamma}$ or $G_\epsilon^c$. To do so, $(\text{ST})^2$ maintains anytime confidence widths, $C_{\delta/n}(t)$ such that for an empirical mean $\widehat{\mu}_i(t)$ of $t$ samples, we have $\mathbb{P}(\bigcup_{t=1}^\infty |\widehat{\mu}_i(t) - \mu_i| > C_{\delta/n}(t)) \leq \delta/n$. For this work, we take $C_\delta(t) = \sqrt{\frac{c_\phi \log(\log_2(2t)/\delta)}{t}}$ for a constant $c_\phi$. It suffices to take $c_\phi = 4$, though tighter bounds are known and should be used in practice, e.g. [6, 23, 24].

---

**Algorithm 1** $(\text{ST})^2$: **S**ample the **T**hreshold, **S**plit the **T**hreshold

**Require:** $\epsilon, \delta > 0$, $\gamma \geq 0$, instance $\nu$
1: Pull each arm once, initialize $T_i \leftarrow 1$, update $\widehat{\mu}_i$ for each $i \in \{1, 2, \ldots, n\}$
2: Empirically good arms: $\widehat{G} = \{i : \widehat{\mu}_i \geq \max_j \widehat{\mu}_j - \epsilon\}$, $\widehat{G} = \{i : \widehat{\mu}_i \geq (1 - \epsilon) \max_j \widehat{\mu}_j\}$
3: $U_t = \max_j \widehat{\mu}_j(T_j) + C_{\delta/n}(T_j) - \epsilon - \gamma$ and $L_t = \max_j \widehat{\mu}_j(T_j) - C_{\delta/n}(T_j) - \epsilon$
4: $U_t = (1 - \epsilon - \gamma)\left(\max_j \widehat{\mu}_j(t) + C_{\delta/n}(T_j)\right)$ and $L_t = (1 - \epsilon)\left(\max_j \widehat{\mu}_j(t) - C_{\delta/n}(T_j)\right)$
5: Known arms: $K = \{i : \widehat{\mu}_i(T_i) + C_{\delta/n}(T_i) < L_t \text{ or } \widehat{\mu}_i(T_i) - C_{\delta/n}(T_i) > U_t\}$
6: **while** $K \neq [n]$ **do**
7:     Pull arm $i_1(t) = \arg\min_{i \in \widehat{G} \setminus K} \widehat{\mu}_i(T_i) - C_{\delta/n}(T_i)$, update $T_{i_1}, \widehat{\mu}_{i_1}$
8:     Pull arm $i_2(t) = \arg\max_{i \in \widehat{G}^c \setminus K} \widehat{\mu}_i(T_i) + C_{\delta/n}(T_i)$, update $T_{i_2}, \widehat{\mu}_{i_2}$
9:     Pull arm $i^*(t) = \arg\max_i \widehat{\mu}_i(T_i) + C_{\delta/n}(T_i)$, update $T_{i^*}, \widehat{\mu}_{i^*}$
10:     Update bounds $L_t, U_t$, sets $\widehat{G}, K$
    **return** The set of good arms $\{i : \widehat{\mu}_i(T_i) - C_{\delta/n}(T_i) > U_t\}$

---

## 3.1 Theoretical guarantees

Next we present a pair of theorems on the sample complexity of $(\text{ST})^2$. For clarity, we omit doubly logarithmic terms and defer such statements to Appendix B. Below we denote $a \wedge b := \min\{a, b\}$.

**Theorem 3.1** (Additive Case). *Fix $\epsilon > 0$, $0 < \delta \leq 1/2$, $\gamma \leq 16$ and an instance $\nu$ such that $\max(\Delta_i, |\epsilon - \Delta_i|) \leq 8$ for all $i$. With probability at least $1 - \delta$, there is a constant $c_1$ such that $(\text{ST})^2$ returns a set $\widehat{G}$ such that $G_\epsilon \subset \widehat{G} \subset G_{(\epsilon+\gamma)}$ in at most the following number of samples.*

$$c_1 \log\left(\frac{n}{\delta}\right) \sum_{i=1}^n \max\left\{\frac{1}{(\mu_1 - \epsilon - \mu_i)^2}, \frac{1}{(\mu_1 + \alpha_\epsilon - \mu_i)^2}, \frac{1}{(\mu_1 + \beta_\epsilon - \mu_i)^2}\right\} \wedge \frac{1}{\gamma^2} \quad (3)$$

Given a positive slack $\gamma$, we are allowed to return an arm that is $(\epsilon + \gamma)$-good. Thus a confidence width less than $\Omega(\gamma)$ on any arm is not needed, resulting in the $1/\gamma^2$ term in Theorem 3.1. In particular this prevents unbounded sample complexities if there is an arm at the threshold $\mu_1 - \epsilon$. For $\gamma = 0$, the first two terms inside the max are also present in the lower bound (Theorem 2.1). When $\alpha_\epsilon$ is within a constant factor of $\beta_\epsilon$, the second and third term in the max have the same order, and the upper bound matches the lower bound up to a $\log(n)$ factor.

If $\beta_\epsilon \ll \alpha_\epsilon$, (3) has a different scaling than the lower bound. In such restrictive settings the upper bound above can be significantly larger than the lower bound. In the next section, we provide an algorithm that overcomes these issues and is optimal over all parameter regimes. The multiplicative case has different terms but follows the same intuition.

**Theorem 3.2** (Multiplicative Case). *Fix $\epsilon \in (0, 1/2]$, $\gamma \in [0, \min(16/\mu_1, 1/2)]$ and $0 < \delta \le 1/2$ and an instance $\nu$ such that $\mu_1 \ge 0$ and $\max(\Delta_i, |\epsilon \mu_1 - \Delta_i|) \le 2$ for all $i$. With probability at least $1 - \delta$, for a constant $c_1$ $(ST)^2$ returns a set $G$ such that $M_\epsilon \subset G \subset M_{(\epsilon+\gamma)}$ with sample complexity:*

$$c_1 \log\left(\frac{n}{\delta}\right) \sum_{i=1}^n \max\left\{ \frac{1}{((1-\epsilon)\mu_1 - \mu_i)^2}, \frac{1}{(\mu_1 + \frac{\tilde{\alpha}_\epsilon}{1-\epsilon} - \mu_i)^2}, \frac{1}{(\mu_1 + \frac{\tilde{\beta}_\epsilon}{1-\epsilon} - \mu_i)^2} \right\} \wedge \frac{1}{\gamma^2 \mu_1^2}.$$

## 4 Surprising Complexity of Finding All $\epsilon$-Good arms

When $\alpha_\epsilon$ and $\beta_\epsilon$ are not of the same order, $(ST)^2$ is not optimal. In this section we present an algorithm that is optimal for all parameter regimes. We focus on the additive case here, and defer the multiplicative case to Appendix E. We first state an improved sample complexity lower bound for a family of problem instances that makes explicit the *moderate confidence* terms.

**Theorem 4.1.** *Fix $\delta \le 1/16$, $n \ge 2/\delta$, and $\epsilon > 0$. Let $\nu$ be an instance of $n$ arms such that the $i^{th}$ is distributed as $\mathcal{N}(\mu_i, 1)$, $|G_{2\beta_\epsilon}| = 1$, and $\beta_\epsilon < \epsilon/2$. Select a permutation $\pi : [n] \to [n]$ uniformly from the set of $n!$ permutations, and consider the permuted instance $\pi(\nu)$. Any algorithm that returns $G_\epsilon(\pi(\nu))$ on $\pi(\nu)$ correctly with probability at least $1 - \delta$ requires at least the following number of samples in expectation over randomness in $\nu$ and $\pi$ for a universal constant $c_2$.*

$$\left[ c_2 \sum_{i=1}^n \max\left\{ \frac{1}{(\mu_1 - \epsilon - \mu_i)^2}, \frac{1}{(\mu_1 + \alpha_\epsilon - \mu_i)^2} \right\} \log\left(\frac{1}{2.4\delta}\right) \right] + c_2 \sum_{i=1}^n \frac{1}{(\mu_1 + \beta_\epsilon - \mu_i)^2} \quad (4)$$

*Proof.* (Sketch) To give a tight lower bound in the setting where $|G_{2\beta_\epsilon}| = 1$ and $\beta_\epsilon < \epsilon/2$, we break our argument into pieces performing a series of reductions that link the ALL-$\epsilon$ problem to a hypothesis test, and then the hypothesis test to the problem of identifying the best-arm. We apply the Simulator technique from [9] to compute precise moderate confidence bounds. Other works that prove strong lower bounds in moderate confidence include [25]. We extend the Simulator technique via a novel reduction to composite hypothesis testing in order to connect to ALL-$\epsilon$. In all cases, we consider sample complexity in expectation with respect to the randomness in the outcomes and a randomly chosen permutation of the means.

**Step 1. Finding an isolated best arm:** Consider the problem of finding the best arm where $\mu_1 = \beta > 0$ and $\mu_2, \cdots, \mu_n \le -\beta$. This relates to the problem of finding a $\beta$-good arm when $\mu_1$ is known, studied by [11]. We use the Simulator technique, [9], to show that any algorithm requires $\Omega\left(\sum_{i=2}^n \Delta_i^{-2}\right)$ samples in expectation. This can be significantly larger than the asymptotically optimal rate of $O(\beta^{-2} \log(1/\delta))$ (which was proven by [11]) for non-asymptotic $\delta$, e.g. $\delta = 0.05$.

**Step 2. Deciding if Any mean is positive:** We then consider a composite hypothesis test on $n$ distributions where the null hypothesis, $H_0$, is that the mean of each distribution is less that $-\beta$ and the alternate hypothesis, $H_1$, is that there exists a *single* distribution $i^*$ with mean $\beta$ and the remainder have mean less than $-\beta$. Importantly, an algorithm does not need to declare which arm is $i^*$, otherwise the bound from step 1 applies immediately. Instead, to link this to step 1, we develop a novel extension of the simulator technique and use this to show that if an algorithm can solve this composite hypothesis test in fewer than $o\left(\sum_{i=2}^n \Delta_i^{-2}\right)$ samples, then one may design a method to solve the problem in step 1 in $o\left(\sum_{i=2}^n \Delta_i^{-2}\right)$ samples which is a contradiction. Hence any algorithm for this hypothesis test requires $\Omega\left(\sum_{i=2}^n \Delta_i^{-2}\right)$ samples in expectation.

**Step 3: Reducing ALL-$\epsilon$ to Step 2:** Finally, we show that a generic algorithm for ALL-$\epsilon$ can be used to solve the hypothesis test in step 2. Hence the lower bound from step 2 applies to finding all $\epsilon$-good arms as well. In the case of the instances considered in the theorem statement, $O\left(\sum_{i=2}^n \Delta_i^{-2}\right) = O\left(\sum_{i=2}^n (\mu_1 + \beta_\epsilon - \mu_i)^{-2}\right)$. Combining this bound, which is independent of $\delta$ with the result from Theorem 2.1 gives the result. $\qquad\square$

Theorem 4.1 states that an additional $\Omega(\sum_{i=1}^{n}(\mu_1 + \beta_\epsilon - \mu_i)^{-2})$ samples are necessary for instances where no arm is within $2\beta_\epsilon$ of $\mu_1$ compared to the lower bound Theorem 2.1. Somewhat surprisingly, these samples are *necessary in moderate confidence*, independent of $\delta$ and negligible as $\delta \to 0$. For non-asymptotic values of $\delta$, such as the common choice of $\delta = .05$ in scientific applications, this term is present and can even dominate the sample complexity when $\beta_\epsilon \ll \alpha_\epsilon$. As an extreme example, if $\mu_1 = \beta > 0$, $\mu_2 \cdots, \mu_{n-1} = -\beta, \mu_n = -\epsilon$, the first term in 4 scales like $((n-1)/\epsilon^2 + 1/\beta^2) \log(1/\delta)$ but the second term scales like $n/\beta^2$, which is $O(n)$ larger than the first term for small $\beta$ and fixed $\delta$. Furthermore, we point out that Theorem 4.1 highlights that $(\text{ST})^2$ is optimal on these instances up to a $\log$ factor! The algorithm we present next, FAREAST, improves $(\text{ST})^2$'s dependence on $\delta$ and matches the lower bound in Theorem 4.1 for certain instances. Though moderate confidence terms can dominate the sample complexity in practice, few works have focused on understanding their effect.

## 4.1 FAREAST

We focus on the additive case with $\gamma = 0$ in Algorithm 4.1, FAREAST, and defer the more general case (multiplicative and $\gamma > 0$) to Algorithm E.1 in the supplementary. FAREAST matches the instance dependent lower bound in Theorem 2.1 as $\delta \to 0$. At a high level, FAREAST (**F**ast **A**rm **R**emoval **E**limination **A**lgorithm for a **S**ampled **T**hreshold) proceeds in rounds $r$ and maintains sets $\widehat{G}_r$ and $\widehat{B}_r$ of arms thus far declared to be good or bad. It sorts unknown arms into either set through use of a good filter to detect arms in $G_\epsilon$ and a bad filter to detect arms in $G_\epsilon^c$.

**Good Filter:** The good filter is a simple elimination scheme. It maintains an upper bound $U_t$ and lower bound $L_t$ on $\mu_1 - \epsilon$. If an arm's upper bound drops below $L_t$ (line 20), the good filter eliminates that arm, otherwise, if an arm's lower bound rises above $U_t$ (19), the good filter adds the arm to $\widehat{G}_r$, but only eliminates this arm if its upper bound falls below the highest lower bound. This ensures that $\mu_1$ is never eliminated and $U_t$ and $L_t$ are always valid bounds [1]. As the sampling is split across rounds, the good filter always samples the least sampled arm, breaking ties arbitrarily. The number of samples given to the good filter in each round is such that both filters receive identically many samples. This prevents the good filter from over-sampling bad arms and vice versa. In our proof we show that in an unknown round, $\widehat{G}_r = G_\epsilon$, ie all good arms have been found, having used fewer than $O\left(\sum_{i=1}^{n} \max\left\{(\mu_1 - \epsilon - \mu_i)^{-2}, (\mu_1 + \alpha_\epsilon - \mu_i)^{-2}\right\} \log(n/\delta)\right)$ samples, matching the lower bound.

FAREAST cannot yet terminate, however, as it must also verify that any remaining arms are in $G_\epsilon^c$.

**Bad Filter:** The bad filter removes arms that are not $\epsilon$-good. To show an arm $i$ is in $G_\epsilon^c$, it suffices to find any $j$ such that $\mu_j - \mu_i > \epsilon$. To motivate the idea of lines 9-12, consider the following procedure in the special case where $\beta_i = \mu_1 - \epsilon - \mu_i$ is known. In each round we first run Median-Elimination, [12], with failure probability $1/16$, to find an arm $\widehat{i}$ that is $\beta_i/2$-good in $O(n/\beta_i^2)$ samples [2]. We then pull both $i$ and $\widehat{i}$ roughly $O(1/\beta_i^2 \log(1/\delta))$ times and can check whether $\mu_{\widehat{i}} - \mu_i > \epsilon$ with probability greater than $1 - \delta$. This procedure relies on Median-Elimination succeeding, which happens with probability $15/16$. In the case that it fails and we declare $\mu_{\widehat{i}} - \mu_i < \epsilon$, we merely repeat this process until it succeeds-- on average $O(1)$ times. This gives an expected sample complexity of $O(n/\beta_i^2 + 1/\beta_i^2 \log(1/\delta))$ for any $i \in G_\epsilon^c$. Of course, $\beta_i$ is unknown to the algorithm. Instead, in each round $r$, the bad filter guesses that $\beta_i \geq 2^{-r}$ for all unknown arms $i \notin \widehat{G}_r \cup \widehat{B}_r$ and performs the above procedure. The following theorem demonstrates that this algorithm matches our lower bounds asymptotically as $\delta \to 0$.

**Theorem 4.2.** *Fix $0 < \epsilon$, $0 < \delta < 1/8$, and an instance $\nu$ of $n$ arms such that $\max(\Delta_i, |\epsilon - \Delta_i|) \leq 8$ for all $i$. There exists an event $E$ such that $\mathbb{P}(E) \geq 1 - \delta$ and on $E$, FAREAST terminates and returns $G_\epsilon$. Letting $T$ denote the number of samples taken, for a constant $c_3$*

$$\mathbb{E}[\mathbb{1}_E T] \leq \left[c_3 \sum_{i=1}^{n} \max\left\{\frac{1}{(\mu_1 - \epsilon - \mu_i)^2}, \frac{1}{(\mu_1 + \alpha_\epsilon - \mu_i)^2}\right\} \log\left(\frac{n}{\delta}\right)\right] + c_3 \sum_{i \in G_\epsilon^c} \frac{c''n}{(\mu_1 - \epsilon - \mu_i)^2}.$$

*Additionally for $\gamma \leq 16$ FAREAST terminates on $E$ and returns a set $\widehat{G}$ such that $G_\epsilon \subset \widehat{G} \subset G_{\epsilon+\gamma}$ in a number of samples no more than a constant times (3), the complexity of $(\text{ST})^2$.*

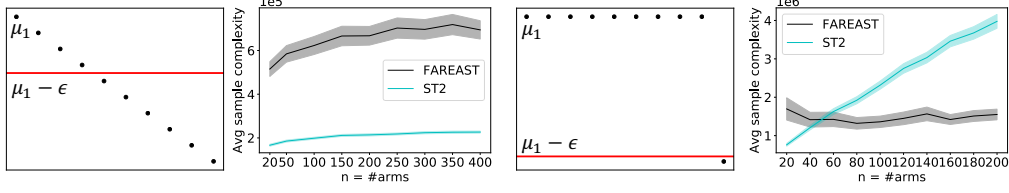

(a) A typical setting of diverse means (black dots).  (b) A more challenging setting

Figure 3: Comparison of $(\texttt{ST})^2$ and $\texttt{FAREAST}$ averaged over 250 trials plotted with 3 standard errors.

---

**Algorithm 4.1: additive $\texttt{FAREAST}$ with $\gamma = 0$**

1  **Algorithm 4.1: additive $\texttt{FAREAST}$ with $\gamma = 0$**
2  **Input**: $\epsilon$, $\delta$, instance $\nu$
3  Let $\widehat{G}_0 = \emptyset$ be the set of arms declared as good and $\widehat{B}_0 = \emptyset$ the set of arms declared as bad.
4  Let $\mathcal{A} = [n]$ be the active set, $N_i = 0$ track the total number of samples of arm $i$ by the Good Filter.
5  Let $t = 0$ denote the total number of times that line 16 is true in the Good Filter.
6  **for** $r = 1, 2, \cdots$
7   Let $\delta_r = \delta/2r^2$, $\tau_r = \left\lceil 2^{2r+3} \log\left(\frac{8n}{\delta_r}\right)\right\rceil$, Initialize $\widehat{G}_r = \widehat{G}_{r-1}$ and $\widehat{B}_r = \widehat{B}_{r-1}$
8   // **Bad Filter: find bad arms in** $G_\epsilon^c$
9   Let $i_r = \texttt{MedianElimination}(\nu, 2^{-r}, 1/16)$, sample $i_r$ $\tau_r$ times and compute $\widehat{\mu}_{i_r}$
10   **for** $i \notin \widehat{G}_{r-1} \cup \widehat{B}_{r-1}$:
11    Sample $\mu_i$ $\tau_r$ times and compute $\widehat{\mu}_i$
12    **If** $\widehat{\mu}_{i_r} - \widehat{\mu}_i \geq \epsilon + 2^{-r+1}$: Add $i$ to $\widehat{B}_r$                // Bad arm detected
13   // **Good Filter: find good arms in** $G_\epsilon$
14   **for** $s = 1, \cdots, H_{\mathbf{ME}}(n, 2^{-r}, 1/16) + (|(\widehat{G}_{r-1} \cup \widehat{B}_{r-1})^c| + 1)\tau_r$:
15    Pull arm $I_s \in \arg\min_{j \in \mathcal{A}}\{N_j\}$ and set $N_{I_s} \leftarrow N_{I_s} + 1$.
16    **if** $\min_{j \in \mathcal{A}}\{N_j\} = \max_{j \in \mathcal{A}}\{N_j\}$:
17     Update $t = t + 1$. Let $U_t = \max_{j \in \mathcal{A}}\widehat{\mu}_i(t) + C_{\delta/2n}(t) - \epsilon$ and $L_t = \max_{j \in \mathcal{A}}\widehat{\mu}_i(t) - C_{\delta/2n}(t) - \epsilon$
18     **for** $i \in \mathcal{A}$:
19      **if** $\widehat{\mu}_i(t) - C_{\delta/2n}(t) \geq U_t$: Add $i$ to $\widehat{G}_r$                // Good arm detected
20      **if** $\widehat{\mu}_i(t) + C_{\delta/2n}(t) \leq L_t$: Remove $i$ from $\mathcal{A}$ and add $i$ to $\widehat{B}_r$    // Bad arms removed
21      **if** $i \in \widehat{G}_r$ **and** $\widehat{\mu}_i(t) + C_{\delta/2n}(t) \leq \max_{j \in \mathcal{A}}\widehat{\mu}(t) - C_{\delta/2n}(t)$:    // Good arms removed
22       Remove $i$ from $\mathcal{A}$
23      **if** $\mathcal{A} \subset \widehat{G}_r$ or $\widehat{G}_r \cup \widehat{B}_r = [n]$: **Return the set** $\widehat{G}_r$

---

## 5  Empirical Performance

We begin by comparing $(\texttt{ST})^2$ and $\texttt{FAREAST}$ on simulated data. $\texttt{FAREAST}$ is asymptotically optimal, but suffers worse constant factors compared to $(\texttt{ST})^2$ [3]. $(\texttt{ST})^2$ is optimal *except* when $\beta_\epsilon \ll \alpha_\epsilon$. We compare $(\texttt{ST})^2$ and $\texttt{FAREAST}$ on two instances in the additive case, shown in Figure 3. All arms are Gaussian with $\sigma = 1$. In the first example on the left, $\delta = 0.1$, $\alpha_\epsilon = \beta_\epsilon = 0.05$. Both $(\texttt{ST})^2$ and $\texttt{FAREAST}$ are optimal in this setting; we show the scaling of their sample complexity as the number of arms increases while keeping the threshold, $\alpha_\epsilon$, and $\beta_\epsilon$ constant. In the second example, $\alpha_\epsilon = \epsilon = 0.99$, and $\beta = 0.01$. When $1/\beta_\epsilon^2 \gg n/\epsilon^2$, Theorem 2.1 suggests that $O(1/\beta_\epsilon^2 \log(1/\delta))$ samples are necessary, independent of $n$. Indeed, in Figure 3, for $\delta = 0.01$, the average complexity of $\texttt{FAREAST}$ is constant, but $(\texttt{ST})^2$ scales linearly with $n$ as Theorem 3.1 suggests. Finally, a naive uniform sampling strategy performed very poorly - additional experiments including the uniform sampling method and with $\gamma > 0$ are in the Appendix A.

### 5.1  Finding all $\epsilon$-good arms in real world data – *fast*

As discussed in the introduction, in many applications such as the New Yorker Cartoon Caption Contest (NYCCC), the ALL-$\epsilon$ objective returns a set of good arms which can then be screened further to choose a favorite. We considered Contest 651, which had 9250 captions whose means we estimated from a total of 2.2 million ratings. We set $\epsilon = 0.1$ and focus on the multiplicative setting, i.e., the objective of recovering all captions within 10% of the funniest one. In this experiment, we contrast $(\texttt{ST})^2$ with several other methods including two *oracle* methods (marked with ▲): LUCB1 [4] with

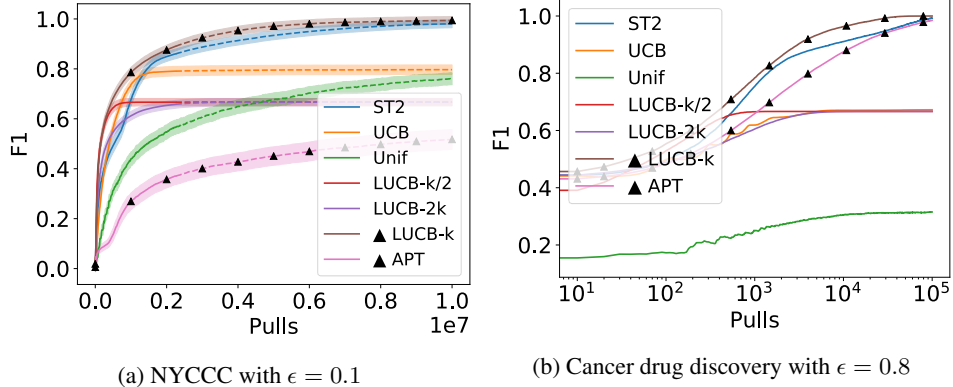

(a) NYCCC with $\epsilon = 0.1$       (b) Cancer drug discovery with $\epsilon = 0.8$

Figure 4: F1 scores averaged over 600 trials with $95\%$ confidence widths for each dataset.

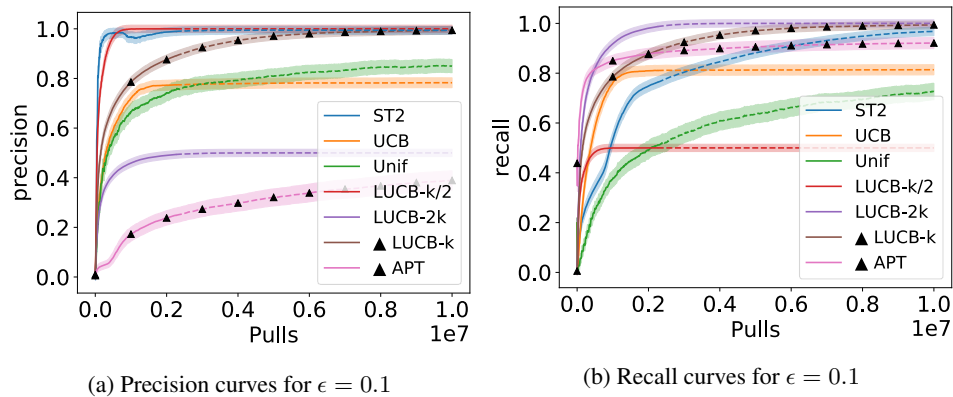

(a) Precision curves for $\epsilon = 0.1$       (b) Recall curves for $\epsilon = 0.1$

Figure 5: Precision and recall averaged over 600 trials with $95\%$ confidence widths on NYCCC data.

$k$ set to the number of $\epsilon$-good arms (here it was 46), and a threshold-bandit, APT [1] given the value of $0.9\mu_1$. We focus on a common practical requirement, each algorithm's ability to balance precision and recall as it samples. With every new sample, each method recommends an empirical set of $\epsilon$-good arms based on the empirical means, and we consider the F1 score of this set[4]. We focus on the F1 score as it is practically relevant and provides a continuous measure of performance of each method. $F1 = 1$ indicates that an algorithm has found all $\epsilon$-good arms. As can be seen in Figure 4a, $(\text{ST})^2$ outperforms all baselines including the oracle APT, and almost matches the performance of the TOP-$k$ oracle! We transition from a solid line to a dashed one at 2.2M pulls to mark the number of samples drawn in the real contest from which we gather the data. To illustrate the importance of knowing the correct value of $k$, we also plot LUCB1 given $k = 46/2 = 23$ and $k = 46 \times 2 = 92$, settings where the experimenter under or over estimates the number of $\epsilon$ good arms by as little as a factor of 2. Both cases result in a poor performance. We have also included UCB, currently being used for the contest [26]; the plot shows that UCB is not able to estimate the $\epsilon$-good set. In Figure 5, we show precision and recall curves for each method on the NYCCC data. $(\text{ST})^2$ achieves near-perfect precision quickly, matched only by UCB. APT's poor performance is a consequence of having low-precision, shown in Figure 5a. $(\text{ST})^2$ achieves high recall more slowly, but is still competitive with other methods. In practical experiments, high precision early on may be more important than high recall, as it guarantees that practitioners can trust the declarations that the algorithm has made, even if some arms are yet to be found. In the Supplementary we show plots for more values of $\epsilon$. Additionally, motivated by drug discovery, we performed an experiment on a dataset [27] of 189 inhibitors whose activities were tested against ACVRL1, a kinase associated with cancer [28]. In this experiment, we use the multiplicative case of ALL-$\epsilon$ with $\epsilon = 0.8$ and $\delta = 0.001$, to promote high precision. In this experiment as well, $(\text{ST})^2$ performs best (Figure 4b), with only the oracle methods are competitive with it. We plot on a log-scale to emphasize the early regime.

# 6 Broader Impacts and Funding Transparency Statement

## 6.1 Broader Impacts

The application of machine learning (ML) in domains such as advertising, biology, or medicine brings the possibility of utilizing large computational power and large datasets to solve new problems. It is tempting to use powerful, if not fully understood, ML tools to maximize scientific discovery. However, at times the gap between a tool's theoretical guarantees and its practical performance can lead to sub-optimal behavior. This is especially true in *adaptive data collection* where misspecifying the model or desired output (e.g., "return the top $k$ performing compounds" vs. "return all compounds with a potency about a given threshold") may bias data collection and hinder post-hoc consideration of different objectives. In this paper we highlight several such instances in real-life data collection using multi-armed bandits where such a phenomenon occurs. We believe that the objective studied in this work, that of returning all arms whose mean is quantifiably near-best, more naturally aligns with practical objectives as diverse as finding funny captions to performing medical tests. We point out that methods from adaptive data collection and multi-armed bandits can also be used on content-recommendation platforms such as social media or news aggregator sites. In these scenarios, time and again, we have seen that recommendation systems can be greedy, attempting purely to maximize clickthrough with a long term effect of a less informed public. Adjacent to one of the main themes of this paper, we recommend that practitioners not just focus on the objective of recommendation for immediate profit maximization but rather keep track of a more holistic set of metrics. We are excited to see our work used in practical applications and believe it can have a major impact on driving the process of scientific discovery.

## 6.2 Funding Transparency Statement

The work presented in this paper was supported by ARO grant W911NF-15-1-0479. Additionally, this work was partially supported by the MADLab AF Center of Excellence FA9550-18-1-0166.

## Footnotes

[1] This scheme works as an independent algorithm, we analyze it in Appendix E.5.

[2] Median-Elimination is used for ease of analysis. One can use LUCB [4] or another method instead.

[3]Implementations of all algorithms and baselines used in this paper are available on GitHub.

[4]F1 is the harmonic mean of precision (fraction of captions returned that are actually good) and recall (fraction of all good captions that are actually returned).

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
