[Supplementary Material · camera_ready_supp_v2.pdf]

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

# Contents

(a) Plot in Figure 3a with uniform sampling in-cluded.

(b) Plot in Figure 3b with uniform sampling in-cluded.

Figure 6: Simulation results with uniform sampling included.

Figure 7: $(\text{ST})^2$ and FAREAST with different values of $\gamma$

## A   Additional Experimental Results

**Practical change made to FAREAST for simulations:** We make one change to FAREAST that we recommend for practitioners wishing to use FAREAST that improve its empirical performance. In particular, Median-Elimination may instead be replaced by another method, such as LUCB1, [4], to find $\epsilon$-good arms. LUCB1, for instance, has better constant factors and enjoys improved empirical performance versus Median-Elimination. The use of Median-Elimination in this algorithm serves to ease both notation and analysis since it's sample complexity is deterministic. To modify the algorithm, simply track the number of samples given to the bad filter in total, which can be a random variable, and give the good filter the same number in that round. The proof then follows identically, with only the moderate confidence term changing in the result.

**Additional Simulations Results** As mentioned in the Experiments, Section 5, we omitted curves comparing against uniform sampling as they make the plots hard to read with uniform performing much more poorly. For completeness, we include them in Figure 6. Clearly, uniform sampling performs much more poorly than either active method, as expected.

Additionally, we include experiments with $\gamma > 0$ here. For small $\gamma$, the only valid solution is $G_\epsilon$ (resp. $M_\epsilon$) itself. However, for larger $\gamma$, there are many valid solutions. Indeed, any $G$ such that $G_\epsilon \subset G \subset G_{\epsilon+\gamma}$ is valid. To analyze the effect of $\gamma$ on both $(\text{ST})^2$ and FAREAST, we consider the same type of instances studied in Figure 3b. Here, $n-1$ arms have means equal to $\mu_1$, and a single arm is in $G_\epsilon^c$. Again, we take $\epsilon = 0.99$ and $\beta_\epsilon = 0.01$, and additionally, set $n = 150$ arms. Recall that in this setting, FAREAST outperforms $(\text{ST})^2$, as shown in Figure 3b. As we increase $\gamma$, the problem becomes easier. We increase $\gamma$ on an exponential scale, beginning with $\gamma \approx \epsilon/100$ and ending with $\gamma \approx \epsilon/2$. Indeed, for smaller values of $\gamma$, FAREAST is superior as it finds the exact solution fastest. For larger $\gamma$, $(\text{ST})^2$ is able to terminated more quickly. In Figure 7 we plot these results.

**Metrics we consider for real data experiments:** For all methods, we track their precision, recall and F1 score with respect to the true set of $\epsilon$-good arms. To compute these metrics, at each time, the algorithm outputs a set that it guesses are the $\epsilon$-good arms based on the data it has gathered thus far. For UCB, Uniform, and $(\texttt{ST})^2$, this is based directly on empirical means, i.e., $\widehat{G} = \{i : \widehat{\mu}_i \geq \max_j \widehat{\mu}_j - \epsilon\}$ or $\widehat{G} = \{i : \widehat{\mu}_i \geq \max_j (1-\epsilon)\widehat{\mu}_j\}$ in the multiplicative case. Oracle methods may use their additional information to return the set. In particular, APT returns all arms whose empirical means exceed $(1-\epsilon)\mu_1$ (using knowledge of $\mu_1$) and LUCB1 returns the $k$ largest empirical means (using knowledge that $|M_\epsilon| = k$. Let $TP$ (true positives) denote the number of arms that an algorithm declares as $\epsilon$-good that truly are. Let $FN$ (false negatives denote) the number of arms that an algorithm declares as *not* $\epsilon$-good when in fact they are. Recall, $r \in [0,1]$, is computed as $r = \frac{TP}{TP+FN}$. Intuitively, recall is the total number of $\epsilon$-good arms that the algorithm detects. Precision, $p \in [0,1]$, by contrast is the the fraction of the arms that an algorithm predicts as $\epsilon$-good that truly are. It is computed as $p = \max(TP/|\widehat{G}|, 1)$ where the $\max()$ is necessary to avoid the trivial case that $\widehat{G} = \emptyset$. Finally, the F1 is the harmonic mean of precision and recall: $F1 = \frac{2pr}{p+r}$. It balances how precise an algorithm is with how many discoveries it makes. In many cases, F1 may a more relevant metric than the others, as it avoids trivial edge cases. For instance, an algorithm that always declare every arm as $\epsilon$-good independent of the data, achieves perfect recall because it has $0$ false negatives. Similarly, an algorithm that never declares any arms as $\epsilon$-good, again independent of data, achieves perfect precision. Both methods, despite seemingly good performance with respect to their individual metrics, are undesirable in practice. In particular, both would achieve low F1 scores.

**The New Yorker Caption Contest:** In this section we provide additional experimental results adjoining those in Section 5. The data can be downloaded at https://github.com/nextml/caption-contest-data. We chose contest 651 for our experiments, but hundreds of others are available. Captions are rated on a scale of 1 to 3 ("unfunny", "somewhat funny", or "funny"). It is desirable to find all captions that are nearly as good as the best. However, setting a fixed number of captions or fraction of captions to accept is undesirable as the number of truly funny captions varies from week to week and represents a small fraction of the submissions. For instance, in the contest that ran the week of $3/14/16$, only 8 captions were rated within $20\%$ of the funniest caption. In the following week, by contrast, 187 captions were. Similarly, a choosing a fixed threshold of what it means for a caption to be funny is unrealistic. In the same two contests, first week saw $3\%$ of captions be rated at least 1.5 out of 3 whereas the second saw $< 0.1\%$. For this reason, finding all $\epsilon$-good arms is more natural. We consider finding all multiplicative $\epsilon$-good arms with $\epsilon = 0.1, 0.15, 0.2$. To keep the comparison fair, all methods use the same confidence widths from [24]. In Figure 9b we plot the average rating of each caption in sorted order with horizontal lines corresponding to $(1-0.2)\mu_1$, $(1-0.15)\mu_1$, and $(1-0.1)\mu_1$. The arms with means above this line are $0.2, 0.15$, and $0.1$ $\epsilon$-good. The oracle methods tend to achieve high recall, but low precision, and this is especially true for the threshold oracle, APT. In Figures 10, 11, 12 we plot F1, Precision, and Recall curves for all methods tested on $\epsilon = 0.2, 0.15, 0.1$ respectively. As before, all curves are averaged over 600 independent repetitions and plotted with $95\%$ confidence intervals. It is evident from these curves, that $(\texttt{ST})^2$ performs especially well with regard to precision, though it achieves lower recall than some other baselines.

**Protein Kinase Inhibitors for Cancer Drug Discovery**

Additionally, we consider a second, medically focused experiment. In 2013, researchers at Glaxo-SmithKline published a dataset of protein kinase inhibitors different kinases (PKIS1), primarily from humans [29]. Kinases are a family of enzymes present in many cells and researchers are interested in developing targeted kinase inhibitors to as a new way to treat cancer [2]. The dataset contains numerous measures of how strongly each inhibitor reacts with each kinase. A second, larger dataset (PKIS2) was expanded on by [27][5]. For the purpose of our experiment, we selected a single Kinase in the dataset, ACVRL1, which researchers have linked to numerous types of cancer, most prominently bladder and prostate cancers [28]. PKIS2 contains 641 different compounds that were tested as being potential kinase inhibitors, though not every compound was tested against every kinase. In particular, 189 were tested against ACVRL1. For each compound, there is an associated average "percent inhibition" that is reported. All numbers are between 0 and 1 and averaged across multiple trials in a single assay. We subtract each number from 1 to compute the percent control, representing how effective any method is relative to a control, an important metric for estimating how effective

"And you haven't been out of the country recently?"

| UNFUNNY | SOMEWHAT FUNNY | FUNNY |

| DONE |

Figure 8: The user interface for the caption contest with the caption for contest 651. "Unfunny" = 1, "Somewhat funny" = 2, "Funny" = 3

(a) Sorted means different contests, 627, 651, 690

(b) Sorted means for caption contest 651

(a) F1

(b) Precision

(c) Recall

Figure 10: F1, Precision, and Recall scores on the New Yorker Caption Contest with $\epsilon = 0.2$

(a) F1       (b) Precision       (c) Recall

Figure 11: F1, Precision, and Recall scores on the New Yorker Caption Contest with $\epsilon = 0.15$

(a) F1       (b) Precision       (c) Recall

Figure 12: F1, Precision, and Recall scores on the New Yorker Caption Contest with $\epsilon = 0.1$

that compound is against the target, ACRVL1. A meta-analysis, done by [2], reported that these values have log-normal distributions with variance less than 1. Therefore, we compute the log of each percent control and may sample from a normal distribution with that mean and variance 1. As before, we plot F1, precision, and recall for all methods. To simulate being in a medical research regime where a higher level of precision is often desired, we take $\delta = 0.001$. We test each method on returning all multiplicative $\epsilon$-good arms with $\epsilon = 0.8$ and plot the results in Figure 13. Note that these curves are plotted on a log-scale to emphasize the early regime of this experiment. It is likewise true here that the oracle baselines perform better on recall than they do on precision. $(\text{ST})^2$ again performs well with respect to precision, and is more competitive with respect to recall in this experiment. Finally, $(\text{ST})^2$ is competitive versus oracle methods on F1 score and greatly outperforms UCB and uniform sampling.

(a) F1 score

(b) Precision

(c) Recall

Figure 13: Precision and Recall curves for the PKIS2 cancer drug discovery experiment with $\epsilon = 0.8$

# B $(\mathtt{ST})^2$, An optimism based algorithm for all-$\epsilon$

---

**Algorithm 2** The $(\mathtt{ST})^2$ Algorithm

---

**Require:** Instance $\nu$, $\epsilon > 0$, $\delta \in (0, 1/2]$, $\gamma \geq 0$ ($\epsilon \in (0, 1/2]$, and $\gamma \in [0, \min(16/\mu_1, 1/2)]$)
1: Pull each arm once, initialize $T_i \leftarrow 1$, update $\hat{\mu}_i$ for each $i \in \{1, 2, \ldots, n\}$
2: Empirically good arms: $\widehat{G} = \{i : \hat{\mu}_i \geq \max_j \hat{\mu}_j - \epsilon\}$ or $\widehat{G} = \{i : \hat{\mu}_i \geq (1-\epsilon) \max_j \hat{\mu}_j\}$
3: $U_t = \max_j \hat{\mu}_j(T_j) + C_{\delta/n}(T_j) - \epsilon - \gamma$ or $U_t = (1 - \epsilon - \gamma)\left(\max_j \hat{\mu}_j(t) + C_{\delta/n}(T_j)\right)$
4: $L_t = \max_j \hat{\mu}_j(T_j) - C_{\delta/n}(T_j) - \epsilon$ or $L_t = (1-\epsilon)\left(\max_j \hat{\mu}_j(t) - C_{\delta/n}(T_j)\right)$
5: Known arms: $K = \{i : \hat{\mu}_i(T_i) + C_{\delta/n}(T_i) < L_t \text{ or } \hat{\mu}_i(T_i) - C_{\delta/n}(T_i) > U_t\}$
6: **while** $K \neq [n]$ **do**
7:      Pull arm $i_1(t) = \arg\min_{i \in \widehat{G} \setminus K} \hat{\mu}_i(T_i) - C_{\delta/n}(T_i)$, update $T_{i_1}, \hat{\mu}_{i_1}$
8:      Pull arm $i_2(t) = \arg\max_{i \in \widehat{G}_\epsilon^c \setminus K} \hat{\mu}_i(T_i) + C_{\delta/n}(T_i)$, update $T_{i_2}, \hat{\mu}_{i_2}$
9:      Pull arm $i^*(t) = \arg\max_i \hat{\mu}_i(T_i) + C_{\delta/n}(T_i)$, update $T_{i^*}, \hat{\mu}_{i^*}$
10:      Update bounds $L_t, U_t$, sets $\widehat{G}, K$
     **return** The set of good arms $\{i : \hat{\mu}_i(T_i) - C_{\delta/n}(T_i) > U_t\}$

---

## B.1 Optimism with additive $\gamma$

**Theorem B.1.** *Fix $\epsilon \geq 0$, $0 < \delta \leq 1/2$, $\gamma \in [0, 16]$ and an instance $\nu$ such that $\max(\Delta_i, |\epsilon - \Delta_i|) \leq 8$ for all $i$. In the case that $G_\epsilon = [n]$, let $\alpha_\epsilon = \min(\alpha_\epsilon, \beta_\epsilon)$. With probability at least $1 - \delta$, $(ST)^2$ correctly returns a set $G$ such that $G_\epsilon \subset G \subset G_{\epsilon+\gamma}$ in at most*

$$12 \sum_{i=1}^n \min \left\{ \max \left\{ \frac{1024}{(\mu_1 - \epsilon - \mu_i)^2} \log \left( \frac{2n}{\delta} \log_2 \left( \frac{3072n}{\delta(\mu_1 - \epsilon - \mu_i)^2} \right) \right), \right. \right.$$
$$\frac{4096}{(\mu_1 + \alpha_\epsilon - \mu_i)^2} \log \left( \frac{2n}{\delta} \log_2 \left( \frac{12288n}{\delta(\mu_1 + \alpha_\epsilon - \mu_i)^2} \right) \right),$$
$$\left. \frac{4096}{(\mu_1 + \beta_\epsilon - \mu_i)^2} \log \left( \frac{2n}{\delta} \log_2 \left( \frac{12288n}{\delta(\mu_1 + \beta_\epsilon - \mu_i)^2} \right) \right) \right\},$$
$$\left. \frac{1}{\gamma^2} \log \left( \frac{2n}{\delta} \log_2 \left( \frac{3072n}{\delta\gamma^2} \right) \right) \right\}$$

*samples.*

*Proof.* Throughout the proof, recall that $\Delta_i = \mu_1 - \mu_i$ for all $i$, $\alpha_\epsilon = \min_{i \in G_\epsilon} \mu_i - (\mu_1 - \epsilon)$, and $\beta_\epsilon = \min_{i \in G_\epsilon^c}(\mu_1 - \epsilon) - \mu_i$. Additionally, at any time $t$, we will take $T_j(t)$ to denote the number of samples of arm $j$ up to time $t$.

Define the event

$$\mathcal{E} = \left\{ \bigcap_{i \in [n]} \bigcap_{t \in \mathbb{N}} |\hat{\mu}_i(t) - \mu_i| \leq C_{\delta/n}(t) \right\}.$$

Using standard anytime confidence bound results, and recalling that that $C_\delta(t) := \sqrt{\frac{4 \log(\log_2(2t)/\delta)}{t}}$, we have

$$\mathbb{P}(\mathcal{E}^c) = \mathbb{P} \left( \bigcup_{i \in [n]} \bigcup_{t \in \mathbb{N}} |\hat{\mu}_i - \mu_i| > C_{\delta/n}(t) \right)$$
$$\leq \sum_{i=1}^n \mathbb{P} \left( \bigcup_{t \in \mathbb{N}} |\hat{\mu}_i - \mu_i| > C_{\delta/n}(t) \right) \leq \sum_{i=1}^n \frac{\delta}{n} = \delta$$

Hence, $\mathbb{P}(\mathcal{E}) \geq 1 - \delta$. Throughout, we will make use of a function $h(x, \delta)$ such that if $t \geq h(x, \delta)$, then $C_\delta(t) \leq |x|$. We bound $h(\cdot, \cdot)$ in Lemma F.2. $h(\cdot, \cdot)$ is assumed to decrease monotonically in both arguments and is symmetric in its first argument.

### B.1.1 Step 0: Correctness

We begin by showing that on $\mathcal{E}$, if $(\text{ST})^2$ terminates, it returns a set $G$ such that $G_\epsilon \subset G \subset G_{\epsilon+\gamma}$. Since $\mathbb{P}(\mathcal{E}) \geq 1-\delta$, this implies that $(\text{ST})^2$ is correct with high probability.

**Claim 0:** On Event $\mathcal{E}$, at all times $t$, $U_t \geq \mu_1 - \epsilon - \gamma$.

**Proof.**
$$U_t = \max_j \hat{\mu}_j(T_j(t)) + C_{\delta/n}(T_j(t)) - \epsilon - \gamma \geq \hat{\mu}_1(T_1(t)) + C_{\delta/n}(T_1(t)) - \epsilon - \gamma$$
$$\overset{\mathcal{E}}{\geq} \mu_1 - \epsilon - \gamma$$

$\square$

**Claim 1:** On Event $\mathcal{E}$, at all times $t$, $L_t \leq \mu_1 - \epsilon$.

**Proof.**
$$L_t = \max_j \hat{\mu}_j(T_j(t)) - C_{\delta/n}(T_j(t)) - \epsilon \overset{\mathcal{E}}{\leq} \max_j \mu_j - \epsilon = \mu_1 - \epsilon$$

$\square$

**Claim 2:** On event $\mathcal{E}$, if there is a time $t$ such that $\hat{\mu}_i(T_i(t)) - C_{\delta/n}(T_i(t)) > U_t$, then $i \in G_{\epsilon+\gamma}$.

**Proof.** Assume for some $t$, $\hat{\mu}_i(T_i(t)) - C_{\delta/n}(T_i(t)) > U_t$. Then
$$\mu_i \overset{\mathcal{E}}{\geq} \hat{\mu}_i(T_i(t)) - C_{\delta/n}(T_i(t)) \geq U_t \overset{\text{Claim 0}}{\geq} \mu_1 - \epsilon - \gamma$$
which implies $i \in G_{\epsilon+\gamma}$ $\square$

**Claim 3:** On event $\mathcal{E}$, if there is a time $t$ such that $\hat{\mu}_i(T_i(t)) + C_{\delta/n}(T_i(t)) < L_t$, then $i \in G_\epsilon^c$.

**Proof.** Assume that is a $t$ for which $\hat{\mu}_i(T_i(t)) + C_{\delta/n}(T_i(t)) < L_t$. Then
$$\mu_i \overset{\mathcal{E}}{\leq} \hat{\mu}_i(T_i(t)) + C_{\delta/n}(T_i(t)) \leq L_t \overset{\text{Claim 1}}{\leq} \mu_1 - \epsilon$$
which implies $i \in G_\epsilon^c$. $\square$

$(\text{ST})^2$ terminates at any time $t$ such that simultaneously for all arms $i$, either $\hat{\mu}_i(T_i(t)) + C_{\delta/n}(T_i(t)) > U_t$ or $\hat{\mu}_i(T_i(t)) - C_{\delta/n}(T_i(t)) < L_t$. On $\mathcal{E}$, by Claim 3, $G_\epsilon \subset \{i : \hat{\mu}_i(T_i(t)) + C_{\delta/n}(T_i(t)) > U_t\}$. On $\mathcal{E}$, by Claim 2, $\{i : \hat{\mu}_i(T_i(t)) + C_{\delta/n}(T_i(t)) > U_t\} \subset G_{\epsilon+\gamma}$. Hence, on the event $\mathcal{E}$. $(\text{ST})^2$ returns a set $G$ such that $G_\epsilon \subset G \subset G_{\epsilon+\gamma}$.

### B.1.2 Step 1: Complexity of estimating the threshold, $\mu_1 - \epsilon$

Let STOP denote the termination event that for all arms $i$, either $\hat{\mu}_i(T_i(t)) + C_{\delta/n}(T_i(t)) > U_t$ or $\hat{\mu}_i(T_i(t)) - C_{\delta/n}(T_i(t)) < L_t$. Let $\omega$ denote the quantity

$$\omega := \max\{\gamma, \min(\alpha_\epsilon, \beta_\epsilon)\}.$$

Let $T$ denote the random variable of the total number of rounds before $(\text{ST})^2$ terminates. At most 3 samples are drawn in any round. Hence, the total sample complexity is bounded by $3T$. We may write $T$ as

$$T := |\{t : \neg\text{STOP}\}| = |\{t : \neg\text{STOP and } i^* \notin G_\omega\}| + |\{t : \neg\text{STOP and } i^* \in G_\omega\}|$$

Next, we bound the first event in this decomposition.

**Claim 0:** On $\mathcal{E}$,

$$|\{t : \neg\text{STOP and } i^* \notin G_\omega\}| \leq \sum_{i \in G_\omega^c} \min\left\{h\left(\tfrac{\gamma}{2}, \tfrac{\delta}{n}\right), \min\left[h\left(\tfrac{\Delta_i}{2}, \tfrac{\delta}{n}\right), h\left(\tfrac{\min(\alpha_\epsilon, \beta_\epsilon)}{2}, \tfrac{\delta}{n}\right)\right]\right\}.$$

**Proof.** If for each $i \in G_\omega^c$, $\mu_i + 2C_{\delta/n}(T_i(t)) < \mu_1$ is true, which is ensured when $T_i(t) > h\left(\Delta_i/2, \tfrac{\delta}{n}\right)$ for all $i \in G_\omega^c$, then

$$\hat{\mu}_i(T_i(t)) + C_{\delta/n}(T_i(t)) \overset{\mathcal{E}}{\leq} \mu_i + 2C_{\delta/n}(T_i(t)) < \mu_1 \overset{\mathcal{E}}{\leq} \hat{\mu}_1(T_1(t)) + C_{\delta/n}(T_1(t))$$

which implies that $i \neq i^*$. Additionally, since $i \in G_\omega^c$ by assumption, we have that $\mu_1 - \omega - \mu_i \geq 0$, which reduces to $\Delta_i \geq \omega$. Since $\omega = \max(\gamma, \min(\alpha_\epsilon, \beta_\epsilon))$, it is likewise true that

$$h\left(\frac{\Delta_i}{2}, \frac{\delta}{n}\right) = \min\left[h\left(\frac{\gamma}{2}, \frac{\delta}{n}\right), \min\left\{h\left(\frac{\Delta_i}{2}, \frac{\delta}{n}\right), h\left(\frac{\min(\alpha_\epsilon, \beta_\epsilon)}{2}, \frac{\delta}{n}\right)\right\}\right].$$

Summing over all $i \in G_\omega^c$ achieves the result. $\qquad\square$

We may decompose the set $\{t : \neg\text{STOP and } i^* \in G_\omega\}$ as

$$\left\{t : \neg\text{STOP and } i^* \in G_\omega \text{ and } C_{\delta/n}(T_{i^*}(t)) > \frac{\omega}{16}\right\}$$

$$\cup \left\{t : \neg\text{STOP and } i^* \in G_\omega \text{ and } C_{\delta/n}(T_{i^*}(t)) \leq \frac{\omega}{16}\right\}$$

**Claim 1:** $\left|\left\{t : \neg\text{STOP and } i^* \in G_\omega \text{ and } C_{\delta/n}(T_{i^*}(t)) > \frac{\omega}{16}\right\}\right| \leq$ $\sum_{i \in G_\omega} \min\left\{h\left(\frac{\gamma}{16}, \frac{\delta}{n}\right), \min\left[h\left(\frac{\Delta_i}{8}, \frac{\delta}{n}\right), h\left(\frac{\min(\alpha_\epsilon, \beta_\epsilon)}{16}, \frac{\delta}{n}\right)\right]\right\}$

**Proof.** $C_{\delta/n}(T_i(t)) \leq \frac{\omega}{16}$ is true when $T_i(t) \geq h\left(\frac{\omega}{16}, \frac{\delta}{n}\right)$. Since $i^* \in G_\omega$, $\mu_i - (\mu_1 - \omega) \geq 0$, which implies $\Delta_i \leq \omega$. By definition, $\omega = \min(\gamma, \min(\alpha_\epsilon, \beta_\epsilon))$. Hence, by monotonicity of $h(\cdot, \cdot)$,

$$h\left(\frac{\omega}{16}, \frac{\delta}{n}\right) = \min\left[h\left(\frac{\Delta_i}{16}, \frac{\delta}{n}\right), h\left(\frac{\omega}{16}, \frac{\delta}{n}\right)\right]$$

$$= \min\left\{h\left(\frac{\gamma}{16}, \frac{\delta}{n}\right), \min\left[h\left(\frac{\Delta_i}{16}, \frac{\delta}{n}\right), h\left(\frac{\min(\alpha_\epsilon, \beta_\epsilon)}{16}, \frac{\delta}{n}\right)\right]\right\}$$

Summing over all $i \in G_\omega$ achieves the desired result. $\qquad\square$

### B.1.3 Step 2: Controlling "crossing" events

Recall that we sample $i_1(t) \in \widehat{G}$ and $i_2(t) \in \widehat{G}^c$. In this section, we control the number of times that $i_1(t) \in G_{\epsilon+\frac{\gamma}{2}}^c$ and $i_2(t) \in G_{\epsilon+\frac{\gamma}{2}}$.

To do so, we first decompose the set $\left\{t : \neg\text{STOP and } i^* \in G_\omega \text{ and } C_{\delta/n}(T_{i^*}(t)) \leq \frac{\omega}{16}\right\}$ as

$$\left\{t : \neg\text{STOP and } i^* \in G_\omega \text{ and } C_{\delta/n}(T_{i^*}(t)) \leq \frac{\omega}{16} \text{ and } i_1(t) \in G_{\epsilon+\frac{\gamma}{2}}^c\right\}$$

$$\cup \left\{t : \neg\text{STOP and } i^* \in G_\omega \text{ and } C_{\delta/n}(T_{i^*}(t)) \leq \frac{\omega}{16} \text{ and } i_1(t) \in G_{\epsilon+\frac{\gamma}{2}}\right\}$$

**Claim 0:** $\left|\left\{t : \neg\text{STOP and } i^* \in G_\omega \text{ and } C_{\delta/n}(T_{i^*}(t)) \leq \frac{\omega}{16} \text{ and } i_1(t) \in G_{\epsilon+\frac{\gamma}{2}}^c\right\}\right| \leq$ $\sum_{i \in G_{\epsilon+\frac{\gamma}{2}}^c} \min\left[h\left(\frac{\Delta_i - \epsilon}{8}, \frac{\delta}{n}\right), h\left(\frac{\gamma}{8}, \frac{\delta}{n}\right)\right]$.

**Proof.** Recall that $\widehat{G}$ is the set of all arms whose empirical means exceed $\max_i \hat{\mu}_i(T_i(t)) - \epsilon$, and $i_1(t) \in \widehat{G}$ by definition. Note that $\max_i \hat{\mu}_i(T_i(t)) - \epsilon > \max_i \hat{\mu}_i(T_i(t)) - C_{\delta/n}(T_i(t)) - \epsilon = L_t$. Hence, if an arm's upper bound is below $L_t$, then the arm cannot be in $\widehat{G}$ and thus not be $i_1(t)$. By the above event, $C_{\delta/n}(T_{i^*}(t)) \leq \frac{\omega}{16}$. Hence,

$$\mu_i^* + \frac{\omega}{8} \geq \mu_i^* + 2C_{\delta/n}(T_{i^*}(t)) \overset{\mathcal{E}}{\geq} \hat{\mu}_i^*(T_{i^*}(t)) + C_{\delta/n}(T_{i^*}(t)) \geq \hat{\mu}_1(T_1(t)) + C_{\delta/n}(T_1(t)) \overset{\mathcal{E}}{\geq} \mu_1.$$

Therefore, $\mu_{i^*} \geq \mu_1 - \frac{\omega}{8}$ or equivalently, $i^* \in G_{\omega/8}$. Using this,

$$L_t = \max_i \hat{\mu}_i(T_i(t)) - C_{\delta/n}(T_i(t)) - \epsilon \geq \hat{\mu}_{i^*}(T_{i^*}(t)) - C_{\delta/n}(T_{i^*}(t)) - \epsilon$$

$$\overset{\mathcal{E}}{\geq} \mu_{i^*} - 2C_{\delta/n}(T_{i^*}(t)) - \epsilon$$

$$\overset{\mathcal{E}}{\geq} \mu_{i^*} - \frac{\omega}{8} - \epsilon$$

$$\geq \mu_1 - \frac{\omega}{4} - \epsilon$$

Next, we bound the number of times an arm $i \in G^c_{\epsilon + \frac{\gamma}{2}}$ is sampled before its upper bound is below $\mu_1 - \frac{\omega}{4} - \epsilon$. Note that $C_{\delta/n}(T_i(t)) < \frac{1}{2}\left(\mu_1 - \frac{\omega}{4} - \epsilon - \mu_i\right)$, true when $T_i(t) > h\left(\frac{1}{2}\left(\mu_1 - \frac{\omega}{4} - \epsilon - \mu_i\right), \frac{\delta}{n}\right)$ implies that

$$\hat{\mu}_i(T_i(t)) + C_{\delta/n}(T_i(t)) \overset{\mathcal{E}}{\leq} \mu_i + 2C_{\delta/n}(T_i(t)) < \mu_1 - \frac{\omega}{4} - \epsilon \leq L_t.$$

Finally, we turn our attention to the difference $\mu_1 - \frac{\omega}{4} - \epsilon - \mu_i$. Recall that $\omega = \max(\gamma, \min(\alpha_\epsilon, \beta_\epsilon))$.

$$\mu_1 - \frac{\omega}{4} - \epsilon - \mu_i = (\mu_1 - \epsilon) - \mu_i - \frac{1}{4}\omega$$
$$= (\mu_1 - \epsilon) - \mu_i - \frac{1}{4}\max(\gamma, \min(\alpha_\epsilon, \beta_\epsilon)).$$

By definition, $\beta_\epsilon = \min_{i \in G^c_\epsilon}(\mu_1 - \epsilon) - \mu_i$. Hence, $\min(\alpha_\epsilon, \beta_\epsilon) \leq (\mu_1 - \epsilon) - \mu_i$ for all $i \in G^c_{\epsilon + \frac{\gamma}{2}}$. Similarly, since $i \in G^c_{\epsilon + \frac{\gamma}{2}}$ by assumption, $(\mu_1 - \epsilon - \frac{\gamma}{2}) - \mu_i \geq 0$, which rearranges to $\frac{\gamma}{2} \leq (\mu_1 - \epsilon) - \mu_i$. Therefore,

$$(\mu_1 - \epsilon) - \mu_i - \frac{1}{4}\max(\gamma, \min(\alpha_\epsilon, \beta_\epsilon)) \geq \frac{1}{2}((\mu_1 - \epsilon) - \mu_i) = \frac{\Delta_i - \epsilon}{2}.$$

Hence, by monotonicity of $h(\cdot, \cdot)$,

$$h\left(\frac{1}{2}\left(\mu_1 - \frac{\omega}{4} - \epsilon - \mu_i\right), \frac{\delta}{n}\right) \leq h\left(\frac{\Delta_i - \epsilon}{4}, \frac{\delta}{n}\right).$$

Lastly, as above, since $i \in G^c_{\epsilon + \frac{\gamma}{2}}$, we have that $\Delta_i - \epsilon = (\mu_1 - \epsilon) - \mu_i \geq \frac{1}{2}\gamma$. Hence,

$$h\left(\frac{\Delta_i - \epsilon}{4}, \frac{\delta}{n}\right) \leq \min\left[h\left(\frac{\Delta_i - \epsilon}{8}, \frac{\delta}{n}\right), h\left(\frac{\gamma}{8}, \frac{\delta}{n}\right)\right].$$

Putting this together, if $T_i(t) \geq \min\left[h\left(\frac{\Delta_i - \epsilon}{8}, \frac{\delta}{n}\right), h\left(\frac{\gamma}{8}, \frac{\delta}{n}\right)\right]$, then $i \neq i_1(t)$ for all $i \in G^c_{\epsilon + \frac{\gamma}{2}}$. Summing over all such $i$ bounds the size of set stated in the claim. $\square$

We decompose the remaining event

$$\left\{t : \neg\text{STOP and } i^* \in G_\omega \text{ and } C_{\delta/n}(T_{i^*}(t)) \leq \frac{\omega}{16} \text{ and } i_1(t) \in G_{\epsilon + \frac{\gamma}{2}}\right\}$$

as

$$\left\{t : \neg\text{STOP and } i^* \in G_\omega \text{ and } C_{\delta/n}(T_{i^*}(t)) \leq \frac{\omega}{16} \text{ and } i_1(t) \in G_{\epsilon + \frac{\gamma}{2}} \text{ and } i_2(t) \in G_{\epsilon + \frac{\gamma}{2}}\right\}$$
$$\cup \left\{t : \neg\text{STOP and } i^* \in G_\omega \text{ and } C_{\delta/n}(T_{i^*}(t)) \leq \frac{\omega}{16} \text{ and } i_1(t) \in G_{\epsilon + \frac{\gamma}{2}} \text{ and } i_2(t) \in G^c_{\epsilon + \frac{\gamma}{2}}\right\}.$$

We proceed by bounding the size of the first set.

**Claim 1:**

$$\left|\left\{t : \neg\text{STOP and } i^* \in G_\omega \text{ and } C_{\delta/n}(T_{i^*}(t)) \leq \frac{\omega}{16} \text{ and } i_1(t) \in G_{\epsilon + \frac{\gamma}{2}} \text{ and } i_2(t) \in G_{\epsilon + \frac{\gamma}{2}}\right\}\right|$$
$$\leq \sum_{i \in G_{\epsilon + \frac{\gamma}{2}}} \min\left[h\left(\frac{\epsilon \Delta_i}{8}, \frac{\delta}{n}\right), h\left(\frac{\gamma}{8}, \frac{\delta}{n}\right)\right]$$

**Proof.** Recall that $K = \{i : \hat{\mu}(T_i(t)) + C_{\delta/n}(T_i(t)) < L_t \text{ or } \hat{\mu}(T_i(t)) - C_{\delta/n}(T_i(t)) > L_t\}$ and $i_2$ is sampled from the set $\widehat{G}^c \backslash K$, ie all arms in $\widehat{G}^c$ who have not been declared as above $U_t$ or below $L_t$. Hence, if an arm's lower bound exceeds $U_t = \max_i \hat{\mu}(T_i(t)) + C_{\delta/n}(T_i(t)) - \epsilon - \gamma$, it must be in $K$ an thus cannot be $i_2$. Recall that $i^*(t) = \arg\max \hat{\mu}_i(T_i(t)) + C_{\delta/n}(T_i(t))$. By the above event, $i^*(t) \in G_\omega$ and $C_{\delta/n}(T_{i^*}(t)) \leq \frac{\omega}{16}$. Hence,

$$U_t = \max_i \hat{\mu}_i(T_i(t)) + C_{\delta/n}(T_i(t)) - \epsilon - \gamma = \hat{\mu}_{i^*(t)}(T_{i^*(t)}(t)) + C_{\delta/n}(T_{i^*(t)}(t)) - \epsilon - \gamma$$

$$\overset{\mathcal{E}}{\leq} \mu_{i^*(t)} + 2C_{\delta/n}(T_{i^*(t)}(t)) - \epsilon - \gamma$$

$$\leq \mu_{i^*(t)} + \frac{\omega}{8} - \epsilon - \gamma$$

$$\leq \mu_1 + \frac{\omega}{8} - \epsilon - \gamma$$

Next, we bound the number of times an arm $i \in G_{\epsilon + \frac{\gamma}{2}}$ is sampled before its lower bound is above $\mu_1 + \frac{\omega}{8} - \epsilon - \gamma$. Note that $C_{\delta/n}(T_i(t)) < \frac{1}{2}\left(\mu_i - (\mu_1 + \frac{\omega}{8} - \epsilon - \gamma)\right)$, true when $T_i(t) > h\left(\frac{1}{2}\left(\mu_i - (\mu_1 + \frac{\omega}{8} - \epsilon - \gamma)\right), \frac{\delta}{n}\right)$ implies that

$$\hat{\mu}_i(T_i(t)) - C_{\delta/n}(T_i(t)) \overset{\mathcal{E}}{\geq} \mu_i - 2C_{\delta/n}(T_i(t)) > \mu_1 + \frac{\omega}{8} - \epsilon - \gamma.$$

Finally, we turn our attention to the difference $\mu_i - (\mu_1 + \frac{\omega}{8} - \epsilon - \gamma)$. Recall that $\omega = \max(\gamma, \min(\alpha_\epsilon, \beta_\epsilon))$.

$$\mu_i - \left(\mu_1 + \frac{\omega}{8} - \epsilon - \gamma\right) = \mu_i - (\mu_1 - \epsilon) + \gamma - \frac{1}{8}\omega$$

**Case 1a, $\omega = \min(\alpha_\epsilon, \beta_\epsilon)$ and $i \in G_\epsilon$:.**

By definition, $\alpha_\epsilon = \min_{i \in G_\epsilon} \mu_i - (\mu_1 - \epsilon)$. Hence, $\min(\alpha_\epsilon, \beta_\epsilon) \leq \mu_i - (\mu_1 - \epsilon)$ for all $i \in G_\epsilon$. Therefore,

$$\mu_i - (\mu_1 - \epsilon) + \gamma - \frac{1}{8}\omega = \mu_i - (\mu_1 - \epsilon) + \gamma - \frac{1}{8}\min(\alpha_\epsilon, \beta_\epsilon)$$

$$\geq \max\left(\mu_i - (\mu_1 - \epsilon) - \frac{1}{8}\min(\alpha_\epsilon, \beta_\epsilon), \gamma\right)$$

$$\geq \max\left(\frac{7}{8}(\mu_i - (\mu_1 - \epsilon)), \gamma\right)$$

**Case 1b, $\omega = \min(\alpha_\epsilon, \beta_\epsilon)$ and $i \in G_\epsilon^c \cap G_{\epsilon + \frac{\gamma}{2}}$**

Since $\omega = \max(\gamma, \min(\alpha_\epsilon, \beta_\epsilon))$, if $\omega = \min(\alpha_\epsilon, \beta_\epsilon)$, then $\frac{1}{2}\gamma < \min(\alpha_\epsilon, \beta_\epsilon)$. Since $\min(\alpha_\epsilon, \beta_\epsilon) = \min|\mu_i - (\mu_1 - \epsilon)|$, the set $G_\epsilon^c \cap G_{\epsilon + \frac{\gamma}{2}}$ is empty and there is nothing to prove.

**Case 2a, $\omega = \gamma$ and $i \in G_\epsilon$:**

$$\mu_i - (\mu_1 - \epsilon) + \gamma - \frac{1}{8}\omega = \mu_i - (\mu_1 - \epsilon) + \frac{7}{8}\gamma \geq \max\left(\mu_i - (\mu_1 - \epsilon), \frac{7}{8}\gamma\right)$$

**Case 2b, $\omega = \gamma$ and $i \in G_\epsilon^c \cap G_{\epsilon + \frac{\gamma}{2}}$:**

For $i \in G_\epsilon^c \cap G_{\epsilon + \frac{\gamma}{2}}$, we have that $\mu_i - (\mu_1 - \epsilon - \gamma/2) \geq 0$. Hence $\mu_i - (\mu_1 - \epsilon) \geq \frac{-\gamma}{2}$. Therefore,

$$\mu_i - (\mu_1 - \epsilon) + \gamma - \frac{1}{8}\omega \geq \frac{3}{8}\gamma = \max\left(\frac{3}{8}((\mu_1 - \epsilon) - \mu_i), \frac{3}{8}\gamma\right).$$

Applying the above cases and using monotonicity of $h(\cdot, \cdot)$, we see that for $i \in G_{\epsilon + \frac{\gamma}{2}}$,

$$h\left(\frac{1}{2}\left(\mu_i - \left(\mu_1 + \frac{\omega}{8} - \epsilon\right)\right), \frac{\delta}{n}\right) \leq \min\left[h\left(\frac{\epsilon - \Delta_i}{8}, \frac{\delta}{n}\right), h\left(\frac{\gamma}{8}, \frac{\delta}{n}\right)\right].$$

Hence, if any $i \in G_{\epsilon + \frac{\gamma}{2}}$ has received this many samples, then its lower bound exceeds $U_t$ and thus the arm must be in $\widehat{G}$. Putting this together, if $T_i(t) \geq \min\left[h\left(\frac{\epsilon - \Delta_i}{8}, \frac{\delta}{n}\right), h\left(\frac{\gamma}{8}, \frac{\delta}{n}\right)\right]$, then $i \neq i_2(t)$ for all $i \in G_{\epsilon + \frac{\gamma}{2}}$. Summing over all such $i$ bounds the size of set stated in the claim. $\qquad\square$

### B.1.4   Step 3: Controlling the complexity until stopping occurs

In this step, we turn our attention to the final event to control:

$$S := \left\{ t : \neg\text{STOP and } i^* \in G_\omega \text{ and } C_{\delta/n}(T_{i^*}(t)) \leq \frac{\omega}{16} \text{ and } i_1(t) \in G_{\epsilon+\frac{\gamma}{2}} \text{ and } i_2(t) \in G^c_{\epsilon+\frac{\gamma}{2}} \right\}.$$

For brevity, we will refer to this set as $S$ for this step. The objective will be to bound the time before each arms lower bound either clears $U_t$ or its upper bound clears $L_t$ which implies the stopping condition. To do so, we introduce, two events:

$$E_1(t) := \{\hat{\mu}_{i_1(t)}(T_{i_1(t)}(t)) - C_{\delta/n}(T_{i_1(t)}(t)) > U_t\} \tag{5}$$

and

$$E_2(t) := \{\hat{\mu}_{i_2(t)}(T_{i_2(t)}(t)) + C_{\delta/n}(T_{i_2(t)}(t)) < L_t\}. \tag{6}$$

If $E_1(t)$ is true, then $\hat{\mu}_i(T_i) - C_{\delta/n}(T_i(t)) > L_t$ for all $i \in \widehat{G}$. If $E_2(t)$ is true, then $\hat{\mu}_i(T_i) + C_{\delta/n}(T_i(t)) < U_t$ for all $i \in \widehat{G}^c$. Hence, by line 6 of $(\text{ST})^2$, if both $E_1(t)$ and $E_2(t)$ are true, then $(\text{ST})^2$ terminates.

**Claim 0:** $|S \cap \{t : \neg E_1(t)\}| \leq \sum_{i \in G_{\epsilon+\frac{\gamma}{2}}} \min\left[h\left(\frac{\epsilon-\Delta_i}{8}, \frac{\delta}{n}\right), h\left(\frac{\gamma}{8}, \frac{\delta}{n}\right)\right].$

**Proof.** Recall that by the set $S$, we have that $i_1(t) \in G_{\epsilon+\frac{\gamma}{2}}$. Furthermore, by the set $S$, we have that $i^*(t) \in G_\omega$ and $C_{\delta/n}(T_{i^*}(t)) \leq \omega/16$. Hence,

$$\begin{aligned}
U_t &= \max_i \hat{\mu}_i(T_i(t)) + C_{\delta/n}(T_i(t)) - \epsilon - \gamma \\
&= \hat{\mu}_{i^*(t)}(T_{i^*(t)}(t)) + C_{\delta/n}(T_{i^*(t)}(t)) - \epsilon - \gamma \\
&\overset{\mathcal{E}}{\leq} \mu_{i^*(t)} + 2C_{\delta/n}(T_{i^*(t)}(t)) - \epsilon - \gamma \\
&\leq \mu_{i^*(t)} + \frac{\omega}{8} - \epsilon - \gamma \\
&\leq \mu_1 + \frac{\omega}{8} - \epsilon - \gamma
\end{aligned}$$

If $C_{\delta/n}(T_i) \leq \frac{1}{2}\left(\mu_i - \left(\mu_1 + \frac{\omega}{8} - \epsilon - \gamma\right)\right)$ which is true when $T_i \geq h\left(\frac{1}{2}\left(\mu_i - \left(\mu_1 + \frac{\omega}{8} - \epsilon - \gamma\right)\right), \frac{\delta}{n}\right)$, then

$$\hat{\mu}_i(T_i) - C_{\delta/n}(T_i) \geq \mu_i - 2C_{\delta/n}(T_i) \geq \mu_1 + \frac{\omega}{8} - \epsilon - \gamma \geq U_t.$$

The remainder of the proof of this claim focuses on controlling the difference: $\mu_i - \left(\mu_1 + \frac{\omega}{8} - \epsilon - \gamma\right)$ in the case that $\omega = \min(\alpha_\epsilon, \beta_\epsilon)$ and $\omega = \gamma$. Recall that $\omega = \max(\gamma, \min(\alpha_\epsilon, \beta_\epsilon))$. Hence, if any possible $i \in G_{\epsilon+\frac{\gamma}{2}}$ has received sufficiently many samples, since $i_1(t) \in G_{\epsilon+\frac{\gamma}{2}}$, this implies $E_1(t)$.

**Case 1a,** $\omega = \min(\alpha_\epsilon, \beta_\epsilon)$ **and** $i \in G_\epsilon$

We focus on the difference $\mu_i - \left(\mu_1 + \frac{\omega}{8} - \epsilon - \gamma\right)$.

$$\begin{aligned}
\mu_i - \left(\mu_1 + \frac{\omega}{8} - \epsilon - \gamma\right) &= \mu_i - \left(\mu_1 + \frac{\min(\alpha_\epsilon, \beta_\epsilon)}{8} - \epsilon - \gamma\right) \\
&= \mu_i - (\mu_1 - \epsilon) + \gamma - \frac{1}{8}\min(\alpha_\epsilon, \beta_\epsilon) \\
&\overset{(\gamma\geq 0)}{\geq} \frac{1}{2}(\mu_i - (\mu_1 - \epsilon)) = \frac{\epsilon - \Delta_i}{2}
\end{aligned}$$

where the final step follows since $\min(\alpha_\epsilon, \beta_\epsilon) \leq \alpha_\epsilon \leq \mu_i - (\mu_1 - \epsilon)$ by definition for all $i \in G_\epsilon$. Then by monotonicity of $h(\cdot, \cdot)$,

$$h\left(\frac{1}{2}\left(\mu_i - \left(\mu_1 + \frac{\omega}{8} - \epsilon - \gamma\right)\right), \frac{\delta}{n}\right) \leq h\left(\frac{\epsilon - \Delta_i}{4}, \frac{\delta}{n}\right).$$

Lastly, in this setting, $\gamma \leq \min(\alpha_\epsilon, \beta_\epsilon) \leq \epsilon - \Delta_i$ since $\omega = \min(\alpha_\epsilon, \beta_\epsilon)$. Hence, it is trivially true that

$$h\left(\frac{\epsilon - \Delta_i}{4}, \frac{\delta}{n}\right) = \min\left[h\left(\frac{\epsilon - \Delta_i}{4}, \frac{\delta}{n}\right), h\left(\frac{\gamma}{4}, \frac{\delta}{n}\right)\right]$$

**Case 1b,** $\omega = \min(\alpha_\epsilon, \beta_\epsilon)$ **and** $i \in G_\epsilon^c \cap G_{\epsilon+\frac{\gamma}{2}}$

Since $\omega = \max(\gamma, \min(\alpha_\epsilon, \beta_\epsilon))$, if $\omega = \min(\alpha_\epsilon, \beta_\epsilon)$, then $\frac{1}{2}\gamma < \min(\alpha_\epsilon, \beta_\epsilon)$. Since $\min(\alpha_\epsilon, \beta_\epsilon) = \min|\mu_i - (\mu_1 - \epsilon)|$, the set $G_\epsilon^c \cap G_{\epsilon+\frac{\gamma}{2}}$ is empty and there is nothing to prove.

**Case 2a,** $\omega = \gamma$ **and** $i \in G_\epsilon$

Again, we bound the difference $\mu_i - \left(\mu_1 + \frac{\omega}{4} - \epsilon - \gamma\right)$.

$$\mu_i - \left(\mu_1 + \frac{\omega}{8} - \epsilon - \gamma\right) = \mu_i - (\mu_1 - \epsilon) + \frac{7}{8}\gamma$$

Since $i \in G_\epsilon$, $\mu_i - (\mu_1 - \epsilon) \ge 0$. Hence,

$$\mu_i - (\mu_1 - \epsilon) + \frac{7}{8}\gamma \ge \max\left(\mu_i - (\mu_1 - \epsilon), \frac{7}{8}\gamma\right)$$

$$\ge \frac{1}{2}\max\left(\epsilon - \Delta_i, \gamma\right)$$

Therefore, we have that

$$h\left(\frac{1}{2}\left(\mu_i - \left(\mu_1 + \frac{\omega}{8} - \epsilon - \gamma\right)\right), \frac{\delta}{n}\right) \le h\left(\frac{\epsilon - \Delta_i}{4}, \frac{\delta}{n}\right)$$

and

$$h\left(\frac{1}{2}\left(\mu_i - \left(\mu_1 + \frac{\omega}{8} - \epsilon - \gamma\right)\right), \frac{\delta}{n}\right) \le h\left(\frac{\gamma}{4}, \frac{\delta}{n}\right).$$

Hence,

$$h\left(\frac{1}{2}\left(\mu_i - \left(\mu_1 + \frac{\omega}{4} - \epsilon - \gamma\right)\right), \frac{\delta}{n}\right) \le \min\left[h\left(\frac{\epsilon - \Delta_i}{4}, \frac{\delta}{n}\right), h\left(\frac{\gamma}{4}, \frac{\delta}{n}\right)\right].$$

**Case 2b,** $\omega = \gamma$ **and** $i \in G_\epsilon^c \cap G_{\epsilon+\frac{\gamma}{2}}$

As before,

$$\mu_i - \left(\mu_1 + \frac{\omega}{8} - \epsilon - \gamma\right) = \mu_i - (\mu_1 - \epsilon) + \frac{7}{8}\gamma$$

Since $i \in G_\epsilon^c \cap G_{\epsilon+\frac{\gamma}{2}}$, we have that $\mu_i - (\mu_1 - \epsilon - \frac{\gamma}{2}) \ge 0$. Rearranging implies that $\mu_i - (\mu_1 - \epsilon) \ge \frac{-1}{2}\gamma$. Hence,

$$\mu_i - (\mu_1 - \epsilon) + \frac{7}{8}\gamma \ge \frac{3}{8}\gamma.$$

Hence,

$$h\left(\frac{1}{2}\left(\mu_i - \left(\mu_1 + \frac{\omega}{8} - \epsilon - \gamma\right)\right), \frac{\delta}{n}\right) \le h\left(\frac{\gamma}{8}, \frac{\delta}{n}\right).$$

Additionally, as above, if $i \in G_\epsilon^c \cap G_{\epsilon+\frac{\gamma}{2}}$, we have that $\mu_i - (\mu_1 - \epsilon - \frac{\gamma}{2}) \ge 0$ which implies that $(\mu_1 - \epsilon) - \mu_i \le \gamma$. Hence

$$h\left(\frac{\gamma}{8}, \frac{\delta}{n}\right) = \min\left[h\left(\frac{\Delta_i - \epsilon}{8}, \frac{\delta}{n}\right), h\left(\frac{\gamma}{8}, \frac{\delta}{n}\right)\right].$$

Therefore, if $T_i$ exceeds the above, then $E_1(t)$ is true for an $i_1 \in G_\epsilon^c \cap G_{\epsilon+\frac{\gamma}{2}}$. Combining all cases, and noting that $h(x, \delta) \ge h(x/2, \delta) \, \forall x$, we see that for $i_1 \in G_{\epsilon+\frac{\gamma}{2}}$, if

$$T_{i_1(t)}(t) > \min\left[h\left(\frac{\epsilon - \Delta_i}{8}, \frac{\delta}{n}\right), h\left(\frac{\gamma}{8}, \frac{\delta}{n}\right)\right],$$

Then $E_1(t)$ is true. Summing over all possible $i_1 \in G_{\epsilon+\frac{\gamma}{2}}$ proves the claim. $\qquad\square$

**Claim 1:** $|\mathcal{S} \cap \{t : E_1(t)\} \cap \{t : \neg E_2(t)\}| \le \sum_{i \in G_{\epsilon+\frac{\gamma}{2}}^c} \min\left[h\left(\frac{\epsilon - \Delta_i}{8}, \frac{\delta}{n}\right), h\left(\frac{\gamma}{8}, \frac{\delta}{n}\right)\right].$

**Proof.** By the events in set $\mathcal{S}$, $C_{\delta/n}(T_{i^*}(t)) \leq \frac{\omega}{16}$. Hence,

$$\mu_i^* + \frac{\omega}{8} \geq \mu_i^* + 2C_{\delta/n}(T_{i^*}(t)) \overset{\mathcal{E}}{\geq} \hat{\mu}_i^*(T_{i^*}(t)) + C_{\delta/n}(T_{i^*}(t)) \geq \hat{\mu}_1(T_1(t)) + C_{\delta/n}(T_1(t)) \overset{\mathcal{E}}{\geq} \mu_1.$$

Therefore, $\mu_{i^*} \geq \mu_1 - \frac{\omega}{8}$ or equivalently, $i^* \in G_{\omega/8}$. Using this,

$$L_t = \max_i \hat{\mu}_i(T_i(t)) - C_{\delta/n}(T_i(t)) - \epsilon \geq \hat{\mu}_{i^*}(T_{i^*}(t)) - C_{\delta/n}(T_{i^*}(t)) - \epsilon$$

$$\overset{\mathcal{E}}{\geq} \mu_{i^*} - 2C_{\delta/n}(T_{i^*}(t)) - \epsilon$$

$$\overset{\mathcal{E}}{\geq} \mu_{i^*} - \frac{\omega}{8} - \epsilon$$

$$\geq \mu_1 - \frac{\omega}{4} - \epsilon$$

For $i \in G^c_{\epsilon+\frac{\gamma}{2}}$, if $C_{\delta/n}(T_i) \leq \frac{1}{2}\left(\left(\mu_1 - \frac{\omega}{4} - \epsilon\right) - \mu_i\right)$, true when $T_i \geq h\left(\frac{1}{2}\left(\left(\mu_1 - \frac{\omega}{4} - \epsilon\right) - \mu_i\right), \frac{\delta}{n}\right)$, then

$$\hat{\mu}_i(T_i) + C_{\delta/n}(T_i) \leq \mu_i + 2C_{\delta/n}(T_i) \leq \mu_1 - \frac{\omega}{4} - \epsilon \leq L_t.$$

As before, we seek a lower bound for the difference $\left(\mu_1 - \frac{\omega}{4} - \epsilon\right) - \mu_i$.

**Case 1:** $\omega = \min(\alpha_\epsilon, \beta_\epsilon)$

$$\left(\mu_1 - \frac{\omega}{4} - \epsilon\right) - \mu_i = (\mu_1 - \epsilon) - \mu_i - \frac{1}{4}\min(\alpha_\epsilon, \beta_\epsilon)$$

$$\geq \frac{1}{2}\left((\mu_1 - \epsilon) - \mu_i\right)$$

since $(\mu_1 - \epsilon) - \mu_i \geq \min(\alpha_\epsilon, \beta_\epsilon)$. Therefore, we have that

$$h\left(\frac{1}{2}\left(\left(\mu_1 - \frac{\omega}{4} - \epsilon\right) - \mu_i\right), \frac{\delta}{n}\right) \leq h\left(\frac{\Delta_i - \epsilon}{4}, \frac{\delta}{n}\right).$$

Lastly, in this setting, $\gamma \leq \min(\alpha_\epsilon, \beta_\epsilon) \leq \epsilon - \Delta_i$ since $\omega = \min(\alpha_\epsilon, \beta_\epsilon)$. Hence, it is trivially true that

$$h\left(\frac{\Delta_i - \epsilon}{4}, \frac{\delta}{n}\right) = \min\left[h\left(\frac{\Delta_i - \epsilon}{4}, \frac{\delta}{n}\right), h\left(\frac{\gamma}{4}, \frac{\delta}{n}\right)\right].$$

**Case 2:** $\omega = \gamma$

Assume that $\gamma > \min(\alpha_\epsilon, \beta_\epsilon)$, as equality is covered by the previous case. Hence,

$$\left(\mu_1 - \frac{\omega}{4} - \epsilon\right) - \mu_i = (\mu_1 - \epsilon) - \mu_i - \frac{1}{4}\gamma$$

Recall that we seek to control $i_2 \in G^c_{\epsilon+\frac{\gamma}{2}}$. For any $i \in G^c_{\epsilon+\frac{\gamma}{2}}$, we have that $\mu_1 - \epsilon - \frac{\gamma}{2} - \mu_i \geq 0$. Rearranging, we see that $(\mu_1 - \epsilon) - \mu_i \geq \frac{1}{2}\gamma$ which implies that

$$(\mu_1 - \epsilon) - \mu_i - \frac{1}{4}\gamma \geq \frac{1}{2}((\mu_1 - \epsilon) - \mu_i).$$

Therefore, we have that

$$h\left(\frac{1}{2}\left(\left(\mu_1 - \frac{\omega}{4} - \epsilon\right) - \mu_i\right), \frac{\delta}{n}\right) \leq h\left(\frac{\Delta_i - \epsilon}{4}, \frac{\delta}{n}\right)$$

is this setting as well. Similarly, since $\Delta_i - \epsilon \geq \frac{1}{2}\gamma$, we likewise have that

$$h\left(\frac{\Delta_i - \epsilon}{4}, \frac{\delta}{n}\right) \leq \min\left[h\left(\frac{\Delta_i - \epsilon}{8}, \frac{\delta}{n}\right), h\left(\frac{\gamma}{8}, \frac{\delta}{n}\right)\right].$$

Hence, if $T_i$ exceeds the right-hand side of the preceding inequality, then for any $i \in G^c_{\epsilon+\frac{\gamma}{2}}$, its upper bound is below $L_t$. Hence for $i_2(t) \in G^c_{\epsilon+\frac{\gamma}{2}}$, this implies event $E_2(t)$. Summing over all possible values of $i_2(t) \in G^c_{\epsilon+\frac{\gamma}{2}}$ proves the claim. $\qquad\square$

**Claim 2:** The cardinality of $\mathcal{S}$ is bounded as $|\mathcal{S}| \leq \sum_{i=1}^{n} \min\left[h\left(\frac{\Delta_i - \epsilon}{8}, \frac{\delta}{n}\right), h\left(\frac{\gamma}{8}, \frac{\delta}{n}\right)\right]$.

**Proof.** First, $\mathcal{S}$ may be decomposed as

$$|\mathcal{S}| = |\mathcal{S} \cap \{t : \neg E_1(t)\}| + |\mathcal{S} \cap \{t : E_1(t)\} \cap \{t : \neg E_2(t)\}| + |\mathcal{S} \cap \{t : E_1(t)\} \cap \{t : E_2(t)\}|$$

Note that $|\mathcal{S} \cap \{t : E_1(t)\} \cap \{t : E_2(t)\}| = 0$ because we have assumed in set $\mathcal{S}$ that $(\text{ST})^2$ has not stopped, and $\{t : E_1(t)\} \cap \{t : E_2(t)\}$ implies termination. By Claim 0, $|\mathcal{S} \cap \{t : \neg E_1(t)\}| \leq \sum_{i \in G_{\epsilon + \frac{\gamma}{2}}} \min\left[h\left(\frac{\epsilon - \Delta_i}{4}, \frac{\delta}{n}\right), h\left(\frac{\gamma}{4}, \frac{\delta}{n}\right)\right]$. By Claim 1, $|\mathcal{S} \cap \{t : E_1(t)\} \cap \{t : \neg E_2(t)\}| \leq \sum_{i \in G^c_{\epsilon + \frac{\gamma}{2}}} \min\left[h\left(\frac{\epsilon - \Delta_i}{8}, \frac{\delta}{n}\right), h\left(\frac{\gamma}{8}, \frac{\delta}{n}\right)\right]$. Recalling that $h$ is assumed to be symmetric in its first argument proves the claim. $\qquad\square$

### B.1.5  Step 4: Putting it all together

Recall that the total number of rounds $T$ that $(\text{ST})^2$ runs for is given by $T = |\{t : \neg\text{STOP}\}|$. To bound this quantity, we have decomposed the set $\{t : \neg\text{STOP}\}$ into many subsets. Below, we show this decomposition.

$$\{t : \neg\text{STOP}\} =$$
$$\{t : \neg\text{STOP and } i^* \notin G_\omega\}$$
$$\cup \left\{t : \neg\text{STOP and } i^* \in G_\omega \text{ and } C_{\delta/n}(T_{i^*}(t)) > \frac{\omega}{16}\right\}$$
$$\cup \left\{t : \neg\text{STOP and } i^* \in G_\omega \text{ and } C_{\delta/n}(T_{i^*}(t)) \leq \frac{\omega}{16} \text{ and } i_1(t) \in G^c_{\epsilon + \frac{\gamma}{2}}\right\}$$
$$\cup \left\{t : \neg\text{STOP and } i^* \in G_\omega \text{ and } C_{\delta/n}(T_{i^*}(t)) \leq \frac{\omega}{16} \text{ and } i_1(t) \in G_{\epsilon + \frac{\gamma}{2}} \text{ and } i_2(t) \in G_{\epsilon + \frac{\gamma}{2}}\right\}$$
$$\cup \left\{t : \neg\text{STOP and } i^* \in G_\omega \text{ and } C_{\delta/n}(T_{i^*}(t)) \leq \frac{\omega}{16} \text{ and } i_1(t) \in G_{\epsilon + \frac{\gamma}{2}} \text{ and } i_2(t) \in G^c_{\epsilon + \frac{\gamma}{2}}\right\}.$$

Hence, by a union bound and plugging in the results of the above steps,

$$|\{t : \neg\text{STOP}\}| \leq$$
$$|\{t : \neg\text{STOP and } i^* \notin G_\omega\}|$$
$$+ \left|\left\{t : \neg\text{STOP and } i^* \in G_\omega \text{ and } \exists i \in G_\omega : C_{\delta/n}(T_{i^*}(t)) > \frac{\omega}{16}\right\}\right|$$
$$+ \left|\left\{t : \neg\text{STOP and } i^* \in G_\omega \text{ and } C_{\delta/n}(T_{i^*}(t)) \leq \frac{\omega}{16} \text{ and } i_1(t) \in G^c_{\epsilon + \frac{\gamma}{2}}\right\}\right|$$
$$+ \left|\left\{t : \neg\text{STOP and } i^* \in G_\omega \text{ and } C_{\delta/n}(T_{i^*}(t)) \leq \frac{\omega}{16} \text{ and } i_1(t) \in G_{\epsilon + \frac{\gamma}{2}} \text{ and } i_2(t) \in G_{\epsilon + \frac{\gamma}{2}}\right\}\right|$$
$$+ \left|\left\{t : \neg\text{STOP and } i^* \in G_\omega \text{ and } C_{\delta/n}(T_{i^*}(t)) \leq \frac{\omega}{16} \text{ and } i_1(t) \in G_{\epsilon + \frac{\gamma}{2}} \text{ and } i_2(t) \in G^c_{\epsilon + \frac{\gamma}{2}}\right\}\right|$$
$$\leq \sum_{i \in G^c_\omega} \min\left\{h\left(\frac{\gamma}{2}, \frac{\delta}{n}\right), \min\left[h\left(\frac{\Delta_i}{2}, \frac{\delta}{n}\right), h\left(\frac{\min(\alpha_\epsilon, \beta_\epsilon)}{2}, \frac{\delta}{n}\right)\right]\right\}$$
$$+ \sum_{i \in G_\omega} \min\left\{h\left(\frac{\gamma}{16}, \frac{\delta}{n}\right), \min\left[h\left(\frac{\Delta_i}{16}, \frac{\delta}{n}\right), h\left(\frac{\min(\alpha_\epsilon, \beta_\epsilon)}{16}, \frac{\delta}{n}\right)\right]\right\}$$
$$+ \sum_{i \in G^c_{\epsilon + \frac{\gamma}{2}}} \min\left[h\left(\frac{\Delta_i - \epsilon}{8}, \frac{\delta}{n}\right), h\left(\frac{\gamma}{8}, \frac{\delta}{n}\right)\right]$$
$$+ \sum_{i \in G_{\epsilon + \frac{\gamma}{2}}} \min\left[h\left(\frac{\epsilon - \Delta_i}{8}, \frac{\delta}{n}\right), h\left(\frac{\gamma}{8}, \frac{\delta}{n}\right)\right]$$
$$+ \sum_{i=1}^{n} \min\left[h\left(\frac{\Delta_i - \epsilon}{8}, \frac{\delta}{n}\right), h\left(\frac{\gamma}{8}, \frac{\delta}{n}\right)\right]$$
$$\overset{(\epsilon \leq 1/2)}{\leq} \sum_{i=1}^{n} \min\left\{h\left(\frac{\gamma}{16}, \frac{\delta}{n}\right), \min\left[h\left(\frac{\Delta_i}{16}, \frac{\delta}{n}\right), h\left(\frac{\min(\alpha_\epsilon, \beta_\epsilon)}{16}, \frac{\delta}{n}\right)\right]\right\}$$

$$+ 2 \sum_{i=1}^{n} \min \left[ h \left( \frac{\Delta_i - \epsilon}{8}, \frac{\delta}{n} \right), h \left( \frac{\gamma}{8}, \frac{\delta}{n} \right) \right]$$

$$\leq 4 \sum_{i=1}^{n} \min \left\{ \max \left\{ h \left( \frac{\Delta_i - \epsilon}{16}, \frac{\delta}{n} \right), \min \left[ h \left( \frac{\Delta_i}{16}, \frac{\delta}{n} \right), h \left( \frac{\min(\alpha_\epsilon, \beta_\epsilon)}{16}, \frac{\delta}{n} \right) \right] \right\}, \right.$$

$$\left. h \left( \frac{\gamma}{16}, \frac{\delta}{n} \right) \right\}$$

Next, by Lemma F.3, we may bound the minimum of $h(\cdot, \cdot)$ functions.

$$4 \sum_{i=1}^{n} \min \left\{ \max \left\{ h \left( \frac{\Delta_i - \epsilon}{16}, \frac{\delta}{n} \right), \min \left[ h \left( \frac{\Delta_i}{16}, \frac{\delta}{n} \right), h \left( \frac{\min(\alpha_\epsilon, \beta_\epsilon)}{16}, \frac{\delta}{n} \right) \right] \right\}, \right.$$

$$\left. h \left( \frac{\gamma}{16}, \frac{\delta}{n} \right) \right\}$$

$$= 4 \sum_{i=1}^{n} \min \left\{ \max \left\{ h \left( \frac{\Delta_i - \epsilon}{16}, \frac{\delta}{n} \right), \right. \right.$$

$$\left. \min \left[ h \left( \frac{\Delta_i}{16}, \frac{\delta}{n} \right), \max \left[ h \left( \frac{\alpha_\epsilon}{16}, \frac{\delta}{n} \right), h \left( \frac{\beta_\epsilon}{16}, \frac{\delta}{n} \right) \right] \right] \right\},$$

$$\left. h \left( \frac{\gamma}{16}, \frac{\delta}{n} \right) \right\}$$

$$\leq 4 \sum_{i=1}^{n} \min \left\{ \max \left\{ h \left( \frac{\Delta_i - \epsilon}{16}, \frac{\delta}{n} \right), \right. \right.$$

$$\left. \max \left[ h \left( \frac{\Delta_i + \alpha_\epsilon}{32}, \frac{\delta}{n} \right), h \left( \frac{\Delta_i + \beta_\epsilon}{32}, \frac{\delta}{n} \right) \right] \right\},$$

$$\left. h \left( \frac{\gamma}{16}, \frac{\delta}{n} \right) \right\}$$

$$= 4 \sum_{i=1}^{n} \min \left\{ \max \left\{ h \left( \frac{\Delta_i - \epsilon}{16}, \frac{\delta}{n} \right), h \left( \frac{\Delta_i + \alpha_\epsilon}{32}, \frac{\delta}{n} \right), h \left( \frac{\Delta_i + \beta_\epsilon}{32}, \frac{\delta}{n} \right) \right\}, \right.$$

$$\left. h \left( \frac{\gamma}{16}, \frac{\delta}{n} \right) \right\}$$

Finally, we use Lemma F.2 to bound the function $h(\cdot, \cdot)$. Since $\delta \leq 1/2$, $\delta/n \leq 2e^{-e/2}$. Further, $|\epsilon - \Delta_i| \leq 8$ for all $i$ and $\epsilon \leq 1/2$ implies that $\frac{1}{8}|\epsilon - \Delta_i| \leq 2$ and $\frac{1}{8} \min(\alpha_\epsilon, \beta_\epsilon) \leq 2$. $\Delta_i \leq 16$ for all $i$, gives $0.125\Delta_i \leq 2$. Lastly, $\gamma \leq 16$ implies that $\frac{\gamma}{8} \leq 2$. Therefore,

$$4 \sum_{i=1}^{n} \min \left\{ \max \left\{ h \left( \frac{\Delta_i - \epsilon}{16}, \frac{\delta}{n} \right), h \left( \frac{\Delta_i + \alpha_\epsilon}{32}, \frac{\delta}{n} \right), h \left( \frac{\Delta_i + \beta_\epsilon}{32}, \frac{\delta}{n} \right) \right\}, \right.$$

$$\left. h \left( \frac{\gamma}{16}, \frac{\delta}{n} \right) \right\}$$

$$\leq 4 \sum_{i=1}^{n} \min \left\{ \max \left\{ \frac{1024}{(\epsilon - \Delta_i)^2} \log \left( \frac{2n}{\delta} \log_2 \left( \frac{3072n}{\delta(\epsilon - \Delta_i)^2} \right) \right), \right. \right.$$

$$\frac{4096}{(\Delta_i + \alpha_\epsilon)^2} \log \left( \frac{2n}{\delta} \log_2 \left( \frac{12288n}{\delta(\Delta_i + \alpha_\epsilon)^2} \right) \right),$$

$$\left. \frac{4096}{(\Delta_i + \beta_\epsilon)^2} \log \left( \frac{2n}{\delta} \log_2 \left( \frac{12288n}{\delta(\Delta_i + \beta_\epsilon)^2} \right) \right) \right\},$$

$$\left. \frac{1}{\gamma^2} \log \left( \frac{2n}{\delta} \log_2 \left( \frac{3072n}{\delta\gamma^2} \right) \right) \right\}$$

$$= 4\sum_{i=1}^{n} \min\left\{ \max\left\{ \frac{1024}{(\mu_1 - \epsilon - \mu_i)^2} \log\left( \frac{2n}{\delta} \log_2\left( \frac{3072n}{\delta(\mu_1 - \epsilon - \mu_i)^2} \right) \right), \right. \right.$$

$$\frac{4096}{(\mu_1 + \alpha_\epsilon - \mu_i)^2} \log\left( \frac{2n}{\delta} \log_2\left( \frac{12288n}{\delta(\mu_1 + \alpha_\epsilon - \mu_i)^2} \right) \right),$$

$$\left. \frac{4096}{(\mu_1 + \beta_\epsilon - \mu_i)^2} \log\left( \frac{2n}{\delta} \log_2\left( \frac{12288n}{\delta(\mu_1 + \beta_\epsilon - \mu_i)^2} \right) \right) \right\},$$

$$\left. \frac{1}{\gamma^2} \log\left( \frac{2n}{\delta} \log_2\left( \frac{3072n}{\delta\gamma^2} \right) \right) \right\}.$$

The above bounds the number of rounds $T$. Therefore, the total number of samples is at most $3T$. $\quad\square$

## B.2 Optimism with multiplicative $\gamma$

**Theorem B.2.** *Fix $\epsilon \in (0, 1/2]$, $0 < \delta \leq 1/2$, $\gamma \in [0, \min(16/\mu_1, 1/2)]$ and an instance $\nu$ such that $\max(\Delta_i, |\epsilon\mu_1 - \Delta_i|) \leq 8$ for all $i$. In the case that $M_\epsilon = [n]$, let $\tilde{\alpha}_\epsilon = \min(\tilde{\alpha}_\epsilon, \tilde{\beta}_\epsilon)$. With probability at least $1 - \delta$, $(\mathrm{ST})^2$ correctly returns a set $G$ such that $M_\epsilon \subset G \subset M_{\epsilon+\gamma}$ in at most*

$$12\sum_{i=1}^{n} \min\left\{ \max\left\{ \frac{1024}{((1-\epsilon)\mu_1 - \mu_i)^2} \log\left( \frac{2n}{\delta} \log_2\left( \frac{3072n}{\delta((1-\epsilon)\mu_1 - \mu_i)^2} \right) \right), \right. \right.$$

$$\frac{4096}{(\mu_1 + \frac{\tilde{\alpha}_\epsilon}{1-\epsilon} - \mu_i)^2} \log\left( \frac{2n}{\delta} \log_2\left( \frac{12288n}{\delta(\mu_1 + \frac{\tilde{\alpha}_\epsilon}{1-\epsilon})^2} \right) \right),$$

$$\left. \frac{4096}{(\mu_1 + \frac{\tilde{\beta}_\epsilon}{1-\epsilon} - \mu_i)^2} \log\left( \frac{2n}{\delta} \log_2\left( \frac{12288n}{\delta(\mu_1 + \frac{\tilde{\beta}_\epsilon}{1-\epsilon} - \mu_i)^2} \right) \right) \right\}, \right.$$

$$\left. \frac{1024}{\gamma^2\mu_1^2} \log\left( \frac{2n}{\delta} \log_2\left( \frac{3072n}{\delta\gamma^2\mu_1^2} \right) \right) \right\}$$

*samples.*

*Proof.* Throughout the proof, recall that $\Delta_i = \mu_1 - \mu_i$ for all $i$, $\tilde{\alpha}_\epsilon = \min_{i \in M_\epsilon} \mu_i - (1-\epsilon)\mu_1$, and $\tilde{\beta}_\epsilon = \min_{i \in M_\epsilon^c} (1-\epsilon)\mu_1 - \mu_i$. Additionally, at any time $t$, we will take $T_j(t)$ to denote the number of samples of arm $j$ up to time $t$.

Define the event

$$\mathcal{E} = \left\{ \bigcap_{i \in [n]} \bigcap_{t \in \mathbb{N}} |\hat{\mu}_i(t) - \mu_i| \leq C_{\delta/n}(t) \right\}.$$

Using standard anytime confidence bound results, and recalling that that $C_\delta(t) := \sqrt{\frac{4\log(\log_2(2t)/\delta)}{t}}$, we have

$$\mathbb{P}(\mathcal{E}^c) = \mathbb{P}\left( \bigcup_{i \in [n]} \bigcup_{t \in \mathbb{N}} |\hat{\mu}_i - \mu_i| > C_{\delta/n}(t) \right)$$

$$\leq \sum_{i=1}^{n} \mathbb{P}\left( \bigcup_{t \in \mathbb{N}} |\hat{\mu}_i - \mu_i| > C_{\delta/n}(t) \right) \leq \sum_{i=1}^{n} \frac{\delta}{n} = \delta$$

Hence, $\mathbb{P}(\mathcal{E}) \geq 1 - \delta$. Throughout, we will make use of a function $h(x, \delta)$ such that if $t \geq h(x, \delta)$, then $C_\delta(t) \leq |x|$. We bound $h(\cdot, \cdot)$ in Lemma F.2. $h(\cdot, \cdot)$ is assumed to decrease monotonically in both arguments and is symmetric in its first argument.

### B.2.1 Step 0: Correctness

We begin by showing that on $\mathcal{E}$, if $(\mathrm{ST})^2$ terminates, it returns a set $G$ such that $M_\epsilon \subset G \subset M_{(\epsilon+\gamma)}$. Since $\mathbb{P}(\mathcal{E}) \geq 1 - \delta$, this implies that $(\mathrm{ST})^2$ is correct with high probability.

**Claim 0:** On Event $\mathcal{E}$, at all times $t$, $U_t \geq (1 - \epsilon - \gamma)\mu_1$.

**Proof.**

$$U_t = (1 - \epsilon - \gamma)(\max_j \hat{\mu}_j(T_j(t)) + C_{\delta/n}(T_j(t))) \geq (1 - \epsilon - \gamma)(\hat{\mu}_1(T_1(t)) + C_{\delta/n}(T_1(t)))$$

$$\overset{\mathcal{E}}{\geq} (1 - \epsilon - \gamma)\mu_1$$

$\square$

**Claim 1:** On Event $\mathcal{E}$, at all times $t$, $L_t \leq (1 - \epsilon)\mu_1$.

**Proof.**

$$L_t = (1 - \epsilon)\left(\max_j \hat{\mu}_j(T_j(t)) - C_{\delta/n}(T_j(t))\right) \overset{\mathcal{E}}{\leq} (1 - \epsilon)\max_j \mu_j = (1 - \epsilon)\mu_1$$

$\square$

**Claim 2:** On event $\mathcal{E}$, if there is a time $t$ such that $\hat{\mu}_i(T_i(t)) - C_{\delta/n}(T_i(t)) > U_t$, then $i \in M_{\epsilon+\gamma}$.

**Proof.** Assume for some $t$, $\hat{\mu}_i(T_i(t)) - C_{\delta/n}(T_i(t)) > U_t$. Then

$$\mu_i \overset{\mathcal{E}}{\geq} \hat{\mu}_i(T_i(t)) - C_{\delta/n}(T_i(t)) \geq U_t \overset{\text{Claim 0}}{\geq} (1 - \epsilon - \gamma)\mu_1$$

which implies $i \in M_{\epsilon+\gamma}$ $\square$

**Claim 3:** On event $\mathcal{E}$, if there is a time $t$ such that $\hat{\mu}_i(T_i(t)) + C_{\delta/n}(T_i(t)) < L_t$, then $i \in M_\epsilon^c$.

**Proof.** Assume that is a $t$ for which $\hat{\mu}_i(T_i(t)) + C_{\delta/n}(T_i(t)) < L_t$. Then

$$\mu_i \overset{\mathcal{E}}{\leq} \hat{\mu}_i(T_i(t)) + C_{\delta/n}(T_i(t)) \leq L_t \overset{\text{Claim 1}}{\leq} (1 - \epsilon)\mu_1$$

which implies $i \in M_\epsilon^c$. $\square$

$(\texttt{ST})^2$ terminates at any time $t$ such that simultaneously for all arms $i$, either $\hat{\mu}_i(T_i(t)) + C_{\delta/n}(T_i(t)) > U_t$ or $\hat{\mu}_i(T_i(t)) - C_{\delta/n}(T_i(t)) < L_t$. On $\mathcal{E}$, by Claim 3, $M_\epsilon \subset \{i : \hat{\mu}_i(T_i(t)) + C_{\delta/n}(T_i(t)) > U_t\}$. On $\mathcal{E}$, by Claim 2, $\{i : \hat{\mu}_i(T_i(t)) + C_{\delta/n}(T_i(t)) > U_t\} \subset M_{\epsilon+\gamma}$. Hence, on the event $\mathcal{E}$. $(\texttt{ST})^2$ returns a set $G$ such that $M_\epsilon \subset G \subset M_{\epsilon+\gamma}$.

### B.2.2 Step 1: Complexity of estimating the threshold, $(1 - \epsilon)\mu_1$

Let STOP denote the termination event that for all arms $i$, either $\hat{\mu}_i(T_i(t)) + C_{\delta/n}(T_i(t)) > U_t$ or $\hat{\mu}_i(T_i(t)) - C_{\delta/n}(T_i(t)) < L_t$. Let $\omega$ denote the quantity

$$\omega := \max\{\gamma\mu_1, \min(\tilde{\alpha}_\epsilon, \tilde{\beta}_\epsilon)\}.$$

Let $T$ denote the random variable of the total number of rounds before $(\texttt{ST})^2$ terminates. At most 3 samples are drawn in any round. Hence, the total sample complexity is bounded by $3T$. We may write $T$ as

$$T \equiv |\{t : \neg\text{STOP}\}| = |\{t : \neg\text{STOP and } i^* \notin M_{\omega/\mu_1}\}| + |\{t : \neg\text{STOP and } i^* \in M_{\omega/\mu_1}\}|$$

Next, we bound the first event in this decomposition.

**Claim 0:** On $\mathcal{E}$, $|\{t : \neg\text{STOP and } i^* \notin M_{\omega/\mu_1}\}| \leq \sum_{i \in M_{\omega/\mu_1}^c} \min\left\{h\left(\frac{\gamma\mu_1}{2}, \frac{\delta}{n}\right), \min\left[h\left(\frac{\Delta_i}{2}, \frac{\delta}{n}\right), h\left(\frac{\min(\tilde{\alpha}_\epsilon, \tilde{\beta}_\epsilon)}{2}, \frac{\delta}{n}\right)\right]\right\}$.

**Proof.** For each $i \in M_{\omega/\mu_1}^c$, $\mu_i + 2C_{\delta/n}(T_i(t)) < \mu_1$, true when $T_i(t) > h\left(\Delta_i/2, \frac{\delta}{n}\right)$ implies that

$$\hat{\mu}_i(T_i(t)) + C_{\delta/n}(T_i(t)) \overset{\mathcal{E}}{\leq} \mu_i + 2C_{\delta/n}(T_i(t)) < \mu_1 \overset{\mathcal{E}}{\leq} \hat{\mu}_1(T_1(t)) + C_{\delta/n}(T_1(t))$$

which implies that $i \neq i^*$. Additionally, since $i \in M_{\omega/\mu_1}^c$ by assumption, we have that $(1 - \omega/\mu_1)\mu_1 - \mu_i \geq 0$, which reduces to $\Delta_i \geq \omega$. Since $\omega = \max(\gamma\mu_1, \min(\tilde{\alpha}_\epsilon, \tilde{\beta}_\epsilon))$, it is likewise true that

$$h\left(\frac{\Delta_i}{2}, \frac{\delta}{n}\right) = \min\left[h\left(\frac{\gamma\mu_1}{2}, \frac{\delta}{n}\right), \min\left\{h\left(\frac{\Delta_i}{2}, \frac{\delta}{n}\right), h\left(\frac{\min(\tilde{\alpha}_\epsilon, \tilde{\beta}_\epsilon)}{2}, \frac{\delta}{n}\right)\right\}\right].$$

Summing over all $i \in M^c_{\omega/\mu_1}$ achieves the result. □

We may decompose the event $\{t : \neg\text{STOP and } i^* \in M_{\omega/\mu_1}\}$ as

$$\left\{ t : \neg\text{STOP and } i^* \in M_{\omega/\mu_1} \text{ and } \exists i \in M_{\omega/\mu_1} : C_{\delta/n}(T_{i^*}(t)) > \frac{\omega}{16(1-\epsilon)} \right\}$$

$$\cup \left\{ t : \neg\text{STOP and } i^* \in M_{\omega/\mu_1} \text{ and } C_{\delta/n}(T_{i^*}(t)) \le \frac{\omega}{16(1-\epsilon)} \right\}$$

**Claim 1:** $\left| \left\{ t : \neg\text{STOP and } i^* \in M_{\omega/\mu_1} \text{ and } C_{\delta/n}(T_{i^*}(t)) \ge \frac{\omega}{16(1-\epsilon)} \right\} \right| \le$
$\sum_{i \in M_{\omega/\mu_1}} \min \left\{ h\left(\frac{\gamma\mu_1}{16}, \frac{\delta}{n}\right), \min\left[ h\left(\frac{\Delta_i}{16}, \frac{\delta}{n}\right), h\left(\frac{\min(\tilde\alpha_\epsilon, \tilde\beta_\epsilon)}{16(1-\epsilon)}, \frac{\delta}{n}\right) \right] \right\}$

**Proof.** $C_{\delta/n}(T_i(t)) \le \frac{\omega}{16(1-\epsilon)}$ is true when $T_i(t) \ge h\left(\frac{\omega}{16(1-\epsilon)}, \frac{\delta}{n}\right)$. Since $i^* \in M_{\omega/\mu_1}$, $\mu_i - (1 - \omega/\mu_1)\mu_1 \ge 0$, which implies $\Delta_i \le \omega$. By definition, $\omega = \min(\gamma\mu_1, \min(\tilde\alpha_\epsilon, \tilde\beta_\epsilon))$. Hence, by monotonicity of $h(\cdot, \cdot)$,

$$h\left(\frac{\omega}{16(1-\epsilon)}, \frac{\delta}{n}\right) = \min\left[ h\left(\frac{\Delta_i}{16(1-\epsilon)}, \frac{\delta}{n}\right), h\left(\frac{\omega}{16(1-\epsilon)}, \frac{\delta}{n}\right) \right]$$

$$= \min\left\{ h\left(\frac{\gamma\mu_1}{16(1-\epsilon)}, \frac{\delta}{n}\right), \min\left[ h\left(\frac{\Delta_i}{16(1-\epsilon)}, \frac{\delta}{n}\right), h\left(\frac{\min(\tilde\alpha_\epsilon, \tilde\beta_\epsilon)}{16(1-\epsilon)}, \frac{\delta}{n}\right) \right] \right\}$$

$$\le \min\left\{ h\left(\frac{\gamma\mu_1}{16}, \frac{\delta}{n}\right), \min\left[ h\left(\frac{\Delta_i}{16}, \frac{\delta}{n}\right), h\left(\frac{\min(\tilde\alpha_\epsilon, \tilde\beta_\epsilon)}{16(1-\epsilon)}, \frac{\delta}{n}\right) \right] \right\}.$$

Summing over all $i \in M_{\omega/\mu_1}$ achieves the desired result. □

### B.2.3 Step 2: Controlling "crossing" events

Recall that we sample $i_1(t) \in \widehat{G}$ and $i_2(t) \in \widehat{G}^c$. In this section, we control the number of times that $i_1(t) \in M^c_{\epsilon + \frac{\gamma}{2}}$ and $i_2(t) \in M_{\epsilon + \frac{\gamma}{2}}$.

To do so, we first decompose the set $\left\{ t : \neg\text{STOP and } i^* \in M_{\omega/\mu_1} \text{ and } C_{\delta/n}(T_{i^*}(t)) \le \frac{\omega}{16(1-\epsilon)} \right\}$ as

$$\left\{ t : \neg\text{STOP and } i^* \in M_{\omega/\mu_1} \text{ and } C_{\delta/n}(T_{i^*}(t)) \le \frac{\omega}{16(1-\epsilon)} \text{ and } i_1(t) \in M^c_{\epsilon + \frac{\gamma}{2}} \right\}$$

$$\cup \left\{ t : \neg\text{STOP and } i^* \in M_{\omega/\mu_1} \text{ and } C_{\delta/n}(T_{i^*}(t)) \le \frac{\omega}{16(1-\epsilon)} \text{ and } i_1(t) \in M_{\epsilon + \frac{\gamma}{2}} \right\}$$

**Claim 0:** $\left| \left\{ t : \neg\text{STOP and } i^* \in M_{\omega/\mu_1} \text{ and } C_{\delta/n}(T_{i^*}(t)) \le \frac{\omega}{16(1-\epsilon)} \text{ and } i_1(t) \in M^c_{\epsilon + \frac{\gamma}{2}} \right\} \right| \le$
$\sum_{i \in M^c_{\epsilon + \frac{\gamma}{2}}} \min\left[ h\left(\frac{\Delta_i - \epsilon\mu_1}{16}, \frac{\delta}{n}\right), h\left(\frac{\gamma\mu_1}{16}, \frac{\delta}{n}\right) \right]$.

**Proof.** Recall that $\widehat{G}$ is the set of all arms whose empirical means exceed $(1 - \epsilon)\max_i \hat\mu_i(T_i(t))$, and $i_1(t) \in \widehat{G}$ by definition. Note that $(1 - \epsilon)\max_i \hat\mu_i(T_i(t)) > (1 - \epsilon)\left(\max_i \hat\mu_i(T_i(t)) - C_{\delta/n}(T_i(t))\right) = L_t$. Hence, if an arm's upper bound is below $L_t$, then the arm cannot be in $\widehat{G}$ and thus not be $i_1(t)$. By the above event, $C_{\delta/n}(T_{i^*}(t)) \le \frac{\omega}{16(1-\epsilon)}$. Therefore,

$$\mu_{i^*} + \frac{\omega}{8(1-\epsilon)} \ge \mu_{i^*} + 2C_{\delta/n}(T_{i^*}(t)) \overset{\mathcal{E}}{\ge} \hat\mu_{i^*}(T_{i^*}(t)) + C_{\delta/n}(T_{i^*}(t)) \ge \hat\mu_1(T_1(t)) + C_{\delta/n}(T_1(t))$$

$$\overset{\mathcal{E}}{\ge} \mu_1.$$

Hence, $\mu_{i^*} \ge \mu_1 - \frac{\omega}{8(1-\epsilon)}$. Rearranging this, we see that $\mu_{i^*} - \left(1 - \frac{\omega}{8\mu_1(1-\epsilon)}\right)\mu_1 \ge 0$ which implies that $i^* \in M_{\frac{\omega}{8\mu_1(1-\epsilon)}}$. Hence,

$$L_t = (1 - \epsilon)\left(\max_i \hat\mu_i(T_i(t)) - C_{\delta/n}(T_i(t))\right)(1 - \epsilon)\left(\hat\mu_{i^*}(T_{i^*}(t)) - C_{\delta/n}(T_{i^*}(t))\right)$$

$$\overset{\mathcal{E}}{\geq} (1-\epsilon)\left(\mu_{i^*} - 2C_{\delta/n}(T_{i^*}(t))\right)$$

$$\geq (1-\epsilon)\left(\mu_{i^*} - \frac{\omega}{8(1-\epsilon)}\right)$$

$$\geq (1-\epsilon)\left(\mu_1 - \frac{\omega}{4(1-\epsilon)}\right)$$

Next, we bound the number of times an arm $i \in M_{\epsilon+\frac{\gamma}{2}}^c$ is sampled before its upper bound is below $(1-\epsilon)\left(\mu_1 - \frac{\omega}{4(1-\epsilon)}\right)$. Note that $C_{\delta/n}(T_i(t)) < \frac{1}{2}\left((1-\epsilon)\left(\mu_1 - \frac{\omega}{4(1-\epsilon)}\right) - \mu_i\right)$, true when $T_i(t) > h\left(\frac{1}{2}\left((1-\epsilon)\left(\mu_1 - \frac{\omega}{4(1-\epsilon)}\right) - \mu_i\right), \frac{\delta}{n}\right)$ implies that

$$\hat{\mu}_i(T_i(t)) + C_{\delta/n}(T_i(t)) \overset{\mathcal{E}}{\leq} \mu_i + 2C_{\delta/n}(T_i(t)) < (1-\epsilon)\left(\mu_1 - \frac{\omega}{4(1-\epsilon)}\right) \leq L_t.$$

Finally, we turn our attention to the difference $(1-\epsilon)\left(\mu_1 - \frac{\omega}{4(1-\epsilon)}\right) - \mu_i$. Recall that $\omega = \max(\gamma\mu_1, \min(\tilde{\alpha}_\epsilon, \tilde{\beta}_\epsilon))$.

$$(1-\epsilon)\left(\mu_1 - \frac{\omega}{4(1-\epsilon)}\right) - \mu_i = (1-\epsilon)\mu_1 - \mu_i - \frac{1}{4}\omega$$

$$= (1-\epsilon)\mu_1 - \mu_i - \frac{1}{4}\max(\gamma\mu_1, \min(\tilde{\alpha}_\epsilon, \tilde{\beta}_\epsilon)).$$

By definition, $\tilde{\beta}_\epsilon = \min_{i \in M_\epsilon^c}(1-\epsilon)\mu_1 - \mu_i$. Hence, $\min(\tilde{\alpha}_\epsilon, \tilde{\beta}_\epsilon) \leq (1-\epsilon)\mu_1 - \mu_i$ for all $i \in M_{\epsilon+\frac{\gamma}{2}}^c$. Similarly, since $i \in M_{\epsilon+\frac{\gamma}{2}}^c$ by assumption, $(1-\epsilon-\frac{\gamma}{2})\mu_1 - \mu_i \geq 0$, which rearranges to $\frac{\gamma\mu_1}{2} \leq (1-\epsilon)\mu_1 - \mu_i$. Therefore,

$$(1-\epsilon)\mu_1 - \mu_i - \frac{1}{4}\max(\gamma\mu_1, \min(\tilde{\alpha}_\epsilon, \tilde{\beta}_\epsilon)) \geq \frac{1}{2}\left((1-\epsilon)\mu_1 - \mu_i\right) = \frac{\Delta_i - \epsilon\mu_1}{2}.$$

Hence, by monotonicity of $h(\cdot, \cdot)$,

$$h\left(\frac{1}{2}\left((1-\epsilon)\left(\mu_1 - \frac{\omega}{4(1-\epsilon)}\right) - \mu_i\right), \frac{\delta}{n}\right) \leq h\left(\frac{\Delta_i - \epsilon\mu_1}{4}, \frac{\delta}{n}\right).$$

Lastly, as above, since $i \in M_{\epsilon+\frac{\gamma}{2}}^c$, we have that $\Delta_i - \epsilon\mu_1 = (1-\epsilon)\mu_1 - \mu_i \geq \frac{1}{2}\gamma\mu_1$. Hence,

$$h\left(\frac{\Delta_i - \epsilon\mu_1}{4}, \frac{\delta}{n}\right) \leq \min\left[h\left(\frac{\Delta_i - \epsilon\mu_1}{8}, \frac{\delta}{n}\right), h\left(\frac{\gamma\mu_1}{8}, \frac{\delta}{n}\right)\right].$$

Putting this together, if $T_i(t) \geq \min\left[h\left(\frac{\Delta_i - \epsilon\mu_1}{8}, \frac{\delta}{n}\right), h\left(\frac{\gamma\mu_1}{8}, \frac{\delta}{n}\right)\right]$, then $i \neq i_1(t)$ for all $i \in M_{\epsilon+\frac{\gamma}{2}}^c$. Summing over all such $i$ bounds the size of set stated in the claim. $\square$

We decompose the remaining event

$$\left\{t : \neg\text{STOP and } i^* \in M_{\omega/\mu_1} \text{ and } C_{\delta/n}(T_{i^*}(t)) \leq \frac{\omega}{16(1-\epsilon)} \text{ and } i_1(t) \in M_{\epsilon+\frac{\gamma}{2}}\right\}$$

as

$$\left\{t : \neg\text{STOP and } i^* \in M_{\omega/\mu_1} \text{ and } C_{\delta/n}(T_{i^*}(t)) \leq \frac{\omega}{16(1-\epsilon)} \text{ and } i_1(t) \in M_{\epsilon+\frac{\gamma}{2}}\right.$$
$$\left. \text{and } i_2(t) \in M_{\epsilon+\frac{\gamma}{2}}\right\}$$
$$\cup \left\{t : \neg\text{STOP and } i^* \in M_{\omega/\mu_1} \text{ and } C_{\delta/n}(T_{i^*}(t)) \leq \frac{\omega}{16(1-\epsilon)} \text{ and } i_1(t) \in M_{\epsilon+\frac{\gamma}{2}}\right.$$
$$\left. \text{and } i_2(t) \in M_{\epsilon+\frac{\gamma}{2}}^c\right\}.$$

We proceed by bounding the cardinality of the first set.

**Claim 1:**

$$\left| \left\{ t : \neg\text{STOP and } i^* \in M_{\omega/\mu_1} \text{ and } C_{\delta/n}(T_{i^*}(t)) \le \frac{\omega}{16(1-\epsilon)} \text{ and } i_1(t) \in M_{\epsilon+\frac{\gamma}{2}} \right.\right.$$

$$\left.\left. \text{and } i_2(t) \in M_{\epsilon+\frac{\gamma}{2}} \right\} \right|$$

$$\le \sum_{i \in M_{\epsilon+\frac{\gamma}{2}}} \min\left[ h\left( \frac{\epsilon\mu_1 - \Delta_i}{8}, \frac{\delta}{n} \right), h\left( \frac{\gamma\mu_1}{8}, \frac{\delta}{n} \right) \right]$$

**Proof.** Recall that $K = \{ i : \hat{\mu}_i(T_i(t)) + C_{\delta/n}(T_i(t)) < L_t \text{ or } \hat{\mu}_i(T_i(t)) - C_{\delta/n}(T_i(t)) > U_t \}$ is the set of known arms and $i_2$ is sampled from $\widehat{G}^c \backslash K$. Hence, if an arm's lower bound exceeds $U_t$, it must be in $K$ and therefore cannot be $i_2$. Recall that $i^*(t) = \arg\max \hat{\mu}_i(T_i(t)) + C_{\delta/n}(T_i(t))$. By the above event, $i^*(t) \in M_{\omega/\mu_1}$ and $C_{\delta/n}(T_{i^*}(t)) \le \frac{\omega}{16(1-\epsilon)}$. Hence,

$$U_t = (1 - \epsilon - \gamma)\left( \max_i \hat{\mu}_i(T_i(t)) + C_{\delta/n}(T_i(t)) \right)$$

$$= (1 - \epsilon - \gamma)\left( \hat{\mu}_{i^*(t)}(T_{i^*(t)}(t)) + C_{\delta/n}(T_{i^*(t)}(t)) \right)$$

$$\overset{\mathcal{E}}{\le} (1 - \epsilon - \gamma)\left( \mu_{i^*(t)} + 2C_{\delta/n}(T_{i^*(t)}(t)) \right)$$

$$\le (1 - \epsilon - \gamma)\left( \mu_{i^*(t)} + \frac{\omega}{8(1-\epsilon)} \right)$$

$$\le (1 - \epsilon - \gamma)\left( \mu_1 + \frac{\omega}{8(1-\epsilon)} \right)$$

Next, we bound the number of times an arm $i \in M_{\epsilon+\frac{\gamma}{2}}$ is sampled before its lower bound is above $(1-\epsilon-\gamma)\left(\mu_1 + \frac{\omega}{8(1-\epsilon)}\right)$. Note that $C_{\delta/n}(T_i(t)) < \frac{1}{2}\left( \mu_i - (1-\epsilon-\gamma)\left(\mu_1 + \frac{\omega}{8(1-\epsilon)}\right) \right)$, true when $T_i(t) > h\left( \frac{1}{2}\left( \mu_i - (1-\epsilon-\gamma)\left(\mu_1 + \frac{\omega}{8(1-\epsilon)}\right) \right), \frac{\delta}{n} \right)$ implies that

$$\hat{\mu}_i(T_i(t)) - C_{\delta/n}(T_i(t)) \overset{\mathcal{E}}{\ge} \mu_i - 2C_{\delta/n}(T_i(t)) > (1-\epsilon-\gamma)\left( \mu_1 + \frac{\omega}{8(1-\epsilon)} \right) \ge U_t.$$

Finally, we turn our attention to the difference $\mu_i - (1-\epsilon)\left(\mu_1 + \frac{\omega}{8}\right)$. Recall that $\omega = \max(\gamma\mu_1, \min(\tilde{\alpha}_\epsilon, \tilde{\beta}_\epsilon))$. Additionally, recall $\epsilon + \gamma \le 1$.

$$\mu_i - (1-\epsilon-\gamma)\left( \mu_1 + \frac{\omega}{8(1-\epsilon)} \right) = \mu_i - (1-\epsilon)\mu_1 + \gamma\mu_1 - \frac{1}{8}\left( \frac{1-\epsilon-\gamma}{1-\epsilon} \right)\omega$$

$$\ge \mu_i - (1-\epsilon)\mu_1 + \gamma\mu_1 - \frac{1}{8}\omega$$

**Case 1a, $\omega = \min(\tilde{\alpha}_\epsilon, \tilde{\beta}_\epsilon)$ and $i \in M_\epsilon$:**

By definition, $\tilde{\alpha}_\epsilon = \min_{i \in M_\epsilon} \mu_i - (1-\epsilon)\mu_1$. Hence, $\min(\tilde{\alpha}_\epsilon, \tilde{\beta}_\epsilon) \le \mu_i - (1-\epsilon)\mu_1$ for all $i \in M_\epsilon$. Therefore,

$$\mu_i - (1-\epsilon)\mu_1 + \gamma\mu_1 - \frac{1}{8}\omega = \mu_i - (1-\epsilon)\mu_1 + \gamma\mu_1 - \frac{1}{8}\min(\tilde{\alpha}_\epsilon, \tilde{\beta}_\epsilon)$$

$$\ge \max\left( \mu_i - (1-\epsilon)\mu_1 - \frac{1}{8}\min(\tilde{\alpha}_\epsilon, \tilde{\beta}_\epsilon), \gamma\mu_1 \right)$$

$$\ge \max\left( \frac{7}{8}(\mu_i - (1-\epsilon)\mu_1), \gamma\mu_1 \right)$$

**Case 1b, $\omega = \min(\tilde{\alpha}_\epsilon, \tilde{\beta}_\epsilon)$ and $i \in M_\epsilon^c \cap M_{\epsilon+\frac{\gamma}{2}}$**

Since $\omega = \max(\gamma\mu_1, \min(\tilde{\alpha}_\epsilon, \tilde{\beta}_\epsilon))$, if $\omega = \min(\tilde{\alpha}_\epsilon, \tilde{\beta}_\epsilon)$, then $\frac{1}{2}\gamma\mu_1 < \min(\tilde{\alpha}_\epsilon, \tilde{\beta}_\epsilon)$. Since $\min(\tilde{\alpha}_\epsilon, \tilde{\beta}_\epsilon) = \min|\mu_i - (1-\epsilon)\mu_1|$, the set $M_\epsilon^c \cap M_{\epsilon+\frac{\gamma}{2}}$ is empty and there is nothing to prove.

**Case 2a,** $\omega = \gamma\mu_1$ and $i \in M_\epsilon$

$$\mu_i - (1-\epsilon)\mu_1 + \gamma\mu_1 - \frac{1}{8}\omega = \mu_i - (1-\epsilon)\mu_1 + \frac{7}{8}\gamma\mu_1 \geq \max\left(\mu_i - (1-\epsilon)\mu_1, \frac{7}{8}\gamma\mu_1\right).$$

**Case 2b,** $\omega = \gamma\mu_1$ and $i \in M_\epsilon^c \cap M_{\epsilon+\frac{\gamma}{2}}$

For $i \in M_\epsilon^c \cap M_{\epsilon+\frac{\gamma}{2}}$, $\mu_i - (1-\epsilon-\frac{\gamma}{2})\mu_1 \geq 0$. Hence, $\mu_i - (1-\epsilon)\mu_1 \geq \frac{-\gamma\mu_1}{2}$. Therefore,

$$\mu_i - (1-\epsilon)\mu_1 + \gamma\mu_1 - \frac{1}{8}\omega = \mu_i - (1-\epsilon)\mu_1 + \frac{7}{8}\gamma\mu_1 \geq \frac{3}{8}\gamma\mu_1 \geq \max\left(\frac{1}{4}\gamma\mu_1, \frac{(1-\epsilon)\mu_1 - \mu_i}{4}\right).$$

Combining all cases, by monotonicity of $h(\cdot,\cdot)$ and symmetry in its first argument, we see that

$$h\left(\frac{1}{2}\left(\mu_i - (1-\epsilon-\gamma)\left(\mu_1 + \frac{\omega}{8(1-\epsilon)}\right)\right), \frac{\delta}{n}\right) \leq \min\left[h\left(\frac{\gamma\mu_1}{8}, \frac{\delta}{n}\right), h\left(\frac{\epsilon\mu_1 - \Delta_i}{8}, \frac{\delta}{n}\right)\right].$$

Putting this together, if $T_i(t) \geq \min\left[h\left(\frac{\epsilon\mu_1 - \Delta_i}{8}, \frac{\delta}{n}\right), h\left(\frac{\gamma\mu_1}{8}, \frac{\delta}{n}\right)\right]$, then $i \neq i_2(t)$ for all $i \in M_{\epsilon+\frac{\gamma}{2}}$. Summing over all such $i$ bounds the size of set stated in the claim. $\square$

### B.2.4 Step 3: Controlling the complexity until stopping occurs

In this step, we turn our attention to the final event to control:

$$\mathcal{S} := \left\{ t : \neg\text{STOP and } i^* \in M_{\omega/\mu_1} \text{ and } C_{\delta/n}(T_{i^*}(t)) \leq \frac{\omega}{16(1-\epsilon)} \right. \tag{7}$$
$$\left. \text{and } i_1(t) \in M_{\epsilon+\frac{\gamma}{2}} \text{ and } i_2(t) \in M_{\epsilon+\frac{\gamma}{2}}^c \right\}.$$

For brevity, we will refer to this set as $\mathcal{S}$ for this step. The objective will be to bound the time before each arms lower bound either clears $U_t$ or its upper bound clears $L_t$ which implies the stopping condition. To do so, we introduce, two events:

$$E_1(t) := \{\hat{\mu}_{i_1(t)}(T_{i_1(t)}(t)) - C_{\delta/n}(T_{i_1(t)}(t)) > U_t\} \tag{8}$$

and

$$E_2(t) := \{\hat{\mu}_{i_2(t)}(T_{i_2(t)}(t)) + C_{\delta/n}(T_{i_2(t)}(t)) < L_t\}. \tag{9}$$

If $E_1(t)$ is true, then $\hat{\mu}_i(T_i) - C_{\delta/n}(T_i(t)) > L_t$ for all $i \in \widehat{G}$. If $E_2(t)$ is true, then $\hat{\mu}_i(T_i) + C_{\delta/n}(T_i(t)) < U_t$ for all $i \in \widehat{G}^c$. Hence, by line 6 of $(\text{ST})^2$, if both $E_1(t)$ and $E_2(t)$ are true, then $(\text{ST})^2$ terminates.

**Claim 0:** $|\mathcal{S} \cap \{t : \neg E_1(t)\}| \leq \sum_{i \in M_{\epsilon+\frac{\gamma}{2}}} \min\left[h\left(\frac{\epsilon\mu_1 - \Delta_i}{4}, \frac{\delta}{n}\right), h\left(\frac{\gamma\mu_1}{4}, \frac{\delta}{n}\right)\right]$.

**Proof.** Recall that by the set $\mathcal{S}$, we have that $i_1(t) \in M_{\epsilon+\frac{\gamma}{2}}$. Furthermore, by the set $\mathcal{S}$, we have that $i^*(t) \in M_{\omega/\mu_1}$ and $C_{\delta/n}(T_{i^*}(t)) \leq \omega/16(1-\epsilon)$. Hence,

$$
\begin{aligned}
U_t &= (1-\epsilon-\gamma)\left(\max_i \hat{\mu}_i(T_i(t)) + C_{\delta/n}(T_i(t))\right) \\
&= (1-\epsilon-\gamma)\left(\hat{\mu}_{i^*(t)}(T_{i^*(t)}(t)) + C_{\delta/n}(T_{i^*(t)}(t))\right) \\
&\overset{\mathcal{E}}{\leq} (1-\epsilon-\gamma)\left(\mu_{i^*(t)} + 2C_{\delta/n}(T_{i^*(t)}(t))\right) \\
&\leq (1-\epsilon-\gamma)\left(\mu_{i^*(t)} + \frac{\omega}{8(1-\epsilon)}\right) \\
&\leq (1-\epsilon-\gamma)\left(\mu_1 + \frac{\omega}{8(1-\epsilon)}\right)
\end{aligned}
$$

If $C_{\delta/n}(T_i) \leq \frac{1}{2}\left(\mu_i - (1-\epsilon-\gamma)\left(\mu_1 + \frac{\omega}{8(1-\epsilon)}\right)\right)$, true when $T_i \geq h\left(\frac{1}{2}\left(\mu_i - (1-\epsilon-\gamma)\left(\mu_1 + \frac{\omega}{8(1-\epsilon)}\right)\right), \frac{\delta}{n}\right)$, then

$$\hat{\mu}_i(T_i) - C_{\delta/n}(T_i) \geq \mu_i - 2C_{\delta/n}(T_i) \geq (1-\epsilon-\gamma)\left(\mu_1 + \frac{\omega}{8(1-\epsilon)}\right) \geq U_t.$$

The remainder of the proof of this claim focuses on controlling the difference: $\mu_i - (1 - \epsilon - \gamma)\left(\mu_1 + \frac{\omega}{8(1-\epsilon)}\right)$ in the case that $\omega = \min(\tilde{\alpha}_\epsilon, \tilde{\beta}_\epsilon)$ and $\omega = \gamma\mu_1$. Recall that $\omega = \max(\gamma\mu_1, \min(\tilde{\alpha}_\epsilon, \tilde{\beta}_\epsilon))$. Hence, if any possible $i \in M_{\epsilon+\frac{\gamma}{2}}$ has received sufficiently many samples, since $i_1(t) \in M_{\epsilon+\frac{\gamma}{2}}$, this implies $E_1(t)$.

**Case 1a,** $\omega = \min(\tilde{\alpha}_\epsilon, \tilde{\beta}_\epsilon)$ **and** $i \in M_\epsilon$

We focus on the difference $\mu_i - (1 - \epsilon - \gamma)\left(\mu_1 + \frac{\omega}{8(1-\epsilon)}\right)$. Recall that $\epsilon + \gamma \leq 1$.

$$\mu_i - (1 - \epsilon - \gamma)\left(\mu_1 + \frac{\omega}{8(1-\epsilon)}\right) = \mu_i - (1 - \epsilon - \gamma)\left(\mu_1 + \frac{\min(\tilde{\alpha}_\epsilon, \tilde{\beta}_\epsilon)}{8(1-\epsilon)}\right)$$

$$= \mu_i - (1 - \epsilon)\mu_1 + \gamma\mu_1 - \frac{1}{8}\left(\frac{1 - \epsilon - \gamma}{1 - \epsilon}\right)\min(\tilde{\alpha}_\epsilon, \tilde{\beta}_\epsilon)$$

$$\overset{\gamma \geq 0 \text{ and } \epsilon + \gamma \leq 1}{\geq} \mu_i - (1 - \epsilon)\mu_1 - \frac{1}{8}\min(\tilde{\alpha}_\epsilon, \tilde{\beta}_\epsilon)$$

$$\geq \frac{1}{2}(\mu_i - (1 - \epsilon)\mu_1) = \frac{\epsilon\mu_1 - \Delta_i}{2}$$

where the final step follows since $\min(\tilde{\alpha}_\epsilon, \tilde{\beta}_\epsilon) \leq \tilde{\alpha}_\epsilon \leq \mu_i - (1 - \epsilon)\mu_1$ by definition for all $i \in M_\epsilon$. Then by monotonicity of $h(\cdot, \cdot)$,

$$h\left(\frac{1}{2}\left(\mu_i - (1 - \epsilon - \gamma)\left(\mu_1 + \frac{\omega}{8(1-\epsilon)}\right)\right), \frac{\delta}{n}\right) \leq h\left(\frac{\epsilon\mu_1 - \Delta_i}{4}, \frac{\delta}{n}\right).$$

Lastly, in this setting, $\gamma\mu_1 \leq \min(\tilde{\alpha}_\epsilon, \tilde{\beta}_\epsilon) \leq \epsilon\mu_1 - \Delta_i$ since $\omega = \min(\tilde{\alpha}_\epsilon, \tilde{\beta}_\epsilon)$. Hence, it is trivially true that

$$h\left(\frac{\epsilon\mu_1 - \Delta_i}{4}, \frac{\delta}{n}\right) = \min\left[h\left(\frac{\epsilon\mu_1 - \Delta_i}{4}, \frac{\delta}{n}\right), h\left(\frac{\gamma\mu_1}{4}, \frac{\delta}{n}\right)\right]$$

**Case 1b,** $\omega = \min(\tilde{\alpha}_\epsilon, \tilde{\beta}_\epsilon)$ **and** $i \in M_\epsilon^c \cap M_{\epsilon+\frac{\gamma}{2}}$

Since $\omega = \max(\gamma\mu_1, \min(\tilde{\alpha}_\epsilon, \tilde{\beta}_\epsilon))$, if $\omega = \min(\tilde{\alpha}_\epsilon, \tilde{\beta}_\epsilon)$, then $\frac{1}{2}\gamma\mu_1 < \min(\tilde{\alpha}_\epsilon, \tilde{\beta}_\epsilon)$. Since $\min(\tilde{\alpha}_\epsilon, \tilde{\beta}_\epsilon) = \min|\mu_i - (1 - \epsilon)\mu_1|$, the set $M_\epsilon^c \cap M_{\epsilon+\frac{\gamma}{2}}$ is empty and there is nothing to prove.

**Case 2a,** $\omega = \gamma\mu_1$ **and** $i \in M_\epsilon$

Next, we bound the difference $\mu_i - (1 - \epsilon - \gamma)\left(\mu_1 + \frac{\omega}{4(1-\epsilon)}\right)$.

$$\mu_i - (1 - \epsilon - \gamma)\left(\mu_1 + \frac{\omega}{8(1-\epsilon)}\right) = \mu_i - (1 - \epsilon)\mu_1 + \gamma\mu_1 - \frac{1}{8}\left(\frac{1 - \epsilon - \gamma}{1 - \epsilon}\right)\gamma\mu_1$$

$$\geq \mu_i - (1 - \epsilon)\mu_1 + \gamma\mu_1\left(1 - \frac{1}{8}\left(\frac{1 - \epsilon - \gamma}{1 - \epsilon}\right)\right)$$

Since $i \in M_\epsilon$, $\mu_i - (1 - \epsilon)\mu_1 \geq 0$. Using this and the fact that $\epsilon, \gamma \geq 0$ and $\epsilon + \gamma \leq 1$,

$$\mu_i - (1 - \epsilon)\mu_1 + \gamma\mu_1\left(1 - \frac{1}{8}\left(\frac{1 - \epsilon - \gamma}{1 - \epsilon}\right)\right) \geq \mu_i - (1 - \epsilon)\mu_1 + \frac{7}{8}\gamma\mu_1$$

$$\geq \max\left(\mu_i - (1 - \epsilon)\mu_1, \frac{7}{8}\gamma\mu_1\right)$$

$$\geq \frac{1}{2}\max\left(\epsilon\mu_1 - \Delta_i, \gamma\mu_1\right)$$

Therefore, we have that

$$h\left(\frac{1}{2}\left(\mu_i - (1 - \epsilon - \gamma)\left(\mu_1 + \frac{\omega}{8(1-\epsilon)}\right)\right), \frac{\delta}{n}\right) \leq h\left(\frac{\epsilon\mu_1 - \Delta_i}{4}, \frac{\delta}{n}\right)$$

and

$$h\left(\frac{1}{2}\left(\mu_i - (1 - \epsilon - \gamma)\left(\mu_1 + \frac{\omega}{8(1-\epsilon)}\right)\right), \frac{\delta}{n}\right) \leq h\left(\frac{\gamma\mu_1}{4}, \frac{\delta}{n}\right).$$

Hence,

$$h\left(\frac{1}{2}\left(\mu_i - (1-\epsilon-\gamma)\left(\mu_1 + \frac{\omega}{8(1-\epsilon)}\right)\right), \frac{\delta}{n}\right) \leq \min\left[h\left(\frac{\epsilon\mu_1 - \Delta_i}{4}, \frac{\delta}{n}\right), h\left(\frac{\gamma\mu_1}{4}, \frac{\delta}{n}\right)\right].$$

**Case 2b,** $\omega = \gamma\mu_1$ **and** $i \in M_\epsilon^c \cap M_{\epsilon+\frac{\gamma}{2}}$

As before,

$$\mu_i - (1-\epsilon-\gamma)\left(\mu_1 + \frac{\omega}{8(1-\epsilon)}\right) = \mu_i - (1-\epsilon)\mu_1 + \gamma\mu_1 - \frac{1}{8}\left(\frac{1-\epsilon-\gamma}{1-\epsilon}\right)\gamma\mu_1$$

Since $i \in M_\epsilon^c \cap M_{\epsilon+\frac{\gamma}{2}}$, we have that $\mu_i - (1-\epsilon-\frac{\gamma}{2})\mu_1 \geq 0$. Rearranging implies that $\mu_i - (1-\epsilon)\mu_1 \geq \frac{-1}{2}\gamma\mu_1$. Hence,

$$\mu_i - (1-\epsilon)\mu_1 + \gamma\mu_1 - \frac{1}{8}\left(\frac{1-\epsilon-\gamma}{1-\epsilon}\right)\gamma\mu_1 \geq \frac{1}{2}\gamma\mu_1 - \frac{1}{8}\left(\frac{1-\epsilon-\gamma}{1-\epsilon}\right)\gamma\mu_1 \geq \frac{3}{8}\gamma\mu_1.$$

Hence,

$$h\left(\frac{1}{2}\left(\mu_i - (1-\epsilon-\gamma)\left(\mu_1 + \frac{\omega}{8(1-\epsilon)}\right)\right), \frac{\delta}{n}\right) \leq h\left(\frac{3\gamma\mu_1}{8}, \frac{\delta}{n}\right).$$

Additionally, as above, if $i \in M_\epsilon^c \cap M_{\epsilon+\frac{\gamma}{2}}$, we have that $\mu_i - (1-\epsilon-\frac{\gamma}{2})\mu_1 \geq 0$ which implies that $(1-\epsilon)\mu_1 - \mu_i \leq \frac{1}{2}\gamma\mu_1$. Hence

$$h\left(\frac{3\gamma\mu_1}{8}, \frac{\delta}{n}\right) \leq \min\left[h\left(\frac{\Delta_i - \epsilon\mu_1}{4}, \frac{\delta}{n}\right), h\left(\frac{\gamma\mu_1}{4}, \frac{\delta}{n}\right)\right].$$

Therefore, if $T_i$ exceeds the above, then $E_1(t)$ is true for an $i_1 \in M_\epsilon^c \cap M_{\epsilon+\frac{\gamma}{2}}$. Combining all cases, we see that for $i_1 \in M_{\epsilon+\frac{\gamma}{2}}$, if

$$T_{i_1(t)}(t) > \min\left[h\left(\frac{\epsilon\mu_1 - \Delta_i}{4}, \frac{\delta}{n}\right), h\left(\frac{\gamma\mu_1}{4}, \frac{\delta}{n}\right)\right],$$

Then $E_1(t)$ is true. Summing over all possible $i_1 \in M_{\epsilon+\frac{\gamma}{2}}$ proves the claim. $\qquad\square$

**Claim 1:** $|\mathcal{S} \cap \{t : E_1(t)\} \cap \{t : \neg E_2(t)\}| \leq \sum_{i \in M_{\epsilon+\frac{\gamma}{2}}^c} \min\left[h\left(\frac{\epsilon\mu_1-\Delta_i}{8}, \frac{\delta}{n}\right), h\left(\frac{\gamma\mu_1}{8}, \frac{\delta}{n}\right)\right].$

**Proof.** By the events in set $\mathcal{S}$, $C_{\delta/n}(T_{i^*}(t)) \leq \frac{\omega}{16(1-\epsilon)}$. Therefore,

$$\mu_{i^*} + \frac{\omega}{8(1-\epsilon)} \geq \mu_{i^*} + 2C_{\delta/n}(T_{i^*}(t)) \overset{\mathcal{E}}{\geq} \hat{\mu}_{i^*}(T_{i^*}(t)) + C_{\delta/n}(T_{i^*}(t)) \geq \hat{\mu}_1(T_1(t)) + C_{\delta/n}(T_1(t))$$

$$\overset{\mathcal{E}}{\geq} \mu_1.$$

Hence, $\mu_{i^*} \geq \mu_1 - \frac{\omega}{8(1-\epsilon)}$. Rearranging this, we see that $\mu_{i^*} - \left(1 - \frac{\omega}{8\mu_1(1-\epsilon)}\right)\mu_1 \geq 0$ which implies that $i^* \in M_{\frac{\omega}{8\mu_1(1-\epsilon)}}$. Hence,

$$L_t = (1-\epsilon)\left(\max_i \hat{\mu}_i(T_i(t)) - C_{\delta/n}(T_i(t))\right)(1-\epsilon)\left(\hat{\mu}_{i^*}(T_{i^*}(t)) - C_{\delta/n}(T_{i^*}(t))\right)$$

$$\overset{\mathcal{E}}{\geq} (1-\epsilon)\left(\mu_{i^*} - 2C_{\delta/n}(T_{i^*}(t))\right)$$

$$\geq (1-\epsilon)\left(\mu_{i^*} - \frac{\omega}{8(1-\epsilon)}\right)$$

$$\geq (1-\epsilon)\left(\mu_1 - \frac{\omega}{4(1-\epsilon)}\right)$$

As before, we seek a lower bound for the difference $(1-\epsilon)\left(\mu_1 - \frac{\omega}{4(1-\epsilon)}\right) - \mu_i$.

**Case 1:** $\omega = \min(\tilde{\alpha}_\epsilon, \tilde{\beta}_\epsilon)$

$$(1 - \epsilon)\left(\mu_1 - \frac{\omega}{4(1 - \epsilon)}\right) - \mu_i = (1 - \epsilon)\mu_1 - \mu_i - \frac{1}{4}\min(\tilde{\alpha}_\epsilon, \tilde{\beta}_\epsilon)$$

$$\geq \frac{1}{2}\left((1 - \epsilon)\mu_1 - \mu_i\right)$$

since $(1 - \epsilon)\mu_1 - \mu_i \geq \min(\tilde{\alpha}_\epsilon, \tilde{\beta}_\epsilon)$. Therefore, we have that

$$h\left(\frac{1}{2}\left((1 - \epsilon)\left(\mu_1 - \frac{\omega}{4(1 - \epsilon)}\right) - \mu_i\right), \frac{\delta}{n}\right) \leq h\left(\frac{\Delta_i - \epsilon\mu_1}{4}, \frac{\delta}{n}\right).$$

Lastly, in this setting, $\gamma\mu_1 \leq \min(\tilde{\alpha}_\epsilon, \tilde{\beta}_\epsilon) \leq \epsilon\mu_1 - \Delta_i$ since $\omega = \min(\tilde{\alpha}_\epsilon, \tilde{\beta}_\epsilon)$. Hence, it is trivially true that

$$h\left(\frac{\Delta_i - \epsilon\mu_1}{4}, \frac{\delta}{n}\right) = \min\left[h\left(\frac{\Delta_i - \epsilon\mu_1}{4}, \frac{\delta}{n}\right), h\left(\frac{\gamma\mu_1}{4}, \frac{\delta}{n}\right)\right].$$

**Case 2:** $\omega = \gamma\mu_1$

Assume that $\gamma\mu_1 > \min(\tilde{\alpha}_\epsilon, \tilde{\beta}_\epsilon)$, as equality is covered by the previous case. Hence,

$$(1 - \epsilon)\left(\mu_1 - \frac{\omega}{4(1 - \epsilon)}\right) - \mu_i = (1 - \epsilon)\mu_1 - \mu_i - \frac{1}{4}\gamma\mu_1$$

Recall that we seek to control $i_2 \in M^c_{\epsilon + \frac{\gamma}{2}}$. For any $i \in M^c_{\epsilon + \frac{\gamma}{2}}$, we have that $(1 - \epsilon - \frac{\gamma}{2})\mu_1 - \mu_i \geq 0$. Rearranging, we see that $(1 - \epsilon)\mu_1 - \mu_i \geq \frac{1}{2}\gamma\mu_1$ which implies that

$$(1 - \epsilon)\mu_1 - \mu_i - \frac{1}{4}\gamma\mu_1 \geq \frac{1}{2}((1 - \epsilon)\mu_1 - \mu_i).$$

Therefore, we have that

$$h\left(\frac{1}{2}\left((1 - \epsilon)\left(\mu_1 - \frac{\omega}{4(1 - \epsilon)}\right) - \mu_i\right), \frac{\delta}{n}\right) \leq h\left(\frac{\Delta_i - \epsilon\mu_1}{4}, \frac{\delta}{n}\right)$$

is this setting as well. Similarly, since $\Delta_i - \epsilon\mu_1 \geq \frac{1}{2}\gamma\mu_1$, we likewise have that

$$h\left(\frac{\Delta_i - \epsilon\mu_1}{4}, \frac{\delta}{n}\right) \leq \min\left[h\left(\frac{\Delta_i - \epsilon\mu_1}{8}, \frac{\delta}{n}\right), h\left(\frac{\gamma\mu_1}{8}, \frac{\delta}{n}\right)\right].$$

Hence, if $T_i$ exceeds the right-hand side of the preceding inequality, then for any $i \in M^c_{\epsilon + \frac{\gamma}{2}}$, its upper bound is below $L_t$. Hence, for $i_2(t) \in M^c_{\epsilon + \frac{\gamma}{2}}$, this implies event $E_2(t)$. Summing over all possible values of $i_2(t) \in M^c_{\epsilon + \frac{\gamma}{2}}$ proves the claim. □

**Claim 2:** The cardinality of $\mathcal{S}$ is bounded as $|\mathcal{S}| \leq \sum_{i=1}^n \min\left[h\left(\frac{\Delta_i - \epsilon\mu_1}{8}, \frac{\delta}{n}\right), h\left(\frac{\gamma\mu_1}{8}, \frac{\delta}{n}\right)\right]$.

**Proof.** First, $\mathcal{S}$ may be decomposed as

$$|\mathcal{S}| = |\mathcal{S} \cap \{t : \neg E_1(t)\}| + |\mathcal{S} \cap \{t : E_1(t)\} \cap \{t : \neg E_2(t)\}| + |\mathcal{S} \cap \{t : E_1(t)\} \cap \{t : E_2(t)\}|$$

Note that $|\mathcal{S} \cap \{t : E_1(t)\} \cap \{t : E_2(t)\}| = 0$ because we have assumed in set $\mathcal{S}$ that $(\mathtt{ST})^2$ has not stopped, and $\{t : E_1(t)\} \cap \{t : E_2(t)\}$ implies termination. By Claim 0, $|\mathcal{S} \cap \{t : \neg E_1(t)\}| \leq \sum_{i \in M_{\epsilon + \frac{\gamma}{2}}} \min\left[h\left(\frac{\epsilon\mu_1 - \Delta_i}{4}, \frac{\delta}{n}\right), h\left(\frac{\gamma\mu_1}{4}, \frac{\delta}{n}\right)\right]$. By Claim 1, $|\mathcal{S} \cap \{t : E_1(t)\} \cap \{t : \neg E_2(t)\}| \leq \sum_{i \in M^c_{\epsilon + \frac{\gamma}{2}}} \min\left[h\left(\frac{\epsilon\mu_1 - \Delta_i}{8}, \frac{\delta}{n}\right), h\left(\frac{\gamma\mu_1}{8}, \frac{\delta}{n}\right)\right]$. Recalling that $h$ is assumed to be symmetric in its first argument and summing the two terms proves the claim. □

### B.2.5 Step 4: Putting it all together

Recall that the total number of rounds $T$ that $(\mathtt{ST})^2$ runs for is given by $T = |\{t : \neg\mathtt{STOP}\}|$. To bound this quantity, we have decomposed the set $\{t : \neg\mathtt{STOP}\}$ into many subsets. Below, we show this decomposition.

$$\{t : \neg\mathtt{STOP}\} =$$

$$\{t : \neg \text{STOP and } i^* \notin M_{\omega/\mu_1}\}$$

$$\cup \left\{ t : \neg \text{STOP and } i^* \in M_{\omega/\mu_1} \text{ and } C_{\delta/n}(T_{i^*}(t)) > \frac{\omega}{16(1-\epsilon)} \right\}$$

$$\cup \left\{ t : \neg \text{STOP and } i^* \in M_{\omega/\mu_1} \text{ and } C_{\delta/n}(T_{i^*}(t)) \leq \frac{\omega}{16(1-\epsilon)} \text{ and } i_1(t) \in M^c_{\epsilon+\frac{\gamma}{2}} \right\}$$

$$\cup \left\{ t : \neg \text{STOP and } i^* \in M_{\omega/\mu_1} \text{ and } C_{\delta/n}(T_{i^*}(t)) \leq \frac{\omega}{16(1-\epsilon)} \text{ and } i_1(t) \in M_{\epsilon+\frac{\gamma}{2}} \right.$$
$$\left. \text{and } i_2(t) \in M_{\epsilon+\frac{\gamma}{2}} \right\}$$

$$\cup \left\{ t : \neg \text{STOP and } i^* \in M_{\omega/\mu_1} \text{ and } C_{\delta/n}(T_{i^*}(t)) \leq \frac{\omega}{16(1-\epsilon)} \text{ and } i_1(t) \in M_{\epsilon+\frac{\gamma}{2}} \right.$$
$$\left. \text{and } i_2(t) \in M^c_{\epsilon+\frac{\gamma}{2}} \right\}.$$

Hence, by a union bound and plugging in the results of the above steps,

$$|\{t : \neg \text{STOP}\}| \leq$$

$$|\{t : \neg \text{STOP and } i^* \notin M_{\omega/\mu_1}\}|$$

$$+ \left| \left\{ t : \neg \text{STOP and } i^* \in M_{\omega/\mu_1} \text{ and } \exists i \in M_{\omega/\mu_1} : C_{\delta/n}(T_i(t)) > \frac{\omega}{8(1-\epsilon)} \right\} \right|$$

$$+ \left| \left\{ t : \neg \text{STOP and } i^* \in M_{\omega/\mu_1} \text{ and } C_{\delta/n}(T_{i^*}(t)) \leq \frac{\omega}{16(1-\epsilon)} \text{ and } i_1(t) \in M^c_{\epsilon+\frac{\gamma}{2}} \right\} \right|$$

$$+ \left| \left\{ t : \neg \text{STOP and } i^* \in M_{\omega/\mu_1} \text{ and } C_{\delta/n}(T_{i^*}(t)) \leq \frac{\omega}{16(1-\epsilon)} \text{ and } i_1(t) \in M_{\epsilon+\frac{\gamma}{2}} \right. \right.$$
$$\left. \left. \text{and } i_2(t) \in M_{\epsilon+\frac{\gamma}{2}} \right\} \right|$$

$$+ \left| \left\{ t : \neg \text{STOP and } i^* \in M_{\omega/\mu_1} \text{ and } C_{\delta/n}(T_{i^*}(t)) \leq \frac{\omega}{16(1-\epsilon)} \text{ and } i_1(t) \in M_{\epsilon+\frac{\gamma}{2}} \right. \right.$$
$$\left. \left. \text{and } i_2(t) \in M^c_{\epsilon+\frac{\gamma}{2}} \right\} \right|$$

$$\leq \sum_{i \in M^c_{\omega/\mu_1}} \min \left\{ h\left(\frac{\gamma\mu_1}{2}, \frac{\delta}{n}\right), \min\left[ h\left(\frac{\Delta_i}{2}, \frac{\delta}{n}\right), h\left(\frac{\min(\tilde{\alpha}_\epsilon, \tilde{\beta}_\epsilon)}{2}, \frac{\delta}{n}\right) \right] \right\}$$

$$+ \sum_{i \in M_{\omega/\mu_1}} \min \left\{ h\left(\frac{\gamma\mu_1}{16}, \frac{\delta}{n}\right), \min\left[ h\left(\frac{\Delta_i}{16}, \frac{\delta}{n}\right), h\left(\frac{\min(\tilde{\alpha}_\epsilon, \tilde{\beta}_\epsilon)}{16(1-\epsilon)}, \frac{\delta}{n}\right) \right] \right\}$$

$$+ \sum_{i \in M^c_{\epsilon+\frac{\gamma}{2}}} \min \left[ h\left(\frac{\Delta_i - \epsilon\mu_1}{8}, \frac{\delta}{n}\right), h\left(\frac{\gamma\mu_1}{8}, \frac{\delta}{n}\right) \right]$$

$$+ \sum_{i \in M_{\epsilon+\frac{\gamma}{2}}} \min \left[ h\left(\frac{\epsilon\mu_1 - \Delta_i}{8}, \frac{\delta}{n}\right), h\left(\frac{\gamma\mu_1}{8}, \frac{\delta}{n}\right) \right]$$

$$+ \sum_{i=1}^{n} \min \left[ h\left(\frac{\Delta_i - \epsilon\mu_1}{8}, \frac{\delta}{n}\right), h\left(\frac{\gamma\mu_1}{8}, \frac{\delta}{n}\right) \right]$$

$$\overset{(\epsilon \leq 1/2)}{\leq} \sum_{i=1}^{n} \min \left\{ h\left(\frac{\gamma\mu_1}{16}, \frac{\delta}{n}\right), \min\left[ h\left(\frac{\Delta_i}{16}, \frac{\delta}{n}\right), h\left(\frac{\min(\tilde{\alpha}_\epsilon, \tilde{\beta}_\epsilon)}{16(1-\epsilon)}, \frac{\delta}{n}\right) \right] \right\}$$

$$+ 2\sum_{i=1}^{n} \min \left[ h\left(\frac{\Delta_i - \epsilon\mu_1}{8}, \frac{\delta}{n}\right), h\left(\frac{\gamma\mu_1}{8}, \frac{\delta}{n}\right) \right]$$

$$\leq 4\sum_{i=1}^{n} \min \left\{ \max\left\{ h\left(\frac{\Delta_i - \epsilon\mu_1}{16}, \frac{\delta}{n}\right), \min\left[ h\left(\frac{\Delta_i}{16}, \frac{\delta}{n}\right), h\left(\frac{\min(\tilde{\alpha}_\epsilon, \tilde{\beta}_\epsilon))}{16(1-\epsilon)}, \frac{\delta}{n}\right) \right] \right\} \right\},$$

$$h\left(\frac{\gamma\mu_1}{16},\frac{\delta}{n}\right)\Big\}$$

Next, by Lemma F.3, we may bound the minimum of $h(\cdot,\cdot)$ functions.

$$4\sum_{i=1}^{n}\min\left\{\max\left\{h\left(\frac{\Delta_i-\epsilon\mu_1}{16},\frac{\delta}{n}\right),\min\left[h\left(\frac{\Delta_i}{16},\frac{\delta}{n}\right),h\left(\frac{\min(\tilde{\alpha}_\epsilon,\tilde{\beta}_\epsilon)}{16(1-\epsilon)},\frac{\delta}{n}\right)\right]\right\},\right.$$
$$\left.h\left(\frac{\gamma\mu_1}{16},\frac{\delta}{n}\right)\right\}$$

$$=4\sum_{i=1}^{n}\min\left\{\max\left\{h\left(\frac{\Delta_i-\epsilon\mu_i}{16},\frac{\delta}{n}\right),\right.\right.$$
$$\left.\min\left[h\left(\frac{\Delta_i}{16},\frac{\delta}{n}\right),\max\left[h\left(\frac{\tilde{\alpha}_\epsilon}{16(1-\epsilon)},\frac{\delta}{n}\right),h\left(\frac{\tilde{\beta}_\epsilon}{16(1-\epsilon)},\frac{\delta}{n}\right)\right]\right]\right\},$$
$$\left.h\left(\frac{\gamma\mu_i}{16},\frac{\delta}{n}\right)\right\}$$

$$\leq4\sum_{i=1}^{n}\min\left\{\max\left\{h\left(\frac{\Delta_i-\epsilon\mu_i}{16},\frac{\delta}{n}\right),\right.\right.$$
$$\left.\max\left[h\left(\frac{\Delta_i+\frac{\tilde{\alpha}_\epsilon}{1-\epsilon}}{32},\frac{\delta}{n}\right),h\left(\frac{\Delta_i+\frac{\tilde{\beta}_\epsilon}{1-\epsilon}}{32},\frac{\delta}{n}\right)\right]\right\},$$
$$\left.h\left(\frac{\gamma\mu_i}{16},\frac{\delta}{n}\right)\right\}$$

$$=4\sum_{i=1}^{n}\min\left\{\max\left\{h\left(\frac{\Delta_i-\epsilon\mu_i}{16},\frac{\delta}{n}\right),h\left(\frac{\Delta_i+\frac{\tilde{\alpha}_\epsilon}{1-\epsilon}}{32},\frac{\delta}{n}\right),h\left(\frac{\Delta_i+\frac{\tilde{\beta}_\epsilon}{1-\epsilon}}{32},\frac{\delta}{n}\right)\right\},\right.$$
$$\left.h\left(\frac{\gamma\mu_i}{16},\frac{\delta}{n}\right)\right\}$$

Finally, we use Lemma F.2 to bound the function $h(\cdot,\cdot)$. Since $\delta\leq1/2$, $\delta/n\leq2e^{-e/2}$. Further, $|\epsilon\mu_1-\Delta_i|\leq8$ for all $i$ and $\epsilon\leq1/2$ implies that $\frac{1}{8(1-\epsilon)}|\epsilon\mu_1-\Delta_i|\leq2$ and $\frac{1}{8(1-\epsilon)}\min(\tilde{\alpha}_\epsilon,\tilde{\beta}_\epsilon)\leq2$. $\Delta_i\leq16$ for all $i$, gives $0.125\Delta_i\leq2$. Lastly, $\gamma\leq16/\mu_1$ implies that $\frac{\gamma\mu_1}{8}\leq2$. Therefore,

$$4\sum_{i=1}^{n}\min\left\{\max\left\{h\left(\frac{\Delta_i-\epsilon\mu_i}{16},\frac{\delta}{n}\right),h\left(\frac{\Delta_i+\frac{\tilde{\alpha}_\epsilon}{1-\epsilon}}{32},\frac{\delta}{n}\right),h\left(\frac{\Delta_i+\frac{\tilde{\beta}_\epsilon}{1-\epsilon}}{32},\frac{\delta}{n}\right)\right\},\right.$$
$$\left.h\left(\frac{\gamma\mu_i}{16},\frac{\delta}{n}\right)\right\}$$

$$\leq4\sum_{i=1}^{n}\min\left\{\max\left\{\frac{1024}{(\epsilon\mu_1-\Delta_i)^2}\log\left(\frac{2n}{\delta}\log_2\left(\frac{3072n}{\delta(\epsilon\mu_1-\Delta_i)^2}\right)\right),\right.\right.$$
$$\frac{4096}{(\Delta_i+\frac{\tilde{\alpha}_\epsilon}{1-\epsilon})^2}\log\left(\frac{2n}{\delta}\log_2\left(\frac{12288n}{\delta(\Delta_i+\frac{\tilde{\alpha}_\epsilon}{1-\epsilon})^2}\right)\right),$$
$$\left.\frac{4096}{(\Delta_i+\frac{\tilde{\beta}_\epsilon}{1-\epsilon})^2}\log\left(\frac{2n}{\delta}\log_2\left(\frac{12288n}{\delta(\Delta_i+\frac{\tilde{\beta}_\epsilon}{1-\epsilon})^2}\right)\right)\right\}$$
$$\frac{1024}{\gamma^2\mu_1^2}\log\left(\frac{2n}{\delta}\log_2\left(\frac{3072n}{\delta\gamma^2\mu_1^2}\right)\right)\right\}$$

$$=4\sum_{i=1}^{n}\min\left\{\max\left\{\frac{1024}{((1-\epsilon)\mu_1-\mu_i)^2}\log\left(\frac{2n}{\delta}\log_2\left(\frac{3072n}{\delta((1-\epsilon)\mu_1-\mu_i)^2}\right)\right),\right.\right.$$

$$\frac{4096}{(\mu_1 + \frac{\tilde{\alpha}_\epsilon}{1-\epsilon} - \mu_i)^2} \log\left(\frac{2n}{\delta} \log_2\left(\frac{12288n}{\delta(\mu_1 + \frac{\tilde{\alpha}_\epsilon}{1-\epsilon})^2}\right)\right),$$

$$\frac{4096}{(\mu_1 + \frac{\tilde{\beta}_\epsilon}{1-\epsilon} - \mu_i)^2} \log\left(\frac{2n}{\delta} \log_2\left(\frac{12288n}{\delta(\mu_1 + \frac{\tilde{\beta}_\epsilon}{1-\epsilon} - \mu_i)^2}\right)\right)\Bigg\},$$

$$\frac{1024}{\gamma^2\mu_1^2} \log\left(\frac{2n}{\delta} \log_2\left(\frac{3072n}{\delta\gamma^2\mu_1^2}\right)\right)\Bigg\}.$$

The above bounds the number of rounds $T$. Therefore, the total number of samples is at most $3T$. □

## C Proof of instance dependent lower bounds, Theorem 2.1

First we restate and prove the lower bound.

**Theorem C.1.** *(additive and multiplicative lower bound) Fix $\delta, \epsilon > 0$. Consider $n$ arms, such that the $i^{th}$ is distributed according to $\mathcal{N}(\mu_i, 1)$. Any $\delta$-PAC algorithm for the additive setting satisfies*

$$\mathbb{E}[\tau] \geq 2\sum_{i=1}^n \max\left\{\frac{1}{(\mu_1 - \epsilon - \mu_i)^2}, \frac{1}{(\mu_1 + \alpha_\epsilon - \mu_i)^2}\right\} \log\left(\frac{1}{2.4\delta}\right)$$

*and if $\mu_1 > 0$ any $\delta$-PAC algorithm for the multiplicative algorithm satisfies,*

$$E[\tau] \geq 2\sum_{i=1}^n \max\left\{\frac{1}{((1-\epsilon)\mu_1 - \mu_i)^2}, \frac{1}{(\mu_1 + \frac{\tilde{\alpha}_\epsilon}{1-\epsilon} - \mu_i)^2}\right\} \log\left(\frac{1}{2.4\delta}\right)$$

*Proof of Theorem 2.1 in the additive case.* Recall that $\nu$ denotes the given instance, and without loss of generality we have assumed that $\mu_1 \geq \mu_2 \geq \cdots \geq \mu_n$. Then $G_\epsilon(\nu) = \{1, \cdots, k\}$. Consider the event $E$ that an algorithm returns $\{1, \cdots, k\}$. For any $\delta$-PAC algorithm, $E$ occurs with probability at least $1 - \delta$. For each arm $i \in [n]$ we consider two alternative instances

$$\nu_i' = \{\mu_1, \cdots, \mu_i', \cdots, \mu_n\}$$

and

$$\nu_i'' = \{\mu_1, \cdots, \mu_i'', \cdots, \mu_n\}$$

such that only the mean of arm $i$ differs compared to $\nu$ but $G_\epsilon(\nu) \neq G_\epsilon(\nu_i')$ and $G_\epsilon(\nu) \neq G_\epsilon(\nu_i'')$. Therefore, on these alternate instances, $E$ occurs with probability at most $\delta$.

For $\nu_i'$, if $i \leq k$, let $\mu_i' = \mu_1 - \epsilon - \eta$. Then $i \in G_\epsilon(\nu)$ but $i \notin G_\epsilon(\nu_i')$. If $k < n$ and $i \geq k+1$, let $\mu_i' = \mu_1 - \epsilon + \eta$. Then $i \notin G_\epsilon(\nu)$ but $i \in G_\epsilon(\nu_i')$.

More subtly, for $\nu_i''$, for any $i \in [n]\backslash\{k\}$, let $\mu_i'' = \mu_k + \epsilon + \eta$. In particular, arm $i$ is now the best arm. Under this definition, $\mu_i'' - \epsilon > \mu_k$. Therefore, $k \notin G_\epsilon(\nu_i'')$ but $k \in G_\epsilon(\nu)$.

The above holds for all $\eta > 0$. Let $N_i$ denote the random variable of the number of samples of arm $i$ and $\mathbb{E}_\nu$ denote expectation with respect to instance $\nu$. Using the fact that we have assumed the distributions are Gaussian, considering $\nu_i'$, by Lemma 1 of [6], taking $\eta \to 0$ we have that for any $\delta$-PAC algorithm,

$$\mathbb{E}_\nu[N_i] \geq \frac{2\log(1/2.4\delta)}{(\mu_i - (\mu_1 - \epsilon))^2}.$$

Furthermore, considering $\nu_i''$, and again taking $\eta \to 0$, we have by the same lemma that for $i \neq k$

$$\mathbb{E}_\nu[N_i] \geq \frac{2\log(1/2.4\delta)}{(\mu_k + \epsilon - \mu_i)^2} = \frac{2\log(1/2.4\delta)}{(\mu_1 + \alpha_\epsilon - \mu_i)^2},$$

where the later equality holds since $\mu_k + \epsilon = \mu_1 + \alpha_\epsilon$ by definition of $\alpha_\epsilon$. For $i = k$, note that $\frac{1}{(\mu_k - (\mu_1 - \epsilon))} = \frac{1}{\alpha_\epsilon^2} \geq \frac{1}{\epsilon^2} = \frac{1}{(\mu_k - \mu_k - \epsilon)^2}$ since $\alpha_\epsilon = \min_{i\in G_\epsilon} \mu_i - (\mu_1 - \epsilon) = \min_{i\in G_\epsilon} \epsilon - \Delta_i$. Putting these pieces together, we see that for any $i$,

$$\mathbb{E}_\nu[N_i] \geq \max\left(\frac{1}{(\mu_i - (\mu_1 - \epsilon))^2}, \frac{1}{(\mu_k + \epsilon - \mu_i)^2}\right) 2\log(1/2.4\delta).$$

Summing over all $i$ establishes a lower bound in the additive case. □

*Proof of Theorem 2.1 in the multiplicative case.* Recall that $\nu$ denotes the given instance, and without loss of generality we have assumed that $\mu_1 \geq \mu_2 \geq \cdots \geq \mu_n$. Let $M_\epsilon(\nu) = \{1, \cdots, k\}$. Consider the event $E$ that an algorithm returns $\{1, \cdots, k\}$. For any $\delta$-PAC algorithm, $E$ occurs with probability at least $1 - \delta$. For each arm $i \in [n]$ we consider two alternative instances

$$\nu_i' = \{\mu_1, \cdots, \mu_i', \cdots, \mu_n\}$$

and

$$\nu_i'' = \{\mu_1, \cdots, \mu_i'', \cdots, \mu_n\}$$

such that only the mean of arm $i$ differs compared to $\nu$ but $M_\epsilon(\nu) \neq M_\epsilon(\nu_i')$ and $M_\epsilon(\nu) \neq M_\epsilon(\nu_i'')$. Therefore, $E$ occurs with probability at most $\delta$ on these alternate instances.

For $\nu_i'$, if $i \leq k$, let $\mu_i' = (1 - \epsilon - \eta)\mu_1$. Then $i \in M_\epsilon(\nu)$ but $i \notin M_\epsilon(\nu_i')$. If $k < n$ and $i \geq k+1$, let $\mu_i' = (1 - \epsilon + \eta)\mu_1$. Then $i \notin M_\epsilon(\nu)$ but $i \in M_\epsilon(\nu_i')$.

More subtly, for $\nu_i''$, for any $i \in [n] \setminus \{k\}$, let $\mu_i'' = \frac{\mu_k}{1 - \epsilon - \eta}$. In particular, arm $i$ is now the best arm. Under this definition, $\mu_i'' - \epsilon > \mu_k$. Therefore, $k \notin M_\epsilon(\nu_i'')$ but $k \in M_\epsilon(\nu)$.

The above holds for all $\eta > 0$. Let $N_i$ denote the random variable of the number of samples of arm $i$ and $\mathbb{E}_\nu$ denote expectation with respect to instance $\nu$. Using the fact that we have assumed the distributions are Gaussian, considering $\nu_i'$, by Lemma 1 of [6], taking $\eta \to 0$, we have that for any $\delta$-PAC algorithm,

$$\mathbb{E}_\nu[N_i] \geq \frac{2 \log(1/2.4\delta)}{(\mu_i - (1-\epsilon)\mu_1)^2} = \frac{2 \log(1/2.4\delta)}{(\epsilon\mu_1 - \Delta_i)^2}.$$

Additionally, by the same Lemma, considering $\nu_i''$ and again taking $\eta \to 0$ we have that for $i \neq k$

$$\mathbb{E}_\nu[N_i] \geq \frac{2 \log(1/2.4\delta)}{\left(\mu_i - \frac{\mu_k}{1-\epsilon}\right)^2} = \frac{2 \log(1/2.4\delta)}{\left(\mu_1 + \frac{\tilde{\alpha}_\epsilon}{1-\epsilon} - \mu_i\right)^2},$$

where the later equality holds since $\frac{\mu_k}{1-\epsilon} = \mu_1 + \frac{\tilde{\alpha}_\epsilon}{1-\epsilon}$ by definition of $\tilde{\alpha}_\epsilon$. Next recall that $\tilde{\alpha}_\epsilon := \min_{i \in M_\epsilon} \mu_i - (1-\epsilon)\mu_1 = \mu_k - (1-\epsilon)\mu_1$, we have that $\mu_k = \tilde{\alpha}_\epsilon + (1-\epsilon)\mu_1$. Hence, $\frac{\mu_k}{1-\epsilon} = \mu_1 + \frac{\tilde{\alpha}_\epsilon}{1-\epsilon}$. Then, for $i = k$

$$\frac{1}{\left(\frac{\mu_k}{1-\epsilon} - \mu_k\right)^2} \leq \frac{1}{(\mu_k - (1-\epsilon)\mu_1)^2} = \frac{1}{\tilde{\alpha}_\epsilon^2}$$

$$\iff \tilde{\alpha}_\epsilon \leq \frac{\mu_k}{1-\epsilon} - \mu_k = \frac{\tilde{\alpha}_\epsilon}{1-\epsilon} + \mu_1 - \mu_k = \frac{\tilde{\alpha}_\epsilon}{1-\epsilon} + \Delta_k$$

$$\overset{(\Delta_k \geq 0)}{\Longleftarrow} \tilde{\alpha}_\epsilon \leq \frac{\tilde{\alpha}_\epsilon}{1-\epsilon}$$

which is always true since $\epsilon > 0$. Therefore,

$$\frac{1}{(\mu_k - (1-\epsilon)\mu_1)^2} = \max\left(\frac{1}{(\mu_k - (1-\epsilon)\mu_1)^2}, \frac{1}{\left(\frac{\mu_k}{1-\epsilon} - \mu_k\right)^2}\right).$$

Hence, for all arms $i$,

$$\mathbb{E}_\nu[N_i] \geq 2 \max\left(\frac{1}{(\mu_i - (1-\epsilon)\mu_1)^2}, \frac{1}{\left(\mu_1 + \frac{\tilde{\alpha}_\epsilon}{1-\epsilon} - \mu_i\right)^2}\right) \log(1/2.4\delta).$$

Summing over all $i$ gives a lower bound for this problem in the multiplicative case. $\square$

## D  Theorem 4.1: Lower bounds in the moderate confidence regime

In this section, we prove a tighter lower bound that includes *moderate confidence* terms independent of the value of $\delta$ similar to those that appear in the upper bound on the sample complexity of FAREAST, Theorem 4.2.

(a) A non-isolated instance ($H_0$ is true)    (b) An isolated instance ($H_0$ is true)

Figure 14: Example of an isolated and non-isolated instance

**Outline.** To give a tight lower bound in the isolated setting, we break our argument into pieces performing a series of reductions that link the all-$\epsilon$ problem to a hypothesis test, and then the hypothesis test to the problem of identifying the best-arm.

**Step 1. Finding an isolated arm.** We first consider the following problem. Imagine that you are given an *isolated* instance, depicted in Figure 14b where there are $n$ distributions, with one of them at mean $\beta$ and the rest with mean $-\beta$. Theorem D.3, captures the sample complexity of any algorithm that can return $i^*$ with probability greater than $1 - \delta$.

**Step 2. Deciding if an instance is isolated.** We then consider a composite hypothesis test on $n$ distributions where the null hypothesis, $H_0$, is that the mean of each distribution is less that $-\beta$ and the alternate hypothesis, $H_1$, is that there exists *single* distribution $i^*$ with mean $\beta$ and the remainder have mean less than $-\beta$ (i.e. the instance is isolated). In Figure 14, we show a picture of an instance where the null is true and where the alternate is true. In Theorem D.6 we lower bound the complexity of performing this test. To link this to Step 1, we show that if you can solve this composite hypothesis test then you can find $i^*$, hence the lower bound of step 1 is a lower bound for the hypothesis test.

**Step 3: Reducing ALL-$\epsilon$ to Step 2** Finally in step 3 we link this to the all-$\epsilon$ problem. Using the above, we lower bound the complexity of ALL-$\epsilon$ in Theorem 4.1 when $|G_{2\beta_\epsilon}| = 1$. The key insight of our proof is that any algorithm that can solve the ALL-$\epsilon$ problem can be used to solve the hypothesis test in Step 2.

## D.1 Step 1: Finding an Isolated Arm

Fix $n \in \mathbb{N}$, $0 < \beta$, and $\delta > 0$. We refer to a $\beta$-*isolated instance* $\nu = \{\rho_1, \cdots, \rho_n\}$, as a collection of $n$, Gaussian distributions with variance one satisfying two properties. Firstly, there exists a single arm $i^* \in [n]$ with $\rho_{i^*} = \mathcal{N}(\beta, 1)$. We refer to this as the *isolated arm*. Secondly, for $i \neq i^*$, $\rho_i = \mathcal{N}(\mu_i, 1) \; \forall \; i \in [n] \backslash \{i^*\}$ have means $\mu_i \leq -\beta$. We introduce the additional notation $\Delta_{i,j} = \mu_i - \mu_j$.

**Lemma D.1.** *Fix $n$, $0 < \beta$ and consider a set $\nu$ of $n$ Gaussian random variables such that for a uniformly random chosen $i^* \in [n]$, $\rho_{i^*} = \mathcal{N}(\beta, 1)$ and $\rho_i = \mathcal{N}(\mu_i, 1)$ for $\mu_i \leq -\beta$ for all $i \neq i^*$. Any algorithm that correctly returns $i^*$ with probability at least $1 - \delta$, pulls arm $i^*$ at least*

$$\frac{1}{2\beta^2} \log(1/2.4\delta)$$

*times in expectation.*

*Proof.* Consider the oracle setting where the value of $i^*$ is known and the algorithm only seeks to confirm that $\mu_{i^*} > -\beta$. Lemma 1 of [6] implies that any $\delta$-PAC algorithm requires at least $\frac{1}{2\beta^2} \log(1/2.4\delta)$ samples in expectation. $\qquad \square$

The above bound controls the number of samples that any algorithm must gather from $i^*$, and is independent of $n$. The proof considered an oracle setting where the value of $i^*$ is known, and

one only wishes to confirm that $\mu_{i^*} > -\beta$ with probability at least $1 - \delta$. To lower bound the number of samples drawn from $[n]\backslash\{i^*\}$, we need significantly more powerful tools. In particular, to rule out trivial algorithms that always output a fixed index, we consider a permutation model, as in [9, 11, 15, 25]. Informally, we consider an additional expectation in the lower bound over a random permutation $\pi$ of the arms where $\pi$ is sampled uniformly from the set of all permutations. In particular, we with use a *Simulator* argument, as in [9, 15]. In what follows, we will let $\pi : [n] \to [n]$ denote a permutation selected uniformly at random from the set of $n!$ permutations. For instance $\nu$, let $\pi(\nu)$ denote the permuted instance such that the $i^{\text{th}}$ distribution is mapped to $\pi(i)$, by a slight overloading of the definition of $\pi(\cdot)$. In what follows, we proceed similarly to the proof of Theorem 1 in [15].

**Theorem D.2.** *Fix $n$, $0 < \beta$, and $\delta < 1/16$ and consider a set $\nu$ of $n$ Gaussian random variables with variance 1 such that for $i^* \in [n]$, $\rho_{i^*} = \mathcal{N}(\beta, 1)$ and $\rho_i = \mathcal{N}(\mu_i, 1)$ for $\mu_i \leq -\beta$ for all $i \neq i^*$. Let $\pi$ be a uniformly chosen permutation of $[n]$ and $\pi(\nu)$ be the permutation applied to instance $\nu$. Let $T$ be the random variable denoting the total number of samples at termination by an algorithm. Any $\delta$-PAC algorithm to detect $\pi(i^*)$ on $\pi(\nu)$ requires*

$$\mathbb{E}_\pi \mathbb{E}_{\pi(\nu)} [T] \geq \frac{1}{16} \sum_{k \neq i^*} \frac{1}{\Delta_{i^*,k}^2}$$

*samples in expectation from arms in $[n]\backslash\{i^*\}$.*

*Proof.* Fix a permutation $\pi$. Let $\pi(\nu)$ be the permutation applied to $\nu$ and $\pi(i)$ be the index of $i$ under the permuted instance, $\pi(\nu)$. Let $\mathcal{A}$ be any algorithm that detects and returns $\pi(i^*)$ on $\pi(\nu)$ with probability at least $1 - \delta$. We will take $\mathbb{P}_\mathcal{A}$ and $\mathbb{E}_\mathcal{A}$ to denote probability and expectation with respect any internal randomness in $\mathcal{A}$. Throughout, we will take $\rho_i = \mathcal{N}(\mu_i, 1)$ to denote the $i^{\text{th}}$ distribution of $\nu$. $\mu_{i^*} > 0$ and $\mu_i < 0$ for all $i \neq i^*$. Additionally, let $\Delta_{ij} = \mu_i - \mu_j$

Fix $k \neq i^*$. To bound the necessary number of samples for arm $k$, we turn to the Simulator [9]. We begin by defining an alternate instance $\nu_k' = \{\rho_1', \cdots, \rho_n'\}$ as

$$\rho_j' = \begin{cases} \rho_j, & j \neq i^* \\ \rho_k, & j = i^* \\ \rho_{i^*}, & j = k \end{cases}$$

Note that $\nu_k'$ is identical to $\nu$ except that the distributions of $i^*$ and $k$ are swapped.

Let $E$ be the event that $\mathcal{A}$ returns $\pi(i^*)$. We may bound the total variation distance between the joint distribution on $\mathcal{A} \times \pi(\nu)$ and $\mathcal{A} \times \pi(\nu_k')$ as

$$\begin{aligned} TV(\mathbb{P}_{\mathcal{A} \times \pi(\nu)}, \mathbb{P}_{\mathcal{A} \times \pi(\nu_k')}) &= \sup_A \left| \mathbb{P}_{\mathcal{A} \times \pi(\nu)}(A) - \mathbb{P}_{\mathcal{A} \times \pi(\nu_k')}(A) \right| \\ &\geq \left| \mathbb{P}_{\mathcal{A} \times \pi(\nu)}(E) - \mathbb{P}_{\mathcal{A} \times \pi(\nu_k')}(E) \right| \\ &\geq 1 - 2\delta. \end{aligned}$$

Let $\Omega_t$ denote the multiset of the transcript of samples up to time $t$.

$$\Omega_t = \{i_s \in [n] \text{ for } 1 \leq s \leq t\}$$

and define the events

$$W_j(\Omega_t) := \left\{ \sum_{i_t \in \Omega_t} \mathbb{1}(i_t = j) \leq \tau \right\}$$

for a $\tau$ to be defined later. With the definitions of $W_j(\Omega_t)$, we define a simulator $\text{Sim}(\nu, \Omega_t)$ with respect to $\nu$. Let $\text{Sim}(\nu, \Omega_t)_i$ denote the distribution of arm $i$ on $\text{Sim}(\nu, \Omega_t)$.

$$\text{Sim}(\nu, \Omega_t)_j = \begin{cases} \rho_j, & \text{if } j \notin \{i^*, k\} \\ \rho_j, & \text{if } j \in \{i^*, k\} \text{ and } W_{i^*}(\Omega_t) \cap W_k(\Omega_t) \\ \rho_{i^*}, & \text{if } j \in \{i^*, k\} \text{ and } (W_{i^*}(\Omega_t) \cap W_k(\Omega_t))^c \end{cases}$$

Furthermore, we define $\text{Sim}(\nu_k', \Omega_t)$ with respect to $\nu_k'$ as

$$\text{Sim}(\nu_k', \Omega_t)_j = \begin{cases} \rho_j', & \text{if } j \notin \{i^*, k\} \\ \rho_j', & \text{if } j \in \{i^*, k\} \text{ and } W_{i^*}(\Omega_t) \cap W_k(\Omega_t) \\ \rho_{i^*}, & \text{if } j \in \{i^*, k\} \text{ and } (W_{i^*}(\Omega_t) \cap W_k(\Omega_t))^c \end{cases}$$

For ease of notation, let $\text{Sim}(\pi(\nu), \Omega_t)$ be the same simulator defined on $\pi(\nu)$ and with respect to events $W_{\pi(i^*)}(\Omega_t)$ and $W_{\pi(k)}(\Omega_t)$. Note that in the simulator of $\nu_k'$, if $(W_{i^*}(\Omega_t) \cap W_k(\Omega_t))^c$ is true, then $i^*$ and $k$ draw samples according to instance $\nu$ not $\nu_k'$.

**Definition 4.** *(Truthfulness of an event $W$, [15]) For an algorithm $\mathcal{A}$, we say that an event $W$ is truthful on a simulator $Sim(\eta)$ with respect to an instance $\eta$ if for all events $E$ in the filtration $\mathcal{F}_T$ generated by playing algorithm $\mathcal{A}$ on instance $\eta$*

$$\mathbb{P}_\eta(E \cap W) = \mathbb{P}_{Sim(\eta)}(E \cap W)$$

By our definition of both simulators, if $(W_{i^*}(\Omega_t) \cap W_k(\Omega_t))^c$ is true, then $\text{Sim}(\nu, \Omega_t)_i = \text{Sim}(\nu_k', \Omega_t)_i \ \forall i \in [n]$. Contrarily, if $W_{i^*}(\Omega_t) \cap W_k(\Omega_t)$ is true, then $\text{Sim}(\nu, \Omega_t) = \nu$ and $\text{Sim}(\nu_k', \Omega_t) = \nu_k'$. Similarly, on $W_{\pi(i^*)}(\Omega_t) \cap W_{\pi(k)}(\Omega_t)$, $\text{Sim}(\pi(\nu), \Omega_t) = \pi(\nu)$ and $\text{Sim}(\pi(\nu_k'), \Omega_t) = \pi(\nu_k')$. Therefore, by the proof of Theorem 1 of [15], $W_{\pi(k)}(\Omega_t)$ is truthful on $\text{Sim}(\pi(\nu), \Omega_t)$ and $W_{\pi(i^*)}(\Omega_t)$ is truthful on $\text{Sim}(\pi(\nu_k'), \Omega_t)$.

Let $i_t$ be the arm queried at time $t \in \mathbb{N}$ by $\mathcal{A}$. Following the proof of Theorem 1 of [15], we may bound the KL-Divergence between $\text{Sim}(\pi(\nu), \Omega_t)$ and $\text{Sim}(\pi(\nu_k'), \Omega_t)$ as

$$\begin{aligned}
\max_{i_1, \cdots, i_T} \sum_{t=1}^{T} \quad & KL\left(\text{Sim}(\pi(\nu), \{i_s\}_{s=1}^t), \text{Sim}(\pi(\nu_k'), \{i_s\}_{s=1}^t)\right) \\
\leq \quad & \tau KL(\pi(\nu)_{\pi(i^*)}, \pi(\nu_k')_{\pi(i^*)}) + \tau KL(\pi(\nu)_{\pi(k)}, \pi(\nu_k')_{\pi(k)}) \\
= \quad & \tau \frac{\Delta_{i^*,k}^2}{2} + \tau \frac{\Delta_{i^*,k}^2}{2} \\
= \quad & \tau \Delta_{i^*,k}^2.
\end{aligned}$$

For any instance $\eta$, an algorithm $\mathcal{A}$ is defined to be symmetric if

$$\mathbb{P}_{\mathcal{A}, \eta}((i_1, \cdots, i_T) = (I_1, \cdots, I_T)) = \mathbb{P}_{\mathcal{A}, \pi(\eta)}((\pi(i_1), \cdots, \pi(i_T)) = (\pi(I_1), \cdots, \pi(I_T))).$$

Semantically, this implies that the proportion of times $\mathcal{A}$ pulls any arm $i$ on the non-permuted instance $\eta$ is the same as the proportion of times it pulls $\pi(i)$ on the permuted instance, $\pi(\eta)$.

In particular, the expected complexity of a symmetric algorithm is independent of the permutation $\pi$. By Lemma 1 of [9], if any algorithm $\mathcal{B}$ (not necessarily symmetric) achieves an expected stopping time $\tau$ where the expectation is taken over all the randomness in the permutation and in the instance, then there is a symmetric algorithm that achieves the same expected stopping time. Hence, we may assume that $\mathcal{A}$ is symmetric and capture the same set of possible stopping times. If $\mathcal{A}$ is not symmetric, we may form an algorithm $\mathcal{A}'$ by permuting the input, passing it to $\mathcal{A}$, getting the output of $\mathcal{A}$ on the permuted input, and then undoing the permutation before return an answer.

Since $W_{\pi(k)}(\Omega_t)$ and $W_{\pi(i^*)}(\Omega_t)$ are truthful on $\text{Sim}(\pi(\nu), \Omega_t)$ and $\text{Sim}(\pi(\nu_k'), \Omega_t)$ respectively, by Lemma 2 of [9], we have that

$$\begin{aligned}
\mathbb{P}_{\mathcal{A}, \pi(\nu)}&(W_{\pi(k)}(\Omega_t)) + \mathbb{P}_{\mathcal{A}, \pi(\nu_k')}(W_{\pi(i^*)}(\Omega_t)) \\
&\geq TV(\mathbb{P}_{\mathcal{A}, \pi(\nu)}, \mathbb{P}_{\mathcal{A}, \pi(\nu_k')}) - Q\left(KL(\text{Sim}(\pi(\nu), \Omega_t), \text{Sim}(\pi(\nu_k'), \Omega_t))\right)
\end{aligned}$$

for $Q(x) = \min\{1 - 1/2e^{-x}, \sqrt{x/2}\}$. Since $\mathcal{A}$ is symmetric, for any permutation $\pi$, we have that

$$\mathbb{P}_{\mathcal{A}, \pi(\nu)}(W_{\pi(k)}(\Omega_t)) + \mathbb{P}_{\mathcal{A}, \pi(\nu_k')}(W_{\pi(i^*)}(\Omega_t)) = \mathbb{P}_{\mathcal{A}, \nu}(W_k(\Omega_t)) + \mathbb{P}_{\mathcal{A}, \nu_k'}(W_{i^*}(\Omega_t)) = 2\mathbb{P}_{\mathcal{A}, \nu}(W_k(\Omega_t)).$$

The first equality holds since event $W_i$ depend only on the number of times that arm $i$ is pulled. Since $\mathcal{A}$ is symmetric, the probability that $\mathcal{A}$ pulls arm $i$ at most $\tau$ times on instance $\nu$ is equal to the probability that $\mathcal{A}$ pulls $\pi(i)$ at most $\tau$ times on instance $\pi(\nu)$. The second equality is true using symmetry as well since instances $\nu$ and $\nu_k'$ are themselves equal up to a permutation.

Combining the above with the previous bounds on the total variation and KL divergence, we have that

$$\mathbb{P}_{\mathcal{A}\times\nu}(N_k > \tau) = \mathbb{P}_{\mathcal{A}\times\nu}(W_k(\Omega_t)) \geq \frac{1}{2}\left(1 - 2\delta - \sqrt{\frac{\tau\Delta_{i^*,k}^2}{2}}\right)$$

Plugging in $\tau = 1/(2\Delta_{i^*,k}^2)$, we see that $\mathbb{P}_{\mathcal{A}\times\nu}(N_k > 1/(2\Delta_{i^*,k}^2)) \geq 1/2(1/2 - 2\delta)$. Since $k$ was arbitrary, we may repeat this argument for each $k$ in $[n]\backslash\{i^*\}$. Combining this with Markov's inequality, we see that

$$\begin{aligned}
\mathbb{E}_{\mathcal{A}\times\nu}\left[\sum_{k\neq i^*} N_k\right] &\geq \frac{1}{4}(1/2 - 2\delta)\sum_{k\neq i^*}\frac{1}{\Delta_{i^*,k}^2} \\
&> \frac{1}{16}\sum_{k\neq i^*}\frac{1}{\Delta_{i^*,k}^2}
\end{aligned}$$

where the final inequality follows from $\delta < 1/16$. The above holds for any $\delta$-PAC algorithm $\mathcal{A}$. $\square$

We now state our strong lower bound on the expected number of samples for any algorithm that can find an isolated arm.

**Theorem D.3.** *Fix $n$, $0 < \beta$, and $\delta < 1/16$ and consider a set $\nu$ of $n$ Gaussian random variables with variance 1 such that for a uniformly random chosen $i^* \in [n]$, $\rho_{i^*} = \mathbb{N}(\beta, 1)$ and $\rho_i = \mathcal{N}(\mu_i, 1)$ for $\mu_i \leq -\beta$ for all $i \neq i^*$. Let $\pi$ be a uniformly chosen permutation of $[n]$ and $\pi(\nu)$ be the permutation applied to instance $\nu$. Any $\delta$-PAC algorithm to detect $\pi(i^*)$ on $\pi(\nu)$ requires*

$$\frac{1}{16}\sum_{k\neq i^*}\frac{1}{\Delta_{i^*,k}^2} + \frac{1}{2\beta^2}\log(1/2.4\delta)$$

*samples in expectation, where the expectation is taken both over the randomness in the permutation, the randomness in $\pi(\nu)$, and any internal randomness to the algorithm.*

*Proof.* By Lemma D.1, arm $i^*$ must be sampled $\frac{1}{2\beta^2}\log(1/2.4\delta)$ times. By Theorem D.2, arms in $[n]\backslash\{i^*\}$ must collectively be sampled $\frac{1}{16}\sum_{k\neq i^*}\frac{1}{\Delta_{i^*,k}^2}$ times. Joining these two results gives the stated result. $\square$

## D.2 Step 2. Deciding if an instance is isolated

Next, we consider a composite hypothesis test that is related to the question of finding an isolated arm. As we will show, this test has the interesting property that the alternate hypothesis may be declared in significantly fewer samples than the null.

**Definition 5** ($\beta$-Isolated Hypothesis Test). *Fix $0 < \epsilon$ and $0 < \beta$. Consider an instance $\nu = \{\rho_1, \cdots, \rho_n\}$ where $\rho_i = \mathcal{N}(\mu_i, 1)$. By sampling individual distributions $\rho_i$, one wishes to perform the following composite hypothesis test:*

***Null Hypothesis $H_0$:*** *$\mu_i < -\beta$ for all $i \in [n]$.*

***Alternate Composite Hypothesis $H_1$:*** *$\exists i^* : \mu_{i^*} = \beta > 0$ and $\mu_i \leq -\beta$ for all $i \neq i^*$.*

For any instance $\nu$, we say "$H_1$ is true on $\nu$" if $\exists i^* : \mu_{i^*} = \beta > 0$ and otherwise we say "$H_0$ is true on $\nu$." Next, we bound the sample complexity of any algorithm to perform the $\beta$-isolated hypothesis test with probability at least $1 - \delta$ in the case that $H_0$ is true.

Figure 2 shows an two example instance. One where $H_0$ is true and one where $H_1$ is true.

**Lemma D.4.** *Fix $n$, $\beta$, and $\delta$ and consider a set $\nu$ of $n$ standard normal random variables where $H_0$ is true. Any algorithm to correctly declare $H_0$ in the $\beta$-isolated hypothesis test problem with probability at least $1 - \delta$ requires*

$$\sum_{i=1}^{n}\frac{2}{(\beta - \mu_i)^2}\log\left(\frac{1}{2.4\delta}\right)$$

*samples in expectation.*

*Proof.* Notice that for each $i \in [n]$, we may construct an alternate instance $\nu_i$ by changing the distribution of $\rho_i$ to be $\mathcal{N}(\beta, 1)$ and leaving others unchanged. On $\nu_i$, $H_1$ is instead true. To distinguish between $\nu$ and $\nu_i$, necessary to declare $H_0$ versus $H_1$, by Lemma 1 of [6], any $\delta$-PAC algorithm requires $\mathbb{E}_\nu[N_i] \geq \frac{2}{(\beta - \mu_i)^2} \log(1/2.4\delta)$ where $\mathbb{E}_\nu$ denotes expectation with respect to the instance $\nu$ and $N_i$ denotes the number of samples of arm $i$. Repeating this argument for each $i \in [n]$ gives the desired result. $\qquad\qquad\square$

To lower bound the expected sample complexity of any algorithm to perform the $\beta$-isolated hypothesis test in the setting where $H_1$ is true, we consider a reduction to the problem studied in Step 1, Section D.1. For the reduction to an algorithm that can find an isolated arm, we show that if there is an algorithm to declare $H_1$ in fewer than $O\left(\sum_{i=1}^n \frac{1}{\Delta_{i^*,k}^2}\right)$ samples, then one can design an algorithm akin to binary search that returns $i^*$ in fewer than $O\left(\sum_{i=1}^n \frac{1}{\Delta_{i^*,k}^2}\right)$ samples, contradicting Lemma D.2.

**Lemma D.5.** *Fix $n$, $\beta$, and $\delta < 1/16$. Let $\pi$ be a random permutation. Consider an instance $\nu$ where $H_1$ is true. In this setting, any algorithm to correctly declare $H_1$ in the $\beta$-Isolated Hypothesis Testing problem on $\pi(\nu)$ with probability at least $1 - \delta$ requires $\frac{1}{32}\sum_{j \neq i^*} \Delta_{i^*,k}^{-2}$ samples in expectation.*

*Proof.* Fix $\delta > 0$ and let $i^*$ denote the single distribution such that $\rho_{i^*} = \mathcal{N}(\beta, 1)$ where $\beta > 0$. In particular, only $i^*$ has a positive mean. Assume for contradiction that there is an algorithm $\mathcal{A}(\pi(\nu), \delta, \beta)$ that correctly declares $H_1$ on $\pi(\nu)$ in at most $\frac{1}{32}\sum_{k \neq i^*} \Delta_{i^*,k}^{-2}$ samples in expectation with probability at least $1 - \delta$ on any instance $\pi(\nu)$ of $n$ distributions if $H_1$ is true. Otherwise, if $H_0$ is true, assume that $\mathcal{A}$ correctly declares $H_0$ in an arbitrary number of samples in expectation, $N_{H_0}(\nu)$ lower bounded by Lemma D.4. As in the proof of Theorem D.2, if any algorithm $\mathcal{B}$ (not necessarily symmetric) achieves an expected stopping time $\tau$ where the expectation is taken over all the randomness in the permutation and in the instance, by Lemma 1 of [9], there is a symmetric algorithm that achieves the same expected stopping time. Hence, we may assume that $\mathcal{A}$ is symmetric and capture the same set of possible stopping times. For the remainder of this proof, we assume $\mathcal{A}$ is symmetric. Therefore, its expected complexity is independent of the permutation $\pi$. Without loss of generality, assume that $n = 2^k$ for some $k \in \mathbb{N}$. Otherwise, we may hallucinate $(2^{\lceil \log_2(n) \rceil} - n)$ normal distributions, $\mathcal{N}(-\beta, 1)$, and form an instance $\nu'$ comprised of these additional distribution and those in $\nu$. If so, anytime $\mathcal{A}$ requests a sample from a distribution in $\nu' \backslash \nu$, draw a sample from $\mathcal{N}(-\beta, 1)$ and pass it to $\mathcal{A}$, only tracking the number of samples drawn from $\nu$.

**Step a).** In what follows, we use $\mathcal{A}$ to develop a method for isolated-arm identification. To do so, we show that one may use $\mathcal{A}$ to perform binary search for the distribution $i^*$ such that $\rho_{i^*} = \mathcal{N}(\beta, 1)$ and this leads to a contradiction of Theorem D.2. For ease of exposition, for a set $\mathcal{S} \subset [n]$, let $\nu(\mathcal{S}) := \{i \in \mathcal{S} : \rho_i\}$, the subset of instance $\nu$ of distributions whose indices are in $\mathcal{S}$.

If $H_1$ is true on $\nu(S)$, by assumption, with probability at least $1 - \delta$, $\mathcal{A}$ correctly declares $H_1$ on $\nu(\mathcal{S})$ in at most $\frac{1}{32}\sum_{i \in \mathcal{S} \backslash \{i^*\}} \Delta_{i^*,k}^{-2}$ samples in expectation. Similarly, if $H_0$ is true on $\nu(\mathcal{S})$, the sample complexity is $N_{H_0}(\nu(\mathcal{S}))$ in expectation.

---

**Algorithm 3** Binary search for Isolated Arm Identification

---

**Require:** $\delta > 0, \beta > 0$, instance $\nu$ such that $H_1$ is true, algorithm $\mathcal{A}$
1: Let Low $= 1$ and High $= n$
2: **for** $i = 1, \cdots, \log_2(n)$ **do**
3:     1) Choose sets $\mathcal{S}_1, \mathcal{S}_2$ uniformly at random such that $\mathcal{S}_1 \cup \mathcal{S}_2 = \mathcal{S}$, $\mathcal{S}_1 \cap \mathcal{S}_2 = \emptyset$, and
    $\mathbb{P}(i \in \mathcal{S}_1) = \mathbb{P}(i \in \mathcal{S}_2)$ for all $i \in \mathcal{S}$
4:     2) In parallel, run $\mathcal{A}_1 = \mathcal{A}(\nu(\mathcal{S}_1), \beta, \delta/2\log_2(n))$ and $\mathcal{A}_2 = \mathcal{A}(\nu(\mathcal{S}_2), \beta, \delta/2\log_2(n))$
5:     3) If either terminates, terminate the other
6:     **if** $\mathcal{A}_1$ declares $H_1$ or $\mathcal{A}_2$ declares $H_0$ **then**
7:         $\mathcal{S} = \mathcal{S}_1$
8:     **else**
9:         $\mathcal{S} = \mathcal{S}_2$
    **return** $i^* \in \mathcal{S}$ (note: $|\mathcal{S}| = 1$ at this point)

---

In step 1, we choose 2 random subsets of $\mathcal{S}$, $\mathcal{S}_1$ and $\mathcal{S}_2$ that partition $\mathcal{S}$ such that each arm is assigned with equal probability to either $\mathcal{S}_1$ or $\mathcal{S}_2$ independently.

In step 2) if the loop, we separately run $\mathcal{A}$ in parallel on $\nu(\mathcal{S}_1)$ and $\nu(\mathcal{S}_2)$, each with failure probability $\delta/2\log(n)$. We alternate between passing a sample to $\mathcal{A}_1$ and to $\mathcal{A}_2$.

In Step 3), we terminate $\mathcal{A}_1$ if $\mathcal{A}_2$ terminates and vice versa. If, for instance, $\mathcal{A}_1$ terminates and declares $H_0$, we may infer $H_1$ on $\mathcal{S}_2$. Alternately, if $\mathcal{A}_1$ declares $H_1$ on $S_1$, we may infer $H_0$ on $S_2$ as there is a single positive mean, $\mu_{i^*}$. This process continues until $|\mathcal{S}_1| = |\mathcal{S}_2| = 1$, when there is a single distribution remaining in each. At this point, if $\mathcal{A}_1$ declares $H_1$, then the single arm $i \in \mathcal{S}_1$ is the positive mean $i^*$. Otherwise, the single arm $j \in \mathcal{S}_2$ is.

First, we show that this algorithm is correct with probability at least $1 - \delta$. The algorithm errs if and only if in any round $i$, either $\mathcal{A}_1$ or $\mathcal{A}_2$ errs, each with occurs with probability at most $\delta/2\log_2(n)$. Union bounding over the $\log_2(n)$ rounds, we see that the algorithm errs with probability at most $\delta$. For the remainder of the proof, we will assume that in no round does either $\mathcal{A}_1$ or $\mathcal{A}_2$ incorrectly declare $H_0$ or $H_1$ if the reverse is true for the given instances $\nu(\mathcal{S}_1)$ and $\nu(\mathcal{S}_2)$.

Now we introduce some notation for the remainder of this proof. As the set $\mathcal{S}$, $\mathcal{S}_1$, and $\mathcal{S}_2$ change in each round, let $\mathcal{S}(r)$, $\mathcal{S}_1(r)$, and $\mathcal{S}_2(r)$ denote their values in round $r$ for $r = 1, \cdots, \log_2(n)$. Define $\mathcal{A}_1(r)$ and $\mathcal{A}_2(r)$ similarly. We stop $\mathcal{A}_1(r)$ if $\mathcal{A}_2(r)$ terminates and vice versa.

Let $T_r$ denote the random variable of the total number of samples of drawn in round $r$. Let $T_{r,1}$ be the number of samples drawn by $\mathcal{A}_1(r)$, and $T_{r,2}$ be the number of samples drawn by $\mathcal{A}_2(r)$.

Next, define $\mathcal{S}^*(r)$ be the set in $\{\mathcal{S}_1(r), \mathcal{S}_2(r)\}$ that contains $i^*$, i.e. let $\mathcal{S}^*(r)$ denote $\mathcal{S}_1(r)$ if $i^* \in \mathcal{S}_1(r)$ and $S_2(r)$ otherwise for all $r$. Similarly, let $\mathcal{A}^*(r)$ denote $\mathcal{A}_1(r)$ if $i^* \in \mathcal{S}_1(r)$ and $\mathcal{A}_2(r)$ otherwise. Define $T_{r,\mathcal{A}^*}$ to be the random number of samples given to $\mathcal{A}^*(r)$. Hence, $T_{r,\mathcal{A}^*} = T_{r,1}$ or $T_{r,\mathcal{A}^*} = T_{r,2}$.

By Step 2, $\mathcal{A}_1(r)$ and $\mathcal{A}_2(r)$ are run in parallel. Hence, $T_{r,1} = T_{r,2}$ deterministically. Furthermore, $T_r = T_{r,1} + T_{r,2}$ deterministically. Therefore,

$$T_{r,\mathcal{A}^*} = \frac{T_{r,1} + T_{r,2}}{2} = \frac{T_r}{2}.$$

Therefore, the expected number of samples in round $r$, taken over the randomness in the set $\mathcal{S}^*(r)$, the randomness in the instance $\nu(\mathcal{S}^*(r))$, and any randomness in $\mathcal{A}^*(r)$ is

$$
\begin{aligned}
\mathbb{E}_{\mathcal{S}^*(1),\cdots,\mathcal{S}^*(r),\nu(\mathcal{S}^*(r))}[T_r] &= 2\mathbb{E}_{\mathcal{S}^*(1),\cdots,\mathcal{S}^*(r),\nu(\mathcal{S}^*(r))}\left[T_{r,\mathcal{A}^*}\right] \\
&= 2\mathbb{E}_{\mathcal{S}^*(1),\cdots,\mathcal{S}^*(r)}\left[\mathbb{E}_{\nu(\mathcal{S}^*(r))}\left[T_{r,\mathcal{A}^*}|S^*(r)\right]\right] \\
&= 2\mathbb{E}_{\mathcal{S}^*(1),\cdots,\mathcal{S}^*(r)}\left[\min\left(\frac{1}{32}\sum_{j\in\mathcal{S}^*(r)\setminus\{i^*\}}\frac{1}{\Delta_{i^*,j}^2}, N_{H_0}(\nu(\mathcal{S}^*(r)^c))\right)\right]
\end{aligned}
$$

$$\leq 2\mathbb{E}_{\mathcal{S}^*(1),\cdots,\mathcal{S}^*(r)}\left[\frac{1}{32}\sum_{j\in\mathcal{S}^*(r)\setminus\{i^*\}}\frac{1}{\Delta_{i^*,j}^2}\right]$$

$$= 2\mathbb{E}_{\mathcal{S}^*(1),\cdots,\mathcal{S}^*(r-1)}\left[\mathbb{E}_{\mathcal{S}^*(r)}\left[\frac{1}{32}\sum_{j\neq i^*}\mathbb{1}[j\in\mathcal{S}^*(r)]\frac{1}{\Delta_{i^*,j}^2}\bigg|\mathcal{S}^*(r-1)\right]\right]$$

$$= 2\mathbb{E}_{\mathcal{S}^*(1),\cdots,\mathcal{S}^*(r-1)}\left[\frac{1}{32}\cdot\left(\frac{1}{2}\right)\sum_{j\neq i^*}\mathbb{1}[j\in\mathcal{S}^*(r-1)]\frac{1}{\Delta_{i^*,j}^2}\right]$$

$$\vdots$$

$$= 2\mathbb{E}_{\mathcal{S}^*(1)}\left[\frac{1}{32}\cdot\left(\frac{1}{2}\right)^{r-1}\sum_{j\neq i^*}\mathbb{1}[j\in\mathcal{S}^*(1)]\frac{1}{\Delta_{i^*,j}^2}\right]$$

$$= \frac{1}{16}\cdot\left(\frac{1}{2}\right)^r\sum_{j\neq i^*}\frac{1}{\Delta_{i^*,j}^2}.$$

Therefore, we may bound the expected total number of samples for the above binary search algorithm to return $i^*$ as

$$\mathbb{E}\left[\sum_{r=1}^{\lceil\log_2(n)\rceil}T_r\right] = \sum_{r=1}^{\log_2(n)}\mathbb{E}[T_r]\leq\frac{1}{16}\sum_{j\neq i^*}\frac{1}{\Delta_{i^*,j}^2}\sum_{r=1}^{\log_2(n)}\left(\frac{1}{2}\right)^r\leq\frac{1}{16}\sum_{j\neq i^*}\frac{1}{\Delta_{i^*,j}^2}.$$

However, this contradicts Theorem D.2 for $\delta < 1/16$. Hence no such algorithm $\mathcal{A}$ exists and any algorithm to declare $H_1$ on instance $\nu$ requires at least $\frac{1}{32}\sum_{j\neq i^*}\frac{1}{\Delta_{i^*,j}^2}$ samples in expectation. $\qquad\square$

**Theorem D.6.** *Fix $n$, $\beta$, and $\delta < 1/16$ and consider an instance $\nu$. If $H_0$ is true on $\nu$, any algorithm requires at least*

$$\sum_{j=1}^{n}\frac{2}{(\beta-\mu_j)^2}\log\left(\frac{1}{2.4\delta}\right)$$

*samples in expectation to perform the $\beta$-isolated Hypothesis Test. If $H_1$ is true on $\nu$, any algorithm requires at least*

$$\frac{1}{4\beta^2}\log\left(\frac{1}{2.4\delta}\right) + \frac{1}{64}\sum_{j\neq i^*}\frac{1}{\Delta_{i^*,j}^2}$$

*samples in expectation to perform the $\beta$-isolated Hypothesis Test.*

*Proof.* If $H_0$ is true for $\nu$, the result follows immediately from Lemma D.4. Otherwise, assume $H_1$ is true for $\nu$ and let $i^*$ be the single distribution such that $\rho_{i^*} = \mathcal{N}(\beta, 1)$. Similar to the proof of Lemma D.1, one may consider an alternate instance $\nu'$ where $\rho_{i^*} = \mathcal{N}(-\beta, 1)$ and all other distributions are unchanged. Therefore, on $\nu'$, $H_0$ is true and any algorithm that is correct with probability at least $1 - \delta$ must be able to distinguish between these two instances. By Lemma 1 of [6], any algorithm that is correct with probability at least $1 - \delta$ must therefore sample $i^*$ $\frac{1}{2\beta^2}\log\left(\frac{1}{2.4\delta}\right)$ times in expectation. Combining this with the result of Lemma D.5, any algorithm that is correct with probability at least $1 - \delta$ must collect at least

$$\max\left\{\frac{1}{32}\sum_{j\neq i^*}\frac{1}{\Delta_{i^*,j}^2}, \frac{1}{2\beta^2}\log\left(\frac{1}{2.4\delta}\right)\right\}\geq\frac{1}{4\beta^2}\log\left(\frac{1}{2.4\delta}\right) + \frac{1}{64}\sum_{j\neq i^*}\frac{1}{\Delta_{i^*,j}^2}$$

samples in expectation. $\qquad\square$

## D.3 Step 3: Reducing ALL-$\epsilon$ to isolated instance detection

In this section, we prove that for any instance $\nu$ for ALL-$\epsilon$ such that $|G_{\beta_\epsilon}(\nu)| = 1$ requires at least $O\left(\sum_{2=1}^{n} \frac{1}{\Delta_i^2}\right)$ samples in expectation. To do so, we prove a reduction from finding all $\epsilon$-good arms to the $\beta$-Isolated Hypothesis Testing. In particular, we show that if one has a generic method to find all $\epsilon$-good arms (with slack $\gamma = 0$), then one may use this to develop a method to perform the $\beta$-Isolated Hypothesis Test. Therefore, lower bounds on the this test apply to the problem of finding all $\epsilon$-good arms as well.

**Lemma D.7.** *Fix $\delta \leq 1/16$, $n \geq 2/\delta$, $\epsilon > 0$, $\beta \in (0, \epsilon/2)$. Let $\nu$ be an instance of $n$ arms such that the $i^{th}$ is distributed as $\mathcal{N}(\mu_i, 1)$, $|G_{2\beta_\epsilon}| = 1$, and there exists an arm in $G_\epsilon^c$ such that $\mu_1 - \epsilon - \mu_i = \beta$. Select a permutation $\pi : [n] \to [n]$ uniformly from the set of $n!$ permutations, and consider the permuted instance $\pi(\nu)$. Any algorithm that returns $G_\epsilon(\pi(\nu))$ on $\pi(\nu)$ with correctly probability at least $1 - \delta$ requires at least*

$$\frac{1}{64} \sum_{i=2}^{n} \frac{1}{\Delta_i^2} + \frac{1}{4\beta_\epsilon^2} \log\left(\frac{1}{2.4\delta}\right)$$

*samples in expectation, where the expectation is taken jointly over the randomness in $\nu$ and $\pi$.*

*Proof.* Fix $0 < \delta < 1/16$, $n > 2/\delta$, $\epsilon > 0$, $0 < \beta < \epsilon/2$, and an arbitrary constant $c \in \mathbb{R}$. Consider a given instance $\nu = \{\rho_1, \cdots, \rho_n\}$ such that $\mu_1 \in \{-\beta, \beta\}$, and $\mu_2, \cdots, \mu_n < -\beta$. We wish to perform the $\beta$-isolated hypothesis test on $\pi(\nu)$. Assume for contradiction that there exists a generic algorithm $\mathcal{A}(\nu', \epsilon, \delta)$ such that if given a generic instance $\nu'$ where $|G_{2\beta_\epsilon}(\nu')| = 1$, it returns $G_\epsilon(\nu')$ with probability at least $1 - \delta$ in at most $\frac{1}{64} \sum_{i=2}^{n} \frac{1}{\Delta_i^2}$ samples where $\mu_1'$ is the largest mean in $\nu'$. Consider the following procedure that uses $\mathcal{A}$ to perform the hypothesis test:

---
**Algorithm 4** Using All-$\epsilon$ for $\beta$-isolated hypothesis test

---
**Require:** $\delta > 0$, $\epsilon > 0$, $0 < \beta$, instance $\pi(\nu)$, constant $c$, and algorithm $\mathcal{A}$
1: **Step 1:** Choose an index $\hat{i} \in [n]$ uniformly
2: **Step 2:** Let $\nu'$ be the instance

$$\nu' = \begin{cases} \rho_{\pi(i)} + c & \text{if } i \neq \hat{i} \\ \mathcal{N}(c - \epsilon, 1) & \text{if } i = \hat{i} \end{cases}$$

3: **Step 3:** $G = \mathcal{A}(\nu', \epsilon, \delta/2)$
4: **if** $\hat{i} \in G$ **then**:
5:     Declare $H_0$ and terminate
6: **else**
7:     Declare $H_1$ and terminate

---

Note that as $n \geq 2/\delta$, $\mathbb{P}(\hat{i} = \pi(1)) \leq \delta/2$. The method replaces $\rho_{\hat{i}}$ with $\mathcal{N}(c - \epsilon, 1)$. All other means $\mu_i$ are shifted up by $c$. The test then runs $\mathcal{A}$ on this new instance $\nu'$ with failure probability $\delta/2$. If $H_0$ is true on $\pi(\nu)$, all distributions have means less than $-\beta$, and $\hat{i}$ therefore is $\epsilon$-good on instance $\nu'$. If $H_1$ is true on $\pi(\nu)$, then $\rho_{\pi(1)} = \mathcal{N}(\beta, 1)$ and $\hat{i}$ is not $\epsilon$-good on instance $\nu'$. This method correctly performs the test if $\hat{i} \neq \pi(1)$ and $\mathcal{A}$ does not fail, the joint event of which occurs with probability at most $2\delta$. Therefore, this test is correct with probability at least $1 - \delta$.

Let $\mathcal{T}_\mathcal{A}(\nu')$ denote the random variable of the number of samples drawn by $\mathcal{A}$ on instance $\nu'$ and let $T$ denote the random variable of the total number of samples drawn by this procedure before it terminates and declares $H_0$ or $H_1$ on $\nu'$. Therefore, $\mathbb{E}_{\pi,\nu}[T] = \mathbb{E}_{\pi,\nu}[\mathcal{T}_\mathcal{A}(\nu')]$.

By Lemma 1 of [9], averaging over all permutations is equivalent to first permuting the instance $\nu$ and then passing it to $\mathcal{A}$ and undoing the permutation when returning the answer. We therefore assume that $\mathcal{A}$ is *symmetric* in that its expected sample complexity of $\mathcal{A}$ is invariant to the permutation $\pi$. Otherwise, we may use $\mathcal{A}$ to form a symmetric algorithm. Therefore, $\mathbb{E}_{\pi,\nu}[T] = \mathbb{E}_{\pi,\nu}[\mathcal{T}_\mathcal{A}(\nu')] =$

$\mathbb{E}_\nu[\mathcal{T}_\mathcal{A}(\nu')]$. By Theorem D.6, if $H_1$ is true,

$$\mathbb{E}_{\pi,\nu}[T] \geq \frac{1}{64} \sum_{i=2}^n \frac{1}{\Delta_i^2} + \frac{1}{4\beta^2} \log\left(\frac{1}{2.4\delta}\right).$$

Hence,

$$\mathbb{E}_\nu[\mathcal{T}_\mathcal{A}(\nu')] \geq \frac{1}{64} \sum_{i=2}^n \frac{1}{\Delta_i^2} + \frac{1}{4\beta^2} \log\left(\frac{1}{2.4\delta}\right).$$

Lastly, as the constant $c$ was chosen arbitrarily, and $\beta$ is an number in $(0, \epsilon/2)$ this argument applies to any ALL-$\epsilon$ instance $\nu'$ such that $\beta_\epsilon \in (0, \epsilon/2)$ and $|G_{2\beta_\epsilon}| = 1$ for n appropriate choice of $c$. $\qquad \square$

With the above proof, we restate the following moderate confidence lower bound on the sample complexity of returning all $\epsilon$-good stated in Section 4. In particular, this bound highlights *moderate confidence* terms that are independent of $\delta$. Moderate confidence terms have been studied in works such as [9, 25]. Despite being independent of $\delta$, these terms can have important effects in real world scenarios. The following bound demonstrates that there are instances for which moderate confidence terms are necessary for finding all $\epsilon$-good arms. Moderate confidence terms likewise appear in the upper bound of the complexity of FAREAST, Theorem 4.2.

**Theorem D.8.** *Fix $\delta \leq 1/16$, $n \geq 2/\delta$, and $\epsilon > 0$. Let $\nu$ be an instance of $n$ arms such that the $i^{th}$ is distributed as $\mathcal{N}(\mu_i, 1)$, $|G_{2\beta_\epsilon}| = 1$, and $\beta_\epsilon < \epsilon/2$. Select a permutation $\pi : [n] \to [n]$ uniformly from the set of $n!$ permutations, and consider the permuted instance $\pi(\nu)$. Any algorithm that returns $G_\epsilon(\pi(\nu))$ on $\pi(\nu)$ with correctly probability at least $1 - \delta$ requires at least*

$$c_2 \sum_{i=1}^n \max\left(\frac{1}{(\mu_1 - \epsilon - \mu_i)^2}, \frac{1}{(\mu_1 + \alpha_\epsilon - \mu_i)^2}\right) \log\left(\frac{1}{2.4\delta}\right) + c_2 \sum_{i=1}^n \frac{1}{(\mu_1 + \beta_\epsilon - \mu_i)^2}$$

*samples in expectation over the randomness in $\nu$ and $\pi$ for a universal constant $c_2$.*

*Proof.* We may equivalently consider the same instance with all means shifted down by $\epsilon - 2\beta$ since a method for that instance could be used to return all $\epsilon$ good arms in the stated instance. By Lemma D.7, $c_2 \frac{n}{\beta^2}$ samples are necessary in expectation. By Theorem 2.1,

$$2 \sum_{i=1}^n \max\left(\frac{1}{(\mu_1 - \epsilon - \mu_i)^2}, \frac{1}{(\mu_1 + \alpha_\epsilon - \mu_i)^2}\right) \log\left(\frac{1}{2.4\delta}\right)$$

samples are necessary in expectation. By Lemma D.7,

$$\frac{1}{64} \sum_{i=2}^n \frac{1}{\Delta_i^2} + \frac{1}{4\beta_\epsilon^2} \log\left(\frac{1}{2.4\delta}\right) \geq \frac{1}{64} \sum_{i=2}^n \frac{1}{(\mu_1 + \beta_\epsilon - \mu_i)^2} + \frac{1}{4\beta_\epsilon^2} \log\left(\frac{1}{2.4\delta}\right)$$

$$\geq \frac{1}{64} \sum_{i=1}^n \frac{1}{(\mu_1 + \beta_\epsilon - \mu_i)^2}$$

samples are necessary in expectation taken over the randomness in the permutation and in the instance. In particular, the maximum and therefore the average is a valid bound. Therefore, any algorithm requires

$$\sum_{i=1}^n \max\left(\frac{1}{(\mu_1 - \epsilon - \mu_i)^2}, \frac{1}{(\mu_1 + \alpha_\epsilon - \mu_i)^2}\right) \log\left(\frac{1}{2.4\delta}\right) + \frac{1}{128} \sum_{i=1}^n \frac{1}{(\mu_1 + \beta_\epsilon - \mu_i)^2}$$

samples in expectation. $\qquad \square$

# E  An optimal method for finding all additive and multiplicative $\epsilon$-good arms

## E.1  The FAREAST Algorithm

Below, we present an algorithm called FAREAST (**F**ast **A**rm **R**emoval **E**limination **A**lgorithm for a **S**ampled **T**hreshold) that achieves the lower bound when $\gamma = 0$. Similar to $(\text{ST})^2$, it relies on

anytime-correct confidence widths, $C_\delta(t) := \sqrt{\frac{4\log(\log_2(2t)/\delta)}{t}}$. The algorithm proceeds in rounds, and creates a filter for *good* arms and a filter for *bad* arms. The good filter detects arms in $G_\epsilon$ of $M_\epsilon$ and adds them to a set $G_k$. Similarly, the bad filter detects arms in $G_\epsilon^c$ or $M_\epsilon^c$ and adds them to a set $B_k$. At any given time, we may represent the set of arms that have *not* been declared as either in $G_\epsilon/M_\epsilon$ or $G_\epsilon^c/M_\epsilon^c$ as $(G_k \cup B_k)^c$. In either the additive or multiplicative case, the algorithm terminates when it can certify that $G_\epsilon \subset G_k$ and $G_k \cap G_{\epsilon+\gamma}^c = \emptyset$ or $M_\epsilon \subset G_k$ and $G_k \cap M_{\epsilon+\gamma}^c = \emptyset$, respectively– i.e., when $G_k$ contains all additive or multiplicative $\epsilon$-good arms and none worse than $(\epsilon + \gamma)$-good.

In each round, the bad filter uses MedianElimination [12] which given an instance $\nu$, a value of $\epsilon$, and a failure probability $\kappa$, returns an $\epsilon$-good arm with probability at least $1 - \kappa$. In the $k^{\text{th}}$ round, for an arm $i$ in $(G_k \cup B_k)^c$, the bad filter uses MedianElimination to find a $2^{-k}$ good arm $i_k$ with failure probability $\kappa = O(1)$ and then samples both arms $i$ and $i_k$ $\tilde{O}(2^{2k}\log(1/\delta))$ times. Let $\hat{\mu}_i$ and $\hat{\mu}_{i_k}$ denote the empirical means. For instance, in the additive case, if $\hat{\mu}_{i_k} - \hat{\mu}_i \geq \epsilon + 2^{-k+1}$, we may declare that $i \in G_\epsilon^c$, and the bad filter adds $i$ to the set $B_k$. This allows the bad filter to commit to a single arm and sample it sufficiently to remove arms in $G_\epsilon^c$.

The good filter is a simple elimination scheme. It maintains an upper bound $U_t$ and lower bound $L_t$ on $\mu_1 - \epsilon$. If an arm's upper bound drops below $L_t$ (line 20), the good filter eliminates that arm, otherwise, if an arm's lower bound rises above $U_t$ (19), the good filter adds the arm to $G_k$, but only eliminates this arm if its upper bound falls below the highest lower bound. This ensures that $\mu_1$ is never eliminated and $U_t$ and $L_t$ are always valid bounds This scheme works as an independent algorithm and achieves the sample complexity as $(\text{ST})^2$, though worse empirical performance. We analyze this method in Appendix E.5. Indeed, this gives an additional high probability guarantee on the number of samples drawn by FAREAST in both the additive and multiplicative regimes. As the sampling is split across rounds, the good filter always samples the least sampled arm, breaking ties arbitrarily. The number of samples given to the good filter in each round is such that both filters receive identically many samples. Note that this is a random quantity since the number of arms in $(G_k \cup B_k)^c$ in round $k$ is random. Despite this, we prove a lower bound on the number of samples drawn per round which ensures the Good Filter always receives a positive number of samples in each round. Note that by design elimination only occurs when all arms in the active set have received equal numbers of samples. This is crucial as it prevents the good filter from over-sampling bad arms and vice versa. In our proof, we show that in some round, unknown to the algorithm, $G_k = G_\epsilon$, ie all good arms have been found, and this takes no more than $O\left(\sum_{i=1}^n \max\left\{(\mu_1 - \epsilon - \mu_i)^{-2}, (\mu_1 + \alpha_\epsilon - \mu_i)^{-2}\right\}\log(n/\delta)\right)$ samples, matching the lower bound.

The algorithm stops on either of three conditions. First, if $G_k \cup B_k = [n]$, every arm has been declared as either in $G_\epsilon$ or $G_\epsilon^c$ (or $M_\epsilon$ or $M_\epsilon^c$). Second, if $\mathcal{A} \subset G_k$, the Good Filter has found every arm in $G_\epsilon$ and FAREAST can terminate. This is the same stopping condition as EAST itself. In either case, FAREAST returns the set $G_k = G_\epsilon$ exactly. The third condition allows for $\gamma$ slack. The good filter maintains upper and lower bounds $U_t$ and $L_t$ on the threshold in both the additive an multiplicative cases. In the additive case, if $U_t - L_t < \gamma/2$, then all arms in $G_{\epsilon+\gamma}^c$ have been added to $B_k$, and FAREAST may return $G_k \cup \mathcal{A}$. The condition for the multiplicative case is similar, though slightly more complicated. Throughout, we will use red text to denote pieces specific to the additive case and blue text to denote pieces specific to the multiplicative case.

```
1   FAREAST
2   Input: ε, δ, Instance ν, slack γ ≥ 0. If multiplicative, ε ∈ (0, 1/2]
3   Let G₀ = ∅ be the set of arms declared as good and B₀ = ∅ the set of arms declared as bad.
4   Let 𝒜 = [n] be the active set, Nᵢ = 0 track the total number of samples of arm i by the Good Filter.
5   Let t = 0 denote the total number of times that line 19 is true in the Good Filter.
6   Let C_{δ/2n}(t) be an anytime δ/2n-correct confidence width on t samples.
7   Let H_ME(n, ε, κ) = ⌈c' (n/ε²) log(1/κ)⌉ be the complexity of MedianElimination.
8   for k = 1, 2, ⋯
9       Let δ_k = δ/2k², τ_k = ⌈2^{2k+3} log(8n/δ_k)⌉, Initialize G_k = G_{k-1} and B_k = B_{k-1}
10      // Bad Filter: find bad arms in G_ε^c or M_ε^c
11      Let i_k = MedianElimination(ν, 2^{-k}, 1/16), sample i_k τ_k times, and compute μ̂_{i_k}
12      for i ∉ G_{k-1} ∪ B_{k-1}:
13          Sample μ_i τ_k times and compute μ̂_i
14          If μ̂_{i_k} - μ̂_i ≥ ε + 2^{-k+1} or (1-ε)μ̂_{i_k} - μ̂_i > 2^{-(k+1)}(2-ε):
15              Add i to B_k
16      // Good Filter: find good arms in G_ε or M_ε
17      for s = 1, ⋯, H_ME(n, 2^{-k}, 1/16) + τ_k · (|(G_{k-1} ∪ B_{k-1})^c| + 1):
18          Pull arm I_s ∈ arg min_{j∈𝒜}{N_j} and set N_{I_s} ← N_{I_s} + 1.
19          if min_{j∈𝒜}{N_j} = max_{j∈𝒜}{N_j}:
20              t = t + 1
21              For i ∈ 𝒜 denote μ̂_i(t) the average of the first t samples of arm i.
22              Let U_t = max_{j∈𝒜} μ̂_i(t) + C_{δ/2n}(t) - ε or U_t = (1-ε)(max_{j∈𝒜} μ̂_i(t) + C_{δ/2n}(t))
23              Let L_t = max_{j∈𝒜} μ̂_i(t) - C_{δ/2n}(t) - ε or L_t = (1-ε)(max_{j∈𝒜} μ̂_i(t) - C_{δ/2n}(t))
24              for i ∈ 𝒜:
25                  if μ̂_i(t) - C_{δ/2n}(t) ≥ U_t:
26                      Add i to G_k
27                  if μ̂_i(t) + C_{δ/2n}(t) ≤ L_t: // Bad arms are removed from 𝒜
28                      Remove i from 𝒜
29                  if i ∈ G_k and μ̂_i(t) + C_{δ/2n}(t) ≤ max_{j∈𝒜} μ̂(t) - C_{δ/2n}(t): // Good arms removed
30                      Remove i from 𝒜
31              If 𝒜 ⊂ G_k or G_k ∪ B_k = [n]:
32                  Output: the set G_k // Stopping condition for returning G_ε exactly.
33              If U_t - L_t < ½γ or U_t - L_t < (γ/(2-ε))L_t:
34                  Output: the set 𝒜 ∪ G_k // Stopping condition for γ > 0.
```

**Remark 1.** *Note that the active set 𝒜 defined in line 4 of FAREAST is only used and updated internally by the Good Filter. In particular, it is not necessarily true that $(G_k \cup B_k)^c = \mathcal{A}$. Furthermore, a bad arm $i \in G_\epsilon^c$ maybe removed from 𝒜 even though it is not in $B_k$ and vice versa as the Good Filter only seeks to detect good arms in $G_\epsilon$ and the Bad Filter only seeks to detect arms in $G_\epsilon^c$. The same is true in the multiplicative case.*

**Remark 2.** *It is possible that when the loop in line 17 finishes in any given round, some arms in 𝒜 have received more samples than others. Because $I_s \in \arg\min_{j\in\mathcal{A}}\{N_j\}$ in line 18, this difference is no more than 1, and the arms with fewer samples are the first to be sampled in the next round. The condition on line 19 ensures that all arms have equal numbers of samples by the Good Filter (e.g., the $N_i$'s) when the Good Filter identifies good arms or eliminates arms from 𝒜.*

Now, we restate Theorem 4.2 for reference.

**Theorem E.1.** *Fix $0 < \epsilon$, $0 < \delta < 1/8$, slack $\gamma \in [0, 8]$ and an instance $\nu$ of $n$ arms such that $\max(\Delta_i, |\epsilon - \Delta_i|) \le 8$ for all $i$. There exists an event $E$ such that $\mathbb{P}(E) \ge 1 - \delta$, and on $E$, FAREAST terminates and returns $G$ such that $G_\epsilon \subset G \subset G_{\epsilon+\gamma}$ in at most*

$$c_4 \sum_{i=1}^{n} \min \left\{ \max \left\{ \frac{1}{(\mu_1 - \epsilon - \mu_i)^2} \log \left( \frac{n}{\delta} \log_2 \left( \frac{n}{\delta(\mu_1 - \epsilon - \mu_i)^2} \right) \right), \right. \right.$$

$$\frac{1}{(\mu_1 + \alpha_\epsilon - \mu_i)^2} \log \left( \frac{n}{\delta} \log_2 \left( \frac{n}{\delta(\mu_1 + \alpha_\epsilon - \mu_i)^2} \right) \right),$$

$$\left. \frac{1}{(\mu_1 + \beta_\epsilon - \mu_i)^2} \log \left( \frac{n}{\delta} \log_2 \left( \frac{n}{\delta(\mu_1 + \beta_\epsilon - \mu_i)^2} \right) \right) \right\},$$

$$\left. \frac{1}{\gamma^2} \log \left( \frac{n}{\delta} \log_2 \left( \frac{n}{\delta\gamma^2} \right) \right) \right\}$$

*samples for a constant $c_4$. Furthermore*

$$\mathbb{E}[\mathbb{1}_E T] \leq c_3 \sum_{i \in G_\epsilon} \max\left\{ \frac{1}{(\mu_1 - \epsilon - \mu_i)^2} \log\left(\frac{n}{\delta} \log_2\left(\frac{n}{\delta(\mu_1 - \epsilon - \mu_i)^2}\right)\right),\right.$$

$$\left. \frac{1}{(\mu_1 + \alpha_\epsilon - \mu_i)^2} \log\left(\frac{n}{\delta} \log_2\left(\frac{n}{\delta(\mu_1 + \alpha_\epsilon - \mu_i)^2}\right)\right)\right\}$$

$$+ c_3 \sum_{i \in G_\epsilon^c} \frac{n}{(\mu_1 - \epsilon - \mu_i)^2} + \frac{1}{(\mu_1 - \epsilon - \mu_i)^2} \log\left(\frac{n}{\delta} \log_2\left(\frac{n}{\delta(\mu_1 - \epsilon - \mu_i)^2}\right)\right)$$

*for a sufficiently large constant $c_3$ where $T$ denotes the number of samples.*

Next, we present a theorem bounding the sample complexity of `FAREAST` for returning multiplicative $\epsilon$-good arms. Recall that $\tilde{\alpha}_\epsilon := \min_{i \in M_\epsilon} \mu_i - (1 - \epsilon)\mu_1$ and $\tilde{\beta}_\epsilon := \min_{i \in M_\epsilon^c}(1 - \epsilon)\mu_1 - \mu_i$, the distance for the smallest good arm and best arm that is not good to the threshold $(1 - \epsilon)\mu_1$.

**Theorem E.2.** *Fix $\epsilon \in (0, 1/2]$, $\gamma \in [0, \min(1, 6/\mu_1))$, $0 < \delta < 1/8$ and an instance $\nu$ of $n$ arms such that $\max(\Delta_i, |\epsilon\mu_1 - \Delta_i|) \leq 6$. Assume that the highest mean is non-negative, i.e., $\mu_1 \geq 0$. There exists an event $E$ such that $\mathbb{P}(E) \geq 1 - \delta$, and on $E$, `FAREAST` terminates and returns $G$ such that $M_\epsilon \subset G \subset M_{\epsilon+\gamma}$ in at most*

$$c_5 \sum_{i=1}^n \min\left\{ \max\left\{ \frac{1}{((1-\epsilon)\mu_1 - \mu_i)^2} \log\left(\frac{n}{\delta} \log_2\left(\frac{n}{\delta((1-\epsilon)\mu_1 - \mu_i)^2}\right)\right),\right.\right.$$

$$\frac{1}{(\mu_1 + \frac{\tilde{\alpha}_\epsilon}{1-\epsilon} - \mu_i)^2} \log\left(\frac{n}{\delta} \log_2\left(\frac{n}{\delta(\mu_1 + \frac{\tilde{\alpha}_\epsilon}{1-\epsilon})^2}\right)\right),$$

$$\left.\left. \frac{1}{(\mu_1 + \frac{\tilde{\beta}_\epsilon}{1-\epsilon} - \mu_i)^2} \log\left(\frac{n}{\delta} \log_2\left(\frac{n}{\delta(\mu_1 + \frac{\tilde{\beta}_\epsilon}{1-\epsilon} - \mu_i)^2}\right)\right)\right\},\right.$$

$$\left. \frac{(1-\epsilon+\gamma)^2}{\gamma^2\mu_1^2} \log\left(\frac{n}{\delta} \log_2\left(\frac{(1-\epsilon+\gamma)^2 n}{\delta\gamma^2\mu_1^2}\right)\right)\right\}$$

*samples for a sufficiently large constant $c_5$. Furthermore*

$$\mathbb{E}[\mathbb{1}_E T] \leq c_6 \sum_{i=1}^n \max\left\{ \frac{1}{((1-\epsilon)\mu_1 - \mu_i)^2} \log\left(\frac{n}{\delta} \log_2\left(\frac{n}{\delta((1-\epsilon)\mu_1 - \mu_i)^2}\right)\right),\right.$$

$$\left. \frac{1}{\left(\mu_1 + \frac{\tilde{\alpha}_\epsilon}{1-\epsilon} - \mu_i\right)^2} \log\left(\frac{n}{\delta} \log_2\left(\frac{n}{\delta\left(\mu_1 + \frac{\tilde{\alpha}_\epsilon}{1-\epsilon} - \mu_i\right)^2}\right)\right)\right\}$$

$$+ c_6 \sum_{i \in M_\epsilon^c} \frac{n}{((1-\epsilon)\mu_1 - \mu_i)^2}$$

*for a sufficiently large constant $c_6$, where $T$ denotes the number of samples.*

### E.2  Key ideas of the proof

The proof revolves around a central idea: there is an event in unknown round $K_{\text{Good}}$ in which the final arm from $G_\epsilon$ or $M_\epsilon$ is added to $G_k$. We may split the total number of samples drawn as the number taken through round $K_{\text{Good}}$ and the number taken from $K_{\text{Good}} + 1$ until termination if the algorithm does not terminate in round $K_{\text{Good}}$. Note that the Good filter and Bad filter are given the same number of samples in each round. The proof of `FAREAST` in the multiplicative regime is similar and deferred to Appendix E.4.

We begin by bounding the number of samples given to the Good filter when this event occurs that $G_k = G_\epsilon$. Next, since this happens at a random time within round $K_{\text{Good}}$, we bound the total number of additional samples in this round. Collectively, this gives us control over the number of samples drawn through round $K_{\text{Good}}$.

Next, we bound the number of samples from $K_{\text{Good}} + 1$ until termination. To do so, we analyze the expected number of samples drawn by the Bad filter before all arms in $G_\epsilon^c$ have been added to $B_k$. The total number of samples from $K_{\text{Good}} + 1$ until termination is no worse than twice this value. The proof is split into 12 steps and logically are organized as follows:

1. Step 0: We show that $G_k \subset G_\epsilon$ and $B_k \subset G_\epsilon^c$. In particular, this is implies that $G_k \cup B_k = [n] \implies G_k = G_\epsilon$ so FAREAST terminates correctly.

2. Step 1: We split the total number of samples drawn by FAREAST into two sums that we will control individually.

3. Steps 2-4: We control the number of samples given to the Good filter before $G_k = G_\epsilon$.

4. Steps 5-6: Using the result of steps 2-4, we bound the total number of samples through round $K_{\text{Good}}$

5. Steps 7-8: We use the result of step 6 to bound the total *expected* number of samples drawn by FAREAST, simplifying slightly in the process.

6. Step 9: We bound the number of samples that the Bad filter draws in adding a single bad arm to $B_k$.

7. Step 10: Repeating the argument in step 9, for every $i \in G_\epsilon^c$, we bound the total number of samples from round $K_{\text{Good}} + 1$ until termination. We finish by combining the bound on the number of samples drawn through $K_{\text{Good}}$ with the bound from $K_{\text{Good}} + 1$ until termination. This controls the expected sample complexity of FAREAST.

8. Step 11: We provide a high probability bound on the sample complexity of FAREAST.

### E.3 Proof of Theorem 4.2, FAREAST in the additive regime

*Proof.* **Notation for the proof:** Throughout, recall $\Delta_i = \mu_1 - \mu_i$. Recall that $t$ counts the number of times the conditional in line 19 is true. By Line 19 of FAREAST, all arms in $\mathcal{A}$ have received $t$ samples when the loop in line 23 is executed for the $t^{\text{th}}$ time. Within any round $k$, let $\mathcal{A}(t)$ and $G_k(t)$ denote the sets $\mathcal{A}$ and $G_k$ at this time since both sets can change in lines 27 and 29 and 25 respectively. Let $t_k$ denote the maximum value of $t$ in round $k$. By Lines 18 and 19 of FAREAST, the total number of samples given to the good filter when the conditional in line 19 is true for the $t^{\text{th}}$ time is $\sum_{s=1}^{t} |\mathcal{A}(s)|$.

For $i \in G_\epsilon$, let $T_i$ denote the random variable of the number of times arm $i$ is sampled by the good filter before it is added to $G_k$ in Line 25. For $i \in G_\epsilon^c$, let $T_i$ denote the random variable of the number of times arm $i$ is sampled by the good filter before it is removed from $\mathcal{A}$ in Line 27. For any arm $i$, let $T_i'$ denote the random variable of the of the number of times $i$ is sampled by the good filter before $\hat{\mu}_i(t) + C_{\delta/2n}(t) \leq \max_{j \in \mathcal{A}} \hat{\mu}_j(t) - C_{\delta/2n}(t)$. Lastly, let $T_\gamma$ denote the random variable of the number of times any arm is sampled by the good filter before $U_t - L_t < \gamma/2$.

Define the event

$$\mathcal{E}_1 = \left\{ \bigcap_{i \in [n]} \bigcap_{t \in \mathbb{N}} |\hat{\mu}_i(t) - \mu_i| \leq C_{\delta/2n}(t) \right\}.$$

Using standard anytime confidence bound results, and recalling that that $C_\delta(t) := \sqrt{\frac{4 \log(\log_2(2t)/\delta)}{t}}$, we have

$$\mathbb{P}(\mathcal{E}_1^c) = \mathbb{P}\left( \bigcup_{i \in [n]} \bigcup_{t \in \mathbb{N}} |\hat{\mu}_i - \mu_i| > C_{\delta/2n}(t) \right)$$

$$\leq \sum_{i=1}^{n} \mathbb{P}\left( \bigcup_{t \in \mathbb{N}} |\hat{\mu}_i - \mu_i| > C_{\delta/2n}(t) \right) \leq \sum_{i=1}^{n} \frac{\delta}{2n} = \frac{\delta}{2}$$

Next, recall that $\hat{\mu}_i(t)$ denotes the empirical average of $t$ samples of $\rho_i$. Consider the event,

$$\mathcal{E}_2 = \bigcap_{i \in G_\epsilon} \bigcap_{k \in \mathbb{N}} |(\hat{\mu}_{i_k}(\tau_k) - \hat{\mu}_i(\tau_k)) - (\mu_{i_k} - \mu_i)| \leq 2^{-k}$$

By Hoeffding's inequality,

$$\mathbb{P}\left(|(\hat{\mu}_j\left(\tau_k\right) - \hat{\mu}_i\left(\tau_k\right)) - (\mu_j - \mu_i)| > 2^{-k}\big|i_k = j\right) \le \frac{\delta}{4nk^2}.$$

Then

$$\mathbb{P}\big(|(\hat{\mu}_j\left(\tau_k\right) - \hat{\mu}_i\left(\tau_k\right)) - (\mu_j - \mu_i)| > 2^{-k}\big)$$

$$= \sum_{j=1}^{n} \mathbb{P}\left(|(\hat{\mu}_j\left(\tau_k\right) - \hat{\mu}_i\left(\tau_k\right)) - (\mu_j - \mu_i)| > 2^{-k}\big|i_k = j\right)\mathbb{P}(i_k = j)$$

$$\le \frac{\delta}{4nk^2} \sum_{j=1}^{n} \mathbb{P}(i_k = j)$$

$$= \frac{\delta}{4nk^2}$$

Therefore, union bounding over the rounds $k \in \mathbb{N}$, $\mathbb{P}(\mathcal{E}_2^c) \le \sum_{i \in G_\epsilon} \sum_{k=1}^{\infty} \frac{\delta}{4nk^2} \le \frac{\delta}{2}$. Hence, $\mathbb{P}\left(\mathcal{E}_1 \cap \mathcal{E}_2\right) \ge 1 - \delta$.

### E.3.1   Step 0: Correctness.

On $\mathcal{E}_1 \cap \mathcal{E}_2$, first we prove that if there exists a random round $k$ at which $G_k \cup B_k = [n]$ then $G_k = G_\epsilon$. Additionally, we prove that on $\mathcal{E}_1 \cap \mathcal{E}_2$, if $\mathcal{A} \subset G_k$, then $G_k = G_\epsilon$. Therefore, for either stopping condition for FAREAST in line 31, on the event $\mathcal{E}_1 \cap \mathcal{E}_2$, FAREAST correctly returns the set $G_\epsilon$.

**Claim 0:** On $\mathcal{E}_1 \cap \mathcal{E}_2$, for all $k \in \mathbb{N}$, $G_k \subset G_\epsilon$.

**Proof.** Firstly we show $1 \in \mathcal{A}$ for all $t \in \mathbb{N}$, namely the best arm is never removed from $\mathcal{A}$. Note for any $i$

$$\hat{\mu}_1 + C_{\delta/2n}(t) \ge \mu_1 \ge \mu_i \ge \hat{\mu}_i(t) - C_{\delta/2n}(t) > \hat{\mu}_i(t) - C_{\delta/2n}(t) - \epsilon.$$

In particular this shows, $\hat{\mu}_1 + C_{\delta/2n}(t) > \max_{i \in \mathcal{A}} \hat{\mu}_i(t) - C_{\delta/2n}(t) - \epsilon = L_t$ and $\hat{\mu}_1 + C_{\delta/2n}(t) \ge \max_{i \in \mathcal{A}} \hat{\mu}_i(t) - C_{\delta/2n}(t)$ showing that 1 will never exit $\mathcal{A}$ in line 28.

Secondly, we show that at all times $t$, $\mu_1 - \epsilon \in [L_t, U_t]$. By the above, since $\mu_1$ never leaves $\mathcal{A}$,

$$U_t = \max_{i \in \mathcal{A}} \hat{\mu}_i(t) + C_{\delta/2n}(t) - \epsilon \ge \hat{\mu}_1(t) + C_{\delta/2n}(t) - \epsilon \ge \mu_1 - \epsilon$$

and for any $i$,

$$\mu_1 - \epsilon \ge \mu_i - \epsilon \ge \hat{\mu}_i(t) - C_{\delta/2n}(t) - \epsilon$$

Hence $\mu_1 - \epsilon \ge \max_i \hat{\mu}_i(t) - C_{\delta/2n}(t) - \epsilon = L_t$.

Next, we show that $G_k \subset G_\epsilon$ for all $k \ge 1, t \ge 1$. Suppose not. Then $\exists, k, t \in N$ and $\exists i \in G_\epsilon^c \cap G_k(t)$ such that,

$$\mu_i \ge \hat{\mu}_i(t) - C_{\delta/2n}(t) \ge U_t \ge \mu_1 - \epsilon > \mu_i,$$

with the last inequality following from the previous assertion, giving a contradiction.  $\square$

**Claim 1:** On $\mathcal{E}_1 \cap \mathcal{E}_2$, for all $k \in \mathbb{N}$, $B_k \subset G_\epsilon^c$.

**Proof.**  Next, we show $B_k \subset G_\epsilon^c$. Suppose not. Either a good arm was added to the bad set by the bad filter or by the good filter. First, consider the case, that the bad filter added an arm in $G_\epsilon$ to $B_k$ for some $k$. By definition, $B_0 = \emptyset$ and $B_{k-1} \subset B_k$ for all $k$. Then there must exist $k \in \mathbb{N}$ and an $i \in G_\epsilon$ such that $i \in B_k$ and $i \notin B_{k-1}$. Following line 14 of the algorithm, this occurs if and only if

$$\hat{\mu}_{i_k} - \hat{\mu}_i \ge \epsilon + 2^{-k+1}.$$

On the event $\mathcal{E}_2$, the above implies

$$\mu_{i_k} - \mu_i + 2^{-k} \ge \epsilon + 2^{-k+1},$$

and simplifying, we see that $\epsilon + 2^{-k} \le \mu_{i_k} - \mu_i \le \mu_1 - \mu_i$ which contradicts the assertion that $i \in G_\epsilon$.

Next, consider the case that the good filter incorrectly adds a good arm $i \in G_\epsilon$ to $B_k$ in some round $k$. Then there must be a $t \in \mathbb{N}$ such that.

$$\mu_i \overset{\mathcal{E}_1}{\leq} \hat{\mu}_i + C_{\delta/2n}(t) < L_t \overset{\mathcal{E}_1}{\leq} \mu_1 - \epsilon$$

which contradicts $i \in G_\epsilon$. Hence, in both cases $B_k \subset G_\epsilon^c$ for all $k$. $\qquad$ $\square$Combining the above claims, we see that $\mathcal{E}_1 \cap \mathcal{E}_2$ implies $(G_k \cup B_k = [n])$ and $G_k \cap B_k = \emptyset \implies G_k = G_\epsilon$. Since $\mathbb{P}(\mathcal{E}_1 \cap \mathcal{E}_2) \geq 1 - \delta$, if FAREAST terminates, with probability at least $1 - \delta$, it correctly returns the set $G_\epsilon$.

**Claim 2:** Next, we show that on $\mathcal{E}_1$, $G_\epsilon \subset \mathcal{A}(t) \cup G(t)$ for all $t \in \mathbb{N}$.

In particular this implies that if $\mathcal{A} \subset G$, then $G_\epsilon \subset G$. Combining this with the previous claim gives $G \subset G_\epsilon \subset G$, hence $G = G_\epsilon$. On this condition, FAREAST terminates by line 33 and returns the set $\mathcal{A} \cup G = G$. Note that by definition, $G_\epsilon \subset G_{(\epsilon+\gamma)}$ for all $\gamma \geq 0$. Therefore FAREAST terminates correctly on this condition.

**Proof.** Suppose for contradiction that there exists $i \in G_\epsilon$ such that $i \notin \mathcal{A}(t) \cup G(t)$. This occurs only if $i$ is eliminated in line 28. Hence, there exists a $t' \leq t$ such that $\hat{\mu}_i(t') + C_{\delta/n}(t') < L_{t'}$. Therefore, on the event $\mathcal{E}_1$,

$$\mu_1 - \epsilon \overset{\mathcal{E}_1}{\geq} L_{t'} = \max_{j \in \mathcal{A}} \hat{\mu}_j(t') - C_{\delta/n}(t') - \epsilon > \hat{\mu}_i(t') + C_{\delta/n}(t') \overset{\mathcal{E}_1}{\geq} \mu_i$$

which contradicts $i \in G_\epsilon$. $\qquad \square$

**Claim 3:** Finally, we show that on $\mathcal{E}_1$, if $U_t - L_t \leq \gamma/2$, then $\mathcal{A} \cup G \subset G_{(\epsilon+\gamma)}$.

Combining with Claim 3 that $G_\epsilon \subset \mathcal{A} \cup G$, if FAREAST terminates on this condition by line 33, it does so correctly and returns all arms in $G_\epsilon$.

**Proof.** Assume $U_t - L_t \leq \gamma/2$. Since all arms in $\mathcal{A}(t)$ have received exactly $t$ samples, this implies that

$$\left( \max_{i \in A(t)} \hat{\mu}_i(t) + C_{\delta/n}(t) - \epsilon \right) - \left( \max_{i \in A(t)} \hat{\mu}_i(t) - C_{\delta/n}(t) - \epsilon \right) = 2C_{\delta/n}(t) \leq \gamma/2.$$

Suppose for contradiction that there exists $i \in G_{(\epsilon+\gamma)}^c$ such that $i \in \mathcal{A} \cup G$. Since $G_\epsilon \cap G_{(\epsilon+\gamma)}^c = \emptyset$ and we have previously shown than $G(t) \subset G_\epsilon$ for all $t$, we have that $i \in A \backslash G$. Therefore, by the condition in line 27, $\hat{\mu}_i(t) + C_{\delta/n}(t) \geq L_t$. Hence, $\mu_i + 2C_{\delta/n}(t) \overset{\mathcal{E}_1}{\geq} \hat{\mu}_i(t) + C_{\delta/n}(t) \geq L_t$. By assumption, we have that $U_t - \gamma/2 \leq L_t$, and the event $\mathcal{E}_1$ implies that $U_t \geq \mu_1 - \epsilon$. Therefore, $\mu_i + 2C_{\delta/n}(t) \geq U_t - \gamma/2 \geq \mu_1 - \epsilon - \gamma/2$. Combining this with the inequality $2C_{\delta/n} \leq \gamma/2$, we have that

$$\gamma \geq 2C_{\delta/n}(t) + \gamma/2 \geq \mu_1 - \epsilon - \mu_i \overset{i \in G_{(\epsilon+\gamma)}^c}{>} \gamma$$

which is a contradiction. $\qquad \square$

### E.3.2 Step 1: An expression for the total number of samples drawn and introducing several helper random variables

Next, we write an expression for the total number of samples drawn by FAREAST. In particular, we introduce two sums that we will spend the remainder of the proof controlling. Additionally, we show that the conditional in line 19 in the good filter is true at least once in each round. Based on this, we more precisely define the random variables $T_i$ and $T_i'$ introduces in the notation section in subsection E.3. Additionally, we introduce the time $T_\gamma$ at which $U_t - L_t < \frac{1}{2}\gamma$.

Recall that the largest value of $t$ in round $k$ is denoted $t_k$. Let $E_k^\gamma$ be the event that $U_t - L_t \geq \gamma/2$ for all $t$ in round $k$:

$$E_k^\gamma := \{U_t - L_t \geq \gamma/2 : t \in (t_{k-1}, t_k]\}.$$

Note that if $E_{k-1}^\gamma$ is false, then FAREAST terminates in round $k - 1$ by line 33. We may write the total number of samples drawn by the algorithm as

$$T = \sum_{k=1}^{\infty} 2 \mathbb{1} \left[ \mathcal{A} \not\subset G_{k-1} \text{ and } G_{k-1} \cup B_{k-1} \neq [n] \text{ and } E_{k-1}^\gamma \right]$$

$$\left(H_{\mathrm{ME}}(n, 2^{-k}, 1/16) + \tau_k + \tau_k|(G_{k-1} \cup B_{k-1})^c|\right)$$

Deterministically, $\mathbb{1}\left[\mathcal{A} \not\subset G_{k-1} \text{ and } G_{k-1} \cup B_{k-1} \neq [n] \text{ and } E^{\gamma}_{k-1}\right] \leq \mathbb{1}\left[G_{k-1} \cup B_{k-1} \neq [n]\right]$

Applying this,

$$T \leq \sum_{k=1}^{\infty} 2\mathbb{1}\left[G_{k-1} \cup B_{k-1} \neq [n]\right] \left(H_{\mathrm{ME}}(n, 2^{-k}, 1/16) + \tau_k + \tau_k|(G_{k-1} \cup B_{k-1})^c|\right)$$

$$= \sum_{k=1}^{\infty} 2\mathbb{1}\left[G_{k-1} \neq G_\epsilon\right] \mathbb{1}\left[G_{k-1} \cup B_{k-1} \neq [n]\right] \left(H_{\mathrm{ME}}(n, 2^{-k}, 1/16) + \tau_k + \tau_k|(G_{k-1} \cup B_{k-1})^c|\right)$$

(10)

$$+ \sum_{k=1}^{\infty} 2\mathbb{1}\left[G_{k-1} = G_\epsilon\right] \mathbb{1}\left[G_{k-1} \cup B_{k-1} \neq [n]\right] \left(H_{\mathrm{ME}}(n, 2^{-k}, 1/16) + \tau_k + \tau_k|(G_{k-1} \cup B_{k-1})^c|\right)$$

(11)

In round $k$, line 18 of the Good Filter, whereby an arm is sampled, is evaluated

$$\left(H_{\mathrm{ME}}(n, 2^{-k}, 1/16) + \tau_k + \tau_k|(G_{k-1} \cup B_{k-1})^c|\right) \geq \left(H_{\mathrm{ME}}(n, 2^{-k}, 1/16) + 2\tau_k\right) \geq n$$

times since $H_{\mathrm{ME}}(n, 2^{-k}, 1/16)) \geq n$ for all $k$ and $|(G_{k-1} \cup B_{k-1})^c| \geq 1$ unless $G_{k-1} \cup B_{k-1} = [n]$ which implies termination in round $k - 1$. Each time line 18 is called, $N_{I_s} \leftarrow N_{I_s} + 1$. Since $|\arg\min_{j \in \mathcal{A}}\{N_j\}| \leq |\mathcal{A}| \leq n$, line 18 is called at most $n$ times before $\min_{j \in \mathcal{A}}\{N_j\} = \max_{j \in \mathcal{A}}\{N_j\}$. When this occurs, the conditional in line 19 is true and $t \leftarrow t + 1$.

If $\min_{i \in \mathcal{A}(t)}\{N_i\} = \max_{i \in \mathcal{A}(t)}\{N_i\}$, then $N_i = t$ for any $i \in \mathcal{A}(t)$. By Step 0, only arms in $G_\epsilon$ are added to $G_k$. Therefore, $T_i$ is defined as

$$T_i = \min \left\{ t : \begin{array}{ll} i \in G_k(t+1) & \text{if } i \in G_\epsilon \\ i \notin \mathcal{A}(t+1) & \text{if } i \in G_\epsilon^c \end{array} \right\} \stackrel{\mathcal{E}_1}{=} \min \left\{ t : \begin{array}{ll} \hat{\mu}_i - C_{\delta/2n}(t) \geq U_t & \text{if } i \in G_\epsilon \\ \hat{\mu}_i + C_{\delta/2n}(t) \leq L_t & \text{if } i \in G_\epsilon^c \end{array} \right\} \quad (12)$$

Define $T_i = \infty$ if this never occurs. Note that this may happen if FAREAST terminates due to the condition in line 32 that $U_t - L_t < \gamma/2$. Similarly, recall $T_i'$ denotes the random variable of the of the number of times $i$ is sampled before $\hat{\mu}_i(t) + C_{\delta/2n}(t) \leq \max_{j \in \mathcal{A}} \hat{\mu}_j(t) - C_{\delta/2n}(t)$. Hence,

$$T_i' = \min \left\{ t : \hat{\mu}_i(t) + C_{\delta/2n}(t) \leq \max_{j \in \mathcal{A}(t)} \hat{\mu}_j(t) - C_{\delta/2n}(t) \right\} \quad (13)$$

Define $T_i' = \infty$ if this never occurs. Note that this may happen if FAREAST terminates due to the condition in line 32 that $U_t - L_t < \gamma/2$. Finally, we define the time $T_\gamma$ such that $U_t - L_t < \frac{1}{2}\gamma$.

$$T_\gamma = \min \left\{ t : U_t - L_t < \frac{1}{2}\gamma \right\} \quad (14)$$

By design, no arm is sampled more that $T_\gamma$ times by the good filter, controlling the cases that $T_i$ or $T_i'$ are infinite.

### E.3.3 Step 2: Bounding $T_i$ and $T_i'$ for $i \in G_\epsilon$

**Step 2a:** For $i \in G_\epsilon$, we have that $T_i \leq h(0.25(\epsilon - \Delta_i), \delta/2n)$.

**Proof.** Note that, $4C_{\delta/2n}(t) \leq \mu_i - (\mu_1 - \epsilon)$, true when $t > h\left(0.25(\epsilon - \Delta_i), \frac{\delta}{2n}\right)$, implies that for all $j$,

$$\begin{aligned} \hat{\mu}_i(t) - C_{\delta/2n}(t) &\stackrel{\mathcal{E}_1}{\geq} \mu_i - 2C_{\delta/2n}(t) \\ &\geq \mu_1 + 2C_{\delta/2n}(t) - \epsilon \\ &\geq \mu_j + 2C_{\delta/2n}(t) - \epsilon \\ &\stackrel{\mathcal{E}_1}{\geq} \hat{\mu}_j(t) + C_{\delta/2n}(t) - \epsilon \end{aligned}$$

so in particular, $\hat{\mu}_i(t) - C_{\delta/2n}(t) \geq \max_{j \in \mathcal{A}} \hat{\mu}_j(t) + C_{\delta/2n}(t) - \epsilon = U_t$. $\qquad \square$

Additionally, we define a time $T_{\max}$ when all good arms have entered $G_k$.

**Step 2b:** Defining $T_{\max} := \min\{t : G_k(t) = G_\epsilon\} = \max_{i \in G_\epsilon} T_i$, we also have that $T_{\max} \leq h(0.25\alpha_\epsilon, \delta/2n)$ (in other words, if $t > h(0.25\alpha_\epsilon, \delta/2n)$ (i.e. line 23 has been run $t$ times), then we have that $G_k(t) = G_\epsilon$).

**Proof.** Recall that $\alpha_\epsilon = \min_{i \in G_\epsilon} \mu_i - \mu_1 + \epsilon = \min_{i \in G_\epsilon} \epsilon - \Delta_i$. By Step 1a, $T_i \leq h\left(0.25(\epsilon - \Delta_i), \frac{\delta}{2n}\right)$. Furthermore, $h(\cdot, \cdot)$ is monotonic in its first argument, such that if $0 < x' < x$, then $h(x', \delta) > h(x, \delta)$ for any $\delta > 0$. Therefore $T_{\max} = \max_{i \in G_\epsilon} T_i \leq \max_{i \in G_\epsilon} h\left(0.25(\epsilon - \Delta_i), \frac{\delta}{2n}\right) = h\left(0.25\alpha_\epsilon, \frac{\delta}{2n}\right)$. $\qquad \square$

**Step 2c:** For $i \in G_\epsilon$, we have that $T_i' \leq h(0.25\Delta_i, \delta/2n)$.

**Proof.** Note that $4C_{\delta/2n}(t) \leq \mu_1 - \mu_i$, true when $t > h\left(0.25\Delta_i, \frac{\delta}{2n}\right)$, implies that

$$
\begin{aligned}
\hat{\mu}_i(t) + C_{\delta/2n}(t) &\overset{\mathcal{E}_1}{\leq} \mu_i + 2C_{\delta/2n}(t) \\
&\leq \mu_1 - 2C_{\delta/2n}(t) \\
&\overset{\mathcal{E}_1}{\leq} \hat{\mu}_1(t) - C_{\delta/2n}(t).
\end{aligned}
$$

As shown in Step 0, $1 \in \mathcal{A}(t)$ for all $t \in \mathbb{N}$, and in particular $\hat{\mu}_1(t) \leq \max_{i \in \mathcal{A}(t)} \hat{\mu}_i(t)$. Hence, $\hat{\mu}_i(t) + C_{\delta/2n}(t) \leq \max_{j \in \mathcal{A}(t)} \hat{\mu}_j(t) - C_{\delta/2n}(t)$. $\qquad \square$

### E.3.4   Step 3: Bounding $T_i$ for $i \in G_\epsilon^c$

Next, we bound $T_i$ for $i \in G_\epsilon^c$. $i \in G_\epsilon^c$ is eliminated from $\mathcal{A}$ if it has received at least $T_i$ samples.

**Claim:** $T_i \leq h\left(0.25(\epsilon - \Delta_i), \frac{\delta}{2n}\right)$ for $i \in G_\epsilon^c$

**Proof.** Note that, $4C_{\delta/2n}(t) \leq \mu_1 - \epsilon - \mu_i$, true when $t > h\left(0.25(\epsilon - \Delta_i), \frac{\delta}{2n}\right)$, implies that

$$
\begin{aligned}
\hat{\mu}_i(t) + C_{\delta/2n}(t) &\overset{\mathcal{E}_1}{\leq} \mu_i + 2C_{\delta/2n}(t) \\
&\leq \mu_1 - 2C_{\delta/2n}(t) - \epsilon \\
&\overset{\mathcal{E}_1}{\leq} \hat{\mu}_1(t) - C_{\delta/2n}(t) - \epsilon
\end{aligned}
$$

As shown in Step 0, $1 \in \mathcal{A}(t)$ for all $t \in \mathbb{N}$, and in particular $\hat{\mu}_1(t) \leq \max_{i \in \mathcal{A}(t)} \hat{\mu}_i(t)$. Therefore $\hat{\mu}_i(t) + C_{\delta/2n}(t) \leq \max_{j \in \mathcal{A}} \hat{\mu}_j(t) - C_{\delta/2n}(t) - \epsilon = L_t$. $\qquad \square$

### E.3.5   Step 4: bounding the total number of samples given to the good filter at time $t = T_{\max}$

Note that for a time $t = T$, the total number of samples given to the good filter is $\sum_{s=1}^{T} |\mathcal{A}(s)|$. Therefore, the total number of samples up to time $T_{\max}$ is $\sum_{t=1}^{T_{\max}} |\mathcal{A}(t)|$.

Let $S_i = \min\{t : i \notin A(t+1)\}$. Hence,

$$
\sum_{t=1}^{T_{\max}} |\mathcal{A}(t)| = \sum_{t=1}^{T_{\max}} \sum_{i=1}^{n} \mathbb{1}[i \in \mathcal{A}(t)] = \sum_{i=1}^{n} \sum_{t=1}^{T_{\max}} \mathbb{1}[i \in \mathcal{A}(t)] = \sum_{i=1}^{n} \min\{T_{\max}, S_i\}
$$

For arms $i \in G_\epsilon^c$, $S_i = T_i$ by definition. For $i \in G_\epsilon$, $S_i = \max(T_i, T_i')$ by line 28 of the algorithm. Then

$$
\begin{aligned}
\sum_{i=1}^{n} \min\{T_{\max}, S_i\} &= \sum_{i \in G_\epsilon} \min\{\mathcal{T}_{\max}, \max(T_i, T_i')\} + \sum_{i \in G_\epsilon^c} \min\{T_{\max}, T_i\} \\
&\leq \sum_{i \in G_\epsilon} \min\{T_{\max}, \max(T_i, T_i')\} + |G_\epsilon^c \cap G_{\epsilon + \alpha_\epsilon}| T_{\max} + \sum_{i \in G_{\epsilon + \alpha_\epsilon}^c} T_i
\end{aligned}
$$

$$= \sum_{i \in G_\epsilon} \max \{T_i, \min(T_i', T_{\max})\} + |G_\epsilon^c \cap G_{\epsilon + \alpha_\epsilon}| T_{\max} + \sum_{i \in G_{\epsilon + \alpha_\epsilon}^c} T_i$$

$$\overset{(a)}{\leq} \sum_{i \in G_\epsilon} \max \left\{ h\left(0.25(\epsilon - \Delta_i), \frac{\delta}{2n}\right), \min\left[h\left(0.25\Delta_i, \frac{\delta}{2n}\right), h\left(0.25\alpha_\epsilon, \frac{\delta}{2n}\right)\right]\right\}$$

$$+ \sum_{i \in G_{\epsilon + \alpha_\epsilon}^c} h\left(0.25(\epsilon - \Delta_i), \frac{\delta}{2n}\right) + |G_\epsilon^c \cap G_{\epsilon + \alpha_\epsilon}| h\left(0.25\alpha_\epsilon, \frac{\delta}{2n}\right).$$

Equality $(a)$ follows from $T_{\max} \leq h\left(0.25\alpha_\epsilon, \frac{\delta}{2n}\right)$ by Step 1b, $T_i \leq h\left(0.25(\epsilon - \Delta_i), \frac{\delta}{2n}\right)$ in Steps 2a and 3, and $T_i' \leq h\left(0.25\Delta_i, \frac{\delta}{2n}\right)$ in Step 2c.

### E.3.6   Step 5: Bounding the number of samples in round $k$ versus $k-1$

Now we show that the total number of samples taken in round $k$ is no more than 9 times the number taken in the previous round.

**Claim:** For $k > 1$

$$\left(H_{\mathrm{ME}}(n, 2^{-k}, 1/16) + \tau_k + \tau_k |(G_{k-1} \cup B_{k-1})^c|\right)$$
$$\leq 9\left(H_{\mathrm{ME}}(n, 2^{-k+1}, 1/16) + \tau_{k-1} + \tau_{k-1} |(G_{k-2} \cup B_{k-2})^c|\right)$$

**Proof.** In round $k$, $\left(H_{\mathrm{ME}}(n, 2^{-k}, 1/16) + \tau_k + \tau_k |(G_{k-1} \cup B_{k-1})^c|\right)$ samples are drawn. Since $G_{k-1} \subset G_k$ and $B_{k-1} \subset B_k$ $\forall k$ deterministically, we see that $|(G_{k-1} \cup B_{k-1})^c| \geq |(G_k \cup B_k)^c|$ $\forall k$. By definition, $H_{\mathrm{ME}}(n, 2^{-k-1}, 1/16) = 4H_{\mathrm{ME}}(n, 2^{-k}, 1/16)$.

Next, recall $\tau_k = \left\lceil 2^{2k+3} \log\left(\frac{8}{\delta_k}\right)\right\rceil$. We bound $\tau_k / \tau_{k-1}$ as

$$\frac{\tau_k}{\tau_{k-1}} = \frac{\left\lceil 2^{2k+3} \log\left(\frac{8}{\delta_k}\right)\right\rceil}{\left\lceil 2^{2k+1} \log\left(\frac{8}{\delta_{k-1}}\right)\right\rceil} = \frac{\left\lceil 2^{2k+3} \log\left(\frac{16nk^2}{\delta}\right)\right\rceil}{\left\lceil 2^{2k+1} \log\left(\frac{16n(k-1)^2}{\delta}\right)\right\rceil}$$

$$\leq \frac{2^{2k+3} \log\left(\frac{16nk^2}{\delta}\right) + 1}{2^{2k+1} \log\left(\frac{16n(k-1)^2}{\delta}\right)} \leq \frac{4 \log\left(\frac{16nk^2}{\delta}\right)}{\log\left(\frac{16n(k-1)^2}{\delta}\right)} + 1$$

$$\leq 4 \frac{\log\left(\frac{16n}{\delta}\right) + 2\log(k)}{\log\left(\frac{16n}{\delta}\right) + 2\log(k-1)} + 1 = (*)$$

If $k = 2$, $(*) \leq 1 + 4 * \log(32)/\log(8) \leq 9$. Otherwise,

$$(*) = \frac{4(\log\left(\frac{16n}{\delta}\right) + 2\log(k))}{\log\left(\frac{16n}{\delta}\right) + 2\log(k-1)} + 1$$

$$\leq \frac{4\log(k)}{\log(k-1)} + 1$$

$$\leq 4 \cdot 2 + 1 = 9$$

Putting these pieces together,

$$\left(H_{\mathrm{ME}}(n, 2^{-k}, 1/16) + \tau_k + \tau_k |(G_{k-1} \cup B_{k-1})^c|\right)$$
$$\leq \left(4H_{\mathrm{ME}}(n, 2^{-k+1}, 1/16) + 9\tau_{k-1} + 9\tau_{k-1} |(G_{k-2} \cup B_{k-2})^c|\right)$$
$$\leq 9\left(H_{\mathrm{ME}}(n, 2^{-k+1}, 1/16) + \tau_{k-1} + \tau_{k-1} |(G_{k-2} \cup B_{k-2})^c|\right)$$

$\square$

### E.3.7 Step 6: Bounding Equation (10)

Here, we introduce the round $K_{\text{Good}}$, when $G_{K_{\text{Good}}} = G_\epsilon$ at some point within the round. Using the result of the previous step, we may bound the total number of samples taken though this round, controlling Equation (10).

With the result of Step 5, we prove the following inequality.

**Claim:**

$$\sum_{k=1}^{\infty} 2\mathbb{1}\left[G_{k-1} \neq G_\epsilon\right]\mathbb{1}\left[G_{k-1} \cup B_{k-1} \neq [n]\right]\left(H_{\text{ME}}(n, 2^{-k}, 1/16) + \tau_k + \tau_k|(G_{k-1} \cup B_{k-1})^c|\right)$$

$$\tag{15}$$

$$\leq c \sum_{i \in G_\epsilon} \max\left\{h\left(0.25(\epsilon - \Delta_i), \frac{\delta}{2n}\right), \min\left[h\left(0.25\Delta_i, \frac{\delta}{2n}\right), h\left(0.25\alpha_\epsilon, \frac{\delta}{2n}\right)\right]\right\}$$

$$+ c|G_\epsilon^c \cap G_{\epsilon+\alpha_\epsilon}|h\left(0.25\alpha_\epsilon, \frac{\delta}{2n}\right) + c \sum_{i \in G_{\epsilon+\alpha_\epsilon}^c} h\left(0.25(\epsilon - \Delta_i), \frac{\delta}{2n}\right)$$

for a constant $c$.

**Proof.** Recall $t_k = \max\{t : t \in k\}$ denotes the maximum value of $t$ in round $k$ and $T_{\max} = \max_{\in G_\epsilon} T_i$ denotes the minimum $t$ such that $G_k(t) = G_\epsilon$. Define the random round

$$K_{\text{Good}} := \min\{k : G_k = G_\epsilon\} = \min\{k : t_k \geq T_{\max}\}$$

By definition of $K_{\text{Good}}$,

$$\sum_{k=1}^{\infty} 2\mathbb{1}[G_{k-1} \neq G_\epsilon]\mathbb{1}\left[G_{k-1} \cup B_{k-1} \neq [n]\right]\left(H_{\text{ME}}(n, 2^{-k}, 1/16) + \tau_k + \tau_k|(G_{k-1} \cup B_{k-1})^c|\right)$$

$$= \sum_{k=1}^{K_{\text{Good}}} 2\mathbb{1}\left[G_{k-1} \cup B_{k-1} \neq [n]\right]\left(H_{\text{ME}}(n, 2^{-k}, 1/16) + \tau_k + \tau_k|(G_{k-1} \cup B_{k-1})^c|\right).$$

Next, applying Step 5, if $K_{\text{Good}} > 1$,

$$\sum_{k=1}^{K_{\text{Good}}} 2\mathbb{1}\left[G_{k-1} \cup B_{k-1} \neq [n]\right]\left(H_{\text{ME}}(n, 2^{-k}, 1/16) + \tau_k + \tau_k|(G_{k-1} \cup B_{k-1})^c|\right)$$

$$\leq 18 \sum_{k=1}^{K_{\text{Good}}-1} \mathbb{1}\left[G_{k-1} \cup B_{k-1} \neq [n]\right]\left(H_{\text{ME}}(n, 2^{-k}, 1/16) + \tau_k + \tau_k|(G_{k-1} \cup B_{k-1})^c|\right).$$

Observe that by lines 17 and 20 of FAREAST, for any round $r$ and for any $t > t_{r-1}$,

$$\sum_{k=1}^{r-1} \mathbb{1}\left[G_{k-1} \cup B_{k-1} \neq [n]\right]\left(H_{\text{ME}}(n, 2^{-k}, 1/16) + \tau_k + \tau_k|(G_{k-1} \cup B_{k-1})^c|\right) \leq \sum_{s=1}^{t} |\mathcal{A}(s)|.$$

By definition, for the round $K_{\text{Good}} - 1$, we see that $t_{(K_{\text{Good}}-1)} < T_{\max}$. Applying the above inequality with the inequality proven in Step 4,

$$18 \sum_{k=1}^{K_{\text{Good}}-1} \mathbb{1}\left[G_{k-1} \cup B_{k-1} \neq [n]\right]\left(H_{\text{ME}}(n, 2^{-k}, 1/16) + \tau_k + \tau_k|(G_{k-1} \cup B_{k-1})^c|\right) \leq 18 \sum_{s=1}^{T_{\max}} |\mathcal{A}(s)|$$

$$\leq 18 \sum_{i \in G_\epsilon} \max\left\{h\left(0.25(\epsilon - \Delta_i), \frac{\delta}{2n}\right), \min\left[h\left(0.25\Delta_i, \frac{\delta}{2n}\right), h\left(0.25\alpha_\epsilon, \frac{\delta}{2n}\right)\right]\right\}$$

$$+ 18 \sum_{i \in G_{\epsilon+\alpha_\epsilon}^c} h\left(0.25(\epsilon - \Delta_i), \frac{\delta}{2n}\right) + 18|G_\epsilon^c \cap G_{\epsilon+\alpha_\epsilon}|h\left(0.25\alpha_\epsilon, \frac{\delta}{2n}\right).$$

Otherwise, if $K_{\text{Good}} = 1$, exactly $4c'n\log(16) + 32n\log(16n/\delta)$ samples are given to the good filter in round 1. One may use Lemma F.2 to invert $h(\cdot, \cdot)$ and show that the summation on the right had side of the above inequality is within a constant of this and the claim holds in this case as well for a different constant, potentially larger than 18. $\qquad\square$

### E.3.8 Step 7: Bounding Equation (11)

Next, we bound $\sum_{k=1}^{\infty} 2\mathbb{1}\left[G_{k-1} = G_\epsilon\right] \mathbb{1}\left[G_{k-1} \cup B_{k-1} \neq [n]\right] \left(H_{\text{ME}}(n, 2^{-k}, 1/16) + \tau_k + \tau_k | (G_{k-1} \cup B_{k-1})^c |\right)$.

$$\sum_{k=1}^{\infty} 2\mathbb{1}\left[G_{k-1} = G_\epsilon\right] \mathbb{1}\left[G_{k-1} \cup B_{k-1} \neq [n]\right] \left(H_{\text{ME}}(n, 2^{-k}, 1/16) + \tau_k + \tau_k | (G_{k-1} \cup B_{k-1})^c |\right)$$

$$= \sum_{k=1}^{\infty} 2\mathbb{1}\left[G_{k-1} = G_\epsilon\right] \mathbb{1}\left[G_{k-1} \cup B_{k-1} \neq [n]\right] \left(H_{\text{ME}}(n, 2^{-k}, 1/16) + \tau_k + \tau_k | (G_\epsilon \cup B_{k-1})^c |\right)$$

$$= \sum_{k=1}^{\infty} 2\mathbb{1}\left[G_{k-1} = G_\epsilon\right] \mathbb{1}\left[G_{k-1} \cup B_{k-1} \neq [n]\right] \left(H_{\text{ME}}(n, 2^{-k}, 1/16) + \tau_k + \tau_k | G_\epsilon^c \backslash B_{k-1} |\right)$$

$$= \sum_{k=K_{\text{Good}}+1}^{\infty} 2\mathbb{1}\left[G_{k-1} \cup B_{k-1} \neq [n]\right] \left(H_{\text{ME}}(n, 2^{-k}, 1/16) + \tau_k + \tau_k | G_\epsilon^c \backslash B_{k-1} |\right)$$

$$\stackrel{\mathcal{E}_1, \mathcal{E}_2}{=} \sum_{k=K_{\text{Good}}+1}^{\infty} 2\mathbb{1}\left[B_{k-1} \neq G_\epsilon^c\right] \left(H_{\text{ME}}(n, 2^{-k}, 1/16) + \tau_k + \tau_k | G_\epsilon^c \backslash B_{k-1} |\right)$$

$$= \sum_{k=K_{\text{Good}}+1}^{\infty} 2\mathbb{1}\left[B_{k-1} \neq G_\epsilon^c\right] \left(H_{\text{ME}}(n, 2^{-k}, 1/16) + \tau_k\right) + \sum_{k=K_{\text{Good}}+1}^{\infty} 2\mathbb{1}\left[B_{k-1} \neq G_\epsilon^c\right] \left(\tau_k | G_\epsilon^c \backslash B_{k-1} |\right)$$

$$= \sum_{k=K_{\text{Good}}+1}^{\infty} 2\mathbb{1}\left[B_{k-1} \neq G_\epsilon^c\right] \left(H_{\text{ME}}(n, 2^{-k}, 1/16) + \tau_k\right) + \sum_{k=K_{\text{Good}}+1}^{\infty} 2\tau_k | G_\epsilon^c \backslash B_{k-1} |$$

$$= \sum_{k=K_{\text{Good}}+1}^{\infty} 2\mathbb{1}\left[B_{k-1} \neq G_\epsilon^c\right] \left(H_{\text{ME}}(n, 2^{-k}, 1/16) + \tau_k\right) + \sum_{k=K_{\text{Good}}+1}^{\infty} \sum_{i \in G_\epsilon^c} 2\tau_k \mathbb{1}\left[i \notin B_{k-1}\right]$$

$$\leq \sum_{k=K_{\text{Good}}+1}^{\infty} 2 | G_\epsilon^c \backslash B_{k-1} | \left(H_{\text{ME}}(n, 2^{-k}, 1/16) + \tau_k\right) + \sum_{k=K_{\text{Good}}+1}^{\infty} \sum_{i \in G_\epsilon^c} 2\tau_k \mathbb{1}\left[i \notin B_{k-1}\right]$$

$$= \sum_{k=K_{\text{Good}}+1}^{\infty} \sum_{i \in G_\epsilon^c} 2\mathbb{1}[i \notin B_{k-1}] \left(H_{\text{ME}}(n, 2^{-k}, 1/16) + \tau_k\right) + \sum_{k=K_{\text{Good}}+1}^{\infty} \sum_{i \in G_\epsilon^c} 2\tau_k \mathbb{1}\left[i \notin B_{k-1}\right]$$

$$= \sum_{k=K_{\text{Good}}+1}^{\infty} \sum_{i \in G_\epsilon^c} 2\mathbb{1}[i \notin B_{k-1}] \left(2\tau_k + H_{\text{ME}}(n, 2^{-k}, 1/16)\right)$$

$$= \sum_{i \in G_\epsilon^c} \sum_{k=K_{\text{Good}}+1}^{\infty} 2\mathbb{1}[i \notin B_{k-1}] \left(2\tau_k + H_{\text{ME}}(n, 2^{-k}, 1/16)\right)$$

$$\leq \sum_{i \in G_\epsilon^c} \sum_{k=1}^{\infty} 2\mathbb{1}[i \notin B_{k-1}] \left(2\tau_k + H_{\text{ME}}(n, 2^{-k}, 1/16)\right) \tag{16}$$

### E.3.9 Step 8: Bounding the expected total number of samples drawn by FAREAST

Now we take expectations over the number of samples drawn. These expectations are conditional on the high probability event $\mathcal{E}_1 \cap \mathcal{E}_2$. The bound in step 5 holds deterministically conditioned on this event.

Note $\tau_k$ and $H_{\text{ME}}(n, 2^{-k}, 1/16)$ are deterministic constants for any $k$. Let all expectations are be jointly over the random instance $\nu$ and the randomness in FAREAST.

$$\mathbb{E}[T | \mathbb{1}[\mathcal{E}_1 \cap \mathcal{E}_2] = 1] \leq$$

$$\sum_{k=1}^{\infty} 2\mathbb{E}\left[\mathbb{1}[G_k \cup B_k \neq [n]] \middle| \mathbb{1}[\mathcal{E}_1 \cap \mathcal{E}_2] = 1\right] \left(\tau_k + H_{\text{ME}}(n, 2^{-k}, 1/16) + \tau_k | (G_{k-1} \cup B_{k-1})^c |\right)$$

$$= \sum_{k=1}^{\infty} 2\mathbb{E}\left[\mathbb{1}\left[G_{k-1} \neq G_\epsilon\right] \mathbb{1}[G_{k-1} \cup B_{k-1} \neq [n]] \big| \mathbb{1}[\mathcal{E}_1 \cap \mathcal{E}_2] = 1\right]$$

$$\left(\tau_k + H_{\mathrm{ME}}(n, 2^{-k}, 1/16) + \tau_k \big| (G_{k-1} \cup B_{k-1})^c\big|\right)$$

$$+ \sum_{k=1}^{\infty} 2\mathbb{E}\left[\mathbb{1}\left[G_{k-1} = G_\epsilon\right] \mathbb{1}[G_{k-1} \cup B_{k-1} \neq [n]] \big| \mathbb{1}[\mathcal{E}_1 \cap \mathcal{E}_2] = 1\right]$$

$$\left(\tau_k + H_{\mathrm{ME}}(n, 2^{-k}, 1/16) + \tau_k \big| (G_{k-1} \cup B_{k-1})^c\big|\right)$$

$$\overset{\text{Step 6}}{\leq} c \sum_{i \in G_\epsilon} \max\left\{ h\left(0.25(\epsilon - \Delta_i), \frac{\delta}{2n}\right), \min\left[h\left(0.25\Delta_i, \frac{\delta}{2n}\right), h\left(0.25\alpha_\epsilon, \frac{\delta}{2n}\right)\right]\right\}$$

$$+ c \sum_{i \in G_{\epsilon+\alpha_\epsilon}^c} h\left(0.25(\epsilon - \Delta_i), \frac{\delta}{2n}\right) + c|G_\epsilon^c \cap G_{\epsilon+\alpha_\epsilon}| h\left(0.25\alpha_\epsilon, \frac{\delta}{2n}\right)$$

$$+ \sum_{k=1}^{\infty} 2\mathbb{E}\left[\mathbb{1}\left[G_{k-1} = G_\epsilon\right] \mathbb{1}[G_{k-1} \cup B_{k-1} \neq [n]] \big| \mathbb{1}[\mathcal{E}_1 \cap \mathcal{E}_2] = 1\right]$$

$$\left(\tau_k + H_{\mathrm{ME}}(n, 2^{-k}, 1/16) + \tau_k \big| (G_{k-1} \cup B_{k-1})^c\big|\right)$$

$$\overset{\text{Step 7}}{\leq} c \sum_{i \in G_\epsilon} \max\left\{ h\left(0.25(\epsilon - \Delta_i), \frac{\delta}{2n}\right), \min\left[h\left(0.25\Delta_i, \frac{\delta}{2n}\right), h\left(0.25\alpha_\epsilon, \frac{\delta}{2n}\right)\right]\right\}$$

$$+ c \sum_{i \in G_{\epsilon+\alpha_\epsilon}^c} h\left(0.25(\epsilon - \Delta_i), \frac{\delta}{2n}\right) + c|G_\epsilon^c \cap G_{\epsilon+\alpha_\epsilon}| h\left(0.25\alpha_\epsilon, \frac{\delta}{2n}\right)$$

$$+ \sum_{i \in G_\epsilon^c} \sum_{k=1}^{\infty} 2\mathbb{E}_\nu\left[\mathbb{1}[i \notin B_{k-1}]\big|\mathbb{1}[\mathcal{E}_1 \cap \mathcal{E}_2] = 1\right]\left(2\tau_k + H_{\mathrm{ME}}(n, 2^{-k}, 1/16)\right)$$

$$\overset{(a)}{=} c \sum_{i \in G_\epsilon} \max\left\{ h\left(0.25(\epsilon - \Delta_i), \frac{\delta}{2n}\right), \min\left[h\left(0.25\Delta_i, \frac{\delta}{2n}\right), h\left(0.25\alpha_\epsilon, \frac{\delta}{2n}\right)\right]\right\}$$

$$+ c \sum_{i \in G_{\epsilon+\alpha_\epsilon}^c} h\left(0.25(\epsilon - \Delta_i), \frac{\delta}{2n}\right) + c|G_\epsilon^c \cap G_{\epsilon+\alpha_\epsilon}| h\left(0.25\alpha_\epsilon, \frac{\delta}{2n}\right)$$

$$+ \sum_{i \in G_\epsilon^c} \sum_{k=1}^{\infty} 2\mathbb{E}_\nu\left[\mathbb{1}[i \notin B_{k-1}]\big|\mathcal{E}_1\right]\left(2\tau_k + H_{\mathrm{ME}}(n, 2^{-k}, 1/16)\right)$$

where $(a)$ follows from $\mathbb{E}_\nu\left[\mathbb{1}[i \notin B_{k-1}]\big|\mathcal{E}_1 \cap \mathcal{E}_2\right] = \mathbb{E}_\nu\left[\mathbb{1}[i \notin B_{k-1}]\big|\mathcal{E}_1\right]$ for $i \in G_\epsilon^c$, since the event $\{i \in B_{k-1}\}$ is independent of $\mathcal{E}_2$ for all $i \in G_\epsilon^c$. This can be observed since $\mathcal{E}_2$ deals only with independent samples taken of arms in $G_\epsilon$.

### E.3.10 Step 9: Bounding $\sum_{k=1}^{\infty} \mathbb{E}_\nu\left[\mathbb{1}[i \notin B_{k-1}]\big|\mathcal{E}_1\right]\left(2\tau_k + H_{\mathrm{ME}}(n, 2^{-k}, 1/16)\right)$ for $i \in G_\epsilon^c$

Next, we bound the expectation remaining from step 8. In particular, this is the number of samples drawn by the bad filter to add arm $i \in G_\epsilon^c$ to $B_k$.

First, we bound the probability that for a given $i \in G_\epsilon^c$ and a given $k$ $i \notin B_k$. Note that by Borel-Cantelli, this implies that the probability that $i$ is never added to any $B_k$ is 0.

**Claim 1:** For $i \in G_\epsilon^c$, $k \geq \left\lceil \log_2\left(\frac{4}{\Delta_i - \epsilon}\right)\right\rceil \implies \mathbb{E}_\nu\left[\mathbb{1}[i \notin B_k]\big|\mathcal{E}_1\right] \leq \left(\frac{1}{8}\right)^{k - \left\lceil \log_2\left(\frac{4}{\Delta_i - \epsilon}\right)\right\rceil}$

**Proof.** $i \in B_k$ if either the good filter or the bad filter added it. Note that the behavior of the bad filter is independent of the event $\mathcal{E}_1$. Hence,

$$\mathbb{E}_\nu\left[\mathbb{1}[i \notin B_k]\big|\mathcal{E}_1\right] = \mathbb{E}_\nu\left[\mathbb{1}[\hat\mu_i + C_{\delta/2n}(t_k) \geq L_{t_k}]\mathbb{1}[\hat\mu_{i_k} - \hat\mu_i < \epsilon + 2^{-k+1}]\big|\mathcal{E}_1\right]$$
$$\leq \mathbb{E}_\nu\left[\mathbb{1}[\hat\mu_{i_k} - \hat\mu_i < \epsilon + 2^{-k+1}]\big|\mathcal{E}_1\right]$$
$$= \mathbb{E}_\nu\left[\mathbb{1}[\hat\mu_{i_k} - \hat\mu_i < \epsilon + 2^{-k+1}]\right].$$

Intuitively, the time at which an arm in $G_\epsilon^c$ enters $B_k$, which occurs if either the good filter adds it or the bad filter does, in expectation is at most the time at which the bad filter does on its own in expectation.

If $i \in B_{k-1}$ then $i \in B_k$ by definition. Otherwise, if $i \notin B_{k-1}$, by Hoeffding's Inequality conditional on the value of $i_k$ and a sum over conditional probabilities as in step 0, with probability at least $1 - \frac{\delta}{4nk^2}$

$$|(\hat{\mu}_{i_k} - \hat{\mu}_i) - (\mu_{i_k} - \mu_i)| \leq 2^{-k}$$

If `MedianElimination` also succeeds, the joint event of which occurs with probability $\frac{15}{16}\left(1 - \frac{\delta}{4nk^2}\right)$ by independence[6],

$$\hat{\mu}_{i_k} - \hat{\mu}_i \geq \mu_{i_k} - \mu_i - 2^{-k} \geq \mu_1 - \mu_i - 2^{-k+1} = \Delta_i - 2^{-k+1}.$$

Then for $k \geq \left\lceil \log_2\left(\frac{4}{\Delta_i - \epsilon}\right) \right\rceil$,

$$\hat{\mu}_{i_k} - \hat{\mu}_i \geq \Delta_i - 2^{-k+1} \geq \frac{1}{2}(\Delta_i + \epsilon) \geq \epsilon + 2^{-k+1},$$

which implies that $i \in B_k$ by line 15 of `FAREAST`. In particular, $\mathbb{E}\left[\mathbb{1}[\hat{\mu}_{i_k} - \hat{\mu}_i \geq \epsilon + 2^{-k+1}] \big| i \notin B_{k-1}\right] \geq \frac{15}{16}\left(1 - \frac{\delta}{4nk^2}\right)$. Furthermore, $i \notin B_0$ by definition. Additionally, recall that $\mathbb{1}[\hat{\mu}_{i_k} - \hat{\mu}_i < \epsilon + 2^{-k+1}]$ is independent of $\mathcal{E}_1$. Then for $k \geq \left\lceil \log_2\left(\frac{4}{\Delta_i - \epsilon}\right) \right\rceil$,

$$
\begin{aligned}
\mathbb{E}\left[\mathbb{1}[\hat{\mu}_{i_k} - \hat{\mu}_i < \epsilon + 2^{-k+1}]\right] &= \mathbb{E}\left[\mathbb{1}[\hat{\mu}_{i_k} - \hat{\mu}_i < \epsilon + 2^{-k+1}](\mathbb{1}[i \notin B_{k-1}] + \mathbb{1}[i \in B_{k-1}]) \big| \mathcal{E}_1\right] \\
&= \mathbb{E}\left[\mathbb{1}[\hat{\mu}_{i_k} - \hat{\mu}_i < \epsilon + 2^{-k+1}]\mathbb{1}[i \notin B_{k-1}] \big| \mathcal{E}_1\right] \\
&\quad + \mathbb{E}\left[\mathbb{1}[\hat{\mu}_{i_k} - \hat{\mu}_i < \epsilon + 2^{-k+1}]\mathbb{1}[i \in B_{k-1}] \big| \mathcal{E}_1\right]
\end{aligned}
$$

Deterministically, $\mathbb{1}[i \notin B_k]\mathbb{1}[i \in B_{k-1}] = 0$. Therefore,

$$
\begin{aligned}
&\mathbb{E}\left[\mathbb{1}[\hat{\mu}_{i_k} - \hat{\mu}_i < \epsilon + 2^{-k+1}]\mathbb{1}[i \notin B_{k-1}] \big| \mathcal{E}_1\right] \\
&\qquad + \mathbb{E}\left[\mathbb{1}[\hat{\mu}_{i_k} - \hat{\mu}_i < \epsilon + 2^{-k+1}]\mathbb{1}[i \in B_{k-1}] \big| \mathcal{E}_1\right] \\
&= \mathbb{E}\left[\mathbb{1}[\hat{\mu}_{i_k} - \hat{\mu}_i < \epsilon + 2^{-k+1}]\mathbb{1}[i \notin B_{k-1}] \big| \mathcal{E}_1\right] \\
&= \mathbb{E}\left[\mathbb{1}[\hat{\mu}_{i_k} - \hat{\mu}_i < \epsilon + 2^{-k+1}]\mathbb{1}[i \notin B_{k-1}] \big| i \notin B_{k-1}\mathcal{E}_1\right] \mathbb{P}(i \notin B_{k-1} | \mathcal{E}_1) \\
&\qquad + \mathbb{E}\left[\mathbb{1}[\hat{\mu}_{i_k} - \hat{\mu}_i < \epsilon + 2^{-k+1}]\mathbb{1}[i \notin B_{k-1}] \big| i \in B_{k-1}, \mathcal{E}_1\right] \mathbb{P}(i \in B_{k-1} | \mathcal{E}_1) \\
&= \mathbb{E}\left[\mathbb{1}[\hat{\mu}_{i_k} - \hat{\mu}_i < \epsilon + 2^{-k+1}]\mathbb{1}[i \notin B_{k-1}] \big| i \notin B_{k-1}, \mathcal{E}_1\right] \mathbb{P}(i \notin B_{k-1} | \mathcal{E}_1) \\
&= \mathbb{E}\left[\mathbb{1}[\hat{\mu}_{i_k} - \hat{\mu}_i < \epsilon + 2^{-k+1}] \big| i \notin B_{k-1}, \mathcal{E}_1\right] \mathbb{E}\left[\mathbb{1}[i \notin B_{k-1}] \big| \mathcal{E}_1\right] \\
&= \mathbb{E}\left[\mathbb{1}[\hat{\mu}_{i_k} - \hat{\mu}_i < \epsilon + 2^{-k+1}] \big| i \notin B_{k-1}\right] \mathbb{E}\left[\mathbb{1}[i \notin B_{k-1}] \big| \mathcal{E}_1\right] \\
&\leq \left(\frac{1}{16} + \frac{\delta}{4nk^2}\right) \mathbb{E}\left[\mathbb{1}[i \notin B_{k-1}] \big| \mathcal{E}_1\right] \\
&\leq \left(\frac{1}{16} + \frac{\delta}{4nk^2}\right) \mathbb{E}\left[\mathbb{1}[\hat{\mu}_{i_k} - \hat{\mu}_i < \epsilon + 2^{-k+2}]\right]
\end{aligned}
$$

where the final inequality follows by the same argument upper bounding $\mathbb{E}\left[\mathbb{1}[i \notin B_k] \big| \mathcal{E}_1\right]$. For $k < \left\lceil \log_2\left(\frac{4}{\Delta_i - \epsilon}\right) \right\rceil$, trivially, $\mathbb{E}[\mathbb{1}[i \notin B_k]] \leq 1$. Recall $\delta \leq 1/8$. For $k \geq \left\lceil \log_2\left(\frac{4}{\Delta_i - \epsilon}\right) \right\rceil$,

$$\mathbb{E}\left[\mathbb{1}[i \notin B_k] \big| \mathcal{E}_1\right] \leq \prod_{s = \left\lceil \log_2\left(\frac{4}{\Delta_i - \epsilon}\right) \right\rceil}^{k} \left(\frac{1}{16} + \frac{\delta}{2ns^2}\right) \leq \left(\frac{1}{8}\right)^{k - \left\lceil \log_2\left(\frac{4}{\Delta_i - \epsilon}\right) \right\rceil}.$$

□

**Claim 2:** For $j \in G_\epsilon^c$, $\sum_{k=1}^\infty 2\mathbb{E}_\nu\left[\mathbb{1}[i \notin B_{k-1}]\big|\mathcal{E}_1\right]\left(2\tau_k + H_{\text{ME}}(n, 2^{-k}, 1/16)\right) \leq c''\frac{n}{(\Delta_i - \epsilon)^2} + c''h\left(0.25(\Delta_i - \epsilon), \frac{\delta}{2n}\right)$

**Proof.** This sum decomposes into two terms.

$$\sum_{k=1}^\infty \mathbb{E}_\nu\left[\mathbb{1}[i \notin B_{k-1}]\big|\mathcal{E}_1\right]\left(2\tau_k + H_{\text{ME}}(n, 2^{-k}, 1/16)\right)$$

$$= \sum_{k=1}^{\left\lfloor \log_2\left(\frac{4}{\Delta_i - \epsilon}\right)\right\rfloor} \mathbb{E}_\nu\left[\mathbb{1}[i \notin B_{k-1}]\big|\mathcal{E}_1\right]\left(H_{\text{ME}}(n, 2^{-k}, 1/16) + 2\left\lceil 2^{2k+3}\log\left(\frac{16nk^2}{\delta}\right)\right\rceil\right)$$

$$+ \sum_{k=\left\lceil \log_2\left(\frac{4}{\Delta_i - \epsilon}\right)\right\rceil}^\infty \mathbb{E}_\nu\left[\mathbb{1}[i \notin B_{k-1}]\big|\mathcal{E}_1\right]\left(H_{\text{ME}}(n, 2^{-k}, 1/16) + 2\left\lceil 2^{2k+3}\log\left(\frac{16nk^2}{\delta}\right)\right\rceil\right)$$

We begin by bounding the first term.

$$\sum_{k=1}^{\left\lfloor \log_2\left(\frac{4}{\Delta_i - \epsilon}\right)\right\rfloor} \mathbb{E}_\nu\left[\mathbb{1}[i \notin B_{k-1}]\right]\left(H_{\text{ME}}(n, 2^{-k}, 1/16) + 2\left\lceil 2^{2k+3}\log\left(\frac{16nk^2}{\delta}\right)\right\rceil\right)$$

$$\leq \sum_{k=1}^{\left\lfloor \log_2\left(\frac{4}{\Delta_i - \epsilon}\right)\right\rfloor} \left(H_{\text{ME}}(n, 2^{-k}, 1/16) + 2\left\lceil 2^{2k+3}\log\left(\frac{16nk^2}{\delta}\right)\right\rceil\right)$$

$$\leq \sum_{k=1}^{\left\lfloor \log_2\left(\frac{4}{\Delta_i - \epsilon}\right)\right\rfloor} \left(c'n2^{2k}\log(16) + 2 + 2^{2k+4}\log\left(\frac{16nk^2}{\delta}\right)\right)$$

$$\leq 2\log_2\left(\frac{4}{\Delta_i - \epsilon}\right) + \left(c'n\log(16) + 16\log\left(\frac{16n}{\delta}\right)\right)\sum_{k=1}^{\left\lfloor \log_2\left(\frac{4}{\Delta_i - \epsilon}\right)\right\rfloor} 2^{2k}$$

$$+ 32\sum_{k=1}^{\left\lfloor \log_2\left(\frac{4}{\Delta_i - \epsilon}\right)\right\rfloor} 2^{2k}\log(k)$$

$$\leq 2\log_2\left(\frac{4}{\Delta_i - \epsilon}\right) + \left(c'n\log(16) + 16\log\left(\frac{16n}{\delta}\right) + 32\log\log_2\left(\frac{4}{\Delta_i - \epsilon}\right)\right)\sum_{k=1}^{\left\lfloor \log_2\left(\frac{4}{\Delta_i - \epsilon}\right)\right\rfloor} 2^{2k}$$

$$\leq 2\log_2\left(\frac{4}{\Delta_i - \epsilon}\right) + \frac{16}{(\Delta_i - \epsilon)^2}\left(c'n\log(16) + 32\log\left(\frac{16n}{\delta}\log_2\left(\frac{4}{\Delta_i - \epsilon}\right)\right)\right)$$

Next, we plug in the bound from claim 1 controlling the probability that $i \notin B_k$.

Using Claim 1, we bound the second sum as follows:

$$\sum_{r=\left\lceil \log_2\left(\frac{4}{\Delta_i - \epsilon}\right)\right\rceil}^\infty \mathbb{E}_\nu\left[\mathbb{1}[i \notin B_{k-1}]\big|\mathcal{E}_1\right]\left(H_{\text{ME}}(n, 2^{-k}, 1/16) + 2\left\lceil 2^{2k+3}\log\left(\frac{16nk^2}{\delta}\right)\right\rceil\right)$$

$$\leq \sum_{k=\left\lceil \log_2\left(\frac{4}{\Delta_i - \epsilon}\right)\right\rceil}^\infty \left(\frac{1}{8}\right)^{k - \left\lceil \log_2\left(\frac{4}{\Delta_i - \epsilon}\right)\right\rceil - 1}\left(c'n2^{2k}\log(16) + 2 + 2^{2k+4}\log\left(\frac{16nk^2}{\delta}\right)\right)$$

$$= c'n\log(16)\sum_{k=1}^\infty \left(\frac{1}{8}\right)^{k-1} 2^{2\left(k + \left\lceil \log_2\left(\frac{4}{\Delta_i - \epsilon}\right)\right\rceil\right)} + 2\sum_{k=1}^\infty \left(\frac{1}{8}\right)^{k-1}$$

$$+ 16\sum_{k=1}^\infty \left(\frac{1}{8}\right)^{k-1} 2^{2\left(k + \left\lceil \log_2\left(\frac{4}{\Delta_i - \epsilon}\right)\right\rceil\right)}\log\left(\frac{16n\left(k + \left\lceil \log_2\left(\frac{4}{\Delta_i - \epsilon}\right)\right\rceil\right)^2}{\delta}\right)$$

$$\leq 3 + c'n \log(16) \sum_{k=1}^{\infty} 2^{-3k+3} 2^{2\left(k+\log_2\left(\frac{4}{\Delta_i - \epsilon}\right)+1\right)}$$

$$+ 16 \sum_{k=1}^{\infty} 2^{-3k+3} 2^{2\left(k+\log_2\left(\frac{4}{\Delta_i - \epsilon}\right)+1\right)} \log\left(\frac{16n\left(k+\left\lceil\log_2\left(\frac{4}{\Delta_i - \epsilon}\right)\right\rceil\right)^2}{\delta}\right)$$

$$= 3 + \left(\frac{2^9 c'n \log(16)}{(\Delta_i - \epsilon)^2} + \frac{2^{13}}{(\Delta_i - \epsilon)^2} \log\left(\frac{16n}{\delta}\right)\right) \sum_{k=1}^{\infty} 2^{-k}$$

$$+ \frac{2^{13}}{(\Delta_i - \epsilon)^2} \sum_{k=1}^{\infty} 2^{-k} \log\left(\left(k+\left\lceil\log_2\left(\frac{4}{\Delta_i - \epsilon}\right)\right\rceil\right)^2\right)$$

$$\leq 3 + \frac{2^9 c'n \log(16)}{(\Delta_i - \epsilon)^2} + \frac{2^{13}}{(\Delta_i - \epsilon)^2} \log\left(\frac{16n}{\delta}\right)$$

$$+ \frac{2^{14}}{(\Delta_i - \epsilon)^2} \sum_{k=1}^{\infty} 2^{-k} \log\left(k+\left\lceil\log_2\left(\frac{4}{\Delta_i - \epsilon}\right)\right\rceil\right)$$

$$= (**)$$

We may bound the final summand, $\sum_{k=1}^{\infty} 2^{-k} \log\left(k+\left\lceil\log_2\left(\frac{4}{\Delta_i - \epsilon}\right)\right\rceil\right)$ as follows:

$$\sum_{k=1}^{\infty} 2^{-k} \log\left(k+\left\lceil\log_2\left(\frac{4}{\Delta_i - \epsilon}\right)\right\rceil\right) \leq \log\left(\frac{e}{2} \log_2\left(\frac{256}{(\Delta_i - \epsilon)^2}\right)\right)$$

Plugging this back into $(**)$, we have that

$$(**) \leq 3 + \frac{2^9 cn \log(16)}{(\Delta_i - \epsilon)^2} + \frac{2^{13}}{(\Delta_i - \epsilon)^2} \log\left(\frac{16n}{\delta}\right) + \frac{2^{14}}{(\Delta_i - \epsilon)^2} \log\left(\frac{e}{2} \log_2\left(\frac{256}{(\Delta_i - \epsilon)^2}\right)\right)$$

Combining the above with the bound on the first sum, we have that

$$\sum_{k=1}^{\infty} \mathbb{E}_\nu\left[\mathbb{1}[i \notin B_{k-1}]\big|\mathcal{E}_1\right]\left(2\tau_k + H_{\text{ME}}(n, 2^{-k}, 1/16)\right)$$

$$\leq c''\left(\frac{n}{(\Delta_i - \epsilon)^2} + \frac{c}{(\Delta_i - \epsilon)^2} \log\left(\frac{2n}{\delta} \log_2\left(\frac{4}{(\Delta_i - \epsilon)^2}\right)\right)\right)$$

$$= \frac{c''n}{(\Delta_i - \epsilon)^2} + c''h\left(0.25(\Delta_i - \epsilon), \frac{\delta}{2n}\right)$$

for a sufficiently large, universal constant $c''$ and $c$ from the definition of $h(\cdot, \cdot)$. $\qquad\square$

### E.3.11 Step 10: Applying the result of Step 9 to the result of Step 8

We may repeat the result of step 9 for every $i \in G_\epsilon^c$ and plug this into the result of Step 8. From this point, we simplify to return the final result.

By Step 8, the total number of samples $T$ drawn by FAREAST is bounded in expectation by

$$\mathbb{E}[T|\mathcal{E}_1 \cap \mathcal{E}_2] \leq c \sum_{i \in G_\epsilon} \max\left\{h\left(0.25(\epsilon - \Delta_i), \frac{\delta}{2n}\right), \min\left[h\left(0.25\Delta_i, \frac{\delta}{2n}\right), h\left(0.25\alpha_\epsilon, \frac{\delta}{2n}\right)\right]\right\}$$

$$+ c \sum_{i \in G_{\epsilon+\alpha_\epsilon}^c} h\left(0.25(\epsilon - \Delta_i), \frac{\delta}{2n}\right) + c|G_\epsilon^c \cap G_{\epsilon+\alpha_\epsilon}|h\left(0.25\alpha_\epsilon, \frac{\delta}{2n}\right)$$

$$+ 2 \sum_{i \in G_\epsilon^c} \sum_{k=1}^{\infty} \mathbb{E}_\nu\left[\mathbb{1}[i \notin B_{k-1}]\big|\mathcal{E}_1\right]\left(2\tau_k + H_{\text{ME}}(n, 2^{-k}, 1/16)\right).$$

Applying the bound from Step 9 to each $i \in G_\epsilon^c$, we have that

$$\mathbb{E}[T|\mathcal{E}_1 \cap \mathcal{E}_2] \leq c \sum_{i \in G_\epsilon} \max\left\{h\left(0.25(\epsilon - \Delta_i), \frac{\delta}{2n}\right), \min\left[h\left(0.25\Delta_i, \frac{\delta}{2n}\right), h\left(0.25\alpha_\epsilon, \frac{\delta}{2n}\right)\right]\right\}$$

$$+ c \sum_{i \in G^c_{\epsilon+\alpha_\epsilon}} h\left(0.25(\epsilon - \Delta_i), \frac{\delta}{2n}\right) + c|G^c_\epsilon \cap G_{\epsilon+\alpha_\epsilon}|h\left(0.25\alpha_\epsilon, \frac{\delta}{2n}\right)$$

$$+ 2c'' \sum_{i \in G^c_\epsilon} \frac{n}{(\Delta_i - \epsilon)^2} + h\left(0.25(\Delta_i - \epsilon), \frac{\delta}{2n}\right).$$

For $i \in G^c_\epsilon \cap G_{\epsilon+\alpha_\epsilon}$, $\alpha_\epsilon = \min_{j \in G_\epsilon} \epsilon - \Delta_j \geq \Delta_i - \epsilon$. By monotonicity of $h(\cdot, \cdot)$, $h\left(0.25\alpha_\epsilon, \frac{\delta}{2n}\right) \leq \frac{c''n}{(\Delta_i-\epsilon)^2} + c''h\left(\Delta_i - \epsilon, \frac{\delta}{2n}\right)$. Therefore,

$$\mathbb{E}[T|\mathcal{E}_1 \cap \mathcal{E}_2] \leq c \sum_{i \in G_\epsilon} \max\left\{ h\left(0.25(\epsilon - \Delta_i), \frac{\delta}{2n}\right), \min\left[h\left(0.25\Delta_i, \frac{\delta}{2n}\right), h\left(0.25\alpha_\epsilon, \frac{\delta}{2n}\right)\right]\right\}$$

$$+ (2c'' + c) \sum_{i \in G^c_\epsilon} \frac{n}{(\Delta_i - \epsilon)^2} + h\left(0.25(\Delta_i - \epsilon), \frac{\delta}{2n}\right).$$

Next, we use Lemma F.3 to bound the minimum of $h(\cdot, \cdots)$ functions.

$$c \sum_{i \in G_\epsilon} \max\left\{ h\left(0.25(\epsilon - \Delta_i), \frac{\delta}{2n}\right), \min\left[h\left(0.25\Delta_i, \frac{\delta}{2n}\right), h\left(0.25\alpha_\epsilon, \frac{\delta}{2n}\right)\right]\right\}$$

$$+ (2c'' + c) \sum_{i \in G^c_\epsilon} \frac{n}{(\Delta_i - \epsilon)^2} + h\left(0.25(\Delta_i - \epsilon), \frac{\delta}{2n}\right)$$

$$\leq c \sum_{i \in G_\epsilon} \max\left\{ h\left(0.25(\epsilon - \Delta_i), \frac{\delta}{2n}\right), h\left(\frac{\Delta_i + \alpha_\epsilon}{8}, \frac{\delta}{2n}\right)\right\}$$

$$+ (2c'' + c) \sum_{i \in G^c_\epsilon} \frac{n}{(\Delta_i - \epsilon)^2} + h\left(0.25(\Delta_i - \epsilon), \frac{\delta}{2n}\right)$$

Finally, we use Lemma F.2 to bound the function $h(\cdot, \cdot)$. Since $\delta \leq 1/2$, $\delta/n \leq 2e^{-e/2}$. Further, $\max(\Delta_i, |\epsilon - \Delta_i|) \leq 8$ for all $i$, we have that $0.25\Delta_i \leq 2$, $0.25|\epsilon - \Delta_i| \leq 2$, and $0.25\min(\alpha_\epsilon, \beta_\epsilon) \leq 2$. Therefore,

$$\mathbb{E}[T|\mathcal{E}_1 \cap \mathcal{E}_2] \leq c \sum_{i \in G_\epsilon} \max\left\{ h\left(0.25(\epsilon - \Delta_i), \frac{\delta}{2n}\right), h\left(\frac{\Delta_i + \alpha_\epsilon}{8}, \frac{\delta}{2n}\right)\right\}$$

$$+ (2c'' + c) \sum_{i \in G^c_\epsilon} \frac{n}{(\Delta_i - \epsilon)^2} + h\left(0.25(\Delta_i - \epsilon), \frac{\delta}{2n}\right)$$

$$\leq c \sum_{i \in G_\epsilon} \max\left\{ \frac{64}{(\epsilon - \Delta_i)^2} \log\left(\frac{4n}{\delta} \log_2\left(\frac{384n}{\delta(\epsilon - \Delta_i)^2}\right)\right), \right.$$

$$\left. \frac{256}{(\Delta_i + \alpha_\epsilon)^2} \log\left(\frac{4n}{\delta} \log_2\left(\frac{768n}{\delta(\Delta_i + \alpha_\epsilon)^2}\right)\right)\right\}$$

$$+ (2c'' + c) \sum_{i \in G^c_\epsilon} \frac{n}{(\Delta_i - \epsilon)^2} + \frac{64}{(\epsilon - \Delta_i)^2} \log\left(\frac{4n}{\delta} \log_2\left(\frac{384n}{\delta(\epsilon - \Delta_i)^2}\right)\right)$$

$$\leq c_3 \sum_{i \in G_\epsilon} \max\left\{ \frac{1}{(\epsilon - \Delta_i)^2} \log\left(\frac{n}{\delta} \log_2\left(\frac{n}{\delta(\epsilon - \Delta_i)^2}\right)\right), \right.$$

$$\left. \frac{1}{(\Delta_i + \alpha_\epsilon)^2} \log\left(\frac{n}{\delta} \log_2\left(\frac{n}{\delta(\Delta_i + \alpha_\epsilon)^2}\right)\right)\right\}$$

$$+ c_3 \sum_{i \in G^c_\epsilon} \frac{n}{(\Delta_i - \epsilon)^2} + \frac{1}{(\epsilon - \Delta_i)^2} \log\left(\frac{n}{\delta} \log_2\left(\frac{n}{\delta(\epsilon - \Delta_i)^2}\right)\right)$$

$$= c_3 \sum_{i \in G_\epsilon} \max\left\{ \frac{1}{(\mu_1 - \epsilon - \mu_i)^2} \log\left(\frac{n}{\delta} \log_2\left(\frac{n}{\delta(\mu_1 - \epsilon - \mu_i)^2}\right)\right), \right.$$

$$\frac{1}{(\mu_1 + \alpha_\epsilon - \mu_i)^2} \log\left(\frac{n}{\delta} \log_2\left(\frac{n}{\delta(\mu_1 + \alpha_\epsilon - \mu_i)^2}\right)\right)\Bigg\}$$

$$+ c_3 \sum_{i \in G_\epsilon^c} \frac{n}{(\mu_1 - \epsilon - \mu_i)^2} + \frac{1}{(\mu_1 - \epsilon - \mu_i)^2} \log\left(\frac{n}{\delta} \log_2\left(\frac{n}{\delta(\mu_1 - \epsilon - \mu_i)^2}\right)\right)$$

for a sufficiently large constant $c_4$.

### E.3.12  Step 11: High probability sample complexity bound

Finally, the Good Filter is equivalent to EAST, Algorithm 5, except split across rounds. Note that the Good Filter is union bounded over $2n$ events whereas the bounds in EAST are union bounded over $n$ events. The Good Filter and Bad Filter are given the same number of samples in each round, and the Good Filter can terminate within a round, conditioned on $\mathcal{E}_1 \cap \mathcal{E}_2$. Therefore, we can bound the complexity of FAREAST in terms of that of EAST run at failure probability $\delta/2$. If FAREAST terminates in the second round or later, the arguments in Steps 4 and 5 can be used to show that FAREAST draws no more than a factor of 18 more samples than EAST, though this estimate is highly pessimistic. If FAREAST terminates in round 1 (when gaps are large), we may still show that this is within a constant factor of the complexity of EAST, but the story is more complicated. In the first round, the bad filter draws at most $c'n \log(16) + 32n \log(8n/\delta)$ samples where $c'$ is the constant from Median Elimination. Since we have assumed that $\max(\Delta_i, |\epsilon - \Delta_i|) \le 8$, this sum is likewise within a constant factor of the complexity of EAST. Hence, by Theorem E.3,

$$T \le c_4 \sum_{i=1}^{n} \min\Bigg\{\max\Bigg\{\frac{1}{(\mu_1 - \epsilon - \mu_i)^2} \log\left(\frac{n}{\delta} \log_2\left(\frac{n}{\delta(\mu_1 - \epsilon - \mu_i)^2}\right)\right),$$

$$\frac{1}{(\mu_1 + \alpha_\epsilon - \mu_i)^2} \log\left(\frac{n}{\delta} \log_2\left(\frac{n}{\delta(\mu_1 + \alpha_\epsilon - \mu_i)^2}\right)\right),$$

$$\frac{1}{(\mu_1 + \beta_\epsilon - \mu_i)^2} \log\left(\frac{n}{\delta} \log_2\left(\frac{n}{\delta(\mu_1 + \beta_\epsilon - \mu_i)^2}\right)\right)\Bigg\}$$

$$\frac{1}{\gamma^2} \log\left(\frac{n}{\delta} \log_2\left(\frac{n}{\delta\gamma^2}\right)\right)\Bigg\}$$

samples. $\qquad\square$

### E.4  Proof of Theorem E.2, FAREAST in the multiplicative regime

*Proof.* **Notation for the proof:** Throughout, recall $\Delta_i = \mu_1 - \mu_i$. Recall that $t$ counts the number of times the conditional in line 19 is true. By Line 19 of FAREAST, all arms in $\mathcal{A}$ have received $t$ samples when the loop in line 23 is executed for the $t^{\text{th}}$ time. Within any round $k$, let $\mathcal{A}(t)$ and $G_k(t)$ denote the sets $\mathcal{A}$ and $G_k$ at this time since both sets can change in lines 27 and 29 and 25 respectively. Let $t_k$ denote the maximum value of $t$ in round $k$. By Lines 18 and 19 of FAREAST, the total number of samples given to the good filter when the conditional in line 19 is true for the $t^{\text{th}}$ time is $\sum_{s=1}^{t} |\mathcal{A}(s)|$.

For $i \in M_\epsilon$, let $T_i$ denote the random variable of the number of times arm $i$ is sampled before it is added to $G_k$ in Line 25. For $i \in M_\epsilon^c$, let $T_i$ denote the random variable of the number of times arm $i$ is sampled before it is removed from $\mathcal{A}$ in Line 27. For any arm $i$, let $T_i'$ denote the random variable of the of the number of times $i$ is sampled before $\hat\mu_i(t) + C_{\delta/2n}(t) \le \max_{j \in \mathcal{A}} \hat\mu_j(t) - C_{\delta/2n}(t)$.

Define the event

$$\mathcal{E}_1 = \left\{\bigcap_{i \in [n]} \bigcap_{t \in \mathbb{N}} |\hat\mu_i(t) - \mu_i| \le C_{\delta/2n}(t)\right\}.$$

Using standard anytime confidence bound results, and recalling that that $C_\delta(t) := \sqrt{\frac{4 \log(\log_2(2t)/\delta)}{t}}$, we have

$$\mathbb{P}(\mathcal{E}_1^c) = \mathbb{P}\left(\bigcup_{i \in [n]} \bigcup_{t \in \mathbb{N}} |\hat\mu_i - \mu_i| > C_{\delta/2n}(t)\right)$$

$$\leq \sum_{i=1}^{n} \mathbb{P}\left(\bigcup_{t \in \mathbb{N}} |\hat{\mu}_i - \mu_i| > C_{\delta/2n}(t)\right) \leq \sum_{i=1}^{n} \frac{\delta}{2n} = \frac{\delta}{2}$$

Next, recall that $\hat{\mu}_i(t)$ denotes the empirical average of $t$ samples of $\rho_i$. Consider the event,

$$\mathcal{E}_2 = \bigcap_{i \in M_\epsilon} \bigcap_{k \in \mathbb{N}} |((1-\epsilon)\hat{\mu}_{i_k}(\tau_k) - \hat{\mu}_i(\tau_k)) - ((1-\epsilon)\mu_{i_k} - \mu_i)| \leq 2^{-(k+1)}(2-\epsilon)$$

By Hoeffding's inequality,

$$\mathbb{P}\left(|((1-\epsilon)\hat{\mu}_{i_k}(\tau_k) - \hat{\mu}_i(\tau_k)) - ((1-\epsilon)\mu_{i_k} - \mu_i)| \leq 2^{-(k+1)}(2-\epsilon)\big| i_k = j\right) \leq \frac{\delta}{4nk^2}.$$

Then

$$\mathbb{P}\left(|((1-\epsilon)\hat{\mu}_{i_k}(\tau_k) - \hat{\mu}_i(\tau_k)) - ((1-\epsilon)\mu_{i_k} - \mu_i)| \leq 2^{-(k+1)}(2-\epsilon)\right)$$

$$= \sum_{j=1}^{n} \mathbb{P}\left(|((1-\epsilon)\hat{\mu}_{i_k}(\tau_k) - \hat{\mu}_i(\tau_k)) - ((1-\epsilon)\mu_{i_k} - \mu_i)| \leq 2^{-(k+1)}(2-\epsilon)\big| i_k = j\right)\mathbb{P}(i_k = j)$$

$$\leq \frac{\delta}{4nk^2} \sum_{j=1}^{n} \mathbb{P}(i_k = j)$$

$$= \frac{\delta}{4nk^2}$$

Therefore, union bounding over the rounds $k \in \mathbb{N}$, $\mathbb{P}(\mathcal{E}_2^c) \leq \sum_{i \in M_\epsilon} \sum_{k=1}^{\infty} \frac{\delta}{4nk^2} \leq \frac{\delta}{2}$. Hence, $\mathbb{P}(\mathcal{E}_1 \cap \mathcal{E}_2) \geq 1 - \delta$.

### E.4.1 Step 0: Correctness.

On $\mathcal{E}_1 \cap \mathcal{E}_2$, first we prove that if there exists a random round $k$ at which $G_k \cup B_k = [n]$ then $G_k = M_\epsilon$. Additionally, we prove that on $\mathcal{E}_1 \cap \mathcal{E}_2$, if $\mathcal{A} \subset G_k$, then $G_k = M_\epsilon$. Therefore, for either stopping condition for FAREAST in line 31, on the event $\mathcal{E}_1 \cap \mathcal{E}_2$, FAREAST correctly returns the set $M_\epsilon$.

**Claim 0:** On $\mathcal{E}_1 \cap \mathcal{E}_2$, for all $k \in \mathbb{N}$, $G_k \subset M_\epsilon$.

**Proof.** Firstly we show $1 \in \mathcal{A}$ for all $t \in \mathbb{N}$, namely the best arm is never removed from $\mathcal{A}$. Note for any $i$ such that $\hat{\mu}_i(t) - C_{\delta/2n}(t) \geq 0$,

$$\hat{\mu}_1 + C_{\delta/2n}(t) \geq \mu_1 \geq \mu_i \geq \hat{\mu}_i(t) - C_{\delta/2n}(t) > (1-\epsilon)(\hat{\mu}_i(t) - C_{\delta/2n}(t)).$$

For $i$ such that $\hat{\mu}_i(t) - C_{\delta/2n}(t) < 0$, if $\hat{\mu}_1 + C_{\delta/2n}(t) \geq 0$, then

$$\hat{\mu}_1 + C_{\delta/2n}(t) \geq 0 > (1-\epsilon)(\hat{\mu}_i(t) - C_{\delta/2n}(t)).$$

Note that $\hat{\mu}_1 + C_{\delta/2n}(t) < 0$ implies on the event $\mathcal{E}_1$ that $\mu_1 < 0$, which contradicts the assumption that $\mu_1 \geq 0$ made in the theorem. In particular this shows, $\hat{\mu}_1 + C_{\delta/2n}(t) > (1-\epsilon)(\max_{i \in \mathcal{A}} \hat{\mu}_i(t) - C_{\delta/2n}(t)) = L_t$ and $\hat{\mu}_1 + C_{\delta/2n}(t) \geq \max_{i \in \mathcal{A}} \hat{\mu}_i(t) - C_{\delta/2n}(t)$ showing that 1 will never exit $\mathcal{A}$ in line 28.

Secondly, we show that at all times $t$, $(1-\epsilon)\mu_1 \in [L_t, U_t]$. By the above, since $\mu_1$ never leaves $\mathcal{A}$,

$$U_t = (1-\epsilon)(\max_{i \in \mathcal{A}} \hat{\mu}_i(t) + C_{\delta/2n}(t)) \geq (1-\epsilon)(\hat{\mu}_1(t) + C_{\delta/2n}(t)) \geq (1-\epsilon)\mu_1$$

and for any $i$,

$$(1-\epsilon)\mu_1 \geq (1-\epsilon)\mu_i \geq (1-\epsilon)(\hat{\mu}_i(t) - C_{\delta/2n}(t))$$

Hence $(1-\epsilon)\mu_1 \geq (1-\epsilon)(\max_i \hat{\mu}_i(t) - C_{\delta/2n}(t)) = L_t$.

Next, we show that $G_k \subset M_\epsilon$ for all $k \geq 1, t \geq 1$. Suppose not. Then $\exists, k, t \in N$ and $\exists i \in M_\epsilon^c \cap G_k(t)$ such that,

$$\mu_i \geq \hat{\mu}_i(t) - C_{\delta/2n}(t) \geq U_t \geq (1-\epsilon)\mu_1 > \mu_i,$$

with the last inequality following from the previous assertion, giving a contradiction. $\square$

**Claim 1:** On $\mathcal{E}_1 \cap \mathcal{E}_2$, for all $k \in \mathbb{N}$, $B_k \subset M_\epsilon^c$.

**Proof.** Next, we show $B_k \subset M_\epsilon^c$. Suppose not. Then either the good filter or the bad filter added an arm in $M_\epsilon$ to $B_k$. Take $i \in M_\epsilon$. In the former, this implies that

$$\mu_i \overset{\mathcal{E}_1}{\leq} \hat{\mu}_i(t) + C_{\delta/2n}(t) < L_t \overset{\mathcal{E}_1}{\leq} (1-\epsilon)\mu_1$$

which contradicts $i \in M_\epsilon$. Consider the alternate case that the bad filter adds $i$ to $B_k$ for some $k$. By definition, $B_0 = \emptyset$ and $B_{k-1} \subset B_k$ for all $k$. Then there must exist $k \in \mathbb{N}$ and an $i \in M_\epsilon$ such that $i \in B_k$ and $i \notin B_{k-1}$. Following line 14 of the algorithm, this occurs if and only if

$$(1-\epsilon)\hat{\mu}_{i_k} - \hat{\mu}_i > 2^{-(k+1)}(2-\epsilon).$$

On the event $\mathcal{E}_2$, the above implies

$$(1-\epsilon)\mu_{i_k} - \mu_i + 2^{-(k+1)}(2-\epsilon) > 2^{-(k+1)}(2-\epsilon),$$

and simplifying, we see that $0 < (1-\epsilon)\mu_{i_k} - \mu_i \leq (1-\epsilon)\mu_1 - \mu_i$ which contradicts the assertion that $i \in M_\epsilon$. Combining the above claims, we see that $\mathcal{E}_1 \cap \mathcal{E}_2$ implies $(G_k \cup B_k = [n])$ and $G_k \cap B_k = \emptyset \implies G_k = M_\epsilon$. Since $\mathbb{P}(\mathcal{E}_1 \cap \mathcal{E}_2) \geq 1 - \delta$, if FAREAST terminates, with probability at least $1 - \delta$, it correctly returns the set $M_\epsilon$. $\square$

**Claim 2:** Next, we show that on $\mathcal{E}_1$, $M_\epsilon \subset \mathcal{A}(t) \cup G(t)$ for all $t \in \mathbb{N}$.

In particular this implies that if $\mathcal{A} \subset G$, then $M_\epsilon \subset G$. Combining this with the previous claim gives $G \subset M_\epsilon \subset G$, hence $G = M_\epsilon$. On this condition, FAREAST terminates by line 33 and returns the set $\mathcal{A} \cup G = G$. Note that by definition, $M_\epsilon \subset M_{(\epsilon+\gamma)}$ for all $\gamma \geq 0$. Therefore FAREAST terminates correctly on this condition.

**Proof.** Suppose for contradiction that there exists $i \in M_\epsilon$ such that $i \notin \mathcal{A}(t) \cup G(t)$. This occurs only if $i$ is eliminated in line 28. Hence, there exists a $t' \leq t$ such that $\hat{\mu}_i(t') + C_{\delta/n}(t') < L_{t'}$. Therefore, on the event $\mathcal{E}_1$,

$$(1-\epsilon)\mu_1 \overset{\mathcal{E}_1}{\geq} L_{t'} = (1-\epsilon)\left(\max_{j \in \mathcal{A}} \hat{\mu}_j(t') - C_{\delta/n}(t')\right) > \hat{\mu}_i(t') + C_{\delta/n}(t') \overset{\mathcal{E}_1}{\geq} \mu_i$$

which contradicts $i \in M_\epsilon$. $\square$

**Claim 3:** Finally, we show that on $\mathcal{E}_1$, if $U_t - L_t \leq \frac{\gamma}{2-\epsilon}L_t$, then $\mathcal{A} \cup G \subset M_{(\epsilon+\gamma)}$.

Combining with Claim 3 that $M_\epsilon \subset \mathcal{A} \cup G$, if FAREAST terminates on this condition by line 33, it does so correctly and returns all arms in $M_\epsilon$ and none in $M_{(\epsilon+\gamma)}^c$.

**Proof.** By Claim 0, $G \subset M_\epsilon \subset M_{\epsilon+\gamma}$. Hence, $G \cap M_{(\epsilon+\gamma)}^c = \emptyset$. Therefore, we wish to show that $\mathcal{A} \cap M_{(\epsilon+\gamma)}^c = \emptyset$ which implies that $G \cap \mathcal{A} \subset M_{\epsilon+\gamma}$. Assume $U_t - L_t < \frac{\gamma}{2-\epsilon}L_t$. Recall that

$$U_t = (1-\epsilon)\left(\max_{i \in \mathcal{A}} \hat{\mu}_i(t) + C_{\delta/2n}(t)\right)$$

and

$$L_t = (1-\epsilon)\left(\max_{i \in \mathcal{A}} \hat{\mu}_i(t) - C_{\delta/2n}(t)\right)$$

All arms in $\mathcal{A}(t)$ have received exactly $t$ samples. Hence, $U_t - L_t = 2(1-\epsilon)C_{\delta/2n}(t)$. On $\mathcal{E}_1$, $L_t \leq (1-\epsilon)\mu_1$ This implies that

$$2(1-\epsilon)C_{\delta/2n}(t) < \frac{\gamma}{2-\epsilon}L_t \leq \frac{1-\epsilon}{2-\epsilon}\gamma\mu_1,$$

and in particular,

$$2C_{\delta/2n}(t) < \frac{\gamma\mu_1}{2-\epsilon}.$$

Therefore, we wish to show that when the above is true, then for any $i \in M_{\epsilon+\gamma}^c$, $L_t - (\hat{\mu}_i(t) + C_{\delta/n}(t)) > 0$, implying that $i \notin \mathcal{A}$.

$$L_t - (\hat{\mu}_i(t) + C_{\delta/n}(t)) = (1-\epsilon)\left(\max_{j \in \mathcal{A}} \hat{\mu}_j - C_{\delta/n}(t)\right) - (\hat{\mu}_i(t) + C_{\delta/n}(t))$$

$$\geq (1-\epsilon)\left(\max_{j\in\mathcal{A}}\mu_j - 2C_{\delta/n}(t)\right) - (\mu_i + 2C_{\delta/n}(t))$$

$$\overset{(a)}{\geq} (1-\epsilon)\left(\mu_1 - 2C_{\delta/n}(t)\right) - ((1-\epsilon-\gamma)\mu_1 + 2C_{\delta/n}(t))$$

$$= \gamma\mu_1 - 2(2-\epsilon)C_{\delta/n}(t)$$

$$> \gamma\mu_1 - (2-\epsilon)\frac{\gamma\mu_1}{2-\epsilon}$$

$$= 0$$

which implies that $i \notin \mathcal{A}$. Inequality $(a)$ follows jointly from the fact that $1 \in \mathcal{A}$ and the fact that all arms in $\mathcal{A}$ have received $t$ samples implies $\max_{j\in\mathcal{A}}\mu_j - 2C_{\delta/n}(t) = \mu_1 - 2C_{\delta/n}(t)$. Additionally, inequality $(a)$ follows from $\mu_i \leq (1-\epsilon-\gamma)\mu_1$ since $i \in M^c_{\epsilon+\gamma}$. $\qquad\square$

### E.4.2 Step 1: An expression for the total number of samples drawn and introducing several helper random variables

Next, we write an expression for the total number of samples drawn by FAREAST. In particular, we introduce two sums that we will spend the remainder of the proof controlling. Additionally, we show that the conditional in line 19 in the good filter is true at least once in each round. Based on this, we more precisely define the random variables $T_i$ and $T'_i$ introduced in the notation section in section E.4. Additionally, we introduce the time $T_\gamma$ at which $U_t - L_t < \frac{\gamma}{2-\epsilon}L_t$.

Recall that the largest value of $t$ in round $k$ is denoted $t_k$. Let $E^\gamma_k$ be the event that $U_t - L_t \geq \frac{\gamma}{2-\epsilon}L_t$ for all $t$ in round $k$:

$$E^\gamma_k := \left\{ U_t - L_t \geq \frac{\gamma}{2-\epsilon}L_t : t \in (t_{k-1}, t_k] \right\}.$$

Note that if $E^\gamma_{k-1}$ is false, then FAREAST terminates in round $k-1$ by line 33. We may write the total number of samples drawn by the algorithm as

$$T = \sum_{k=1}^{\infty} 2\mathbb{1}\left[\mathcal{A} \not\subset G_{k-1} \text{ and } G_{k-1} \cup B_{k-1} \neq [n] \text{ and } E^\gamma_{k-1}\right]$$
$$\left(H_{\mathrm{ME}}(n, 2^{-k}, 1/16) + \tau_k + \tau_k |(G_{k-1} \cup B_{k-1})^c|\right)$$

Deterministically, $\mathbb{1}\left[\mathcal{A} \not\subset G_{k-1} \text{ and } G_{k-1} \cup B_{k-1} \neq [n] \text{ and } E^\gamma_{k-1}\right] \leq \mathbb{1}\left[G_{k-1} \cup B_{k-1} \neq [n]\right].$

Applying this,

$$T \leq \sum_{k=1}^{\infty} 2\mathbb{1}\left[G_{k-1} \cup B_{k-1} \neq [n]\right]\left(H_{\mathrm{ME}}(n, 2^{-k}, 1/16) + \tau_k + \tau_k |(G_{k-1} \cup B_{k-1})^c|\right)$$

$$= \sum_{k=1}^{\infty} 2\mathbb{1}\left[G_{k-1} \neq M_\epsilon\right]\mathbb{1}\left[G_{k-1} \cup B_{k-1} \neq [n]\right]\left(H_{\mathrm{ME}}(n, 2^{-k}, 1/16) + \tau_k + \tau_k |(G_{k-1} \cup B_{k-1})^c|\right)$$

$$\tag{17}$$

$$+ \sum_{k=1}^{\infty} 2\mathbb{1}\left[G_{k-1} = M_\epsilon\right]\mathbb{1}\left[G_{k-1} \cup B_{k-1} \neq [n]\right]\left(H_{\mathrm{ME}}(n, 2^{-k}, 1/16) + \tau_k + \tau_k |(G_{k-1} \cup B_{k-1})^c|\right)$$

$$\tag{18}$$

In round $k$, line 18 of the Good Filter, whereby an arm is sampled, is evaluated

$$\left(H_{\mathrm{ME}}(n, 2^{-k}, 1/16) + \tau_k + \tau_k |(G_{k-1} \cup B_{k-1})^c|\right) \geq (2\tau_k + H_{\mathrm{ME}}(n, 2^{-k}, 1/16)) \geq n$$

times since $H_{\mathrm{ME}}(n, 2^{-k}, 1/16)) \geq n$ for all $k$ and $|(G_{k-1} \cup B_{k-1})^c| \geq 1$ unless $G_{k-1} \cup B_{k-1} = [n]$ which implies termination in round $k-1$. Each time line 18 is called, $N_{I_s} \leftarrow N_{I_s} + 1$. Since $|\arg\min_{j\in\mathcal{A}}\{N_j\}| \leq |\mathcal{A}| \leq n$, line 18 is called at most $n$ times before $\min_{j\in\mathcal{A}}\{N_j\} = \max_{j\in\mathcal{A}}\{N_j\}$. When this occurs, the conditional in line 19 is true and $t \leftarrow t+1$.

If $\min_{i \in \mathcal{A}(t)}\{N_i\} = \max_{i \in \mathcal{A}(t)}\{N_i\}$, then $N_i = t$ for any $i \in \mathcal{A}(t)$. By Step 0, only arms in $M_\epsilon$ are added to $G_k$. Therefore, $T_i$ is defined as

$$T_i = \min\left\{t : \begin{array}{ll} i \in G_k(t+1) & \text{if } i \in M_\epsilon \\ i \notin \mathcal{A}(t+1) & \text{if } i \in M_\epsilon^c \end{array}\right\} \overset{\mathcal{E}_1}{=} \min\left\{t : \begin{array}{ll} \hat{\mu}_i - C_{\delta/2n}(t) \geq U_t & \text{if } i \in M_\epsilon \\ \hat{\mu}_i + C_{\delta/2n}(t) \leq L_t & \text{if } i \in M_\epsilon^c \end{array}\right\} \quad (19)$$

Define $T_i = \infty$ if this never occurs. Note that this may happen if FAREAST terminates due to the conditition in line 32 that $U_t - L_t < \frac{\gamma}{2-\epsilon}L_t$. Similarly, recall $T_i'$ denotes the random variable of the of the number of times $i$ is sampled before $\hat{\mu}_i(t) + C_{\delta/2n}(t) \leq \max_{j \in \mathcal{A}} \hat{\mu}_j(t) - C_{\delta/2n}(t)$. Hence,

$$T_i' = \min\left\{t : \hat{\mu}_i(t) + C_{\delta/2n}(t) \leq \max_{j \in \mathcal{A}(t)} \hat{\mu}_j(t) - C_{\delta/2n}(t)\right\} \quad (20)$$

Define $T_i' = \infty$ if this never occurs. Note that this may happen if FAREAST terminates due to the conditition in line 32 that $U_t - L_t < \frac{\gamma}{2-\epsilon}L_t$. Finally, we define the time $T_\gamma$ such that $U_t - L_t < \frac{\gamma}{2-\epsilon}L_t$.

$$T_\gamma = \min\left\{t : U_t - L_t < \frac{\gamma}{2-\epsilon}L_t\right\} \quad (21)$$

By design, no arm is sampled more that $T_\gamma$ times by the good filter, controlling the cases that $T_i$ or $T_i'$ are infinite.

### E.4.3 Step 2: Bounding $T_i$ and $T_i'$ for $i \in M_\epsilon$

**Step 2a:** For $i \in M_\epsilon$, we have that $T_i \leq h\left(\frac{\epsilon\mu_1 - \Delta_i}{4 - 2\epsilon}, \frac{\delta}{2n}\right)$.

**Proof.** Note that $\mu_i - 2C_{\delta/2n}(t) \geq (1-\epsilon)(\mu_1 + 2C_{\delta/2n}(t))$ may be rearranged as $(4-2\epsilon)C_{\delta/2n}(t) \leq \epsilon\mu_1 - \Delta_i$, and this is true when $t > h\left(\frac{\epsilon\mu_1 - \Delta_i}{4 - 2\epsilon}, \frac{\delta}{2n}\right)$. This condition implies that for all $j$,

$$\begin{aligned} \hat{\mu}_i(t) - C_{\delta/2n}(t) &\overset{\mathcal{E}_1}{\geq} \mu_i - 2C_{\delta/2n}(t) \\ &\geq (1-\epsilon)(\mu_1 + 2C_{\delta/2n}(t)) \\ &\geq (1-\epsilon)(\mu_j + 2C_{\delta/2n}(t)) \\ &\overset{\mathcal{E}_1}{\geq} (1-\epsilon)(\hat{\mu}_j(t) + C_{\delta/2n}(t)) \end{aligned}$$

so in particular, $\hat{\mu}_i(t) - C_{\delta/2n}(t) \geq (1-\epsilon)(\max_{j \in \mathcal{A}} \hat{\mu}_j(t) + C_{\delta/2n}(t)) = U_t$. $\qquad \square$

Additionally, we define a time $T_{\max}$ when all good arms have entered $G_k$.

**Step 2b:** Defining $T_{\max} := \min\{t : G_k(t) = M_\epsilon\} = \max_{i \in M_\epsilon} T_i$, we also have that $T_{\max} \leq h(\tilde{\alpha}_\epsilon/(4-2\epsilon), \delta/2n)$ (in other words, if $t > h(\tilde{\alpha}_\epsilon/(4-2\epsilon), \delta/2n)$ (i.e. line 23 has been run $t$ times, then we have that $G_k(t) = M_\epsilon$).

**Proof.** Recall that $\tilde{\alpha}_\epsilon = \min_{i \in M_\epsilon} \mu_i - \mu_1 + \epsilon = \min_{i \in M_\epsilon} \epsilon\mu_1 - \Delta_i$. By Step 1a, $T_i \leq h\left(\frac{\epsilon\mu_1 - \Delta_i}{4 - 2\epsilon}, \frac{\delta}{2n}\right)$. Furthermore, $h(\cdot, \cdot)$ is monotonic in its first argument, such that if $0 < x' < x$, then $h(x', \delta) > h(x, \delta)$ for any $\delta > 0$. Therefore $T_{\max} = \max_{i \in M_\epsilon} T_i \leq \max_{i \in M_\epsilon} h\left(\frac{\epsilon\mu_1 - \Delta_i}{4 - 2\epsilon}, \frac{\delta}{2n}\right) = h\left(\tilde{\alpha}_\epsilon/(4-2\epsilon), \frac{\delta}{2n}\right)$. $\qquad \square$

**Step 2c:** For $i \in M_\epsilon$, we have that $T_i' \leq h(0.25\Delta_i, \delta/2n)$.

**Proof.** Note that $4C_{\delta/2n}(t) \leq \mu_1 - \mu_i$, true when $t > h\left(0.25\Delta_i, \frac{\delta}{2n}\right)$, implies that

$$\begin{aligned} \hat{\mu}_i(t) + C_{\delta/2n}(t) &\overset{\mathcal{E}_1}{\leq} \mu_i + 2C_{\delta/2n}(t) \\ &\leq \mu_1 - 2C_{\delta/2n}(t) \\ &\overset{\mathcal{E}_1}{\leq} \hat{\mu}_1(t) - C_{\delta/2n}(t). \end{aligned}$$

As shown in Step 0, $1 \in \mathcal{A}(t)$ for all $t \in \mathbb{N}$, and in particular $\hat{\mu}_1(t) \leq \max_{i \in \mathcal{A}(t)} \hat{\mu}_i(t)$. Hence, $\hat{\mu}_i(t) + C_{\delta/2n}(t) \leq \max_{j \in \mathcal{A}(t)} \hat{\mu}_j(t) - C_{\delta/2n}(t)$. $\qquad \square$

### E.4.4  Step 3: Bounding $T_i$ for $i \in M_\epsilon^c$

Next, we bound $T_i$ for $i \in M_\epsilon^c$. $i \in M_\epsilon^c$ is eliminated from $\mathcal{A}$ if it has received at least $T_i$ samples.

**Claim:** $T_i \leq h\left(\frac{\Delta_i - \epsilon\mu_1}{4 - 2\epsilon}, \frac{\delta}{2n}\right)$ for $i \in M_\epsilon^c$

**Proof.** Note that $\mu_i + 2C_{\delta/2n}(t) \leq (1-\epsilon)(\mu_1 - 2C_{\delta/2n}(t))$ may be rearranged as $(4-2\epsilon)C_{\delta/2n}(t) \leq \Delta_i - \epsilon\mu_1$, and this is true when $t > h\left(\frac{\epsilon\mu_1 - \Delta_i}{4 - 2\epsilon}, \frac{\delta}{2n}\right)$. This condition implies that

$$
\begin{aligned}
\hat{\mu}_i(t) + C_{\delta/2n}(t) &\overset{\mathcal{E}_1}{\leq} \mu_i + 2C_{\delta/2n}(t) \\
&\leq (1-\epsilon)(\mu_1 - 2C_{\delta/2n}(t)) \\
&\overset{\mathcal{E}_1}{\leq} (1-\epsilon)(\hat{\mu}_1(t) - C_{\delta/2n}(t))
\end{aligned}
$$

As shown in Step 0, $1 \in \mathcal{A}(t)$ for all $t \in \mathbb{N}$, and in particular $\hat{\mu}_1(t) \leq \max_{i \in \mathcal{A}(t)} \hat{\mu}_i(t)$. Therefore $\hat{\mu}_i(t) + C_{\delta/2n}(t) \leq (1-\epsilon)(\max_{j \in \mathcal{A}} \hat{\mu}_j(t) - C_{\delta/2n}(t)) = L_t$.  $\square$

### E.4.5  Step 4: bounding the total number of samples given to the good filter at time $t = T_{\max}$

Note that for a time $t = T$, the total number of samples given to the good filter is $\sum_{s=1}^{T} |\mathcal{A}(s)|$. Therefore, the total number of samples up to time $T_{\max}$ is $\sum_{t=1}^{T_{\max}} |\mathcal{A}(t)|$.

Let $S_i = \min\{t : i \notin A(t+1)\}$. Hence,

$$
\sum_{t=1}^{T_{\max}} |\mathcal{A}(t)| = \sum_{t=1}^{T_{\max}} \sum_{i=1}^{n} \mathbb{1}[i \in \mathcal{A}(t)] = \sum_{i=1}^{n} \sum_{t=1}^{T_{\max}} \mathbb{1}[i \in \mathcal{A}(t)] = \sum_{i=1}^{n} \min\{T_{\max}, S_i\}
$$

For arms $i \in M_\epsilon^c$, $S_i = T_i$ by definition. For $i \in M_\epsilon$, $S_i = \max(T_i, T_i')$ by line 28 of the algorithm. Then

$$
\begin{aligned}
\sum_{i=1}^{n} \min\{T_{\max}, S_i\} &= \sum_{i \in M_\epsilon} \min\{\mathcal{T}_{\max}, \max(T_i, T_i')\} + \sum_{i \in M_\epsilon^c} \min\{T_{\max}, T_i\} \\
&\leq \sum_{i \in M_\epsilon} \min\{T_{\max}, \max(T_i, T_i')\} + |M_\epsilon^c \cap M_{\epsilon + \tilde{\alpha}_\epsilon}|T_{\max} + \sum_{i \in M_{\epsilon + \tilde{\alpha}_\epsilon}^c} T_i \\
&= \sum_{i \in M_\epsilon} \max\{T_i, \min(T_i', T_{\max})\} + |M_\epsilon^c \cap M_{\epsilon + \tilde{\alpha}_\epsilon/\mu_1}|T_{\max} + \sum_{i \in M_{\epsilon + \tilde{\alpha}_\epsilon/\mu_1}^c} T_i \\
&\overset{(a)}{\leq} \sum_{i \in M_\epsilon} \max\left\{h\left(\frac{\epsilon\mu_1 - \Delta_i}{4 - 2\epsilon}, \frac{\delta}{2n}\right), \min\left[h\left(0.25\Delta_i, \frac{\delta}{2n}\right), h\left(\frac{\tilde{\alpha}_\epsilon}{4 - 2\epsilon}, \frac{\delta}{2n}\right)\right]\right\} \\
&\quad + \sum_{i \in M_{\epsilon + \tilde{\alpha}_\epsilon/\mu_1}^c} h\left(\frac{\epsilon\mu_1 - \Delta_i}{4 - 2\epsilon}, \frac{\delta}{2n}\right) + |M_\epsilon^c \cap M_{\epsilon + \tilde{\alpha}_\epsilon/\mu_1}|h\left(\frac{\tilde{\alpha}_\epsilon}{4 - 2\epsilon}, \frac{\delta}{2n}\right).
\end{aligned}
$$

Equality $(a)$ follows from $T_{\max} \leq h\left(\frac{\tilde{\alpha}_\epsilon}{4 - 2\epsilon}, \frac{\delta}{2n}\right)$ by Step 1b, $T_i \leq h\left(\frac{\epsilon\mu_1 - \Delta_i}{4 - 2\epsilon}, \frac{\delta}{2n}\right)$ in Steps 2a and 3, and $T_i' \leq h\left(0.25\Delta_i, \frac{\delta}{2n}\right)$ in Step 2c.

### E.4.6  Step 5: Bounding the number of samples in round $k$ versus $k-1$

Now we show that the total number of samples taken in round $k$ is no more than 9 times the number taken in the previous round.

**Claim:** For $k > 1$

$$
\left(H_{\text{ME}}(n, 2^{-k}, 1/16) + \tau_k + \tau_k |(G_{k-1} \cup B_{k-1})^c|\right) \leq 9\left(H_{\text{ME}}(n, 2^{-k+1}, 1/16) + \tau_{k-1} + \tau_{k-1} |(G_{k-2} \cup B_{k-2})^c|\right)
$$

**Proof.** In round $k$, $\left(H_{\text{ME}}(n, 2^{-k}, 1/16) + \tau_k + \tau_k |(G_{k-1} \cup B_{k-1})^c|\right)$ samples are drawn. Since $G_{k-1} \subset G_k$ and $B_{k-1} \subset B_k$ $\forall k$ deterministically, we see that $|(G_{k-1} \cup B_{k-1})^c| \geq |(G_k \cup B_k)^c|$ $\forall k$.

By definition,
$$H_{\text{ME}}(n, 2^{-k-1}, 1/16) = 4H_{\text{ME}}(n, 2^{-k}, 1/16).$$

Next, recall $\tau_k = \left\lceil 2^{2k+3} \log\left(\frac{8}{\delta_k}\right) \right\rceil$. We bound $\tau_k/\tau_{k-1}$ as

$$
\begin{aligned}
\frac{\tau_k}{\tau_{k-1}} &= \frac{\left\lceil 2^{2k+3} \log\left(\frac{8}{\delta_k}\right) \right\rceil}{\left\lceil 2^{2k+1} \log\left(\frac{8}{\delta_{k-1}}\right) \right\rceil} = \frac{\left\lceil 2^{2k+3} \log\left(\frac{16nk^2}{\delta}\right) \right\rceil}{\left\lceil 2^{2k+1} \log\left(\frac{16n(k-1)^2}{\delta}\right) \right\rceil} \\
&\leq \frac{2^{2k+3} \log\left(\frac{16nk^2}{\delta}\right) + 1}{2^{2k+1} \log\left(\frac{16n(k-1)^2}{\delta}\right)} \leq \frac{4 \log\left(\frac{16nk^2}{\delta}\right)}{\log\left(\frac{16n(k-1)^2}{\delta}\right)} + 1 \\
&\leq 4\frac{\log\left(\frac{16n}{\delta}\right) + 2\log(k)}{\log\left(\frac{16n}{\delta}\right) + 2\log(k-1)} + 1 = (*)
\end{aligned}
$$

If $k = 2$, $(*) \leq 1 + 4 * \log\{32\}/\log(8) \leq 9$. Otherwise,

$$
\begin{aligned}
(*) &= \frac{4(\log\left(\frac{16n}{\delta}\right) + 2\log(k))}{\log\left(\frac{16n}{\delta}\right) + 2\log(k-1)} + 1 \\
&\leq \frac{4\log(k)}{\log(k-1)} + 1 \\
&\leq 4 \cdot 2 + 1 = 9
\end{aligned}
$$

Putting these pieces together,

$$
\begin{aligned}
&\left(H_{\text{ME}}(n, 2^{-k}, 1/16) + \tau_k + \tau_k|(G_{k-1} \cup B_{k-1})^c|\right) \\
&\leq \left(4H_{\text{ME}}(n, 2^{-k+1}, 1/16) + 9\tau_{k-1} + 9\tau_{k-1}|(G_{k-2} \cup B_{k-2})^c|\right) \\
&\leq 9\left(H_{\text{ME}}(n, 2^{-k+1}, 1/16) + \tau_{k-1} + \tau_{k-1}|(G_{k-2} \cup B_{k-2})^c|\right)
\end{aligned}
$$

$\square$

### E.4.7  Step 6: Bounding Equation (17)

Here, we introduce the round $K_{\text{Good}}$, when $G_{K_{\text{Good}}} = M_\epsilon$ at some point within the round. Using the result of the previous step, we may bound the total number of samples taken though this round, controlling Equation (17).

With the result of Step 5, we prove the following inequality.

**Claim:**

$$
\sum_{k=1}^{\infty} 2\mathbb{1}\left[G_{k-1} \neq M_\epsilon\right] \mathbb{1}[G_{k-1} \cup B_{k-1} \neq [n]] \left(H_{\text{ME}}(n, 2^{-k}, 1/16) + \tau_k + \tau_k|(G_{k-1} \cup B_{k-1})^c|\right)
$$

$$(22)$$

$$
\leq c \sum_{i \in M_\epsilon} \max\left\{h\left(\frac{\epsilon\mu_1 - \Delta_i}{4 - 2\epsilon}, \frac{\delta}{2n}\right), \min\left[h\left(0.25\Delta_i, \frac{\delta}{2n}\right), h\left(0.25\frac{\tilde{\alpha}_\epsilon}{4 - 2\epsilon}, \frac{\delta}{2n}\right)\right]\right\}
$$

$$
+ c|M_\epsilon^c \cap M_{\epsilon+\tilde{\alpha}_\epsilon/\mu_1}|h\left(\frac{\tilde{\alpha}_\epsilon}{4 - 2\epsilon}, \frac{\delta}{2n}\right) + c \sum_{i \in M_{\epsilon+\tilde{\alpha}_\epsilon/\mu_1}^c} h\left(\frac{\epsilon\mu_1 - \Delta_i}{4 - 2\epsilon}, \frac{\delta}{2n}\right)
$$

**Proof.** Recall $t_k = \max\{t : t \in k\}$ denotes the maximum value of $t$ in round $k$ and $T_{\max} = \max_{\in M_\epsilon} T_i$ denotes the minimum $t$ such that $G_k(t) = M_\epsilon$. Define the random round

$$
K_{\text{Good}} := \min\{k : G_k = M_\epsilon\} = \min\{k : t_k \geq T_{\max}\}
$$

By definition of $K_{\mathrm{Good}}$,

$$\sum_{k=1}^{\infty} 2\mathbb{1}[G_{k-1} \neq M_\epsilon]\mathbb{1}[G_{k-1} \cup B_{k-1} \neq [n]] \left(H_{\mathrm{ME}}(n, 2^{-k}, 1/16) + \tau_k + \tau_k|(G_{k-1} \cup B_{k-1})^c|\right)$$

$$= \sum_{k=1}^{K_{\mathrm{Good}}} 2\mathbb{1}[G_{k-1} \cup B_{k-1} \neq [n]] \left(H_{\mathrm{ME}}(n, 2^{-k}, 1/16) + \tau_k + \tau_k|(G_{k-1} \cup B_{k-1})^c|\right).$$

Next, applying Step 5, if $K_{\mathrm{Good}} > 1$

$$\sum_{k=1}^{K_{\mathrm{Good}}} 2\mathbb{1}[G_{k-1} \cup B_{k-1} \neq [n]] \left(H_{\mathrm{ME}}(n, 2^{-k}, 1/16) + \tau_k + \tau_k|(G_{k-1} \cup B_{k-1})^c|\right)$$

$$\leq 18 \sum_{k=1}^{K_{\mathrm{Good}}-1} \mathbb{1}[G_{k-2} \cup B_{k-2} \neq [n]] \left(H_{\mathrm{ME}}(n, 2^{-k+1}, 1/16) + \tau_{k-1} + \tau_{k-1}|(G_{k-2} \cup B_{k-2})^c|\right).$$

Observe that by lines 17 and 20 of FAREAST, for any round $r$ and for any $t > t_{r-1}$,

$$\sum_{k=1}^{r-1} \mathbb{1}[G_{k-1} \cup B_{k-1} \neq [n]] \left(H_{\mathrm{ME}}(n, 2^{-k}, 1/16) + \tau_k + \tau_k|(G_{k-1} \cup B_{k-1})^c|\right) \leq \sum_{s=1}^{t} |\mathcal{A}(s)|.$$

By definition, for the round $K_{\mathrm{Good}} - 1$, we see that $t_{(K_{\mathrm{Good}}-1)} < T_{\max}$. Applying the above inequality with the inequality proven in Step 4,

$$18 \sum_{k=1}^{K_{\mathrm{Good}}-1} |(G_{k-1} \cup B_{k-1})^c| \left(2\tau_k + H_{\mathrm{ME}}(n, 2^{-k}, 1/16)\right) \leq 18 \sum_{s=1}^{T_{\max}} |\mathcal{A}(s)|$$

$$\leq 18 \sum_{i \in M_\epsilon} \max \left\{ h\left(\frac{\epsilon\mu_1 - \Delta_i}{4 - 2\epsilon}, \frac{\delta}{2n}\right), \min\left[h\left(0.25\Delta_i, \frac{\delta}{2n}\right), h\left(\frac{\tilde{\alpha}_\epsilon}{4 - 2\epsilon}, \frac{\delta}{2n}\right)\right] \right\}$$

$$+ 18 \sum_{i \in M^c_{\epsilon+\tilde{\alpha}_\epsilon/\mu_1}} h\left(\frac{\epsilon\mu_1 - \Delta_i}{4 - 2\epsilon}, \frac{\delta}{2n}\right) + 18|M^c_\epsilon \cap M_{\epsilon+\tilde{\alpha}_\epsilon/\mu_1}|h\left(\frac{\tilde{\alpha}_\epsilon}{4 - 2\epsilon}, \frac{\delta}{2n}\right).$$

Otherwise, if $K_{\mathrm{Good}} = 1$, exactly $4c'n\log(16) + 32n\log(16n/\delta)$ samples are given to the good filter in round 1. One may use Lemma F.2 to invert $h(\cdot, \cdot)$ and show that the summation on the right had side of the above inequality is within a constant of this and the claim holds in this case as well for a different constant, potentially larger than 18. $\qquad\square$

### E.4.8 Step 7: Bounding Equation (18)

Next, we bound $\sum_{k=1}^{\infty} 2\mathbb{1}[G_{k-1} = M_\epsilon]\,\mathbb{1}[G_{k-1} \cup B_{k-1} \neq [n]] \left(H_{\mathrm{ME}}(n, 2^{-k}, 1/16) + \tau_k + \tau_k|(G_{k-1} \cup B_{k-1})^c|\right)$.

$$\sum_{k=1}^{\infty} 2\mathbb{1}[G_{k-1} = M_\epsilon]\,\mathbb{1}[G_{k-1} \cup B_{k-1} \neq [n]] \left(H_{\mathrm{ME}}(n, 2^{-k}, 1/16) + \tau_k + \tau_k|(G_{k-1} \cup B_{k-1})^c|\right)$$

$$= \sum_{k=1}^{\infty} 2\mathbb{1}[G_{k-1} = M_\epsilon]\,\mathbb{1}[G_{k-1} \cup B_{k-1} \neq [n]] \left(H_{\mathrm{ME}}(n, 2^{-k}, 1/16) + \tau_k + \tau_k|(M_\epsilon \cup B_{k-1})^c|\right)$$

$$= \sum_{k=1}^{\infty} 2\mathbb{1}[G_{k-1} = M_\epsilon]\,\mathbb{1}[G_{k-1} \cup B_{k-1} \neq [n]] \left(H_{\mathrm{ME}}(n, 2^{-k}, 1/16) + \tau_k + \tau_k|M^c_\epsilon \backslash B_{k-1}|\right)$$

$$= \sum_{k=K_{\mathrm{Good}}+1}^{\infty} 2\mathbb{1}[G_{k-1} \cup B_{k-1} \neq [n]] \left(H_{\mathrm{ME}}(n, 2^{-k}, 1/16) + \tau_k + \tau_k|M^c_\epsilon \backslash B_{k-1}|\right)$$

$$\overset{\mathcal{E}_1, \mathcal{E}_2}{=} \sum_{k=K_{\mathrm{Good}}+1}^{\infty} 2\mathbb{1}[B_{k-1} \neq M^c_\epsilon] \left(H_{\mathrm{ME}}(n, 2^{-k}, 1/16) + \tau_k + \tau_k|M^c_\epsilon \backslash B_{k-1}|\right)$$

$$= \sum_{k=K_{\mathrm{Good}}+1}^{\infty} 2\mathbb{1}[B_{k-1} \neq M^c_\epsilon] \left(H_{\mathrm{ME}}(n, 2^{-k}, 1/16) + \tau_k\right) + \sum_{k=K_{\mathrm{Good}}+1}^{\infty} 2\mathbb{1}[B_{k-1} \neq M^c_\epsilon] \left(\tau_k|M^c_\epsilon \backslash B_{k-1}|\right)$$

$$= \sum_{k=K_{\text{Good}}+1}^{\infty} 2\mathbb{1}\left[B_{k-1} \neq M_\epsilon^c\right]\left(H_{\text{ME}}(n, 2^{-k}, 1/16) + \tau_k\right) + \sum_{k=K_{\text{Good}}+1}^{\infty} 2\tau_k |M_\epsilon^c \backslash B_{k-1}|$$

$$= \sum_{k=K_{\text{Good}}+1}^{\infty} 2\mathbb{1}\left[B_{k-1} \neq M_\epsilon^c\right]\left(H_{\text{ME}}(n, 2^{-k}, 1/16) + \tau_k\right) + \sum_{k=K_{\text{Good}}+1}^{\infty} \sum_{i \in M_\epsilon^c} 2\tau_k \mathbb{1}[i \notin B_{k-1}]$$

$$\leq \sum_{k=K_{\text{Good}}+1}^{\infty} 2|M_\epsilon^c \backslash B_{k-1}|\left(H_{\text{ME}}(n, 2^{-k}, 1/16) + \tau_k\right) + \sum_{k=K_{\text{Good}}+1}^{\infty} \sum_{i \in M_\epsilon^c} 2\tau_k \mathbb{1}[i \notin B_{k-1}]$$

$$= \sum_{k=K_{\text{Good}}+1}^{\infty} \sum_{i \in M_\epsilon^c} 2\mathbb{1}[i \notin B_{k-1}]\left(H_{\text{ME}}(n, 2^{-k}, 1/16) + \tau_k\right) + \sum_{k=K_{\text{Good}}+1}^{\infty} \sum_{i \in M_\epsilon^c} 2\tau_k \mathbb{1}[i \notin B_{k-1}]$$

$$= \sum_{k=K_{\text{Good}}+1}^{\infty} \sum_{i \in M_\epsilon^c} 2\mathbb{1}[i \notin B_{k-1}]\left(2\tau_k + H_{\text{ME}}(n, 2^{-k}, 1/16)\right)$$

$$= \sum_{i \in M_\epsilon^c} \sum_{k=K_{\text{Good}}+1}^{\infty} 2\mathbb{1}[i \notin B_{k-1}]\left(2\tau_k + H_{\text{ME}}(n, 2^{-k}, 1/16)\right)$$

$$\leq \sum_{i \in M_\epsilon^c} \sum_{k=1}^{\infty} 2\mathbb{1}[i \notin B_{k-1}]\left(2\tau_k + H_{\text{ME}}(n, 2^{-k}, 1/16)\right) \tag{23}$$

### E.4.9 Step 8: Bounding the expected total number of samples drawn by FAREAST

Now we take expectations over the number of samples drawn. These expectations are conditional on the high probability event $\mathcal{E}_1 \cap \mathcal{E}_2$. The bound in step 5 holds deterministically conditioned on this event.

Note $\tau_k$ and $H_{\text{ME}}(n, 2^{-k}, 1/16)$ are deterministic constants for any $k$. Let all expectations are be jointly over the random instance $\nu$ and the randomness in FAREAST.

$$\mathbb{E}[T | \mathbb{1}[\mathcal{E}_1 \cap \mathcal{E}_2] = 1] =$$

$$\sum_{k=1}^{\infty} 2\mathbb{E}\left[\mathbb{1}[G_k \cup B_k \neq [n]] \big| \mathbb{1}[\mathcal{E}_1 \cap \mathcal{E}_2] = 1\right]\left(H_{\text{ME}}(n, 2^{-k}, 1/16) + \tau_k + \tau_k \big|(G_{k-1} \cup B_{k-1})^c\big|\right)$$

$$= \sum_{k=1}^{\infty} 2\mathbb{E}\left[\mathbb{1}\left[G_{k-1} \neq M_\epsilon\right]\mathbb{1}[G_k \cup B_k \neq [n]] \big| \mathbb{1}[\mathcal{E}_1 \cap \mathcal{E}_2] = 1\right]$$

$$\left(H_{\text{ME}}(n, 2^{-k}, 1/16) + \tau_k + \tau_k \big|(G_{k-1} \cup B_{k-1})^c\big|\right)$$

$$+ \sum_{k=1}^{\infty} 2\mathbb{E}\left[\mathbb{1}\left[G_{k-1} = M_\epsilon\right]\mathbb{1}[G_k \cup B_k \neq [n]] \big| \mathbb{1}[\mathcal{E}_1 \cap \mathcal{E}_2] = 1\right]$$

$$\left(H_{\text{ME}}(n, 2^{-k}, 1/16) + \tau_k + \tau_k \big|(G_{k-1} \cup B_{k-1})^c\big|\right)$$

$$\overset{\text{Step } 6}{\leq} c \sum_{i \in M_\epsilon} \max\left\{h\left(\frac{\epsilon\mu_1 - \Delta_i}{4 - 2\epsilon}, \frac{\delta}{2n}\right), \min\left[h\left(0.25\Delta_i, \frac{\delta}{2n}\right), h\left(\frac{\tilde{\alpha}_\epsilon}{4 - 2\epsilon}, \frac{\delta}{2n}\right)\right]\right\}$$

$$+ c \sum_{i \in M_{\epsilon + \tilde{\alpha}_\epsilon/\mu_1}^c} h\left(\frac{\epsilon\mu_1 - \Delta_i}{4 - 2\epsilon}, \frac{\delta}{2n}\right) + c|M_\epsilon^c \cap M_{\epsilon + \tilde{\alpha}_\epsilon/\mu_1}| h\left(\frac{\tilde{\alpha}_\epsilon}{4 - 2\epsilon}, \frac{\delta}{2n}\right)$$

$$+ \sum_{k=1}^{\infty} 2\mathbb{E}\left[\mathbb{1}\left[G_{k-1} = M_\epsilon\right]\mathbb{1}[G_k \cup B_k \neq [n]] \big| \mathbb{1}[\mathcal{E}_1 \cap \mathcal{E}_2] = 1\right]$$

$$\left(H_{\text{ME}}(n, 2^{-k}, 1/16) + \tau_k + \tau_k \big|(G_{k-1} \cup B_{k-1})^c\big|\right)$$

$$\overset{\text{Step } 7}{\leq} c \sum_{i \in M_\epsilon} \max\left\{h\left(\frac{\epsilon\mu_1 - \Delta_i}{4 - 2\epsilon}, \frac{\delta}{2n}\right), \min\left[h\left(0.25\Delta_i, \frac{\delta}{2n}\right), h\left(\frac{\tilde{\alpha}_\epsilon}{4 - 2\epsilon}, \frac{\delta}{2n}\right)\right]\right\}$$

$$+ c \sum_{i \in M^c_{\epsilon+\tilde{\alpha}_\epsilon/\mu_1}} h\left(\frac{\epsilon\mu_1 - \Delta_i}{4-2\epsilon}, \frac{\delta}{2n}\right) + c|M^c_\epsilon \cap M_{\epsilon+\tilde{\alpha}_\epsilon/\mu_1}|h\left(\frac{\tilde{\alpha}_\epsilon}{4-2\epsilon}, \frac{\delta}{2n}\right)$$

$$+ \sum_{i \in M^c_\epsilon} \sum_{k=1}^{\infty} 2\mathbb{E}_\nu\left[\mathbb{1}[i \notin B_{k-1}]\big|\mathbb{1}[\mathcal{E}_1 \cap \mathcal{E}_2] = 1\right]\left(2\tau_k + H_{\mathrm{ME}}(n, 2^{-k}, 1/16)\right)$$

$$\overset{(a)}{=} c \sum_{i \in M_\epsilon} \max\left\{h\left(\frac{\epsilon\mu_1 - \Delta_i}{4-2\epsilon}, \frac{\delta}{2n}\right), \min\left[h\left(0.25\Delta_i, \frac{\delta}{2n}\right), h\left(\frac{\tilde{\alpha}_\epsilon}{4-2\epsilon}, \frac{\delta}{2n}\right)\right]\right\}$$

$$+ c \sum_{i \in M^c_{\epsilon+\tilde{\alpha}_\epsilon/\mu_1}} h\left(\frac{\epsilon\mu_1 - \Delta_i}{4-2\epsilon}, \frac{\delta}{2n}\right) + c|M^c_\epsilon \cap M_{\epsilon+\tilde{\alpha}_\epsilon}|h\left(\frac{\tilde{\alpha}_\epsilon/\mu_1}{4-2\epsilon}, \frac{\delta}{2n}\right)$$

$$+ \sum_{i \in M^c_\epsilon} \sum_{k=1}^{\infty} 2\mathbb{E}_\nu\left[\mathbb{1}[i \notin B_{k-1}]\big|\mathbb{1}[\mathcal{E}_1]\right]\left(2\tau_k + H_{\mathrm{ME}}(n, 2^{-k}, 1/16)\right)$$

where $(a)$ follows from $\mathbb{E}_\nu\left[\mathbb{1}[i \notin B_{k-1}]\big|\mathbb{1}[\mathcal{E}_1 \cap \mathcal{E}_2]\right] = \mathbb{E}_\nu\left[\mathbb{1}[i \notin B_{k-1}]\big|\mathbb{1}[\mathcal{E}_1]\right]$ for $i \in M^c_\epsilon$, since the event $\{i \in B_{k-1}\}$ is independent of $\mathcal{E}_2$ for all $i \in M^c_\epsilon$. This can be observed since $\mathcal{E}_2$ deals only with independent samples taken of arms in $M_\epsilon$.

### E.4.10  Step 9: Bounding $\sum_{k=1}^{\infty} \mathbb{E}_\nu\left[\mathbb{1}[i \notin B_{k-1}]|\mathbb{1}[\mathcal{E}_1]\right]\left(2\tau_k + H_{\mathbf{ME}}(n, 2^{-k}, 1/16)\right)$ for $i \in M^c_\epsilon$

Next, we bound the expectation remaining from step 8. In particular, this is the number of samples drawn by the bad filter to add arm $i \in M^c_\epsilon$ to $B_k$.

First, we bound the probability that for a given $i \in M^c_\epsilon$ and a given $k$ $i \notin B_k$. Note that by Borel-Cantelli, this implies that the probability that $i$ is never added to any $B_k$ is 0.

**Claim 1:** For $i \in M^c_\epsilon$, $k \geq \left\lceil \log_2\left(\frac{2-\epsilon}{\Delta_i - \epsilon\mu_1}\right)\right\rceil \implies \mathbb{E}_\nu\left[\mathbb{1}[i \notin B_k]|\mathbb{1}[\mathcal{E}_1]\right] \leq \left(\frac{1}{8}\right)^{k-\left\lceil \log_2\left(\frac{2-\epsilon}{\Delta_i - \epsilon\mu_1}\right)\right\rceil}$

**Proof.** If $i \in B_k$, either the good or the bad filter may have added it. The behavior of the bad filter on arms inn $M^c_\epsilon$ is independent of $\mathcal{E}_1$. Hence.

$$\mathbb{E}_\nu\left[\mathbb{1}[i \notin B_k]|\mathbb{1}[\mathcal{E}_1]\right] = \mathbb{E}_\nu\left[\mathbb{1}[\hat{\mu}_i + C_{\delta/2n}(t) \geq L_{t_k}]\mathbb{1}[\hat{\mu}_{i_k} - \hat{\mu}_i \leq 2^{-(k+1)}(2-\epsilon)]|\mathbb{1}[\mathcal{E}_1]\right]$$

$$\leq \mathbb{E}_\nu\left[\mathbb{1}[\hat{\mu}_{i_k} - \hat{\mu}_i \leq 2^{-(k+1)}(2-\epsilon)]|\mathbb{1}[\mathcal{E}_1]\right]$$

$$= \mathbb{E}_\nu\left[\mathbb{1}[\hat{\mu}_{i_k} - \hat{\mu}_i \leq 2^{-(k+1)}(2-\epsilon)]\right]$$

If $i \in B_{k-1}$ then $i \in B_k$ by definition. Otherwise, if $i \notin B_{k-1}$, by Hoeffding's Inequality conditional on the value of $i_k$ and a sum over conditional probabilities as in step 0, with probability at least $1 - \frac{\delta}{4nk^2}$

$$|((1-\epsilon)\hat{\mu}_{i_k} - \hat{\mu}_i) - ((1-\epsilon)\mu_{i_k} - \mu_i)| \leq 2^{-(k+1)}$$

If `MedianElimination` also succeeds, the joint event of which occurs with probability $\frac{15}{16}\left(1 - \frac{\delta}{4nk^2}\right)$ by independence[7],

$$(1-\epsilon)\hat{\mu}_{i_k} - \hat{\mu}_i \geq (1-\epsilon)\mu_{i_k} - \mu_i - 2^{-(k+1)}$$

$$\geq (1-\epsilon)\mu_1 - \mu_i - 2^{-(k+1)}(2-\epsilon)$$

$$= \Delta_i - \epsilon\mu_1 - 2^{-(k+1)}(2-\epsilon).$$

Then for $k \geq \left\lceil \log_2\left(\frac{2-\epsilon}{\Delta_i - \epsilon\mu_1}\right)\right\rceil$,

$$(1-\epsilon)\hat{\mu}_{i_k} - \hat{\mu}_i \geq \Delta_i - \epsilon\mu_1 - 2^{-(k+1)}(2-\epsilon) \geq 2^{-(k+1)}(2-\epsilon),$$

which implies that $i \in B_k$ by line 15 of FAREAST. In particular,

$$\mathbb{E}\left[\mathbb{1}[i \in B_k]\big|i \notin B_{k-1}\mathbb{1}[\mathcal{E}_1]\right] \geq \mathbb{E}\left[\hat{\mu}_{i_k} - \hat{\mu}_i > 2^{-(k+1)}(2-\epsilon)\big|i \notin B_{k-1}, \mathbb{1}[\mathcal{E}_1]\right] \geq \frac{15}{16}\left(1 - \frac{\delta}{4nk^2}\right).$$

Furthermore, $i \notin B_0$ by definition. Then for $k \geq \left\lceil\log_2\left(\frac{2-\epsilon}{\Delta_i - \epsilon\mu_1}\right)\right\rceil$,

$$\mathbb{E}\left[\mathbb{1}[i \notin B_k]|\mathbb{1}[\mathcal{E}_1]\right] = \mathbb{E}\left[\mathbb{1}[i \notin B_k](\mathbb{1}[i \notin B_{k-1}] + \mathbb{1}[i \in B_{k-1}])|\mathbb{1}[\mathcal{E}_1]\right]$$
$$= \mathbb{E}\left[\mathbb{1}[i \notin B_k]\mathbb{1}[i \notin B_{k-1}]|\mathbb{1}[\mathcal{E}_1]\right] + \mathbb{E}\left[\mathbb{1}[i \notin B_k]\mathbb{1}[i \in B_{k-1}]|\mathbb{1}[\mathcal{E}_1]\right]$$

Deterministically, $\mathbb{1}[i \notin B_k]\mathbb{1}[i \in B_{k-1}] = 0$. Therefore,

$$\mathbb{E}\left[\mathbb{1}[i \notin B_k]\mathbb{1}[i \notin B_{k-1}]|\mathbb{1}[\mathcal{E}_1]\right] + \mathbb{E}\left[\mathbb{1}[i \notin B_k]\mathbb{1}[i \in B_{k-1}]|\mathbb{1}[\mathcal{E}_1]\right]$$
$$= \mathbb{E}\left[\mathbb{1}[i \notin B_k]\mathbb{1}[i \notin B_{k-1}]|\mathbb{1}[\mathcal{E}_1]\right]$$
$$= \mathbb{E}\left[\mathbb{1}[i \notin B_k]\mathbb{1}[i \notin B_{k-1}]\big|i \notin B_{k-1}, \mathbb{1}[\mathcal{E}_1]\right]\mathbb{P}(i \notin B_{k-1}|\mathbb{1}[\mathcal{E}_1])$$
$$\qquad + \mathbb{E}\left[\mathbb{1}[i \notin B_k]\mathbb{1}[i \notin B_{k-1}]\big|i \in B_{k-1}, \mathbb{1}[\mathcal{E}_1]\right]\mathbb{P}(i \in B_{k-1}|\mathbb{1}[\mathcal{E}_1])$$
$$= \mathbb{E}\left[\mathbb{1}[i \notin B_k]\mathbb{1}[i \notin B_{k-1}]\big|i \notin B_{k-1}, \mathbb{1}[\mathcal{E}_1]\right]\mathbb{P}(i \notin B_{k-1}|\mathbb{1}[\mathcal{E}_1])$$
$$= \mathbb{E}\left[\mathbb{1}[i \notin B_k]\big|i \notin B_{k-1}, \mathbb{1}[\mathcal{E}_1]\right]\mathbb{E}\left[\mathbb{1}[i \notin B_{k-1}]|\mathbb{1}[\mathcal{E}_1]\right]$$
$$\leq \left(\frac{1}{16} + \frac{\delta}{4nk^2}\right)\mathbb{E}\left[\mathbb{1}[i \notin B_{k-1}]|\mathbb{1}[\mathcal{E}_1]\right].$$

For $k < \left\lceil\log_2\left(\frac{2-\epsilon}{\Delta_i - \epsilon\mu_1}\right)\right\rceil$, trivially, $\mathbb{E}\left[\mathbb{1}[i \notin B_k]|\mathbb{1}[\mathcal{E}_1]\right] \leq 1$. Recall $\delta \leq 1/8$. For $k \geq \left\lceil\log_2\left(\frac{2-\epsilon}{\Delta_i - \epsilon\mu_1}\right)\right\rceil$,

$$\mathbb{E}\left[\mathbb{1}[i \notin B_k]|\mathbb{1}[\mathcal{E}_1]\right] \leq \prod_{s=\left\lceil\log_2\left(\frac{2-\epsilon}{\Delta_i - \epsilon\mu_1}\right)\right\rceil}^{k}\left(\frac{1}{16} + \frac{\delta}{2ns^2}\right) \leq \left(\frac{1}{8}\right)^{k-\left\lceil\log_2\left(\frac{2-\epsilon}{\Delta_i - \epsilon\mu_1}\right)\right\rceil}.$$

$\square$

**Claim 2:** For $j \in M_\epsilon^c$, $\sum_{k=1}^{\infty} 2\mathbb{E}_\nu\left[\mathbb{1}[i \notin B_{k-1}]|\mathbb{1}[\mathcal{E}_1]\right]\left(2\tau_k + H_{\text{ME}}(n, 2^{-k}, 1/16)\right) \leq c''\frac{4n(2-\epsilon)^2}{(\Delta_i - \epsilon\mu_1)^2} + c''h\left(\frac{\Delta_i - \epsilon\mu_1}{4-2\epsilon}, \frac{\delta}{2n}\right)$

**Proof.** This sum decomposes into two terms.

$$\sum_{k=1}^{\infty}\mathbb{E}_\nu\left[\mathbb{1}[i \notin B_{k-1}]|\mathbb{1}[\mathcal{E}_1]\right]\left(2\tau_k + H_{\text{ME}}(n, 2^{-k}, 1/16)\right)$$

$$= \sum_{k=1}^{\left\lfloor\log_2\left(\frac{2-\epsilon}{\Delta_i - \epsilon\mu_1}\right)\right\rfloor}\mathbb{E}_\nu\left[\mathbb{1}[i \notin B_{k-1}]|\mathbb{1}[\mathcal{E}_1]\right]\left(H_{\text{ME}}(n, 2^{-k}, 1/16) + 2\left\lceil 2^{2k+3}\log\left(\frac{16nk^2}{\delta}\right)\right\rceil\right)$$

$$+ \sum_{k=\left\lceil\log_2\left(\frac{2-\epsilon}{\Delta_i - \epsilon\mu_1}\right)\right\rceil}^{\infty}\mathbb{E}_\nu\left[\mathbb{1}[i \notin B_{k-1}]|\mathbb{1}[\mathcal{E}_1]\right]\left(H_{\text{ME}}(n, 2^{-k}, 1/16) + 2\left\lceil 2^{2k+3}\log\left(\frac{16nk^2}{\delta}\right)\right\rceil\right)$$

We begin by bounding the first term.

$$\sum_{k=1}^{\left\lfloor\log_2\left(\frac{2-\epsilon}{\Delta_i - \epsilon\mu_1}\right)\right\rfloor}\mathbb{E}_\nu\left[\mathbb{1}[i \notin B_{k-1}]|\mathbb{1}[\mathcal{E}_1]\right]\left(H_{\text{ME}}(n, 2^{-k}, 1/16) + 2\left\lceil 2^{2k+3}\log\left(\frac{16nk^2}{\delta}\right)\right\rceil\right)$$

$$\leq \sum_{k=1}^{\left\lfloor\log_2\left(\frac{2-\epsilon}{\Delta_i - \epsilon\mu_1}\right)\right\rfloor}\left(H_{\text{ME}}(n, 2^{-k}, 1/16) + 2\left\lceil 2^{2k+3}\log\left(\frac{16nk^2}{\delta}\right)\right\rceil\right)$$

$$\leq \sum_{k=1}^{\left\lfloor\log_2\left(\frac{2-\epsilon}{\Delta_i - \epsilon\mu_1}\right)\right\rfloor}\left(c'n2^{2k}\log(16) + 2 + 2^{2k+4}\log\left(\frac{16nk^2}{\delta}\right)\right)$$

$$\leq 2\log_2\left(\frac{2-\epsilon}{\Delta_i-\epsilon\mu_1}\right) + \left(c'n\log(16) + 16\log\left(\frac{16n}{\delta}\right)\right)\sum_{k=1}^{\left\lfloor \log_2\left(\frac{2-\epsilon}{\Delta_i-\epsilon\mu_1}\right)\right\rfloor} 2^{2k}$$

$$+ 32\sum_{k=1}^{\left\lfloor \log_2\left(\frac{2-\epsilon}{\Delta_i-\epsilon\mu_1}\right)\right\rfloor} 2^{2k}\log(k)$$

$$\leq 2\log_2\left(\frac{2-\epsilon}{\Delta_i-\epsilon\mu_1}\right) + \left(c'n\log(16) + 16\log\left(\frac{16n}{\delta}\right) + 32\log\log_2\left(\frac{2-\epsilon}{\Delta_i-\epsilon\mu_1}\right)\right)\sum_{k=1}^{\left\lfloor \log_2\left(\frac{2-\epsilon}{\Delta_i-\epsilon\mu_1}\right)\right\rfloor} 2^{2k}$$

$$\leq 2\log_2\left(\frac{2-\epsilon}{\Delta_i-\epsilon\mu_1}\right) + \frac{(2-\epsilon)^2}{(\Delta_i-\epsilon\mu_1)^2}\left(c'n\log(16) + 32\log\left(\frac{16n}{\delta}\log_2\left(\frac{2-\epsilon}{\Delta_i-\epsilon\mu_1}\right)\right)\right)$$

Next, we plug in the bound from claim 1 controlling the probability that $i\notin B_k$.

Using Claim 1, we bound the second sum as follows:

$$\sum_{r=\left\lceil \log_2\left(\frac{2-\epsilon}{\Delta_i-\epsilon\mu_1}\right)\right\rceil}^{\infty} \mathbb{E}_\nu\left[\mathbb{1}[i\notin B_{k-1}]\,|\,\mathbb{1}[\mathcal{E}_1]\right]\left(H_{\mathrm{ME}}(n, 2^{-k}, 1/16) + 2\left\lceil 2^{2k+3}\log\left(\frac{16nk^2}{\delta}\right)\right\rceil\right)$$

$$\leq \sum_{k=\left\lceil \log_2\left(\frac{2-\epsilon}{\Delta_i-\epsilon\mu_1}\right)\right\rceil}^{\infty} \left(\frac{1}{8}\right)^{k-\left\lceil \log_2\left(\frac{2-\epsilon}{\Delta_i-\epsilon\mu_1}\right)\right\rceil-1}\left(c'n2^{2k}\log(16) + 2 + 2^{2k+4}\log\left(\frac{16nk^2}{\delta}\right)\right)$$

$$= c'n\log(16)\sum_{k=1}^{\infty}\left(\frac{1}{8}\right)^{k-1}2^{2\left(k+\left\lceil \log_2\left(\frac{2-\epsilon}{\Delta_i-\epsilon\mu_1}\right)\right\rceil\right)} + 2\sum_{k=1}^{\infty}\left(\frac{1}{8}\right)^{k-1}$$

$$+ 16\sum_{k=1}^{\infty}\left(\frac{1}{8}\right)^{k-1}2^{2\left(k+\left\lceil \log_2\left(\frac{2-\epsilon}{\Delta_i-\epsilon\mu_1}\right)\right\rceil\right)}\log\left(\frac{16n\left(k+\left\lceil \log_2\left(\frac{2-\epsilon}{\Delta_i-\epsilon\mu_1}\right)\right\rceil\right)^2}{\delta}\right)$$

$$+ 16\sum_{k=1}^{\infty}2^{-3k+3}2^{2\left(k+\log_2\left(\frac{2-\epsilon}{\Delta_i-\epsilon\mu_1}\right)+1\right)}\log\left(\frac{16n\left(k+\left\lceil \log_2\left(\frac{2-\epsilon}{\Delta_i-\epsilon\mu_1}\right)\right\rceil\right)^2}{\delta}\right)$$

$$= 3 + \left(c'n\log(16)\frac{2^5(2-\epsilon)^2}{(\Delta_i-\epsilon\mu_1)^2} + \frac{2^9(2-\epsilon)^2}{(\Delta_i-\epsilon\mu_1)^2}\log\left(\frac{16n}{\delta}\right)\right)\sum_{k=1}^{\infty}2^{-k}$$

$$+ \frac{2^9(2-\epsilon)^2}{(\Delta_i-\epsilon\mu_1)^2}\sum_{k=1}^{\infty}2^{-k}\log\left(\left(k+\left\lceil \log_2\left(\frac{2-\epsilon}{\Delta_i-\epsilon\mu_1}\right)\right\rceil\right)^2\right)$$

$$\leq 3 + c'n\log(16)\frac{2^5(2-\epsilon)^2}{(\Delta_i-\epsilon\mu_1)^2} + \frac{2^9(2-\epsilon)^2}{(\Delta_i-\epsilon\mu_1)^2}\log\left(\frac{16n}{\delta}\right)$$

$$+ \frac{2^{10}(2-\epsilon)^2}{(\Delta_i-\epsilon\mu_1)^2}\sum_{k=1}^{\infty}2^{-k}\log\left(k+\left\lceil \log_2\left(\frac{2-\epsilon}{\Delta_i-\epsilon\mu_1}\right)\right\rceil\right)$$

$$= (**)$$

We may bound the final summand, $\sum_{k=1}^{\infty}2^{-k}\log\left(k+\left\lceil \log_2\left(\frac{2-\epsilon}{\Delta_i-\epsilon\mu_1}\right)\right\rceil\right)$ as follows:

$$\sum_{k=1}^{\infty}2^{-k}\log\left(k+\left\lceil \log_2\left(\frac{2-\epsilon}{\Delta_i-\epsilon\mu_1}\right)\right\rceil\right) \leq \log\left(\frac{e}{2}\log_2\left(\frac{16(2-\epsilon)^2}{(\Delta_i-\epsilon\mu_1)^2}\right)\right)$$

Plugging this back into $(**)$, we have that

$$(**) \leq 3 + c'n\log(16)\frac{2^5(2-\epsilon)^2}{(\Delta_i-\epsilon\mu_1)^2} + \frac{2^9(2-\epsilon)^2}{(\Delta_i-\epsilon\mu_1)^2}\log\left(\frac{16n}{\delta}\right)$$

$$+ \frac{2^{10}(2-\epsilon)^2}{(\Delta_i - \epsilon\mu_1)^2} \log\left(\frac{e}{2}\log_2\left(\frac{16(2-\epsilon)^2}{(\Delta_i - \epsilon\mu_1)^2}\right)\right)$$

Combining the above with the bound on the first sum, we have that

$$\sum_{k=1}^{\infty}\mathbb{E}_\nu\left[\mathbb{1}[i \notin B_{k-1}]\mathbb{1}[\mathcal{E}_1]\right]\left(2\tau_k + H_{\text{ME}}(n, 2^{-k}, 1/16)\right)$$

$$\leq c''\left(\frac{4n(2-\epsilon)^2}{(\Delta_i - \epsilon\mu_1)^2} + \frac{4c(2-\epsilon)^2}{(\Delta_i - \epsilon\mu_1)^2}\log\left(\frac{2n}{\delta}\log_2\left(\frac{4-2\epsilon}{(\Delta_i - \epsilon\mu_1)^2}\right)\right)\right)$$

$$= \frac{4c''n(2-\epsilon)^2}{(\Delta_i - \epsilon\mu_1)^2} + c''h\left(\frac{\Delta_i - \epsilon\mu_1}{4-2\epsilon}, \frac{\delta}{2n}\right)$$

for a sufficiently large, universal constant $c''$ and $c$ from the definition of $h(\cdot, \cdot)$. $\qquad\square$

### E.4.11 Step 10: Applying the result of Step 9 to the result of Step 8

We may repeat the result of step 9 for every $i \in M_\epsilon^c$ and plug this into the result of Step 8. From this point, we simplify to return the final result.

By Step 8, the total number of samples $T$ drawn by FAREAST is bounded in expectation by

$$\mathbb{E}[T|\mathcal{E}_1 \cap \mathcal{E}_2] \leq c\sum_{i \in M_\epsilon}\max\left\{h\left(\frac{\epsilon\mu_1 - \Delta_i}{4-2\epsilon}, \frac{\delta}{2n}\right), \min\left[h\left(0.25\Delta_i, \frac{\delta}{2n}\right), h\left(\frac{\tilde{\alpha}_\epsilon/\mu_1}{4-2\epsilon}, \frac{\delta}{2n}\right)\right]\right\}$$

$$+ c\sum_{i \in M_{\epsilon+\tilde{\alpha}_\epsilon/\mu_1}^c}h\left(\frac{\epsilon\mu_1 - \Delta_i}{4-2\epsilon}, \frac{\delta}{2n}\right) + c|M_\epsilon^c \cap M_{\epsilon+\tilde{\alpha}_\epsilon}|h\left(\frac{\tilde{\alpha}_\epsilon}{4-2\epsilon}, \frac{\delta}{2n}\right)$$

$$+ 2\sum_{i \in M_\epsilon^c}\sum_{k=1}^{\infty}\mathbb{E}_\nu\left[\mathbb{1}[i \notin B_{k-1}]\mathbb{1}[\mathcal{E}_1]\right]\left(2\tau_k + H_{\text{ME}}(n, 2^{-k}, 1/16)\right).$$

Applying the bound from Step 9 to each $i \in M_\epsilon^c$, we have that

$$\mathbb{E}[T|\mathcal{E}_1 \cap \mathcal{E}_2] \leq c\sum_{i \in M_\epsilon}\max\left\{h\left(0.25(\epsilon - \Delta_i), \frac{\delta}{2n}\right), \min\left[h\left(0.25\Delta_i, \frac{\delta}{2n}\right), h\left(\frac{\tilde{\alpha}_\epsilon}{4-2\epsilon}, \frac{\delta}{2n}\right)\right]\right\}$$

$$+ c\sum_{i \in M_{\epsilon+\tilde{\alpha}_\epsilon/\mu_1}^c}h\left(0.25(\epsilon - \Delta_i), \frac{\delta}{2n}\right) + c|M_\epsilon^c \cap M_{\epsilon+\tilde{\alpha}_\epsilon/\mu_1}|h\left(\frac{\tilde{\alpha}_\epsilon}{4-2\epsilon}, \frac{\delta}{2n}\right)$$

$$+ 2c''\sum_{i \in M_\epsilon^c}\frac{4n(2-\epsilon)^2}{(\Delta_i - \epsilon\mu_1)^2} + h\left(\frac{\Delta_i - \epsilon\mu_1}{4-2\epsilon}, \frac{\delta}{2n}\right).$$

For $i \in M_\epsilon^c \cap M_{\epsilon+\tilde{\alpha}_\epsilon/\mu_1}$, $\tilde{\alpha}_\epsilon = \min_{j \in M_\epsilon}\epsilon\mu_1 - \Delta_j \geq \Delta_i - \epsilon\mu_1$. By monotonicity of $h(\cdot, \cdot)$, $h\left(\frac{\tilde{\alpha}_\epsilon}{4-2\epsilon}, \frac{\delta}{2n}\right) \leq \frac{c''n(4-2\epsilon)}{(\Delta_i - \epsilon\mu_1)^2} + c''h\left(\frac{\Delta_i - \epsilon\mu_1}{4-2\epsilon}, \frac{\delta}{2n}\right)$. Therefore,

$$\mathbb{E}[T|\mathcal{E}_1 \cap \mathcal{E}_2] \leq c\sum_{i \in M_\epsilon}\max\left\{h\left(\frac{\Delta_i - \epsilon\mu_1}{4-2\epsilon}, \frac{\delta}{2n}\right), \min\left[h\left(0.25\Delta_i, \frac{\delta}{2n}\right), h\left(\frac{\tilde{\alpha}_\epsilon}{4-2\epsilon}, \frac{\delta}{2n}\right)\right]\right\}$$

$$+ (2c'' + c)\sum_{i \in M_\epsilon^c}\frac{n(4-2\epsilon)}{(\Delta_i - \epsilon\mu_1)^2} + h\left(\frac{\Delta_i - \epsilon\mu_1}{4-2\epsilon}, \frac{\delta}{2n}\right).$$

Lastly, note that $\frac{1}{3(1-x)} \leq \frac{1}{2-x}$ for $x \leq 1/2$. By monotonicity of $h$, we may lower bound the denominators $\frac{1}{4-2\epsilon}$ and $\frac{1}{2(2-\epsilon+\gamma)}$ as $\frac{1}{6(1-\epsilon)}$ and $\frac{1}{6(1-\epsilon+\gamma)}$ respectively. Since $\epsilon \in (0, 1/2]$, $\frac{1}{4-2\epsilon} \leq 1/4$. Plugging this in, we see that

$$\mathbb{E}[T|\mathcal{E}_1 \cap \mathcal{E}_2] \leq c\sum_{i \in M_\epsilon}\max\left\{h\left(\frac{\Delta_i - \epsilon\mu_1}{4}, \frac{\delta}{2n}\right), \min\left[h\left(0.25\Delta_i, \frac{\delta}{2n}\right), h\left(\frac{\tilde{\alpha}_\epsilon}{6(1-\epsilon)}, \frac{\delta}{2n}\right)\right]\right\}$$

$$+ (2c'' + c)\sum_{i \in M_\epsilon^c}\frac{4n}{(\Delta_i - \epsilon\mu_1)^2} + h\left(\frac{\Delta_i - \epsilon\mu_1}{4}, \frac{\delta}{2n}\right).$$

Next, we use Lemma F.3 to bound the minimum of $h(\cdot, \cdot)$ functions.

$$c \sum_{i \in M_\epsilon} \max \left\{ h \left( \frac{\Delta_i - \epsilon \mu_1}{4}, \frac{\delta}{2n} \right), \min \left[ h \left( 0.25 \Delta_i, \frac{\delta}{2n} \right), h \left( \frac{\tilde{\alpha}_\epsilon}{6(1 - \epsilon)}, \frac{\delta}{2n} \right) \right] \right\}$$

$$+ (2c'' + c) \sum_{i \in M_\epsilon^c} \frac{4n}{(\Delta_i - \epsilon \mu_1)^2} + h \left( \frac{\Delta_i - \epsilon \mu_1}{4}, \frac{\delta}{2n} \right)$$

$$= c \sum_{i \in M_\epsilon} \max \left\{ h \left( \frac{\Delta_i - \epsilon \mu_1}{4}, \frac{\delta}{2n} \right), h \left( \frac{\Delta_i + \frac{\tilde{\alpha}_\epsilon}{1-\epsilon}}{12}, \frac{\delta}{2n} \right) \right\}$$

$$+ (2c'' + c) \sum_{i \in M_\epsilon^c} \frac{4n}{(\Delta_i - \epsilon \mu_1)^2} + h \left( \frac{\Delta_i - \epsilon \mu_1}{4}, \frac{\delta}{2n} \right)$$

Finally, we use Lemma F.2 to bound the function $h(\cdot, \cdot)$. Since $\delta \leq 1/2$, $\delta/n \leq 2e^{-e/2}$. Further, $|\epsilon \mu_1 - \Delta_i| \leq 6$ for all $i$ and $\epsilon \leq 1/2$ implies that $\frac{1}{6(1-\epsilon)} |\epsilon \mu_1 - \Delta_i| \leq 2$ and $\frac{1}{6(1-\epsilon)} \min(\tilde{\alpha}_\epsilon, \tilde{\beta}_\epsilon) \leq 2$. $\Delta_i \leq 8$ for all $i$, gives $0.25 \Delta_i \leq 2$. Lastly, $\gamma \leq 6/\mu_1$ implies that $\frac{\gamma \mu_1}{6(1-\epsilon+\gamma)} \leq 2$. Therefore,

$$\mathbb{E}[T | \mathcal{E}_1 \cap \mathcal{E}_2] \leq c \sum_{i \in M_\epsilon} \max \left\{ h \left( \frac{\Delta_i - \epsilon \mu_1}{4}, \frac{\delta}{2n} \right), h \left( \frac{\Delta_i + \frac{\tilde{\alpha}_\epsilon}{1-\epsilon}}{12}, \frac{\delta}{2n} \right) \right\}$$

$$+ (2c'' + c) \sum_{i \in M_\epsilon^c} \frac{4n}{(\Delta_i - \epsilon \mu_1)^2} + h \left( \frac{\Delta_i - \epsilon \mu_1}{4}, \frac{\delta}{2n} \right)$$

$$\leq c \sum_{i \in M_\epsilon} \max \left\{ \frac{64}{(\epsilon \mu_1 - \Delta_i)^2} \log \left( \frac{4n}{\delta} \log_2 \left( \frac{384n}{\delta (\epsilon \mu_1 - \Delta_i)^2} \right) \right), \right.$$

$$\left. \frac{576}{\left( \Delta_i + \frac{\tilde{\alpha}_\epsilon}{1-\epsilon} \right)^2} \log \left( \frac{4n}{\delta} \log_2 \left( \frac{1728n}{\delta \left( \Delta_i + \frac{\tilde{\alpha}_\epsilon}{1-\epsilon} \right)^2} \right) \right) \right\}$$

$$+ (2c'' + c) \sum_{i \in M_\epsilon^c} \frac{4n}{(\Delta_i - \epsilon \mu_1)^2} + \frac{64}{(\epsilon \mu_1 - \Delta_i)^2} \log \left( \frac{4n}{\delta} \log_2 \left( \frac{384n}{\delta (\epsilon \mu_1 - \Delta_i)^2} \right) \right)$$

$$\leq c_6 \sum_{i=1}^{n} \max \left\{ \frac{1}{(\epsilon \mu_1 - \Delta_i)^2} \log \left( \frac{n}{\delta} \log_2 \left( \frac{n}{\delta (\epsilon \mu_1 - \Delta_i)^2} \right) \right), \right.$$

$$\left. \frac{1}{\left( \Delta_i + \frac{\tilde{\alpha}_\epsilon}{1-\epsilon} \right)^2} \log \left( \frac{n}{\delta} \log_2 \left( \frac{n}{\delta \left( \Delta_i + \frac{\tilde{\alpha}_\epsilon}{1-\epsilon} \right)^2} \right) \right) \right\}$$

$$+ c_6 \sum_{i \in M_\epsilon^c} \frac{n}{(\Delta_i - \epsilon \mu_1)^2}$$

$$= c_6 \sum_{i=1}^{n} \max \left\{ \frac{1}{((1 - \epsilon)\mu_1 - \mu_i)^2} \log \left( \frac{n}{\delta} \log_2 \left( \frac{n}{\delta ((1 - \epsilon)\mu_1 - \mu_i)^2} \right) \right), \right.$$

$$\left. \frac{1}{\left( \mu_1 + \frac{\tilde{\alpha}_\epsilon}{1-\epsilon} - \mu_i \right)^2} \log \left( \frac{n}{\delta} \log_2 \left( \frac{n}{\delta \left( \mu_1 + \frac{\tilde{\alpha}_\epsilon}{1-\epsilon} - \mu_i \right)^2} \right) \right) \right\}$$

$$+ c_6 \sum_{i \in M_\epsilon^c} \frac{n}{((1 - \epsilon)\mu_1 - \mu_i)^2}$$

for a sufficiently large constant $c_6$.

### E.4.12 Step 11: High probability sample complexity bound

Finally, the Good Filter is equivalent to `EAST`, Algorithm 5, except split across rounds. `EAST` is an elimination algorithm. Note that the Good Filter is union bounded over $2n$ events whereas the bounds in `EAST` are union bounded over $n$ events. The Good Filter and Bad Filter are given the same number of samples in each round, and the Good Filter can terminate within a round, conditioned on $\mathcal{E}_1 \cap \mathcal{E}_2$. Therefore, we can bound the complexity of `FAREAST` in terms of that of `EAST` run at failure probability $\delta/2$. If `FAREAST` terminates in the second round or later, the arguments in Steps 4 and 5 can be used to show that `FAREAST` draws no more than a factor of 18 more samples than `EAST`, though this estimate is highly pessimistic. If `FAREAST` terminates in round 1 (when gaps are large), we may still show that this is within a constant factor of the complexity of `EAST`, but the story is more complicated. In the first round, the bad filter draws at most $c'n \log(16) + 16(n+1)\log(8n/\delta)$ samples where $c'$ is the constant from `Median Elimination`. Since we have assumed that $\max(\Delta_i, |\epsilon\mu_1 - \Delta_i|) \leq 6(1-\epsilon) \leq 6$, this sum is likewise within a constant factor of the complexity of `EAST`. Hence with probability at least $1 - \delta$, by Theorem E.4,

$$
T \leq c_5 \sum_{i=1}^{n} \min \left\{ \max \left\{ \frac{1}{((1-\epsilon)\mu_1 - \mu_i)^2} \log \left( \frac{n}{\delta} \log_2 \left( \frac{n}{\delta((1-\epsilon)\mu_1 - \mu_i)^2} \right) \right), \right. \right.
$$

$$
\frac{1}{(\mu_1 + \frac{\tilde{\alpha}_\epsilon}{1-\epsilon} - \mu_i)^2} \log \left( \frac{n}{\delta} \log_2 \left( \frac{n}{\delta(\mu_1 + \frac{\tilde{\alpha}_\epsilon}{1-\epsilon})^2} \right) \right),
$$

$$
\left. \frac{1}{(\mu_1 + \frac{\tilde{\beta}_\epsilon}{1-\epsilon} - \mu_i)^2} \log \left( \frac{n}{\delta} \log_2 \left( \frac{n}{\delta(\mu_1 + \frac{\tilde{\beta}_\epsilon}{1-\epsilon} - \mu_i)^2} \right) \right) \right\},
$$

$$
\left. \frac{(1-\epsilon+\gamma)^2}{\gamma^2\mu_1^2} \log \left( \frac{n}{\delta} \log_2 \left( \frac{(1-\epsilon+\gamma)^2 n}{\delta\gamma^2\mu_1^2} \right) \right) \right\}
$$

samples for a sufficiently large constant $c_5$.

□

### E.5 An elimination algorithm for all $\epsilon$

First, we state an elimination algorithm `EAST` (**E**limination **A**lgorithm for a **S**ampled **T**hreshold) and bound its sample complexity. `EAST` is equivalent to the good filter in `FAREAST`. At all times, `EAST` maintains an active set $\mathcal{A}$ and samples all arms $i \in \mathcal{A}$, progressively eliminating arms from $\mathcal{A}$ until termination occurs. Additionally, `EAST` maintains upper and lower bounds, denoted $U_t$ and $L_t$, on the the threshold, $\mu_1 - \epsilon$ in the additive case and $(1-\epsilon)\mu_1$ in the multiplicative case. If $\hat{\mu}_i(t) + C_{\delta/n}(t) < L_t$, `EAST` may infer that $i \notin G_\epsilon$ (resp. $i \notin M_\epsilon$) and accordingly removes $i$ from $\mathcal{A}$. If $\hat{\mu}_i(t) - C_{\delta/n}(t) > U_t$, `EAST` may infer that $i \in G_\epsilon$ (resp. $i \in M_\epsilon$) and adds $i$ to a set $G$ of good arms it has found so far. However, a good arm $i \in G$ is only removed from $\mathcal{A}$, if `EAST` can also certify that it is not the best arm, namely if $\hat{\mu}_i(t) + C_{\delta/n}(t) < \max_j \hat{\mu}_j(t) - C_{\delta/n}(t)$. This ensures that $\mu_1 - \epsilon \in [L_t, U_t]$ at all times in the additive case, and similarly, $(1-\epsilon)\mu_1 \in [L_t, U_t]$ in the multiplicative case. If $\mathcal{A} \subset G$, `EAST` may declare that $G = G_\epsilon$ (resp. $G = M_\epsilon$) and terminates. Otherwise, the algorithm terminates when $U_t - L_t < \gamma/2$ and returns $\mathcal{A} \cup G$ in the additive case or when $U_t - L_t < \frac{\gamma}{2-\epsilon}L_t$ in the multiplicative case. This limits the number of samples of any arm and ensures that no arm worse than $(\epsilon + \gamma)$-good is returned. We give pseudocode for `EAST` in Algorithm 5. Pieces specific to the additive case are shown in red, and pieces specific to the multiplicative case are shown in blue.

Recall that $\alpha_\epsilon = \min_{i \in G_\epsilon} \epsilon - \Delta_i$ and $\beta_\epsilon = \min_{i \in G_\epsilon^c} \Delta_i - \epsilon$.

**Theorem E.3.** *Fix $\epsilon > 0$, $0 < \delta \leq 1/2$, $\gamma \in [0, 8]$ and an instance $\nu$ such that $\max(\Delta_i, |\epsilon - \Delta_i|) \leq 8$ for all $i$. In the case that $G_\epsilon = [n]$, let $\alpha_\epsilon = \min(\alpha_\epsilon, \beta_\epsilon)$. With probability at least $1 - \delta$, `EAST` returns a set $G$ such that $G_\epsilon \subset G \subset G_{(\epsilon+\gamma)}$ in at most*

$$
\sum_{i=1}^{n} \min \left\{ \max \left\{ \frac{64}{(\mu_1 - \epsilon - \mu_i)^2} \log \left( \frac{2n}{\delta} \log_2 \left( \frac{768n}{\delta(\mu_1 - \epsilon - \mu_i)^2} \right) \right), \right. \right.
$$

$$
\left. \frac{256}{(\mu_1 + \alpha_\epsilon - \mu_i)^2} \log \left( \frac{2n}{\delta} \log_2 \left( \frac{768n}{\delta(\mu_1 + \alpha_\epsilon - \mu_i)^2} \right) \right) \right),
$$

**Algorithm 5** EAST : **E**limination **A**lgorithm for a **S**ampled **T**hreshold

---

**Require:** $\epsilon, \delta > 0$, slack $\gamma \geq 0$, (if multiplicative, $0 < \epsilon \leq 1/2$)
1: Let $\mathcal{A} \leftarrow [n]$ be the active set, and $G \leftarrow \emptyset$ be the set of $\epsilon$-good arms found so far, Let $t \leftarrow 0$
2: **while** $\mathcal{A} \not\subset G$ and $U_t - L_t \geq \gamma/2$ or $U_t - L_t \geq \frac{\gamma}{2-\epsilon} L_t$ **do**
3:      Pull each arm $i \in \mathcal{A}$ and update its empirical mean $\hat{\mu}_i(t)$ , Update $t \leftarrow t + 1$
4:      Update $U_t \leftarrow \max_j \hat{\mu}_j(t) + C_{\delta/n}(t) - \epsilon$ or $U_t \leftarrow (1 - \epsilon)\left(\max_j \hat{\mu}_j(t) + C_{\delta/n}(t)\right)$
5:      Update $L_t \leftarrow \max_j \hat{\mu}_j(t) - C_{\delta/n}(t) - \epsilon$ or $L_t \leftarrow (1 - \epsilon)\left(\max_j \hat{\mu}_j(t) - C_{\delta/n}(t)\right)$
6:      **for** $i \in \mathcal{A}$ **do**
7:          **if** $\hat{\mu}_i(t) - C_{\delta/n}(t) > U_t$ **then**
8:             add $i$ to $G$                          ▷ Arm $i$ is good
9:          **if** $\hat{\mu}_i(t) + C_{\delta/n}(t) < L_t$ **then**
10:            Remove $i$ from $\mathcal{A}$              ▷ Arms in $G_\epsilon^c$ or $M_\epsilon^c$ are removed
11:          **if** $i \in G$ and $\hat{\mu}_i(t) + C_{\delta/n}(t) < \max_j \hat{\mu}_j(t) - C_{\delta/n}(t)$ **then**
12:            Remove $i$ from $\mathcal{A}$              ▷ Arms in $G_\epsilon$ or $M_\epsilon$ are removed
    **return** $G \cup \mathcal{A}$

---

$$\frac{256}{(\mu_1 + \beta_\epsilon - \mu_i)^2} \log\left(\frac{2n}{\delta} \log_2\left(\frac{768n}{\delta(\mu_1 + \beta_\epsilon - \mu_i)^2}\right)\right)\Bigg\},$$

$$\frac{64}{\gamma^2} \log\left(\frac{2n}{\delta} \log_2\left(\frac{192n}{\delta\gamma^2}\right)\right)\Bigg\}$$

*samples.*

Additionally, in the multiplicative case, recall that $\tilde{\alpha}_\epsilon = \min_{i \in G_\epsilon} \epsilon - \Delta_i$ and $\tilde{\beta}_\epsilon = \min_{i \in G_\epsilon^c} \Delta_i - \epsilon$. Next, we a theorem bounding the complexity of EAST in the multiplicative regime.

**Theorem E.4.** *Fix $\epsilon, \delta \in (0, 1/2]$, $\gamma \in [0, \min(1, 6/\mu_1))$ and an instance $\nu$ such that $\max(\Delta_i, |\epsilon\mu_1 - \Delta_i|) \leq 6$ for all $i$. Assume that $\mu_1 \geq 0$. In the case that $M_\epsilon = [n]$, let $\tilde{\alpha}_\epsilon = \min(\tilde{\alpha}_\epsilon, \tilde{\beta}_\epsilon)$. With probability at least $1 - \delta$, EAST returns a set $G$ such that $M_\epsilon \subset G \subset M_{(\epsilon + \gamma)}$ in at most*

$$\sum_{i=1}^{n} \min\Bigg\{ \max\Bigg\{ \frac{64}{((1 - \epsilon)\mu_1 - \mu_i)^2} \log\left(\frac{2n}{\delta} \log_2\left(\frac{192n}{\delta((1 - \epsilon)\mu_1 - \mu_i)^2}\right)\right),$$

$$\frac{576}{(\mu_1 + \frac{\tilde{\alpha}_\epsilon}{1-\epsilon} - \mu_i)^2} \log\left(\frac{2n}{\delta} \log_2\left(\frac{1728n}{\delta(\mu_1 + \frac{\tilde{\alpha}_\epsilon}{1-\epsilon})^2}\right)\right),$$

$$\frac{576}{(\mu_1 + \frac{\tilde{\beta}_\epsilon}{1-\epsilon} - \mu_i)^2} \log\left(\frac{2n}{\delta} \log_2\left(\frac{1728n}{\delta(\mu_1 + \frac{\tilde{\beta}_\epsilon}{1-\epsilon} - \mu_i)^2}\right)\right)\Bigg\},$$

$$\frac{144(1 - \epsilon + \gamma)}{\gamma^2 \mu_1^2} \log\left(\frac{2n}{\delta} \log_2\left(\frac{432(1 - \epsilon + \gamma)n}{\delta\gamma^2\mu_1^2}\right)\right)\Bigg\}$$

*samples.*

### E.6   Proof of Theorem E.3 EAST in the additive regime

*Proof.* **Notation for the proof:** Throughout, recall $\Delta_i = \mu_1 - \mu_i$. Recall that $t$ counts the number of times each arm in $\mathcal{A}$ has been sampled and thus the number of times that the conditionals in Lines 10 and 11 have been evaluated. Let $\mathcal{A}(t)$ denote the state $\mathcal{A}$ at this time before the arms have been eliminated from $\mathcal{A}$ in lines 10 and 11. Let $G(t)$ be defined similarly. Therefore, the total number of samples drawn by EAST up to time $t$ is $\sum_{s=1}^{t} |\mathcal{A}(s)|$.

For $i \in G_\epsilon$, let $T_i$ denote the random variable of the number of times arm $i$ is sampled before it is added to $G$ in Line 8. For $i \in G_\epsilon^c$, let $T_i$ denote the random variable of the number of times arm $i$ is sampled before it is removed from $\mathcal{A}$ in Line 10. For any arm $i$, let $T_i'$ denote the random variable of the of the number of times $i$ is sampled before $\hat{\mu}_i(t) + C_{\delta/n}(t) \leq \max_{j \in \mathcal{A}} \hat{\mu}_j(t) - C_{\delta/n}(t)$.

Define the event

$$\mathcal{E} = \left\{ \bigcap_{i \in [n]} \bigcap_{t \in \mathbb{N}} |\hat{\mu}_i(t) - \mu_i| \le C_{\delta/n}(t) \right\}.$$

Using standard anytime confidence bound results, and recalling that that $C_\delta(t) := \sqrt{\frac{4 \log(\log_2(2t)/\delta)}{t}}$, we have

$$\mathbb{P}(\mathcal{E}^c) = \mathbb{P}\left( \bigcup_{i \in [n]} \bigcup_{t \in \mathbb{N}} |\hat{\mu}_i - \mu_i| > C_{\delta/n}(t) \right)$$

$$\le \sum_{i=1}^{n} \mathbb{P}\left( \bigcup_{t \in \mathbb{N}} |\hat{\mu}_i - \mu_i| > C_{\delta/n}(t) \right) \le \sum_{i=1}^{n} \frac{\delta}{n} = \delta$$

Hence, $\mathbb{P}(\mathcal{E}) \ge 1 - \delta$.

### E.6.1  Step 0: Correctness

**Claim 0:** On $\mathcal{E}$, first we prove that $G(t) \subset G_\epsilon$ for all $t \in \mathbb{N}$.

In particular, this shows that EAST never incorrectly add arms in $G_\epsilon^c$ to the set $G$.

**Proof.** We begin by showing that on $\mathcal{E}$ the best arm is never removed from $\mathcal{A}$ for all $t$. Note for any $i$

$$\hat{\mu}_1 + C_{\delta/n}(t) \ge \mu_1 \ge \mu_i \ge \hat{\mu}_i(t) - C_{\delta/n}(t) > \hat{\mu}_i(t) - C_{\delta/n}(t) - \epsilon.$$

In particular this shows, $\hat{\mu}_1 + C_{\delta/n}(t) > \max_{i \in \mathcal{A}} \hat{\mu}_i(t) - C_{\delta/n}(t) - \epsilon = L_t^*$ and $\hat{\mu}_1 + C_{\delta/n}(t) \ge \max_{i \in \mathcal{A}} \hat{\mu}_i(t) - C_{\delta/n}(t)$ showing that 1 will never exit $\mathcal{A}$ in line 11.

Secondly, we show that at all times $t$, $\mu_1 - \epsilon \in [L_t, U_t]$. By the above, since $\mu_1$ never leaves $\mathcal{A}$,

$$U_t = \max_{i \in \mathcal{A}} \hat{\mu}_i(t) + C_{\delta/n}(t) - \epsilon \ge \hat{\mu}_1(t) + C_{\delta/n}(t) - \epsilon \ge \mu_1 - \epsilon$$

and for any $i$,

$$\mu_1 - \epsilon \ge \mu_i - \epsilon \ge \hat{\mu}_i(t) - C_{\delta/n}(t) - \epsilon$$

Hence $\mu_1 - \epsilon \ge \max_i \hat{\mu}_i(t) - C_{\delta/n}(t) - \epsilon = L_t$.

Next, we show that $G(t) \subset G_\epsilon$ for all $t \ge 1$. Suppose not. Then $\exists, t \in N$ and $\exists i \in G_\epsilon^c \cap G(t)$ such that,

$$\mu_i \ge \hat{\mu}_i(t) - C_{\delta/n}(t) \ge U_t \ge \mu_1 - \epsilon > \mu_i,$$

with the last inequality following from the previous assertion, giving a contradiction. $\square$

**Claim 1:** Next, we show that on $\mathcal{E}$, $G_\epsilon \subset \mathcal{A}(t) \cup G(t)$ for all $t \in \mathbb{N}$.

In particular this implies that if $\mathcal{A} \subset G$, then $G_\epsilon \subset G$. Combining this with the previous claim gives $G \subset G_\epsilon \subset G$, hence $G = G_\epsilon$. On this condition, EAST terminates by line 2 and returns the set $\mathcal{A} \cup G = G$. Note that by definition, $G_\epsilon \subset G_{(\epsilon+\gamma)}$ for all $\gamma \ge 0$. Therefore EAST terminates correctly on this condition.

**Proof.** Suppose for contradiction that there exists $i \in G_\epsilon$ such that $i \notin \mathcal{A}(t) \cup G(t)$. This occurs only if $i$ is eliminated in line 10. Hence, there exists a $t' \le t$ such that $\hat{\mu}_i(t') + C_{\delta/n}(t') < L_{t'}$. Therefore, on the event $\mathcal{E}$,

$$\mu_1 - \epsilon \overset{\mathcal{E}}{\ge} L_{t'} = \max_{j \in \mathcal{A}} \hat{\mu}_j(t') - C_{\delta/n}(t') - \epsilon > \hat{\mu}_i(t') + C_{\delta/n}(t') \overset{\mathcal{E}}{\ge} \mu_i$$

which contradicts $i \in G_\epsilon$. $\square$

**Claim 2:** Finally, we show that if $U_t - L_t \le \gamma/2$, then $\mathcal{A} \cup G \subset G_{(\epsilon+\gamma)}$.

Combining with the previous that $G_\epsilon \subset \mathcal{A} \cup G$, if EAST terminates on this condition by line 2, it does so correctly.

**Proof.** Assume $U_t - L_t \leq \gamma/2$. This implies that

$$\big(\max_{i \in A(t)} \hat{\mu}_i(t) + C_{\delta/n}(t) - \epsilon\big) - \big(\max_{i \in A(t)} \hat{\mu}_i(t) - C_{\delta/n}(t) - \epsilon\big) = 2C_{\delta/n}(t) \leq \gamma/2.$$

Suppose for contradiction that there exists $i \in G^c_{(\epsilon+\gamma)}$ such that $i \in \mathcal{A} \cup G$. Since $G_\epsilon \cap G^c_{(\epsilon+\gamma)} = \emptyset$ and we have previously shown than $G(t) \subset G_\epsilon$ for all $t$, we have that $i \in A \backslash G$. Therefore, by the condition in line 10, $\hat{\mu}_i(t) + C_{\delta/n}(t) \geq L_t$. Hence, $\mu_i + 2C_{\delta/n}(t) \overset{\mathcal{E}}{\geq} \hat{\mu}_i(t) + C_{\delta/n}(t) \geq L_t$. By assumption, we have that $U_t - \gamma/2 \leq L_t$, and the event $\mathcal{E}$ implies that $U_t \geq \mu_1 - \epsilon$. Therefore, $\mu_i + 2C_{\delta/n}(t) \geq U_t - \gamma/2 \geq \mu_1 - \epsilon - \gamma/2$. Combining this with the inequality $2C_{\delta/n} \leq \gamma/2$, we have that

$$\gamma \geq 2C_{\delta/n}(t) + \gamma/2 \geq \mu_1 - \epsilon - \mu_i \overset{i \in G^c_{(\epsilon+\gamma)}}{>} \gamma$$

which is a contradiction. $\qquad\square$

Therefore, on the event $\mathcal{E}$, if EAST terminates due to either condition in line 2, it returns $\mathcal{A} \cup G$ such that $G_\epsilon \subset \mathcal{A} \cup G \subset G_{(\epsilon+\gamma)}$. Since $\mathbb{P}(\mathcal{E}) \geq 1 - \delta$, EAST terminates correctly with probability at least $1 - \delta$.

### E.6.2  Step 1: Controlling the total number of samples given by EAST to arms in $G_\epsilon$

To keep track of the number of samples that arms are given by EAST, we introduce random variables $T_i$ and $T'_i$ for all $i \in [n]$. When arm $i$ has been given $\max(T_i, T'_i)$ samples it is removed from $\mathcal{A}$ in line 11.

By Step 0, only arms in $G_\epsilon$ are added to $G$. Therefore, $T_i$ is defined as

$$T_i = \min \left\{ t : \begin{matrix} i \in G_k(t+1) & \text{if } i \in G_\epsilon \\ i \notin \mathcal{A}(t+1) & \text{if } i \in G^c_\epsilon \end{matrix} \right\} \overset{\mathcal{E}}{=} \min \left\{ t : \begin{matrix} \hat{\mu}_i - C_{\delta/n}(t) \geq U_t & \text{if } i \in G_\epsilon \\ \hat{\mu}_i + C_{\delta/n}(t) \leq L_t & \text{if } i \in G^c_\epsilon \end{matrix} \right\} \quad (24)$$

Similarly, recall $T'_i$ denotes the random variable of the of the number of times $i$ is sampled before $\hat{\mu}_i(t) + C_{\delta/n}(t) \leq \max_{j \in \mathcal{A}} \hat{\mu}_j(t) - C_{\delta/n}(t)$. Hence,

$$T'_i = \min \left\{ t : \hat{\mu}_i(t) + C_{\delta/n}(t) \leq \max_{j \in \mathcal{A}(t)} \hat{\mu}_j(t) - C_{\delta/n}(t) \right\} \quad (25)$$

**Claim 0:** For $i \in G_\epsilon$, we have that $T_i \leq h(0.25(\epsilon - \Delta_i), \delta/n)$.

**Proof.** Note that, $4C_{\delta/n}(t) \leq \mu_i - (\mu_1 - \epsilon)$, true when $t > h\left(0.25(\epsilon - \Delta_i), \frac{\delta}{n}\right)$, implies that for all $j$,

$$\begin{aligned} \hat{\mu}_i(t) - C_{\delta/n}(t) &\overset{\mathcal{E}}{\geq} \mu_i - 2C_{\delta/n}(t) \\ &\geq \mu_1 + 2C_{\delta/n}(t) - \epsilon \\ &\geq \mu_j + 2C_{\delta/n}(t) - \epsilon \\ &\overset{\mathcal{E}}{\geq} \hat{\mu}_j(t) + C_{\delta/n}(t) - \epsilon \end{aligned}$$

so in particular, $\hat{\mu}_i(t) - C_{\delta/n}(t) \geq \max_{j \in \mathcal{A}} \hat{\mu}_j(t) + C_{\delta/n}(t) - \epsilon = U_t$. $\qquad\square$

**Claim 1:** For $i \in G_\epsilon$, we have that $T'_i \leq h(0.25\Delta_i, \delta/n)$.

**Proof.** Note that $4C_{\delta/n}(t) \leq \mu_1 - \mu_i$, true when $t > h\left(0.25\Delta_i, \frac{\delta}{n}\right)$, implies that

$$\begin{aligned} \hat{\mu}_i(t) + C_{\delta/n}(t) &\overset{\mathcal{E}}{\leq} \mu_i + 2C_{\delta/n}(t) \\ &\leq \mu_1 - 2C_{\delta/n}(t) \\ &\overset{\mathcal{E}}{\leq} \hat{\mu}_1(t) - C_{\delta/n}(t). \end{aligned}$$

As shown in Step 0, $1 \in \mathcal{A}(t)$ for all $t \in \mathbb{N}$, and in particular $\hat{\mu}_1(t) \leq \max_{i \in \mathcal{A}(t)} \hat{\mu}_i(t)$. Hence, $\hat{\mu}_i(t) + C_{\delta/n}(t) \leq \max_{j \in \mathcal{A}(t)} \hat{\mu}_j(t) - C_{\delta/n}(t)$. $\qquad\square$

### E.6.3 Step 2: Controlling the total number of samples given by EAST to arms in $G_\epsilon^c$

**Claim:** Next, we show that $T_i \leq h\left(0.25(\epsilon - \Delta_i), \frac{\delta}{n}\right)$ for $i \in G_\epsilon^c$

**Proof.** Note that, $4C_{\delta/n}(t) \leq \mu_1 - \epsilon - \mu_i$, true when $t > h\left(0.25(\epsilon - \Delta_i), \frac{\delta}{n}\right)$, implies that

$$
\begin{aligned}
\hat{\mu}_i(t) + C_{\delta/n}(t) &\overset{\mathcal{E}}{\leq} \mu_i + 2C_{\delta/n}(t) \\
&\leq \mu_1 - 2C_{\delta/n}(t) - \epsilon \\
&\overset{\mathcal{E}}{\leq} \hat{\mu}_1(t) - C_{\delta/n}(t) - \epsilon
\end{aligned}
$$

As shown in Step 0, $1 \in \mathcal{A}(t)$ for all $t \in \mathbb{N}$, and in particular $\hat{\mu}_1(t) \leq \max_{i \in \mathcal{A}(t)} \hat{\mu}_i(t)$. Therefore $\hat{\mu}_i(t) + C_{\delta/n}(t) \leq \max_{j \in \mathcal{A}} \hat{\mu}_j(t) - C_{\delta/n}(t) - \epsilon = L_t$. $\qquad\square$

### E.6.4 Step 3: Bounding the total number of samples drawn by EAST

With the results of Steps 1 and 2, we may bound the total sample complexity of EAST. Note that independently of the event $\mathcal{E}$, EAST terminates if $U_t - L_t \leq \gamma/2$. Let the random variable of the maximum number of samples given to any arm before this occurs be $T_\gamma$. Additionally, EAST may terminate if $\mathcal{A} \subset G$. Let the random variable of maximum number of samples given to any arm before this occurs be $T_{\alpha_\epsilon \beta_\epsilon}$. Note that due to the sampling procedure, the total number of samples drawn by EAST at termination may be written as $\sum_{t=1}^{\min(T_\gamma, T_{\alpha_\epsilon \beta_\epsilon})} |\mathcal{A}(t)|$.

Now we bound $\sum_{t=1}^{\min(T_\gamma, T_{\alpha_\epsilon \beta_\epsilon})} |\mathcal{A}(t)|$. Let $S_i = \min\{t : i \notin A(t+1)\}$. Hence,

$$
\sum_{t=1}^{\min(T_\gamma, T_{\alpha_\epsilon \beta_\epsilon})} |\mathcal{A}(t)| = \sum_{t=1}^{\min(T_\gamma, T_{\alpha_\epsilon \beta_\epsilon})} \sum_{i=1}^{n} \mathbb{1}[i \in \mathcal{A}(t)] = \sum_{i=1}^{n} \sum_{t=1}^{\min(T_\gamma, T_{\alpha_\epsilon \beta_\epsilon})} \mathbb{1}[i \in \mathcal{A}(t)] = \sum_{i=1}^{n} \min\{T_\gamma, T_{\alpha_\epsilon \beta_\epsilon}, S_i\}
$$

For arms $i \in G_\epsilon^c$, $S_i = T_i$ by definition. For $i \in G_\epsilon$, $S_i = \max(T_i, T_i')$ by line 11 of the algorithm. Then

$$
\begin{aligned}
\sum_{i=1}^{n} \min\{T_\gamma, T_{\alpha_\epsilon \beta_\epsilon}, S_i\} &= \sum_{i \in G_\epsilon} \min\{T_\gamma, T_{\alpha_\epsilon \beta_\epsilon}, \max(T_i, T_i')\} + \sum_{i \in G_\epsilon^c} \min\{T_\gamma, T_{\alpha_\epsilon \beta_\epsilon}, T_i\} \\
&= \sum_{i \in G_\epsilon} \min\{T_\gamma, \min\{T_{\alpha_\epsilon \beta_\epsilon}, \max(T_i, T_i')\}\} + \sum_{i \in G_\epsilon^c} \min\{T_\gamma, T_{\alpha_\epsilon \beta_\epsilon}, T_i\} \\
&= \sum_{i \in G_\epsilon} \min\{T_\gamma, \max\{T_i, \min(T_i', T_{\alpha_\epsilon \beta_\epsilon})\}\} + \sum_{i \in G_\epsilon^c} \min\{T_\gamma, T_{\alpha_\epsilon \beta_\epsilon}, T_i\}
\end{aligned}
$$

We may define $T_\gamma := \min\{t : U_t - L_t \leq \gamma/2\}$. Note that $4C_{\delta/n}(t) \leq \gamma$, true when $t > h(0.25\gamma, \delta/n)$ implies that

$$
U_t - L_t = \left(\max_{i \in A(t)} \hat{\mu}_i(t) + C_{\delta/n}(t) - \epsilon\right) - \left(\max_{i \in A(t)} \hat{\mu}_i(t) - C_{\delta/n}(t) - \epsilon\right) = 2C_{\delta/n}(t) \leq \gamma/2.
$$

Therefore, we have that $T_\gamma \leq h(0.25\gamma, \delta/n)$.

Next, we may define $T_{\alpha_\epsilon \beta_\epsilon} = \min\{t : \mathcal{A}(t) \subset G_\epsilon\}$. By step 0, on the event $\mathcal{E}$, $\mathcal{A} \subset G$ implies that $G = G_\epsilon$. Therefore, $T_{\alpha_\epsilon \beta_\epsilon}$ may be equivalently defined as $T_{\alpha_\epsilon \beta_\epsilon} = \min\{t : G(t) = G_\epsilon \text{ and } G_\epsilon^c \cap \mathcal{A} = \emptyset\}$. Recalling the definition of $T_i$, we see that $T_{\alpha_\epsilon \beta_\epsilon} = \max_i(T_i)$.

Recall that by steps 1 and 2, $T_i \leq h\left(0.25(\epsilon - \Delta_i), \frac{\delta}{n}\right)$ and $T_i' \leq h\left(0.25\Delta_i, \frac{\delta}{n}\right)$. Furthermore, by monotonicity of $h(\cdot, \cdot)$, this implies that $T_{\alpha_\epsilon \beta_\epsilon} = h(0.25 \min(\alpha_\epsilon, \beta_\epsilon), \delta/n)$. Plugging this in, we see that

$$
\begin{aligned}
&\sum_{i \in G_\epsilon} \min\{T_\gamma, \max\{T_i, \min(T_i', T_{\alpha_\epsilon \beta_\epsilon})\}\} + \sum_{i \in G_\epsilon^c} \min\{T_\gamma, T_{\alpha_\epsilon \beta_\epsilon}, T_i\} \\
&= \sum_{i \in G_\epsilon} \min\{T_\gamma, \max\{T_i, \min(T_i', T_{\alpha_\epsilon \beta_\epsilon})\}\} + \sum_{i \in G_\epsilon^c} \min\{T_\gamma, T_i\} \\
&\leq \sum_{i \in G_\epsilon} \min\left\{\max\left\{h\left(0.25(\epsilon - \Delta_i), \frac{\delta}{n}\right), \min\left[h\left(0.25\Delta_i, \frac{\delta}{n}\right), h\left(0.25 \min(\alpha_\epsilon, \beta_\epsilon), \frac{\delta}{n}\right)\right]\right\}\right\},
\end{aligned}
$$

$$h\left(0.25\gamma, \frac{\delta}{n}\right)\Big\}$$

$$+ \sum_{i \in G_\epsilon^c} \min\left\{h\left(0.25(\epsilon - \Delta_i), \frac{\delta}{n}\right), h\left(0.25\min(\alpha_\epsilon, \beta_\epsilon), \frac{\delta}{n}\right)\right\}$$

$$= \sum_{i=1}^{n} \min\left\{\max\left\{h\left(0.25(\epsilon - \Delta_i), \frac{\delta}{n}\right), \min\left[h\left(0.25\Delta_i, \frac{\delta}{n}\right), h\left(0.25\min(\alpha_\epsilon, \beta_\epsilon), \frac{\delta}{n}\right)\right]\right\},$$

$$h\left(0.25\gamma, \frac{\delta}{n}\right)\Big\}$$

where the final equality holds by definition for arms in $G_\epsilon$. Next, by Lemma F.3, we may bound the minimum of $h(\cdot, \cdot)$ functions.

$$\sum_{i=1}^{n} \min\left\{\max\left\{h\left(\frac{\Delta_i - \epsilon}{4}, \frac{\delta}{n}\right), \min\left[h\left(\frac{\Delta_i}{4}, \frac{\delta}{n}\right), h\left(\frac{\min(\alpha_\epsilon, \beta_\epsilon)}{4}, \frac{\delta}{n}\right)\right]\right\},$$

$$h\left(\frac{\gamma}{4}, \frac{\delta}{n}\right)\Big\}$$

$$= \sum_{i=1}^{n} \min\left\{\max\left\{h\left(\frac{\Delta_i - \epsilon}{4}, \frac{\delta}{n}\right),\right.\right.$$

$$\min\left[h\left(\frac{\Delta_i}{4}, \frac{\delta}{n}\right), \max\left[h\left(\frac{\alpha_\epsilon}{4}, \frac{\delta}{n}\right), h\left(\frac{\beta_\epsilon}{4}, \frac{\delta}{n}\right)\right]\right]\Big\},$$

$$h\left(\frac{\gamma}{4}, \frac{\delta}{n}\right)\Big\}$$

$$\leq \sum_{i=1}^{n} \min\left\{\max\left\{h\left(\frac{\Delta_i - \epsilon}{4}, \frac{\delta}{n}\right),\right.\right.$$

$$\max\left[h\left(\frac{\Delta_i + \alpha_\epsilon}{8}, \frac{\delta}{n}\right), h\left(\frac{\Delta_i + \beta_\epsilon}{8}, \frac{\delta}{n}\right)\right]\Big\},$$

$$h\left(\frac{\gamma}{4}, \frac{\delta}{n}\right)\Big\}$$

$$= \sum_{i=1}^{n} \min\left\{\max\left\{h\left(\frac{\Delta_i - \epsilon}{4}, \frac{\delta}{n}\right), h\left(\frac{\Delta_i + \alpha_\epsilon}{8}, \frac{\delta}{n}\right), h\left(\frac{\Delta_i + \beta_\epsilon}{8}, \frac{\delta}{n}\right)\right\},$$

$$h\left(\frac{\gamma}{4}, \frac{\delta}{n}\right)\Big\}$$

Finally, we use Lemma F.2 to bound the function $h(\cdot, \cdot)$. Since $\delta \leq 1/2$, $\delta/n \leq 2e^{-e/2}$. Further, $\max(\Delta_i, |\epsilon - \Delta_i|) \leq 8$ for all $i$, we have that $0.25\Delta_i \leq 2$, $0.25|\epsilon - \Delta_i| \leq 2$, and $0.25\min(\alpha_\epsilon, \beta_\epsilon) \leq 2$. Therefore,

$$\sum_{i=1}^{n} \min\left\{\max\left\{h\left(\frac{\Delta_i - \epsilon}{4}, \frac{\delta}{n}\right), h\left(\frac{\Delta_i + \alpha_\epsilon}{8}, \frac{\delta}{n}\right), h\left(\frac{\Delta_i + \beta_\epsilon}{8}, \frac{\delta}{n}\right)\right\},$$

$$h\left(\frac{\gamma}{4}, \frac{\delta}{n}\right)\Big\}$$

$$\leq \sum_{i=1}^{n} \min\left\{\max\left\{\frac{64}{(\epsilon - \Delta_i)^2} \log\left(\frac{2n}{\delta} \log_2\left(\frac{192n}{\delta(\epsilon - \Delta_i)^2}\right)\right),\right.\right.$$

$$\frac{256}{(\Delta_i + \alpha_\epsilon)^2} \log\left(\frac{2n}{\delta} \log_2\left(\frac{768n}{\delta(\Delta_i + \alpha_\epsilon)^2}\right)\right),$$

$$\frac{256}{(\Delta_i + \beta_\epsilon)^2} \log\left(\frac{2n}{\delta} \log_2\left(\frac{768n}{\delta(\Delta_i + \beta_\epsilon)^2}\right)\right)\Big\},$$

$$\left. \frac{64}{\gamma^2} \log\left( \frac{2n}{\delta} \log_2\left( \frac{192n}{\delta\gamma^2} \right) \right) \right\}$$

$$= \sum_{i=1}^{n} \min\left\{ \max\left\{ \frac{64}{(\mu_1 - \epsilon - \mu_i)^2} \log\left( \frac{2n}{\delta} \log_2\left( \frac{768n}{\delta(\mu_1 - \epsilon - \mu_i)^2} \right) \right), \right.\right.$$

$$\frac{256}{(\mu_1 + \alpha_\epsilon - \mu_i)^2} \log\left( \frac{2n}{\delta} \log_2\left( \frac{768n}{\delta(\mu_1 + \alpha_\epsilon - \mu_i)^2} \right) \right),$$

$$\left. \frac{256}{(\mu_1 + \beta_\epsilon - \mu_i)^2} \log\left( \frac{2n}{\delta} \log_2\left( \frac{768n}{\delta(\mu_1 + \beta_\epsilon - \mu_i)^2} \right) \right) \right\},$$

$$\left. \frac{64}{\gamma^2} \log\left( \frac{2n}{\delta} \log_2\left( \frac{192n}{\delta\gamma^2} \right) \right) \right\}.$$

$\qquad\qquad\qquad\qquad\qquad\qquad\qquad\qquad\qquad\qquad\qquad\qquad\qquad\qquad\qquad\qquad\qquad\Box$

## E.7 Proof of Theorem E.4, EAST in the multiplicative regime

*Proof.* **Notation for the proof:** Throughout, recall $\Delta_i = \mu_1 - \mu_i$. Recall that $t$ counts the number of times each arm in $\mathcal{A}$ has been sampled and thus the number of times that the conditionals in Lines 10 and 11 have been evaluated. Let $\mathcal{A}(t)$ denote the state $\mathcal{A}$ at this time before the arms have been eliminated from $\mathcal{A}$ in lines 10 and 11. Let $G(t)$ be defined similarly. Therefore, the total number of samples drawn by EAST up to time $t$ is $\sum_{s=1}^{t} |\mathcal{A}(s)|$.

For $i \in M_\epsilon$, let $T_i$ denote the random variable of the number of times arm $i$ is sampled before it is added to $G$ in Line 8. For $i \in M_\epsilon^c$, let $T_i$ denote the random variable of the number of times arm $i$ is sampled before it is removed from $\mathcal{A}$ in Line 10. For any arm $i$, let $T_i'$ denote the random variable of the of the number of times $i$ is sampled before $\hat\mu_i(t) + C_{\delta/n}(t) \leq \max_{j \in \mathcal{A}} \hat\mu_j(t) - C_{\delta/n}(t)$.

Define the event

$$\mathcal{E} = \left\{ \bigcap_{i \in [n]} \bigcap_{t \in \mathbb{N}} |\hat\mu_i(t) - \mu_i| \leq C_{\delta/n}(t) \right\}.$$

Using standard anytime confidence bound results, and recalling that that $C_\delta(t) := \sqrt{\frac{4\log(\log_2(2t)/\delta)}{t}}$, we have

$$\mathbb{P}(\mathcal{E}^c) = \mathbb{P}\left( \bigcup_{i \in [n]} \bigcup_{t \in \mathbb{N}} |\hat\mu_i - \mu_i| > C_{\delta/n}(t) \right)$$

$$\leq \sum_{i=1}^{n} \mathbb{P}\left( \bigcup_{t \in \mathbb{N}} |\hat\mu_i - \mu_i| > C_{\delta/n}(t) \right) \leq \sum_{i=1}^{n} \frac{\delta}{n} = \delta$$

Hence, $\mathbb{P}(\mathcal{E}) \geq 1 - \delta$.

### E.7.1 Step 0: Correctness

**Claim 0:** On $\mathcal{E}$, first we prove that $G(t) \subset M_\epsilon$ for all $t \in \mathbb{N}$.

In particular, this shows that EAST never incorrectly add arms in $M_\epsilon^c$ to the set $G$.

**Proof.** Firstly we show $1 \in \mathcal{A}$ for all $t \in \mathbb{N}$, namely the best arm is never removed from $\mathcal{A}$. Note for any $i$ such that $\hat\mu_i(t) - C_{\delta/n}(t) \geq 0$,

$$\hat\mu_1 + C_{\delta/n}(t) \geq \mu_1 \geq \mu_i \geq \hat\mu_i(t) - C_{\delta/n}(t) > (1 - \epsilon)(\hat\mu_i(t) - C_{\delta/n}(t)).$$

For $i$ such that $\hat\mu_i(t) - C_{\delta/n}(t) < 0$, if $\hat\mu_1 + C_{\delta/n}(t) \geq 0$, then

$$\hat\mu_1 + C_{\delta/n}(t) \geq 0 > (1 - \epsilon)(\hat\mu_i(t) - C_{\delta/n}(t)).$$

Note that $\hat\mu_1 + C_{\delta/n}(t) < 0$ implies on the event $\mathcal{E}$ that $\mu_1 < 0$, which contradicts the assumption that $\mu_1 \geq 0$ made in the theorem. In particular this shows, $\hat\mu_1 + C_{\delta/n}(t) > (1 - \epsilon)(\max_{i \in \mathcal{A}} \hat\mu_i(t) -$

$C_{\delta/n}(t)) = L_t$ and $\hat{\mu}_1 + C_{\delta/n}(t) \geq \max_{i \in \mathcal{A}} \hat{\mu}_i(t) - C_{\delta/n}(t)$ showing that $1$ will never exit $\mathcal{A}$ in line 28.

Secondly, we show that at all times $t$, $(1 - \epsilon)\mu_1 \in [L_t, U_t]$. By the above, since $\mu_1$ never leaves $\mathcal{A}$,

$$U_t = (1 - \epsilon)(\max_{i \in \mathcal{A}} \hat{\mu}_i(t) + C_{\delta/n}(t)) \geq (1 - \epsilon)(\hat{\mu}_1(t) + C_{\delta/n}(t)) \geq (1 - \epsilon)\mu_1$$

and for any $i$,

$$(1 - \epsilon)\mu_1 \geq (1 - \epsilon)\mu_i \geq (1 - \epsilon)(\hat{\mu}_i(t) - C_{\delta/n}(t))$$

Hence $(1 - \epsilon)\mu_1 \geq (1 - \epsilon)(\max_i \hat{\mu}_i(t) - C_{\delta/n}(t)) = L_t$.

Next, we show that $G \subset M_\epsilon$ for all $k \geq 1, t \geq 1$. Suppose not. Then $\exists, k, t \in N$ and $\exists i \in M_\epsilon^c \cap G(t)$ such that,

$$\mu_i \geq \hat{\mu}_i(t) - C_{\delta/n}(t) \geq U_t \geq (1 - \epsilon)\mu_1 > \mu_i,$$

with the last inequality following from the previous assertion, giving a contradiction. $\qquad\square$

**Claim 1:** Next, we show that on $\mathcal{E}$, $M_\epsilon \subset \mathcal{A}(t) \cup G(t)$ for all $t \in \mathbb{N}$.

In particular this implies that if $\mathcal{A} \subset G$, then $M_\epsilon \subset G$. Combining this with the previous claim gives $G \subset M_\epsilon \subset G$, hence $G = M_\epsilon$. On this condition, EAST terminates and returns the set $\mathcal{A} \cup G = G$. Note that by definition, $M_\epsilon \subset M_{(\epsilon+\gamma)}$ for all $\gamma \geq 0$. Therefore EAST terminates correctly on this condition.

**Proof.** Suppose for contradiction that there exists $i \in M_\epsilon$ such that $i \notin \mathcal{A}(t) \cup G(t)$. This occurs only if $i$ is eliminated in line 10. Hence, there exists a $t' \leq t$ such that $\hat{\mu}_i(t') + C_{\delta/n}(t') < L_{t'}$. Therefore, on the event $\mathcal{E}$,

$$(1 - \epsilon)\mu_1 \overset{\mathcal{E}}{\geq} L_{t'} = (1 - \epsilon)\left(\max_{j \in \mathcal{A}} \hat{\mu}_j(t') - C_{\delta/n}(t')\right) > \hat{\mu}_i(t') + C_{\delta/n}(t') \overset{\mathcal{E}}{\geq} \mu_i$$

which contradicts $i \in M_\epsilon$. $\qquad\square$

**Claim 2:** Finally, we show that on $\mathcal{E}$, if $U_t - L_t \leq \frac{\gamma}{2-\epsilon} L_t$, then $\mathcal{A} \cup G \subset M_{(\epsilon+\gamma)}$.

Combining with Claim 1 that $M_\epsilon \subset \mathcal{A} \cup G$, if EAST terminates on this condition, it does so correctly and returns all arms in $M_\epsilon$ and none in $M_{(\epsilon+\gamma)}^c$.

**Proof.** By Claim 0, $G \subset M_\epsilon \subset M_{\epsilon+\gamma}$. Hence, $G \cap M_{(\epsilon+\gamma)}^c = \emptyset$. Therefore, we wish to show that $\mathcal{A} \cap M_{(\epsilon+\gamma)}^c = \emptyset$ which implies that $G \cap \mathcal{A} \subset M_{\epsilon+\gamma}$. Assume $U_t - L_t < \frac{\gamma}{2-\epsilon} L_t$. Recall that

$$U_t = (1 - \epsilon)\left(\max_{i \in \mathcal{A}} \hat{\mu}_i(t) + C_{\delta/n}(t)\right)$$

and

$$L_t = (1 - \epsilon)\left(\max_{i \in \mathcal{A}} \hat{\mu}_i(t) - C_{\delta/n}(t)\right)$$

All arms in $\mathcal{A}(t)$ have received exactly $t$ samples. Hence, $U_t - L_t = 2(1 - \epsilon)C_{\delta/n}(t)$. On $\mathcal{E}$, $L_t \leq (1 - \epsilon)\mu_1$ This implies that

$$2(1 - \epsilon)C_{\delta/n}(t) < \frac{\gamma}{2 - \epsilon} L_t \leq \frac{1 - \epsilon}{2 - \epsilon} \gamma \mu_1,$$

and in particular,

$$2C_{\delta/n}(t) < \frac{\gamma \mu_1}{2 - \epsilon}.$$

Therefore, we wish to show that when the above is true, then for any $i \in M_{\epsilon+\gamma}^c$, $L_t - (\hat{\mu}_i(t) + C_{\delta/n}(t)) > 0$, implying that $i \notin \mathcal{A}$.

$$L_t - (\hat{\mu}_i(t) + C_{\delta/n}(t)) = (1 - \epsilon)\left(\max_{j \in \mathcal{A}} \hat{\mu}_j - C_{\delta/n}(t)\right) - (\hat{\mu}_i(t) + C_{\delta/n}(t))$$

$$\geq (1 - \epsilon)\left(\max_{j \in \mathcal{A}} \mu_j - 2C_{\delta/n}(t)\right) - (\mu_i + 2C_{\delta/n}(t))$$

$$\overset{(a)}{\geq} (1-\epsilon)\left(\mu_1 - 2C_{\delta/n}(t)\right) - \left((1-\epsilon-\gamma)\mu_1 + 2C_{\delta/n}(t)\right)$$
$$= \gamma\mu_1 - 2(2-\epsilon)C_{\delta/n}(t)$$
$$> \gamma\mu_1 - (2-\epsilon)\frac{\gamma\mu_1}{2-\epsilon}$$
$$= 0$$

which implies that $i \notin \mathcal{A}$. Inequality $(a)$ follows jointly from the fact that $1 \in \mathcal{A}$ and the fact that all arms in $\mathcal{A}$ have received $t$ samples implies $\max_{j \in \mathcal{A}} \mu_j - 2C_{\delta/n}(t) = \mu_1 - 2C_{\delta/n}(t)$. Additionally, inequality $(a)$ follows from $\mu_i \leq (1-\epsilon-\gamma)\mu_1$ since $i \in M_{\epsilon+\gamma}^c$. $\qquad\square$

Therefore, on the event $\mathcal{E}$, if EAST terminates due to either condition in line 2, it returns $\mathcal{A} \cup G$ such that $M_\epsilon \subset \mathcal{A} \cup G \subset M_{(\epsilon+\gamma)}$. Since $\mathbb{P}(\mathcal{E}) \geq 1 - \delta$, EAST terminates correctly with probability at least $1 - \delta$.

### E.7.2 Step 1: Controlling the total number of samples given by EAST to arms in $M_\epsilon$

To keep track of the number of samples that arms are given by EAST, we introduce random variables $T_i$ and $T_i'$ for all $i \in [n]$. When arm $i$ has been given $\max(T_i, T_i')$ samples it is removed from $\mathcal{A}$ in line 11.

By Step 0, only arms in $M_\epsilon$ are added to $G$. Therefore, $T_i$ is defined as

$$T_i = \min\left\{ t : \begin{array}{ll} i \in G(t+1) & \text{if } i \in M_\epsilon \\ i \notin \mathcal{A}(t+1) & \text{if } i \in M_\epsilon^c \end{array} \right\} \overset{\mathcal{E}}{=} \min\left\{ t : \begin{array}{ll} \hat{\mu}_i - C_{\delta/n}(t) \geq U_t & \text{if } i \in M_\epsilon \\ \hat{\mu}_i + C_{\delta/n}(t) \leq L_t & \text{if } i \in M_\epsilon^c \end{array} \right\} \quad (26)$$

Similarly, recall $T_i'$ denotes the random variable of the of the number of times $i$ is sampled before $\hat{\mu}_i(t) + C_{\delta/n}(t) \leq \max_{j \in \mathcal{A}} \hat{\mu}_j(t) - C_{\delta/n}(t)$. Hence,

$$T_i' = \min\left\{ t : \hat{\mu}_i(t) + C_{\delta/n}(t) \leq \max_{j \in \mathcal{A}(t)} \hat{\mu}_j(t) - C_{\delta/n}(t) \right\} \quad (27)$$

**Claim 0:** For $i \in M_\epsilon$, we have that $T_i \leq h\left(\frac{\epsilon\mu_1 - \Delta_i}{4 - 2\epsilon}, \frac{\delta}{n}\right)$.

**Proof.** Note that $\mu_i - 2C_{\delta/n}(t) \geq (1-\epsilon)(\mu_1 + 2C_{\delta/n}(t))$ may be rearranged as $(4 - 2\epsilon)C_{\delta/n}(t) \leq \epsilon\mu_1 - \Delta_i$, and this is true when $t > h\left(\frac{\epsilon\mu_1 - \Delta_i}{4 - 2\epsilon}, \frac{\delta}{n}\right)$. This condition implies that for all $j$,

$$\hat{\mu}_i(t) - C_{\delta/n}(t) \overset{\mathcal{E}}{\geq} \mu_i - 2C_{\delta/n}(t)$$
$$\geq (1-\epsilon)(\mu_1 + 2C_{\delta/n}(t))$$
$$\geq (1-\epsilon)(\mu_j + 2C_{\delta/n}(t))$$
$$\overset{\mathcal{E}}{\geq} (1-\epsilon)(\hat{\mu}_j(t) + C_{\delta/n}(t))$$

so in particular, $\hat{\mu}_i(t) - C_{\delta/n}(t) \geq (1-\epsilon)(\max_{j \in \mathcal{A}} \hat{\mu}_j(t) + C_{\delta/n}(t)) = U_t$. $\qquad\square$

**Claim 1:** For $i \in M_\epsilon$, we have that $T_i' \leq h(0.25\Delta_i, \delta/n)$.

**Proof.** Note that $4C_{\delta/n}(t) \leq \mu_1 - \mu_i$, true when $t > h\left(0.25\Delta_i, \frac{\delta}{n}\right)$, implies that

$$\hat{\mu}_i(t) + C_{\delta/n}(t) \overset{\mathcal{E}}{\leq} \mu_i + 2C_{\delta/n}(t)$$
$$\leq \mu_1 - 2C_{\delta/n}(t)$$
$$\overset{\mathcal{E}}{\leq} \hat{\mu}_1(t) - C_{\delta/n}(t).$$

As shown in Step 0, $1 \in \mathcal{A}(t)$ for all $t \in \mathbb{N}$, and in particular $\hat{\mu}_1(t) \leq \max_{i \in \mathcal{A}(t)} \hat{\mu}_i(t)$. Hence, $\hat{\mu}_i(t) + C_{\delta/n}(t) \leq \max_{j \in \mathcal{A}(t)} \hat{\mu}_j(t) - C_{\delta/n}(t)$. $\qquad\square$

### E.7.3  Step 2: Controlling the total number of samples given by EAST to arms in $M_\epsilon^c$

Next, we bound $T_i$ for $i \in M_\epsilon^c$. $i \in M_\epsilon^c$ is eliminated from $\mathcal{A}$ if it has received at least $T_i$ samples.

**Claim:** $T_i \leq h\left(\frac{\Delta_i - \epsilon\mu_1}{4 - 2\epsilon}, \frac{\delta}{n}\right)$ for $i \in M_\epsilon^c$

**Proof.** Note that $\mu_i + 2C_{\delta/n}(t) \leq (1 - \epsilon)(\mu_1 - 2C_{\delta/n}(t))$ may be rearranged as $(4 - 2\epsilon)C_{\delta/n}(t) \leq \Delta_i - \epsilon\mu_1$, and this is true when $t > h\left(\frac{\Delta_i - \epsilon\mu_1}{4 - 2\epsilon}, \frac{\delta}{n}\right)$. This condition implies that

$$\hat{\mu}_i(t) + C_{\delta/n}(t) \overset{\mathcal{E}}{\leq} \mu_i + 2C_{\delta/n}(t)$$
$$\leq (1 - \epsilon)(\mu_1 - 2C_{\delta/n}(t))$$
$$\overset{\mathcal{E}}{\leq} (1 - \epsilon)(\hat{\mu}_1(t) - C_{\delta/n}(t))$$

As shown in Step 0, $1 \in \mathcal{A}(t)$ for all $t \in \mathbb{N}$, and in particular $\hat{\mu}_1(t) \leq \max_{i \in \mathcal{A}(t)} \hat{\mu}_i(t)$. Therefore $\hat{\mu}_i(t) + C_{\delta/n}(t) \leq (1 - \epsilon)(\max_{j \in \mathcal{A}} \hat{\mu}_j(t) - C_{\delta/n}(t)) = L_t$.  $\square$

### E.7.4  Step 3: Bounding the total number of samples drawn by EAST

With the results of Steps 1 and 2, we may bound the total sample complexity of EAST. Note that independently of the event $\mathcal{E}$, EAST terminates if $U_t - L_t \leq \frac{\gamma}{2-\epsilon}L_t$. Let the random variable of the maximum number of samples given to any arm before this occurs be $T_\gamma := \min\{t : U_t - L_t \leq \frac{\gamma}{2-\epsilon}L_t\}$. Additionally, EAST may terminate if $\mathcal{A} \subset G$. Let the random variable of maximum number of samples given to any arm before this occurs be $T_{\tilde{\alpha}_\epsilon \tilde{\beta}_\epsilon}$. Note that due to the sampling procedure, the total number of samples drawn by EAST at termination may be written as $\sum_{t=1}^{\min(T_\gamma, T_{\tilde{\alpha}_\epsilon \tilde{\beta}_\epsilon})} |\mathcal{A}(t)|$.

Now we bound $\sum_{t=1}^{\min(T_\gamma, T_{\tilde{\alpha}_\epsilon \tilde{\beta}_\epsilon})} |\mathcal{A}(t)|$. Let $S_i = \min\{t : i \notin A(t+1)\}$. Hence,

$$\sum_{t=1}^{\min(T_\gamma, T_{\tilde{\alpha}_\epsilon \tilde{\beta}_\epsilon})} |\mathcal{A}(t)| = \sum_{t=1}^{\min(T_\gamma, T_{\tilde{\alpha}_\epsilon \tilde{\beta}_\epsilon})} \sum_{i=1}^{n} \mathbb{1}[i \in \mathcal{A}(t)] = \sum_{i=1}^{n} \sum_{t=1}^{\min(T_\gamma, T_{\tilde{\alpha}_\epsilon \tilde{\beta}_\epsilon})} \mathbb{1}[i \in \mathcal{A}(t)] = \sum_{i=1}^{n} \min\left\{T_\gamma, T_{\tilde{\alpha}_\epsilon \tilde{\beta}_\epsilon}, S_i\right\}$$

For arms $i \in M_\epsilon^c$, $S_i = T_i$ by definition. For $i \in M_\epsilon$, $S_i = \max(T_i, T_i')$ by line 11 of the algorithm. Then

$$\sum_{i=1}^{n} \min\left\{T_\gamma, T_{\tilde{\alpha}_\epsilon \tilde{\beta}_\epsilon}, S_i\right\} = \sum_{i \in M_\epsilon} \min\left\{T_\gamma, T_{\tilde{\alpha}_\epsilon \tilde{\beta}_\epsilon}, \max(T_i, T_i')\right\} + \sum_{i \in M_\epsilon^c} \min\left\{T_\gamma, T_{\tilde{\alpha}_\epsilon \tilde{\beta}_\epsilon}, T_i\right\}$$
$$= \sum_{i \in M_\epsilon} \min\left\{T_\gamma, \min\left\{T_{\tilde{\alpha}_\epsilon \tilde{\beta}_\epsilon}, \max(T_i, T_i')\right\}\right\} + \sum_{i \in M_\epsilon^c} \min\left\{T_\gamma, T_{\tilde{\alpha}_\epsilon \tilde{\beta}_\epsilon}, T_i\right\}$$
$$= \sum_{i \in M_\epsilon} \min\left\{T_\gamma, \max\left\{T_i, \min(T_i', T_{\tilde{\alpha}_\epsilon \tilde{\beta}_\epsilon})\right\}\right\} + \sum_{i \in M_\epsilon^c} \min\left\{T_\gamma, T_{\tilde{\alpha}_\epsilon \tilde{\beta}_\epsilon}, T_i\right\}$$

Next we bound $T_\gamma$.

**Claim:** On $\mathcal{E}$, $T_\gamma \leq h\left(\frac{\gamma\mu_1}{2(2-\epsilon+\gamma)}, \frac{\delta}{n}\right)$.

**Proof:** $C_{\delta/n}(t) < \frac{\gamma\mu_1}{2(2-\epsilon+\gamma)}$ is true when $t \geq h\left(\frac{\gamma\mu_1}{2(2-\epsilon+\gamma)}, \frac{\delta}{n}\right)$. Note that

$$C_{\delta/n}(t) < \frac{\gamma\mu_1}{2(2 - \epsilon + \gamma)} \iff 2C_{\delta/n}(t) < \frac{\gamma}{2 - \epsilon}\left(\mu_1 - 2C_{\delta/n}(t)\right).$$

This implies that

$$U_t - L_t = 2(1 - \epsilon)C_{\delta/n}(t)$$
$$< 2\frac{1 - \epsilon}{2 - \epsilon}\gamma\left(\mu_1 - 2C_{\delta/n}(t)\right)$$
$$\leq \frac{1 - \epsilon}{2 - \epsilon}\gamma\left(\hat{\mu}_1(t) - C_{\delta/n}(t)\right)$$

$$\leq \frac{1-\epsilon}{2-\epsilon}\gamma\left(\max_{i\in\mathcal{A}}\hat{\mu}_i - C_{\delta/n}(t)\right)$$

$$= \frac{\gamma}{2-\epsilon}L_t$$

$\square$

Next, we may define $T_{\tilde{\alpha}_\epsilon\tilde{\beta}_\epsilon} = \min\{t : \mathcal{A}(t) \subset M_\epsilon\}$. By step 0, on the event $\mathcal{E}$, $\mathcal{A} \subset G$ implies that $G = M_\epsilon$. Therefore, $T_{\tilde{\alpha}_\epsilon\tilde{\beta}_\epsilon}$ may be equivalently defined as $T_{\tilde{\alpha}_\epsilon\tilde{\beta}_\epsilon} = \min\{t : G(t) = M_\epsilon \text{ and } M_\epsilon^c \cap \mathcal{A} = \emptyset\}$. Recalling the definition of $T_i$, we see that $T_{\tilde{\alpha}_\epsilon\tilde{\beta}_\epsilon} = \max_i(T_i)$.

Recall that by steps 1 and 2, $T_i \leq h\left(\frac{\epsilon\mu_1 - \Delta_i}{4-2\epsilon}, \frac{\delta}{n}\right)$ and $T_i' \leq h\left(0.25\Delta_i, \frac{\delta}{n}\right)$. Furthermore, by monotonicity of $h(\cdot, \cdot)$, this implies that $T_{\tilde{\alpha}_\epsilon\tilde{\beta}_\epsilon} = h\left(\frac{\min(\tilde{\alpha}_\epsilon,\tilde{\beta}_\epsilon)}{4-2\epsilon}, \frac{\delta}{n}\right)$. Plugging this in, we see that

$$\sum_{i\in M_\epsilon} \min\left\{T_\gamma, \max\left\{T_i, \min(T_i', T_{\tilde{\alpha}_\epsilon\tilde{\beta}_\epsilon})\right\}\right\} + \sum_{i\in M_\epsilon^c} \min\left\{T_\gamma, T_{\tilde{\alpha}_\epsilon\tilde{\beta}_\epsilon}, T_i\right\}$$

$$= \sum_{i\in M_\epsilon} \min\left\{T_\gamma, \max\left\{T_i, \min(T_i', T_{\tilde{\alpha}_\epsilon\tilde{\beta}_\epsilon})\right\}\right\} + \sum_{i\in M_\epsilon^c} \min\left\{T_\gamma, T_i\right\}$$

$$\leq \sum_{i\in M_\epsilon} \min\left\{\max\left\{h\left(\frac{\epsilon\mu_1 - \Delta_i}{4-2\epsilon}, \frac{\delta}{n}\right), \min\left[h\left(0.25\Delta_i, \frac{\delta}{n}\right), h\left(\frac{\min(\tilde{\alpha}_\epsilon,\tilde{\beta}_\epsilon)}{4-2\epsilon}, \frac{\delta}{n}\right)\right]\right\},$$

$$h\left(\frac{\gamma\mu_1}{2(2-\epsilon+\gamma)}, \frac{\delta}{n}\right)\right\}$$

$$+ \sum_{i\in M_\epsilon^c} \min\left\{h\left(\frac{\epsilon\mu_1 - \Delta_i}{4-2\epsilon}, \frac{\delta}{n}\right), h\left(\frac{\gamma\mu_1}{2(2-\epsilon+\gamma)}, \frac{\delta}{n}\right)\right\}$$

$$= \sum_{i=1}^{n} \min\left\{\max\left\{h\left(\frac{\epsilon\mu_1 - \Delta_i}{4-2\epsilon}, \frac{\delta}{n}\right), \min\left[h\left(0.25\Delta_i, \frac{\delta}{n}\right), h\left(\frac{\min(\tilde{\alpha}_\epsilon,\tilde{\beta}_\epsilon)}{4-2\epsilon}, \frac{\delta}{n}\right)\right]\right\},$$

$$h\left(\frac{\gamma\mu_1}{2(2-\epsilon+\gamma)}, \frac{\delta}{n}\right)\right\}$$

where the final equality holds by definition for arms in $M_\epsilon$. Lastly, note that $\frac{1}{3(1-x)} \leq \frac{1}{2-x}$ for $x \leq 1/2$. By monotonicity of $h$, we may lower bound the denominators $\frac{1}{4-2\epsilon}$ and $\frac{1}{2(2-\epsilon+\gamma)}$ as $\frac{1}{6(1-\epsilon)}$ and $\frac{1}{6(1-\epsilon+\gamma)}$ respectively. Since $\epsilon \in (0, 1/2]$, we may likewise lower bound $\frac{1}{4-2\epsilon}$ as $1/4$. Plugging this in, we see that

$$\sum_{i=1}^{n} \min\left\{\max\left\{h\left(\frac{\epsilon\mu_1 - \Delta_i}{4-2\epsilon}, \frac{\delta}{n}\right), \min\left[h\left(0.25\Delta_i, \frac{\delta}{n}\right), h\left(\frac{\min(\tilde{\alpha}_\epsilon,\tilde{\beta}_\epsilon)}{4-2\epsilon}, \frac{\delta}{n}\right)\right]\right\},$$

$$h\left(\frac{\gamma\mu_1}{2(2-\epsilon+\gamma)}, \frac{\delta}{n}\right)\right\}$$

$$\leq \sum_{i=1}^{n} \min\left\{\max\left\{h\left(\frac{\epsilon\mu_1 - \Delta_i}{4}, \frac{\delta}{n}\right), \min\left[h\left(0.25\Delta_i, \frac{\delta}{n}\right), h\left(\frac{\min(\tilde{\alpha}_\epsilon,\tilde{\beta}_\epsilon)}{6(1-\epsilon)}, \frac{\delta}{n}\right)\right]\right\},$$

$$h\left(\frac{\gamma\mu_1}{6(1-\epsilon+\gamma)}, \frac{\delta}{n}\right)\right\}$$

Next, by Lemma F.3, we may bound the minimum of $h(\cdot, \cdot)$ functions.

$$\sum_{i=1}^{n} \min\left\{\max\left\{h\left(\frac{\Delta_i - \epsilon\mu_1}{4}, \frac{\delta}{n}\right), \min\left[h\left(\frac{\Delta_i}{4}, \frac{\delta}{n}\right), h\left(\frac{\min(\tilde{\alpha}_\epsilon,\tilde{\beta}_\epsilon)}{6(1-\epsilon)}, \frac{\delta}{n}\right)\right]\right\},$$

$$h\left(\frac{\gamma\mu_1}{6(1-\epsilon+\gamma)}, \frac{\delta}{n}\right)\right\}$$

$$= \sum_{i=1}^{n} \min \left\{ \max \left\{ h\left( \frac{\Delta_i - \epsilon \mu_i}{4}, \frac{\delta}{n} \right), \right. \right.$$

$$\min \left[ h\left( \frac{\Delta_i}{4}, \frac{\delta}{n} \right), \max \left[ h\left( \frac{\tilde{\alpha}_\epsilon}{6(1-\epsilon)}, \frac{\delta}{n} \right), h\left( \frac{\tilde{\beta}_\epsilon}{6(1-\epsilon)}, \frac{\delta}{n} \right) \right] \right] \right\},$$

$$\left. h\left( \frac{\gamma \mu_i}{6(1-\epsilon+\gamma)}, \frac{\delta}{n} \right) \right\}$$

$$\leq \sum_{i=1}^{n} \min \left\{ \max \left\{ h\left( \frac{\Delta_i - \epsilon \mu_i}{4}, \frac{\delta}{n} \right), \right. \right.$$

$$\max \left[ h\left( \frac{\Delta_i + \frac{\tilde{\alpha}_\epsilon}{1-\epsilon}}{12}, \frac{\delta}{n} \right), h\left( \frac{\Delta_i + \frac{\tilde{\beta}_\epsilon}{1-\epsilon}}{12}, \frac{\delta}{n} \right) \right] \right\},$$

$$\left. h\left( \frac{\gamma \mu_i}{6(1-\epsilon+\gamma)}, \frac{\delta}{n} \right) \right\}$$

$$= \sum_{i=1}^{n} \min \left\{ \max \left\{ h\left( \frac{\Delta_i - \epsilon \mu_i}{4}, \frac{\delta}{n} \right), h\left( \frac{\Delta_i + \frac{\tilde{\alpha}_\epsilon}{1-\epsilon}}{12}, \frac{\delta}{n} \right), h\left( \frac{\Delta_i + \frac{\tilde{\beta}_\epsilon}{1-\epsilon}}{12}, \frac{\delta}{n} \right) \right\}, \right.$$

$$\left. h\left( \frac{\gamma \mu_i}{6(1-\epsilon+\gamma)}, \frac{\delta}{n} \right) \right\}$$

Finally, we use Lemma F.2 to bound the function $h(\cdot,\cdot)$. Since $\delta \leq 1/2$, $\delta/n \leq 2e^{-e/2}$. Further, $|\epsilon \mu_1 - \Delta_i| \leq 6$ for all $i$ and $\epsilon \leq 1/2$ implies that $\frac{1}{6(1-\epsilon)}|\epsilon \mu_1 - \Delta_i| \leq 2$ and $\frac{1}{6(1-\epsilon)} \min(\tilde{\alpha}_\epsilon, \tilde{\beta}_\epsilon) \leq 2$. $\Delta_i \leq 8$ for all $i$, gives $0.25\Delta_i \leq 2$. Lastly, $\gamma \leq 6/\mu_1$ implies that $\frac{\gamma \mu_1}{6(1-\epsilon+\gamma)} \leq 2$. Therefore,

$$\sum_{i=1}^{n} \min \left\{ \max \left\{ h\left( \frac{\Delta_i - \epsilon \mu_i}{4}, \frac{\delta}{n} \right), h\left( \frac{\Delta_i + \frac{\tilde{\alpha}_\epsilon}{1-\epsilon}}{12}, \frac{\delta}{n} \right), h\left( \frac{\Delta_i + \frac{\tilde{\beta}_\epsilon}{1-\epsilon}}{12}, \frac{\delta}{n} \right) \right\}, \right.$$

$$\left. h\left( \frac{\gamma \mu_i}{6(1-\epsilon+\gamma)}, \frac{\delta}{n} \right) \right\}$$

$$\leq \sum_{i=1}^{n} \min \left\{ \max \left\{ \frac{64}{(\epsilon \mu_1 - \Delta_i)^2} \log\left( \frac{2n}{\delta} \log_2\left( \frac{192n}{\delta(\epsilon \mu_1 - \Delta_i)^2} \right) \right), \right. \right.$$

$$\frac{576}{(\Delta_i + \frac{\tilde{\alpha}_\epsilon}{1-\epsilon})^2} \log\left( \frac{2n}{\delta} \log_2\left( \frac{1728n}{\delta(\Delta_i + \frac{\tilde{\alpha}_\epsilon}{1-\epsilon})^2} \right) \right),$$

$$\left. \frac{576}{(\Delta_i + \frac{\tilde{\beta}_\epsilon}{1-\epsilon})^2} \log\left( \frac{2n}{\delta} \log_2\left( \frac{1728n}{\delta(\Delta_i + \frac{\tilde{\beta}_\epsilon}{1-\epsilon})^2} \right) \right) \right\},$$

$$\left. \frac{144(1-\epsilon+\gamma)^2}{\gamma^2 \mu_1^2} \log\left( \frac{2n}{\delta} \log_2\left( \frac{432(1-\epsilon+\gamma)^2 n}{\delta \gamma^2 \mu_1^2} \right) \right) \right\}$$

$$= \sum_{i=1}^{n} \min \left\{ \max \left\{ \frac{64}{((1-\epsilon)\mu_1 - \mu_i)^2} \log\left( \frac{2n}{\delta} \log_2\left( \frac{192n}{\delta((1-\epsilon)\mu_1 - \mu_i)^2} \right) \right), \right. \right.$$

$$\frac{576}{(\mu_1 + \frac{\tilde{\alpha}_\epsilon}{1-\epsilon} - \mu_i)^2} \log\left( \frac{2n}{\delta} \log_2\left( \frac{1728n}{\delta(\mu_1 + \frac{\tilde{\alpha}_\epsilon}{1-\epsilon})^2} \right) \right),$$

$$\left. \frac{576}{(\mu_1 + \frac{\tilde{\beta}_\epsilon}{1-\epsilon} - \mu_i)^2} \log\left( \frac{2n}{\delta} \log_2\left( \frac{1728n}{\delta(\mu_1 + \frac{\tilde{\beta}_\epsilon}{1-\epsilon} - \mu_i)^2} \right) \right) \right\},$$

$$\left. \frac{144(1-\epsilon+\gamma)^2}{\gamma^2 \mu_1^2} \log\left( \frac{2n}{\delta} \log_2\left( \frac{432(1-\epsilon+\gamma)^2 n}{\delta \gamma^2 \mu_1^2} \right) \right) \right\}.$$

$\square$

# F Technical Lemmas

**Lemma F.1.** *If $a > 1$, $b > e$, and $t > \max(a \log(2b \log(ab)), e)$, then $\frac{a \log(b \log(t))}{t} \leq 1$*

*Proof.* **Step 1:** Plug in $t = a \log(2b \log(ab))$ to the expression $\frac{a \log(b \log(t))}{t}$.

$$\frac{a \log(b \log(a \log(2b \log(ab))))}{a \log(2b \log(ab))} = \frac{\log(b \log(a \log(2b \log(ab))))}{\log(2b \log(ab))}$$

Since $\log(\cdot)$ increases monotonically, the above is less than 1 if $b \log(a \log(2b \log(ab)) \leq 2b \log(ab)$.

$$b \log(a \log(2b \log(ab))) \leq 2b \log(ab)$$
$$\overset{(b>0)}{\iff} \log(a \log(2b \log(ab))) \leq 2 \log(ab)$$
$$\iff a \log(2b \log(ab)) \leq (ab)^2$$
$$\iff \log(2b \log(ab)) \leq ab^2$$
$$\iff 2b \log(ab) \leq e^{ab^2}$$

which is true if $a, b > 1$.

**Step 2:** Next, for $t > a \log(2b \log(ab))$, we wish to show that the inequality $\frac{a \log(b \log(t))}{t} \leq 1$ still holds. To do so, it suffices to show that $f(t) = \frac{a \log(b \log(t))}{t}$ is decreasing for $t > a \log(2b \log(ab))$. To see this, take the derivative.

$$f'(t) = \frac{a}{t^2 \log(t)} - \frac{a \log(b \log(t))}{t^2} = \frac{a}{t^2} \left( \frac{1}{\log(t)} - \log(b(\log(t))) \right)$$

This is negative when $\frac{1}{\log(t)} < \log(b(\log(t))$. Let $u = b \log(t)$. The previous is equivalent to the condition $b < u \log(u)$. For $t > e$, $u > b$ and $b > e$. Hence $b < u \log(u)$ completing the proof. $\square$

**Lemma F.2.** *For $\delta < 2e^{-e/2}$, $\Delta \leq 2$,*

$$t \geq \frac{4}{\Delta^2} \log \left( \frac{2}{\delta} \log_2 \left( \frac{12}{\delta \Delta^2} \right) \right) \implies C_\delta(t) = \sqrt{\frac{4 \log(\log_2(2t)/\delta)}{t}} \leq \Delta.$$

*Proof.*

$$\sqrt{\frac{4 \log(\log_2(2t)/\delta)}{t}} \leq \Delta \iff \frac{4 \frac{8}{\Delta^2} \log \left( \frac{1}{\delta \log(2)} \log(2t) \right)}{t} \leq 1.$$

If $\Delta \leq 2$, then $8/\Delta^2 \geq 2 > 1$. Similarly, if $\delta < 2e^{-e/2} < \frac{1}{e \log(2)}$, then $\frac{1}{\delta \log(2)} > e$. Hence, by Lemma F.1, setting $a = \frac{8}{\Delta^2}$ and $b = \frac{1}{\delta \log(2)}$, the above is true if

$$2t \geq \max \left( \frac{8}{\Delta^2} \log \left( \frac{2}{\delta \log(2)} \log \left( \frac{8}{\delta \Delta^2 \log(2)} \right) \right), e \right).$$

Trivially, $\delta \log(2) < 2$. Hence, $\delta < 2e^{-e/2}$ and $\Delta \leq 2$ implies

$$\frac{8}{\Delta^2} \log \left( \frac{2}{\delta \log(2)} \log \left( \frac{8}{\delta \Delta^2 \log(2)} \right) \right) \geq 2 \log \left( \frac{2}{\delta} \log_2 \left( \frac{2}{\delta \log(2)} \right) \right) \geq 2 \log(2/\delta) > e.$$

Therefore, we may simplify the maximum as

$$t \geq \frac{4}{\Delta^2} \log \left( \frac{2}{\delta} \log_2 \left( \frac{12}{\delta \Delta^2} \right) \right) \geq \frac{4}{\Delta^2} \log \left( \frac{2}{\delta} \log_2 \left( \frac{8}{\delta \Delta^2 \log(2)} \right) \right)$$

which implies the desired result. $\square$

**Lemma F.3.** *For any function $h(\cdot, \cdot) : \mathbb{R}^+ \times \mathbb{R}^+ \to \mathbb{R}^+$ that decreases monotonically in its first argument, we have that for any $a, b, c, \delta \in \mathbb{R}^+$*

$$\min\left(h(a, \delta), h(b, \delta)\right) \leq h\left(\frac{a+b}{2}, \delta\right)$$

*and*

$$\min\{h(a, \delta), \max[h(b, \delta), h(c, \delta)]\} \leq \max\left\{h\left(\frac{a+b}{2}, \delta\right), h\left(\frac{a+c}{2}, \delta\right)\right\}.$$

*Proof.* First, we bound the expression $\min\left(h(a, c), h(b, c)\right)$.

$$\min\left(h(a, \delta), h(b, \delta)\right) = h\left(\max(a, d), \delta\right) \leq h\left((a+b)/2, \delta\right)$$

Next, we bound, expressions of the form $\min\{h(a, \delta), \max[h(b, \delta), h(c, \delta)]\}$ using the above inequality.

$$\min\{h(a, \delta), \max[h(b, \delta), h(c, \delta)]\} = \max\left\{\min\left[h(a, \delta), h(b, \delta)\right], \min\left[h(a, \delta), h(c, \delta)\right]\right\}$$
$$\leq \max\left\{h\left((a+b)/2, \delta\right), h\left((a+c)/2, \delta\right)\right\}.$$

$\square$