[Reviews · NeurIPS 2020]

Review 1

Summary and Contributions: The paper is tackling the problem of best arm identification in a scenario in which we want to identify all the arms that are at most \eps distant from the optimum. The paper presents lower bounds on the sample required for solving the problem and upper bounds for two novel algorithms, behaving differently depending on the problem instance. Finally, some experimental results have been provided showing the empirical performance of the proposed methods.

Strengths: The problem you are tackling is novel, but derivative from the ones you mentioned in the related works section. None the less, it is an interesting setting and it is interesting for the bandit community.

Weaknesses: The novelty in the methods and setting is limited but significant.

Correctness: Yes. The claims are sound and the empirical evaluation is convincing.

Clarity: The paper is well written and is clear. The structure of the paper is peculiar but is instrumental to the line of thoughts you used to present the setting and the algorithms.

Relation to Prior Work: An extensive comparison with top-K and threshold bandits is discussed. The literature is properly described and critically analysed.

Reproducibility: Yes

Additional Feedback: I think you should mention in the abstract the fact that you are also providing some theoretical results on the number of samples required for solving the All-\eps problem and of the algorithms you proposed. It is not clear to me why you used in Figure 4.a solid lined that at some point become dashed ones. Please provide some insight about it. After authors feedback: The author feedback and the reviews of the other reviewers did not change my positive score about the paper.


Review 2

Summary and Contributions: This paper presents a new multi-armed bandit problem, called the ALL-epsilon problem, where a learner wishes to return all arms that are epsilon-good (relative to the best arm). The authors study this problem in the additive and multiplicative settings. They derive problem-dependent regret lower bounds for both settings. Further, they present two novel algorithms ((ST)^2 and FAREAST) and analyse their sample complexities in both additive and multiplicative settings. (ST)^2 is near-optimal and empirically appealing, and FAREAST is optimal (in view of the provided lower bounds).

Strengths: To the best of my knowledge, the ALL-epsilon problem is new, and I agree with the authors that it poses new challenges, and that it differs from existing similar problems, such as top-k and thresholding bandits. From a practical standpoint, and when compared to the closest counterparts, the ALL-epsilon problem seems to seek more natural objectives as motivated in detail in the paper. Overall I think the presented ALL-epsilon problem nicely expands the applicability of bandit problems. The paper thoroughly studies ALL-epsilon by considering two natural settings (additive and multiplicative) and presenting problem-dependent lower bounds for each setting. The authors provides two algorithms, one of which (FAREAST) is shown to be instance optimal. Finally, a strong aspect of the paper is to provide thorough numerical experiments to empirically assess the performance of the presented algorithms. They conduct numerical experiments on two diverse domains, and in both, the presented algorithms are shown to perform empirically well. I would like to add that the presented lower bounds use the standard techniques used in the literature. Also, both (ST)^2 and FAREAST borrow some key elements from existing algorithms for pure exploration. However, their analyses seem to pose some challenges.

Weaknesses: The lower bounds are stated for Gaussians with fixed and known variance. While at first it looks restricting, the proof techniques seem to allow generalization of the results for other distributions such as Bernoulli, etc. Could you explain whether this is the case? What about Gaussians with unknown variance? The presentation concerns ALL-epsilon in the fixed confidence setting. Would the fixed budget variant of the problem be a reasonable problem definition too? If yes, what about the involved challenges?

Correctness: Formalizing the ALL-epsilon problem, which in my opinion constitutes one core contribution of this paper, is methodologically correct and a natural choice. I have not enough time to check the proofs (in particular, in view of the very long appendix of this paper). The results, however, appear correct to me. I also believe that the empirical evaluation of the presented algorithms is conducted correctly.

Clarity: The paper is overall very well-written and easy-to-follows.

Relation to Prior Work: The presented ALL-epsilon problem falls into the category of pure exploration in bandits (or best arm identification). However, the problem is new, to the best of my knowledge, and the paper clearly states the difference between ALL-epsilon and similar problems (e.g., top-k and thresholding bandits). Overall the authors have done a good job of comparing this new problem to existing counterparts.

Reproducibility: Yes

Additional Feedback: The paper is very well-polished. However, I found a few typos reported below: l. 38: withing ==> Didn’t you mean “within”? l. 85: non empty ==> non-empty l. 126: sampling scheme ==> sampling schemes l. 190: similar ==> similarly l. 207: in the Theorem 3.1 ==> in Theorem 3.1 I. 185 and 228: I believe, O should be replaced by \Omega, since it is a statement about the lower bound. Some questions: 1- The presentation concerns ALL-epsilon in the fixed confidence setting. Would the fixed budget variant of the problem be a reasonable problem definition too? If yes, what about the involved challenges? 2- The lower bounds are stated for Gaussians with fixed and known variance. While at first it looks restricting, the proof techniques seem to allow generalization of the results for other distributions such as Bernoulli, etc. Could you explain whether this is the case? What about Gaussians with unknown variance? ==== AFTER REBUTTAL ==== I thank the authors for their responses to my questions. I have read the other reviews and the authors' response, and will increase my score.


Review 3

Summary and Contributions: This paper proposes a new variant of classical stochastic bandits and provides 2 algorithms to solve the proposed problem. The proposed problem consists of identifying all arms with means higher than a fixed offset from the best arm's mean, ie all $\epsilon$ good arms. The proposed algorithms to solve the problem are accompanied by a sound theoretical and empirical analysis.

Strengths: - To the best of my knowledge the proposed all-\epsilon good arms problem is novel. As the authors show, it is surprisingly quite different from other variations such as top-k arms identification and threshold bandits. - The paper is very well written and provides a lot of intuition behind the theorems/algorithms. - The two proposed algorithms are each accompanied by theorems stating upper bounds on their sample complexities and clear discussions on the tightness of these upper bounds. - The authors confirm their theoretical results with multiple empirical experiments.

Weaknesses: I list a few minor typos below, however, beyond that I do not see any major weaknesses in this work.

Correctness: I did not review the theorem proofs in the appendix. The content in the main body is sound and the empirical analysis is correct and supports the theoretical findings.

Clarity: The paper is very well written with clear and concise language. The short intuitive discussions around theorem statements are very helpful to the reader. I've also found very minor typos which I list below: -line 38: *withing -line 39: and it will make discoveries (unclear what it refers to) -Caption of Fig 3)a): *An typical setting (should be A typical setting)

Relation to Prior Work: According to the authors, and to the best of my knowledge, the proposed problem is novel. Related works are algorithms for top k arm identification and threshold bandits. The authors thoroughly discuss the relationship between their work and other problems as well as strengths/shortcomings of each.

Reproducibility: Yes

Additional Feedback: After reading the authors' feedback and other reviews, I decided to keep my score. (sorry this sentence was cut in my previous iteration).


Review 4

Summary and Contributions: This paper studies the problem trying to find all \eps-good arms in stochastic multi-armed bandits where an \eps-good arm is the arm with mean at least \eps close the the best arm. Two algorithms i.e., (ST)^2 and FAREAST are presented where (ST)^2 has good empirical performance and FAREAST has optimal theoretical guarantee. Comprehensive experiments against previous algorithms are also provided.

Strengths: The solution to the proposed problem seems through. Two algorithms are presented where one is good for use in practical applications and the other one has optimal theoretical guarantee. Both of these algorithms accept slack variables \gamma which makes the algorithm even more powerful for practical usage. Upper bound and lower bound of the algorithms are provided. Comprehensive experiments on both synthetic and real dataset are provided.

Weaknesses: One concern I have is regarding the experiments. The designed algorithms are supposed to maximize the probability of identifying all the \eps-good arms. However, in the experiments the paper is comparing the F1 scores which roughly approximate the fraction of arms returned that are actually good. I understand that these two goals are in the same direction. However, I am wondering what the experiment results look like when comparing the correct probability which is the number of trials an algorithm outputs all the \eps-good arms.

Correctness: I go through the explanation of algorithm ideas and they all look reasonable to me.

Clarity: The paper is well written and the theorems are clearly stated.

Relation to Prior Work: Yes, it is clearly discussed how this work differs from previous contributions.

Reproducibility: Yes

Additional Feedback: Overall the paper is well written with good presentation. And the analysis seems through. Typos: Line 92: "are very close or on \mu_1 - \eps": close to? Line 121: "i.e." -> "i.e.," Line 147: ditto

[Author Response · NeurIPS 2020]

We thank all reviewers for their time taken to review our work and their helpful comments and suggestions to improve this work. As a recap, in our work we studied the problem of returning all $\epsilon$-good arms. We establish lower bounds for the hardness of this problem and present two algorithms with theoretical guarantees, FAREAST and $(\text{ST})^2$. The former is asymptotically optimal on all instances. Additionally, via a novel moderate confidence bound, we are able to show that the latter, which enjoys great empirical performance, is optimal in many settings, such as the common $\delta = 0.05$. Below, we respond to individual review comments individually.

**Reviewer 1:** Thank you for your review. Addressing your comment about weaknesses and novelty: The core challenge of this work stems from the need to estimate the threshold $\mu_1 - \epsilon$ to sufficient precision which is new to this problem. This motivates our algorithms and affects our lower bounds. In particular, we develop a new theoretical technique for the lower bound in Appendix D to handle this that may be of independent interest. We will highlight this in future drafts to make this apparent. Thank you for the note about the abstract. We will be sure to clarify that the nature of our results are to provide sample complexity bounds. Thank you for pointing out the confusion regarding Figure 4a. In the sample data, there were 2.2M total ratings used to estimate the means. Before this number of samples, we draw the performance curves of each algorithm as a solid line, and after this point as a dashed line. This was to highlight that $(\text{ST})^2$ would have performed better in practice as well even at 2.2M ratings. We will clarify this in the final version.

**Reviewer 2:** Thank you for your careful read and comments about our paper. Regarding the lower bounds, Theorem 2.1 follows using standard techniques. To prove Theorem 4.1 however, in section D.2 we develop a novel technique that allows one to reduce a more general composite hypothesis test to the more limited set of tests covered via the Simulator lower bounding technique of [1], and this may be of independent interest. We will highlight and expand upon this in future drafts to make it more readily accessible. Regarding more general distributions for the lower bound: for Gaussians of unknown variance $\sigma^2$, the lower bound of Theorem 2.1 still applies (with an additional factor of $\sigma^2$). Intuitively this is since the known variance case should be an easier problem. Whether is is possible to achieve this bound or if a tighter bound can be proven when $\sigma$ is unknown is an interesting question for future work. For distributions beyond Gaussians, the result can easily be altered for any distribution with a well defined mean. In this case, terms such as $(\mu_1 - \epsilon - \mu_i)^2$ present in the bound will be replaced by KL-divergences between the appropriate distributions. Finally, we agree that a fixed-budget algorithm would be interesting to study and a useful setting in practice. We hope to study it in follow up work. Naively, $(\text{ST})^2$ could be altered to perform anytime fixed budget borrowing from techniques in [2] though this may not perform particularly well in practice and is not optimal. It is not immediately clear how to incorporate the ideas of FAREAST, specifically the technique in the Bad Filter, for an optimal fixed-budget algorithm and would be an interesting challenge for a future work.

**Reviewer 3:** Thank you for your kind review. We will correct the typos that you noted.

**Reviewer 4:** Thank you for taking the time to review our work. Regarding using the F1 score in experiments, F1 equaling 1.0 is equivalent to an algorithm exactly identifying the set of all $\epsilon$-good arms correctly - the objective of this work. Visualizing the F1 score as a function of the number of samples serves to demonstrate the expected sample complexity of the algorithm until $F1 \approx 1$, and also provides a continuous measure of the performance of the algorithm. In particular, even if $(\text{ST})^2$ has not yet found all $\epsilon$-good arms, it still achieves high F1 scores relative to other methods. Addressing the question about the correct probability directly, for synthetic experiments, we observed that the algorithms always correctly returned the set of all $\epsilon$ good arms at termination. For the Caption Contest data experiments in Figure 4a, we ran with $\delta = .1$ and terminated at 10M pulls even if the stopping criteria had not been satisfied. In those experiments, we observed that in almost all ($> 99\%$) cases, the F1 score of $(\text{ST})^2$ was 1.0 around 10M pulls and beyond, indicating that the algorithm correctly identified all $\epsilon$-good arms by this point and would terminate correctly. Hence, $(\text{ST})^2$ would achieve a correctness probability near 1 in this experiment - significantly better than $1 - \delta = .9$. In the case of the cancer dataset, we ran a small fraction to termination as well as with other values of $\epsilon$, and noted that the algorithm always returned the set correctly in all cases.

[1] Simchowitz, M., Jamieson, K., Recht, B. *The Simulator: Understanding Adaptive Sampling in the Moderate-Confidence Regime*. Conference on Learning Theory, 2017.
[2] Jun, K.S., Nowak, R. *Anytime Exploration for Multi-armed Bandits using Confidence Information*. International Conference on Machine Learning, 2016.


[Meta-Review · NeurIPS 2020]

This paper deals with the identification of the set of best arms when the set is defined in reference to a margin wrt. the best arm rather than to a number of best arms. All reviewers consider that this is an original and very complete contribution, which was confirmed during the post authors' feedback discussions. I recommed to accept the paper.